# Landform and lithospheric development contribute to the assembly of mountain floras in China

Wan-Yi Zhao [1,11], Zhong-Cheng Liu [1,2,11], Shi Shi [3,11], Jie-Lan Li[4], Ke-Wang Xu [5], Kang-You Huang [6], Zhi-Hui Chen[1], Ya-Rong Wang[1], Cui-Ying Huang [1], Yan Wang[7], Jing-Rui Chen[1], Xian-Ling Sun[4], Wen-Xing Liang[8], Wei Guo[9], Long-Yuan Wang[9], Kai-Kai Meng[1], Xu-Jie Li [1], Qian-Yi Yin [1], Ren-Chao Zhou[1], Zhao-Dong Wang[4], Hao Wu[4], Da-Fang Cui[3], Zhi-Yao Su[3], Guo-Rong Xin[8], Wei-Qiu Liu[7], Wen-Sheng Shu[1], Jian-Hua Jin[1], David E. Boufford [10], Qiang Fan [1] ✉, Lei Wang[2] ✉, Su-Fang Chen [1] ✉ & Wen-Bo Liao [1,7,8] ✉

Although it is well documented that mountains tend to exhibit high biodiversity, how geological processes affect the assemblage of montane floras is a matter of ongoing research. Here, we explore landform-specific differences among montane floras based on a dataset comprising 17,576 angiosperm species representing 140 Chinese mountain floras, which we define as the collection of all angiosperm species growing on a specific mountain. Our results show that igneous bedrock (granitic and karst-granitic landforms) is correlated with higher species richness and phylogenetic overdispersion, while the opposite is true for sedimentary bedrock (karst, Danxia, and desert landforms), which is correlated with phylogenetic clustering. Furthermore, we show that landform type was the primary determinant of the assembly of evolutionarily older species within floras, while climate was a greater determinant for younger species. Our study indicates that landform type not only affects montane species richness, but also contributes to the composition of montane floras. To explain the assembly and differentiation of mountain floras, we propose the 'floristic geo-lithology hypothesis', which highlights the role of bedrock and landform processes in montane floristic assembly and provides insights for future research on speciation, migration, and biodiversity in montane regions.

Globally, mountains play dual roles as museums and cradles of species diversity[1–5]. It is therefore unsurprising that much of the global biodiversity is concentrated among mountains, especially those within the tropics[6–8]. Worldwide, mountains harbor 39% of the global terrestrial vertebrate biodiversity, with 2.9 times more richness per unit area than lowlands[9]. In China, ten mountainous hotspot ecoregions were found to contain 92% of the plant genera and 91% of the terrestrial mammal species present within the entire country[10]. How such extraordinary diversity occurs across mountains has remained an open question since Humboldt's time[4].

Numerous hypotheses have been proposed to explain both montane and global biodiversity, such as those pertaining to climate stability[11], habitat heterogeneity[12,13], and energetics[14,15]. Along latitudinal gradients, current evidence suggests that biodiversity is affected by environmental energetics, particularly potential evapotranspiration (PET) and average annual temperature[16]. At a finer scale, plant alpha diversity in some extratropical mountain regions (such as Cape Region and East Australia Region) does not substantially differ from that in the tropics[8]. Contemporary climatic regimes do not sufficiently explain the pantropical diversity disparity in Neotropical and Indo-Malayan moist forests[17]. These results strongly suggest that montane species diversity is largely affected by habitat heterogeneity[13], or so-called geodiversity[18,19]. Moreover, the unique evolutionary history of each biological taxon in mountainous regions could exert a profound influence on local biodiversity[20-23]. It is clear that an integrated framework is needed for the prediction of montane biodiversity[20,22,24], and it should include ecological processes (e.g., survival, competition, and niche differentiation)[25], evolutionary processes (e.g., species divergence and extinction)[26,27], and geological processes (e.g., orogeny and lithosphere cycling)[17,28]. A recent attempt at such a framework, the 'mountain geobiodiversity hypothesis' (MGH), was first proposed to explain the biodiversity of the Tibeto-Himalayan region[2,29] and then extended to explain the origin of montane plant diversity at a global scale[3]. The MGH proposes that the evolution of montane biodiversity results from a combination of mountain uplift, geodiversity evolution, and Neogene and Pleistocene climate changes[3,29].

The key to explaining montane biodiversity is understanding the links between biotic processes and topographic erosion, which could contribute to increasing regional habitat heterogeneity[24]. Geological and lithological processes, particularly uplift and erosion, are known to strongly impact montane biodiversity, likely through effects on species formation, immigration, and extinction[15,28-30]. Mountains are cradles of species diversity largely because their formation and subsequent bedrock erosion yield topographic complexities and produce new niches for a wide variety of organisms[5,24,31,32]. Additionally, mountainous geographies facilitate the formation of endemic species specialized to certain types of rock or derived soils[33], especially on limestone (sedimentary rock)[34] or ophiolites (igneous rock)[28,35,36]. For example, previous studies have reported that at least 5%-10% of species exhibit edaphic specialization and, thus, are dependent on specific types of underlying bedrock[33,34]. Although it is well known that climate change forces plant species to migrate, those species that are adapted to local bedrock are constrained in their ability to migrate[33]. The geochemical characteristics of bedrock are on par with climate as regulators of vegetation in granitic mountains[37]. Some studies also suggest that local species diversification processes are consistent with edaphic rather than climatic filtration, such as in the Cape flora[38], Teesdale flora[39] and New Caledonian flora[40], in which approximately 50% of the endemic floristic elements are ultramafic-obligate species[41]. However, empirical studies establishing a relationship between the diversity of edaphic conditions and plant species diversity are still scarce[12]. The unique contributions of geological and lithological processes to local species assembly are often eclipsed by ecological factors (i.e., local climate) and are thus often overlooked[12,41,42].

At the continental scale, montane floras sharing the same underlying bedrock are often highly similar in terms of their plant family and genus compositions. This is the case for the Wuyi, Nanling, and Qinling Mountains in eastern Asia when compared among themselves and to the Appalachian Mountains of eastern North America[43-45]. The underlying bedrock of these mountains is primarily igneous and metamorphic rocks, which are partially responsible for their granitic landforms. The relationship between landform type and floristic composition suggests that the developmental processes of mountains may constrain floristic assembly. Thus, studying the relationship between species diversity and landform type may elucidate the process of floristic assembly in some of the most biodiverse regions of the world[31].

Here we explore the relationship between montane floristic assemblages and the types of landforms in which they occur. To accomplish this, we gather a dataset including 17,576 angiosperm species from 140 Chinese mountain floras representing five landforms categorized based on bedrock: karst, granitic-karst, granitic, Danxia, and desert (Figs. 1 and 2; Supplementary Table 1; see Methods). Here, we use the term 'flora' to refer to the collection of all angiosperm species growing on a specific mountain or in a well-delimited area[46,47]. Then, we calculate species richness, phylogenetic diversity (Faith's PD), phylogenetic structure indices (PDI, NRI, NTI), and mean divergence times (MDT) for each of the 140 floras[48-50] (Supplementary Table 2, see Methods). Finally, we construct regression models (1) using landform as a predictor (landform model hereafter) and (2) using landform along with tectonic, climatic, and geographic explanatory variables as predictors (full model hereafter, see Methods). These estimations enable us to compare the relative importance of landform type, after accounting for other predictors, in explaining the landform process–floristic assembly relationships.

## Results and discussion
### Landform effects on species richness and phylogenetic diversity
The angiosperm species richness in the granitic (median = 1456) and karst-granitic (median = 1458) mountain floras was higher than that in the karst (median = 1137), Danxia (median = 1132), and desert (median = 721) floras (Fig. 3a; Supplementary Fig. 2). The landform model explained 28.95% of the observed species richness deviance according to the generalized linear model (GLM) framework (Akaike Information Criterion, AIC = 116.21), and 31.40% according to the spatial error model (SEM) (AIC = 115; Supplementary Table 3). The full model explained 62.8% of the observed species richness deviance in the GLM and 63.7% in the SEM, and strong interaction effects between landform and mean temperature of the coldest quarter (TCQ) were detected (Supplementary Table 3). We also found weak interactions between landform and annual precipitation (PREC), as well as precipitation of coldest quarter (PCQ) (Supplementary Table 10). The models that integrate landform with any other variable significantly enhanced the explanatory capacity for the deviance of species richness (Supplementary Fig. 9), indicating that species richness is affected by landform effects. Specifically, mountains of sedimentary bedrock (desert, Danxia, and karst landforms) exhibited lower species richness than mountains of igneous bedrock (granitic and karst-granitic landforms).

According to the full model, higher values of longitude, greater differences in elevation (elevdiff), and higher mean TCQ positively affected species richness (Supplementary Tables 3, 10). The mountain floras with higher species richness were mainly located in the monsoon climatic zone of eastern China (Fig. 2). These observed positive relationships were generally consistent with the notion that habitat heterogeneity and precipitation are strong predictors of species richness[51,52]. Notably, high TCQ has a negative effect on species richness in desert landforms (Supplementary Table 3). In fact, when considering the interaction with the landform, high TCQ only positively affected the species richness in granitic landforms (Supplementary Fig. 9l).

We also found that high precipitation seasonality ($P_{var}$) and high mean temperature of the warmest quarter (TWQ) were negatively correlated with species richness, according to the full model (Supplementary Table 3). TWQ is a proxy for environmental energy flux, which has long been regarded as a driver of species richness at continental scales[53]. Interestingly, TWQ was not a significant predictor and explained only 0.9% of the variation in species richness (Supplementary Fig. 9g). The observed negative correlation between

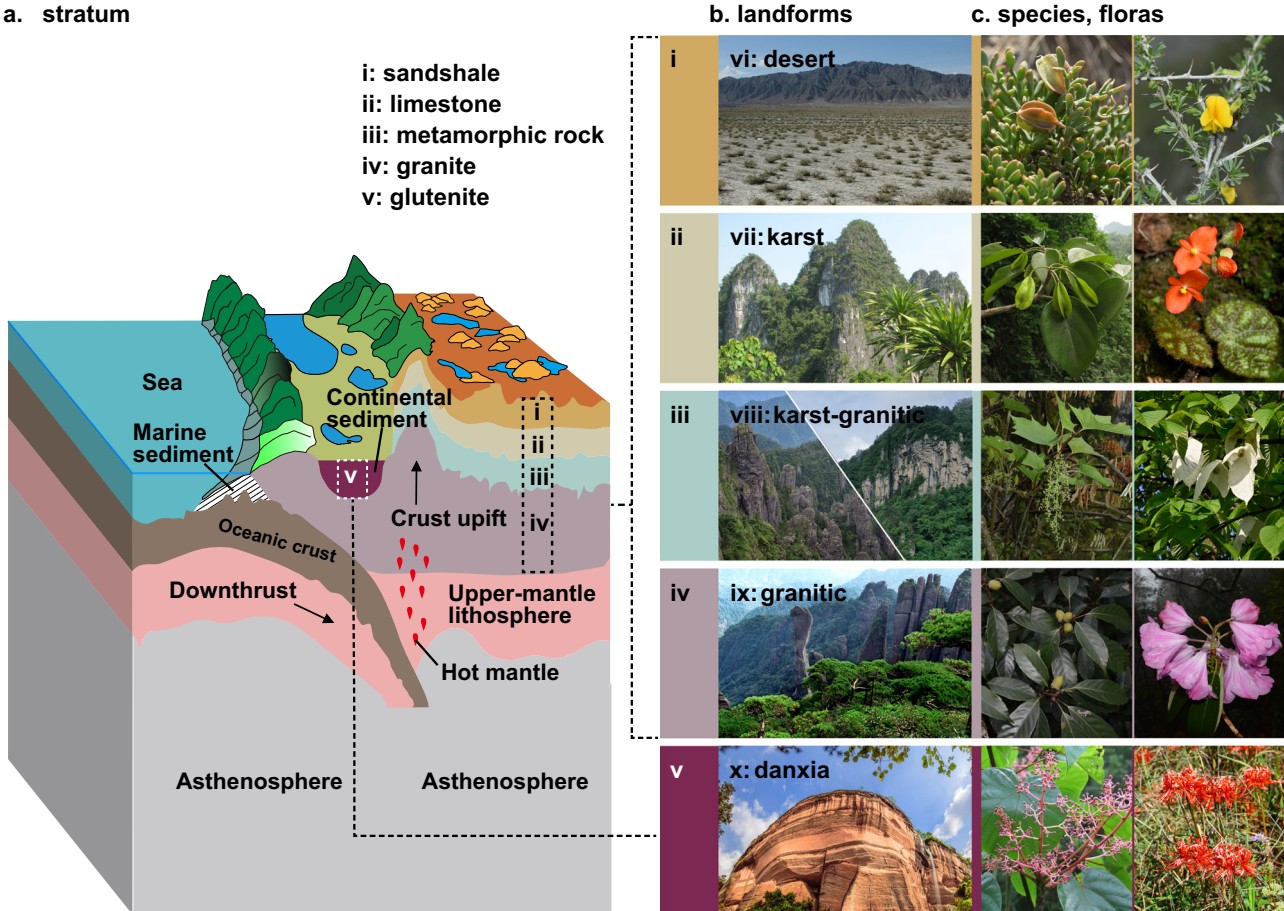

**Fig. 1 | Erosion of strata leading to the diversification of landforms and formation of floras. a** An example of plate tectonic movement, in this case resulting in uplift of marine strata (**i-iii**), invasive granite (**iv**), and subsequent accumulation of continental sedimentary strata (**v**). **b** Bedrock strata are shaped by erosion into different kinds of landforms (**vi-vi**). **c** The differentiation of rock strata and the formation of landforms further promote the diversification and immigration of plant species and the formation of different floras. The desert landform (**vi**) is usually developed on sandshale (**i**), and its floras are represented by species such as *Zygophyllum* and *Caragana*. Karst (**vii**) is developed on limestone (**ii**), and

*Excentrodendron* and *Begonia* are representative of its floras. The karst-granitic landform (**viii**) is developed on metamorphic rock (such as dolomite, quartz sandstone, or slate) (**iii**), and its floras include genera such as *Torricellia* and *Davidia*. The granitic landform (**ix**) is usually developed on granite (**iv**), and its floras contain genera such as *Cyclobalanopsis* and *Rhododendron* sect. *Ponticum*. The Danxia landform (**x**) is usually developed on continental glutenite (**v**), and representative genera include *Firmiana* and *Primulina*. All photographs published with permission according to the image rights agreement between each photographer.

species richness and high TWQ (Supplementary Table 3) may be the result of incorporating landform effects into the regression models (Supplementary Fig. 9i). In particular, the karst, Danxia, and desert landforms are characterized by sedimentary bedrock (e.g., limestone and glutenite), which are more permeable than igneous granite (Fig. 1). This may result in faster water loss from the soil in nongranitic landforms and, thus, a greater moisture deficit yielding lower species richness.

The correlational pattern between phylogenetic diversity and the five landforms was similar to the pattern for species richness (Fig. 2a, e), largely because phylogenetic diversity was positively correlated with species richness (Supplementary Fig. 3). The phylogenetic diversity index (PDI) of the desert (median = −16.59) landform was the lowest, followed by that of the Danxia (median = −3.23) landform, which was consistent with species richness (Fig. 3a, b). Unexpectedly, the PDI was highest for the karst (median = −1.03) landform, which exhibited lower species richness than granitic (median = −1.77) and karst-granitic (median = −1.27) landforms (Fig. 3b; Supplementary Table 5). The landform effect on PDI was significant in both the landform (explaining 70.89% of deviance in GLM, 77.15% in SEM) and full (explaining 88.09% of deviance in GLM, 88.25% in SEM) models (Supplementary Table 5). Thus, species with

the deepest phylogenetic divergences occur in karst landforms, while species with the shallowest divergences occur in desert landforms. The PDI results serve as an indicator of the level of floristic stability. Here, we would like to indicate that the species inhabiting the arid limestone mountains in our data exhibit an earlier divergence age and possess a remarkable capacity for long-term survival. For example, an Oligocene fossil flora discovered in Wenshan basin located in Yunnan, China, revealed a fossil assemblage (e.g., *Burretiodendron*, *Ficus microtrivia*), which clearly indicates that the current local karst vegetation may have existed since the early Oligocene[54,55].

According to the full model, orogenic, high latitude, high temperature annual range (TAR), and high TCQ were negatively correlated with PDI. High TWQ was the only variable positively correlated with PDI (Fig. 4; Supplementary Table 5). This suggests that mountains composed of igneous rocks, such as those with granitic landforms, may have recently undergone higher rates of evolution related to orogeny[24,56]. Furthermore, higher TWQ may have led to higher rates of extinction in mountains composed of sedimentary rock, such as karst landforms. Extinction rates in karst landforms are likely to be higher among closely related species due to their conserved ecological niches[57], thus yielding higher PDI. A low TAR tends to be related to environmental stability and consequently higher species richness,

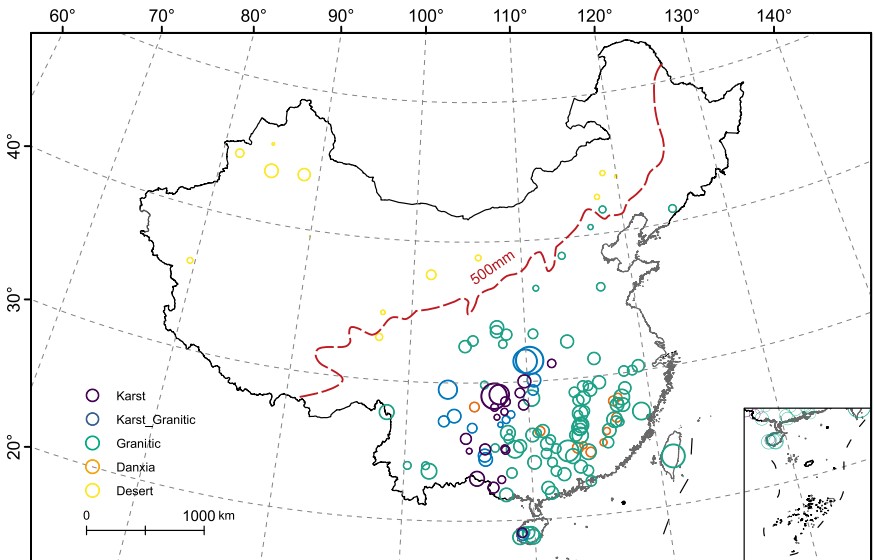

**Fig. 2 | Distribution of the 140 mountain floras in this study.** The dotted red line represents the 500 mm annual precipitation isoline[64], east of which is the monsoon climatic zone. The size of each circle represents the species richness of each mountain flora, with each flora normalized by $(x)/(x_{min})$. Source data are provided as a Source Data file.

especially in tropical areas[58,59]. Nevertheless, the TAR of karst land-forms was quite low (Supplementary Fig. 4), while the karst flora was characterized by lower species richness and higher PDI (Fig. 3a, b). Therefore, the coupling of landform effects and climatic factors in karst landforms may represent a unique mountain environment with a strong environmental filter, sustaining only highly persistent species[54,60,61].

**Landform effects on species age structure of floras**

The assembly history of mountain floras is reflected in both species composition and species age structure. To determine these distinctions between landforms, we calculated the MDTs of all species, mean divergence times of the youngest 25% of species (MDT.$_{youngest}$), and mean divergence times of the oldest 25% of species (MDT.$_{oldest}$) contained within the 140 floras. Among landforms, MDT, MDT.$_{oldest}$, and MDT.$_{youngest}$ exhibited the same patterns, with the median age of karst being highest, followed by karst-granitic, granitic, Danxia, and desert landforms (Fig. 3f–h). For MDT.$_{oldest}$, the results are similar for median age, while the mean age for karst-granitic landforms (24.66 Mya) is higher than that for karst (24.42 Mya). This may imply that species on karst landforms often survive during the transition to granitic despite limestone being a strong driver of the richness of endemic species[27,61] and increased limestone erosion tends to exacerbate the extinction of ancient local endemic species[12,33]. The survival of species within karst-granitic landforms may be due to niche evolution[62], with species adapting from alkaline soil to acidic soil, although this may be infrequent[61]. Our results also show that the MDT of floras on mountains in northern China, in particular desert floras, have significantly younger ages than floras in other landforms (Fig. 3f–h; Supplementary Fig. 5d). This may have resulted from the strong influence of glacial periods in the Pleistocene and necessitated relatively recent recolonizations for the northern mountain floras[63].

We further used regression models to compare the effects of landform and climate variables on the divergence times of floras. The results show that the landform type had significant effects on MDT, MDT.$_{oldest}$, and MDT.$_{youngest}$ in both the landform and full models (Supplementary Tables 6–8). Furthermore, landform effects had greater explanatory power for MDT.$_{oldest}$ than for MDT and MDT.$_{youngest}$ (Fig. 4; Supplementary Table 5). The full model indicated that high TAR and high TCQ were the most important climatic variables and were negatively correlated with the divergence times of floras. The standardized coefficient of TAR was higher for MDT.$_{youngest}$ than for MDT and MDT.$_{oldest}$ (Fig. 4). These results suggest that land-form effects have a greater impact on the assembly of ancient species (the oldest 25% of species) in mountain floras, while modern climates have a greater impact on the assembly of younger species (the youngest 25% of species).

Overall, our results are consistent with several prior studies[2,3,24] suggesting that the species age structure of mountain floras is closely related to landform processes (Supplementary Fig. 6). For example, most Chinese floras, including both montane and lowland floras, dif-ferentiated during the Miocene when the East Asian monsoon climate intensified[50,64]. Accordingly, this period also saw high rates of devel-opment of modern karst, Danxia, and granitic landforms in China[65]. At the intercontinental scale, the ages of floras were largely consistent with regional landform developmental processes. For example, the floras of eastern Asia are older than those of the Andes and Amazonia (Alpine orogeny in the late Cretaceous to Cenozoic), since the former's landform processes are much older while the latter's have occurred more recently[56,66]. Thus, the relationship between the floristic assem-bly of mountains and landform developmental processes seems to be global in scope, at least for angiosperms.

**Landform effects on the phylogenetic structure of floras**

Landform effects can also be observed in the phylogenetic structure of Chinese mountain floras. For example, the net relatedness index (NRI) for desert landforms (median = 9.45) was considerably higher (i.e., largely non-overlapping) than that of other landforms. This is espe-cially true for granitic landforms, which generally have the lowest NRIs (median = −1.47) (Fig. 3c). In fact, landform effects explained 64.91% (GLM) and 77.43% (SEM) of the variance in NRI in the landform model (albeit not significantly; Supplementary Table 9). In the full model, landform effects still had a unique contribution of 1.22% (out of 81.51% of the total variance explained in SEM). Among floras, the NRI results indicate that phylogenetic overdispersion (NRI < 0) occurred mainly in the southeast monsoon region of mainland China (Supplementary Fig. 5a), as has been inferred in prior studies[50,59]. Our results also show that the Danxia (median = −0.41) and karst (median = −0.42) landforms in the southeast monsoon region have higher NRIs than granitic (median = −1.47) and karst-granitic (median = −0.94) landforms

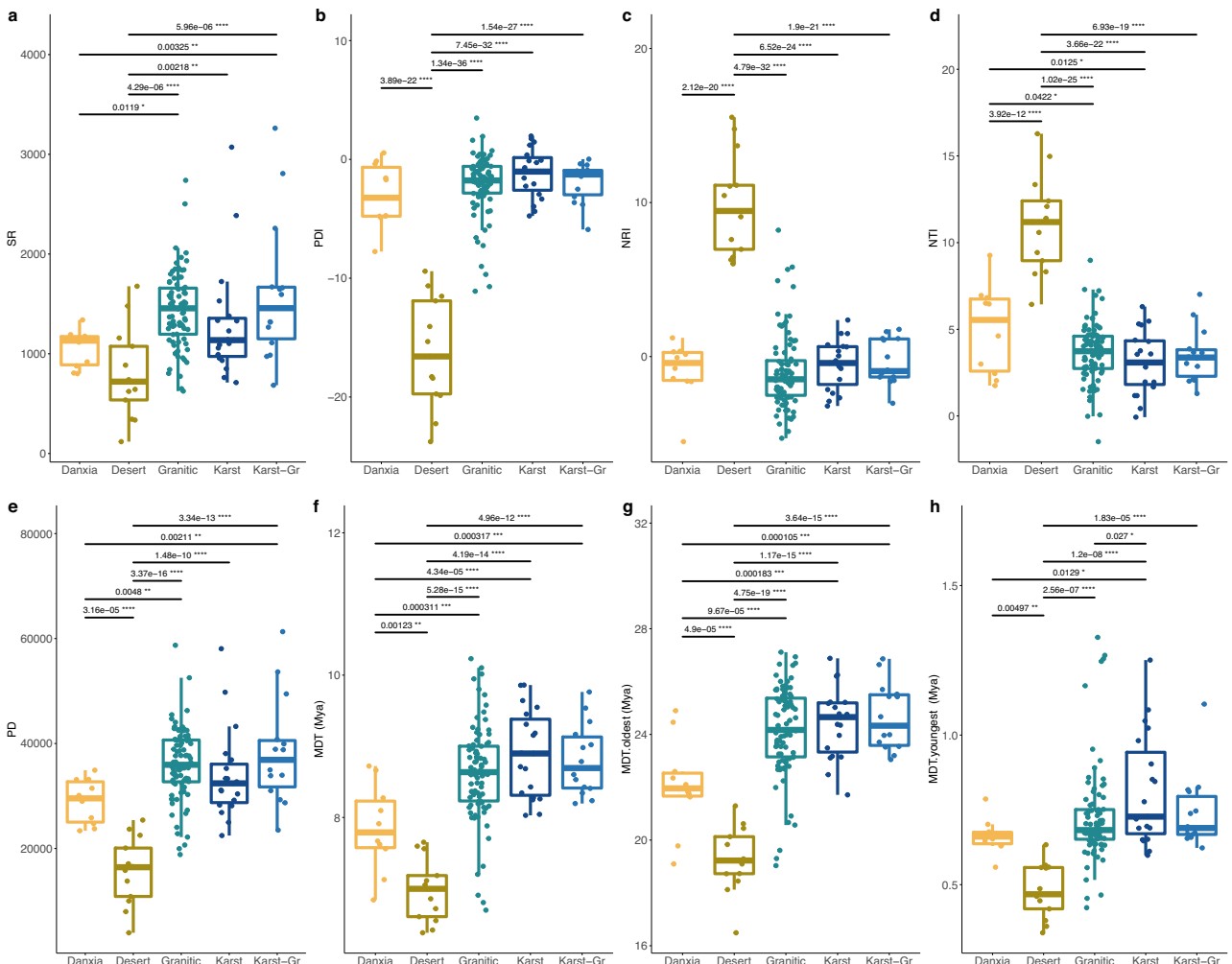

**Fig. 3 | Differences in species richness, phylogenetic diversity, phylogenetic structure, and age of floras among different landforms. a** species richness (SR); **b** phylogenetic diversity index (PDI); **c** net relatedness index (NRI); **d** nearest taxon index (NTI); **e** phylogenetic diversity (PD); **f** mean diversity time of all species (MDT); **g** mean divergence time of the oldest 25% of species (MDT.oldest); **h** mean divergence time of the youngest 25% of species (MDT.youngest). Karst-Gr, karst-granitic. The colors on the x-axis indicate different types of landforms. The sample sizes (n) for Danxia, Desert, Granitic, Karst, Karst-Gr are 10, 13, 84, 19 and 14, respectively. The box plots show the first and third quartiles (box limits), median (center line), and whiskers extend to a maximum of 1.5 times the interquartile range. Differences between each pair of landforms determined by using a two-sided, independent samples t test and P-values shown above the black line. ****$P < 0.00001$; ***$P < 0.0001$; **$P < 0.001$; *$P < 0.05$. Source data are provided as a Source Data file.

(Fig. 3c; Supplementary Fig. 5a). This means that granitic floras exhibited more phylogenetic overdispersion, a pattern also observed in the negative correlation with the nearest taxon index (NTI) in both the landform and full models (Supplementary Table 9). Phylogenetic clustering also occurs in some granitic landform floras, which possibly can be attributed to their occurrence in orogenic belts or colder climate zones (Supplementary Figs. 5a-b, Supplementary Table 1). Orogeny can facilitate rapid in situ evolutionary radiations, thereby promoting the co-occurrence of closely related species in mountains[56,67,68]. Moreover, environmental filtering effects further contribute to the aggregation of those species which could tolerant of cold and alpine environmental conditions, as anticipated by phylogenetic niche conservatism (PNC)[69].

Additionally, in the full model, TAR, which represents climatic instability of the habitat, was positively correlated with NRI (Supplementary Table 4). Highly unstable habitats are known to act as a strong ecological filter and lead to phylogenetic clustering. This is because speciation often occurs ex situ within a regional species pool, from which only certain immigrants can successfully establish[59,62,69]. Aside from their phylogenetic structures, differing landforms also exhibit

differences in their floristic compositions. For example, karst floras have more Malvales, Rosales, and Lamiales species; desert floras are dominated by Poales, Asterales, and Caryophyllales; and granitic floras are dominated by Magnoliales, Saxifragales, and Ericales (Fig. 1; Supplementary Fig. 7). In particular, karst-granitic floras represent a stage of transformation from karst to granitic floras and contain many relict lineages, such as *Torricellia* in Apiales, *Rhoiptelea* and *Platycarya* in Fagales, and *Davidia* and *Nyssa* in Cornales (Fig. 1c).

**Landforms play an important role in shaping floristic diversity**
Our results showed the unique effects of landforms on species richness and phylogenetic diversity. In general, higher habitat heterogeneity in mountain environments is a key driver of higher species richness[9,13]. Montane species richness is also recognized to be positively correlated with high temperature (eg., TWQ, TCQ) and precipitation (Supplementary Table 10)[70], which is consistent with the spatial heat and mass distribution within China. However, our full model indicated that the TWQ is negatively related to species richness (Supplementary Table 3). Specifically, TWQ and TCQ are only positively correlated with species richness in granitic landforms, but

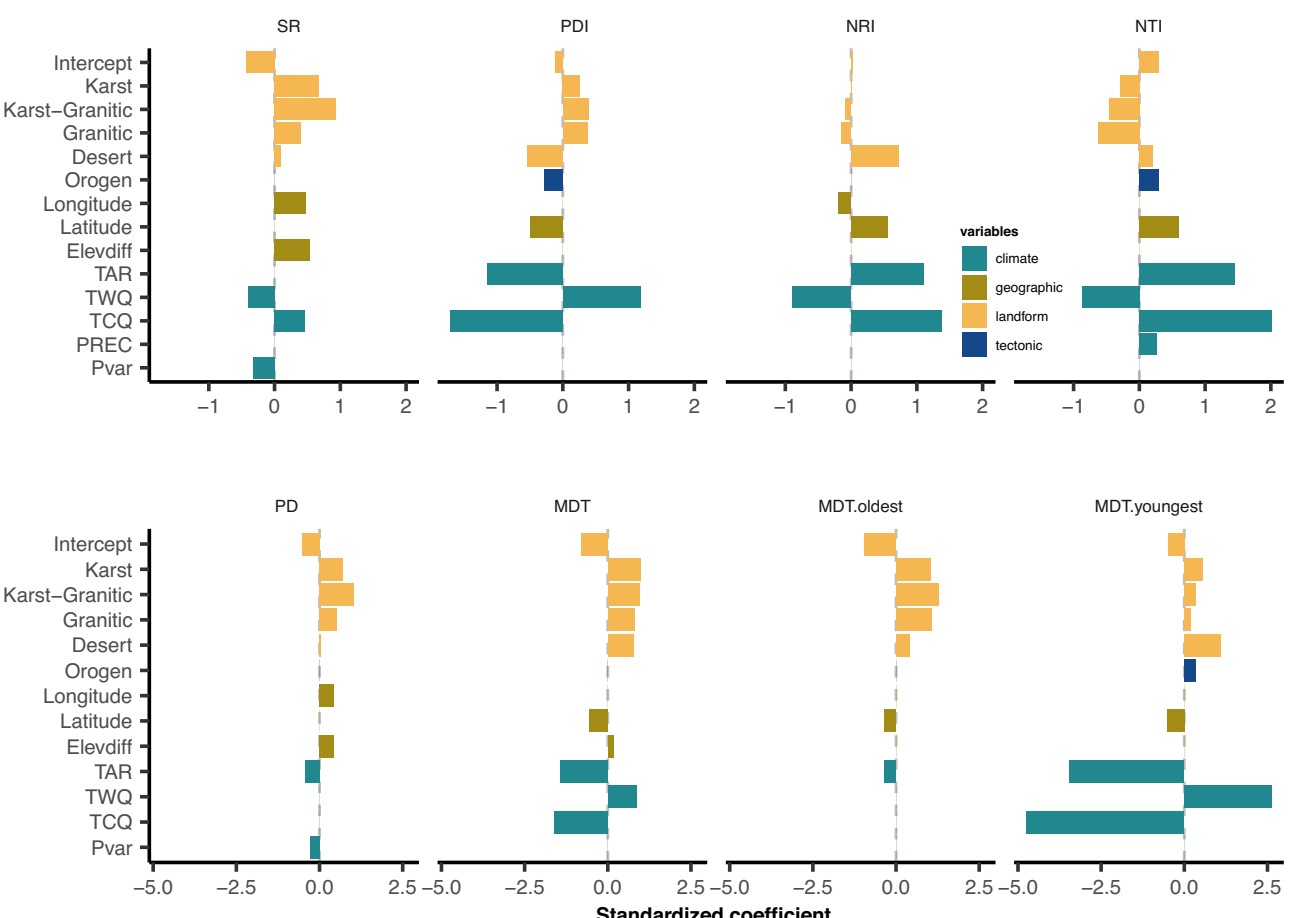

**Fig. 4 | Standardized coefficients of determination for species richness, phylogenetic diversity, phylogenetic structures and divergence times of mountain floras.** Definition of SR, PD, PDI, NRI, NTI, MDT, MDT.youngest, and MDT.oldest are as in Fig. 3. For SR, NRI, PD, MDT and MDT.oldest, the landform effects are coded in reference to Danxia (the intercept). For PDI, NTI, and MDT.youngest, both landform and tectonic effects were coded in reference to the Danxia + craton (the intercept). Compared with Danxia + craton as a baseline, karst, karst-granitic, and granitic mountains located in the orogenic belt have lower NTIs and higher PDIs.

Legend colors indicate different explanatory variables, including landform (karst, karst-granitic, granitic, Danxia, desert), tectonic type (Orogen, mountains located in the orogenic belt), geographic (Longitude; Latitude; Elevdiff, difference between the highest and lowest elevation), and climate (Isoth, isothermality; TAR, temperature annual range; TWQ, mean temperature of the warmest quarter; TCQ, mean temperature of the coldest quarter; PREC, annual precipitation) (Supplementary Table 2). The length of each bar represents the explanatory power of the variables. Source data are provided as a Source Data file.

negatively in other landforms (Supplementary Fig. 9). The special contribution of landform effects was difficult to detect due to interactions with temperature and precipitation (Supplementary Table 9). Landform developmental processes are associated with local bedrock and regional climate[24,71]. The assembly of a local flora is impacted by a combination of geological and climatic processes, as well as biological processes[2,24,56]. However, the species richness in mountains of igneous bedrock (granitic and karst-granitic) is clearly higher than mountains of sedimentary bedrock (karst and Danxia) (Fig. 3a). The underlying reason is that water cycle processes and rock erosion rates differ between igneous and sedimentary mountain ecosystems[72]. In limestone mountains, water can easily be lost through underground river systems, while a greater quantity of overland runoff is available for plants in mountains of igneous bedrock[73,74]. In extreme cases, high temperature can accelerate water loss, further reducing species richness (Supplementary Table 3). This occurs in desert, Danxia, and karst landforms (Supplementary Fig. 9). Although our results highlight the impact of landform on floristic diversity, we do not intend to deny the importance of climate. We hold the opinion that montane species richness is determined by the availability of water and energy (which is combined result of bedrock, temperature, and precipitation) in mountain ecosystems.

## Bedrock promotes local speciation resulting in floristic differentiation between landforms

Based on comparisons between individual floristic phylogenetic structures and MDT, we found that landform effects partially determine the final composition of montane floras. Except in desert landforms, which are dominated by drought[75], the phylogenetic structure of floras associated with mountains of sedimentary bedrock (karst and Danxia) were more clustered than mountains of igneous bedrock (granitic and karst-granitic) (Fig. 3c, d; Supplementary Fig. 7). This may have resulted from both local speciation and strong habitat filtering effects. Bedrocks, from which landforms are derived, promote the evolution of endemic edaphic specialists[33]. For example, the rapid development of karst landforms since the Miocene is thought to have triggered adaptive diversification in several genera, e.g., *Begonia*[76] and *Primulina*[77]. Similarly, the local richness of endemic species in Mount Kinabalu (Malaysia) is primarily made up of preadapted and locally derived species on high-elevation granite pluton[78]. Radiative evolution results in the aggregation of average relatedness among species within a flora[50]. Such effects further promote the phylogenetic relatedness (clustering) of mountain floras because new lineages tend to maintain their ancestral ecological niche[58,62].

On the other hand, adaptive evolution, encompassing both morphological and physiological traits, could play a pivotal role in

facilitating the diversification of plants in novel environments[79]. In plants, radiative evolution often accompanies habitat and landform shifts, as seen in Old World gesneriads[80] and North American desert rock daisies (Compositae tribe Perityleae)[81]. Similar patterns can also be observed in insects. For example, two radiating clades (nodes 14 to 16 and nodes 31 to 33) of *Exocelina* has been suggested as a resulted of ecological niche transition from the transition from uplifted Australian Plate bedrock to ultramafic/ophiolite[82]. Although, many previous studies have documented climate fluctuations as an important driver of species radiations[3,61,83,84]. The contribution of bedrock type on plants adaptive radiation of should not be ignored. For example, the development of a key innovation (lime-secreting hydathodes) may have made *Saxifraga* sect. *Porphyrion* better suited to limestone habitats[85], and the low specific leaf area (SLA) exhibited by *Erica* maybe an adaptation to oligotrophic habitats (quartzite/sandstone) in the Cape[67]. These landform and bedrock effects could strongly promote both species and floristic differentiation between different regions[80,86,87].

### Restricted dispersal with establishment between landforms as the result of environment filtering

For many biogeographers, mountains are regarded as both barriers and bridges of species dispersal[5]. The role of mountains as corridors has been documented in those of North-South orientation, such as the Andes[88] and Hengduan Mountains[89]. However, the contribution of dispersal to montane floristic diversity[2,14,56] largely depends on the ecological and physiological requirements of the species[5,90], since dispersal only affects regional species diversity if it is followed by successful establishment[90]. Our research demonstrates the role of landform constraints on the interaction of different landform floras, which is shown in their species richness, phylogenetic structures, and species age structures (Figs. 3–4; Supplementary Figs. 9–15). The dispersal process is generally less constrained during the initial stages of mountain landform development, which are characterized by gentle slopes and limited geographical barriers[24,56]. This scenario is well demonstrated by the assembly of alpine biotas. Since the Miocene, the colonization rate in the gentle elevation gradient Qinghai-Tibet Plateau (QTP) (0.06-0.25) is always larger than that in the QHM (< 0.05)[56]. The role of local species recruitment is most important during the early stages of mountain floristic assembly, subsequently being supplanted by local adaptation or in situ speciation due to the emergence of heterogeneous mountain environments[24,56]. Mountains in different landforms will recruit different plant species as a result of environmental filtering caused by differences in bedrock exposure[33,91]. For example, mountains composed of limestone bedrock contain more species which are physiologically tolerant of drought and high calcium stress than mountains composed of metamorphic rocks and granites[74,92,93]. The landform restriction effect on species diffusion gradually strengthens when more bedrock is exposed and the connectivity between mountains of different landforms is greatly reduced[42,94,95]. Variation in the species composition of mountain floras between different landforms increases under the combined effects of landform, environmental filtering[96,97], and local endemic speciation[33]. Therefore, we propose that the patchy spatial distribution of different mountain landforms is an important factor in shaping biogeographical zoning.

### Concluding remarks

Our results highlight that bedrock and landform effects play a key role in floristic assembly, especially considering that Earth's bedrock is unevenly distributed[98]. The primary reason why landforms might explain floristic assembly is that they represent or underlie major aspects of environmental filters to which plants respond via the processes of speciation, local extinction, and immigration from the regional species pool[95].

Here, we propose the 'floristic geo-lithology hypothesis' to explain the assembly and differentiation of mountain floras. In this theoretical framework, floristic assembly in mountains is driven by the lithospheric cycle, which refers to the bedrock-constrained developmental processes of landforms. Specifically, under this hypothesis, montane species differentiation is closely related to the type of bedrock and degree of erosion. Both the species richness and species composition of mountain floras result from interactions between the landform and the environment. In addition, the dispersal of plants between different landform types is more restricted than that within the same landform type. Successful diffusion across a landform is often accompanied by the emergence of adaptive traits or speciation.

To explain montane species diversity, this hypothesis differs from those such as the MGH[2,29], which focuses more on the origination of high levels of biodiversity found in mountain systems. The MGH is invoked to explain the cause of alpha diversity. In contrast, our hypothesis is more concerned with the process of mountain flora differentiation, which is a hypothesis of beta diversity. Here, we would like to introduce the concept of 'landform flora' for mountain biodiversity studies, meaning a unique flora formed under the influence of bedrock erosion and mountain landform development processes. Recognizing the differences that exist between different 'landform flora' (e.g., granitic flora, karst flora, Danxia flora) will benefit future studies in the prediction of mountain biodiversity and speciation, and also in species protection[57,80,99]. We argue that the 'floristic geo-lithology hypothesis' presented here could serve as a general explanation for global diversity patterns, as the formation of mountains on the Earth's surface is the result of the cycling of sedimentary, igneous, and metamorphic rocks. In conclusion, our study highlights the floristic patterns of different landforms and provides a framework for studying the mechanisms of plant species diversification within mountains and the distributional patterns of mountain floras worldwide.

## Methods
### Study area and sampling units

The physical geographic environment of modern China has experienced four major orogenic events since the Palaeozoic: the Caledonian, Indosinian, Yanshan, and Himalayan orogenies[100]. These events represent cycles of uplift of marine strata followed by geologic erosion and subsequent formation of a variety of landform types[65]. These Chinese landforms are home to more than 30,000 species of vascular plants[101,102]. China is a natural laboratory for investigating patterns of biodiversity due to its heterogeneous physical geography and range of habitats as well as its large geographic size and considerable biological diversity[50,63]. In this study, the geographic sampling units were the montane floras of protected areas, such as nature reserves and forest parks (Supplementary Table 1). Comprehensive scientific surveys encompassing physical geography and biodiversity have been conducted in these areas, serving as the fundamental cornerstone of our study. A total of 140 mountain floras were included in this study. Maps of China used in this study were adapted from DataV. GeoAtlas (http://datav.aliyun.com/portal/school/atlas/area_selector) and visualized in ArcGIS 10.8 (http://www.esri.com/).

### Dataset generation and reconciliation

We compiled checklists of angiosperm species for each mountain flora from previously published, comprehensive species checklists, white papers, and research papers (Supplementary Table 1). From our initial checklists, we excluded all nonnative species and reconciled the taxonomy with the Leipzig Catalogue of Vascular Plants (LCVP) using the R4.1.0 (http://www.r-project.org/) package lcvplants[103], with infraspecific taxa combined under their respective

species[59]. To determine the generic, familial, and ordinal affinities of the species, we used APG IV[104] and the Angiosperm Phylogeny Website (http://www.mobot.org/MOBOT/research/APweb). Following taxonomic reconciliation and categorization within higher ranks, our dataset comprised a total of 17,576 species in 2,585 genera belonging to 251 families and 56 orders. We used these data to generate a presence-absence (1/0) matrix with 140 mountain sites in columns and the species represented in rows.

## Phylogenetic reconstruction

Using the recently published, dated megaphylogenetic tree GBOTB.extended.LCVP.tre[49] as a backbone, we generated a phylogeny of the study species using the R package V.PhyloMaker2[49]. Of the 2,585 genera and 17,576 species studied here, 2,349 genera and 8,663 species were included in GBOTB.extended.LCVP.tre. Based on previously published megaphylogenies[50,105], we treated each of the 236 missing genera as sisters to their most closely related genera in GBOTB.extended.LCVP.tre using R package V.PhyloMaker2[49]. Although this method resulted in more robust phylogenetic relationships than Phylocom[106], the ultimate phylogenetic relationships should still be considered relative, as complete phylogenetic data are still lacking for many families and genera. We added study species that were absent from GBOTB.extended.LCVP.tre to their respective genera using Phylomatic and generated their branch lengths with BLADJ[106], as implemented in the R package V.PhyloMaker2[49]. Subsequently, in package V.PhyloMaker2, we used build.nodes.1 to extract genus and family information for downstream uses within the algorithm and generated our final megaphylogeny (Fig. S2) using Scenario 3, in which species were added to the backbone topology at the phylogenetic midpoint within their respective genera. Scenario 3 is regarded as the most robust of the three available approaches within the software package[107]. We visualized the resulting megaphylogeny using iTOL v6 (https://itol.embl.de/)[108].

## Indices of phylogenetic diversity and structure

To measure the phylogenetic diversity of each mountain flora, we employed Faith's phylogenetic diversity (PD$_{Faith}$)[109], which is the sum of all phylogenetic branch lengths within the subtree representing the flora and is known to be positively correlated with species richness (SR)[110]. We also used the phylogenetic diversity index (PDI)[111], which standardizes PD$_{Faith}$ using null models. Thus, the PDI allows comparisons among floras with different underlying species richness[59]. To calculate the PDI, we used package PhyloMeasures[112] in R, in which the null model was set as uniform, and the following typical algorithm was implemented:

$$PDI = (PD_{observed} - PD_{randomized})/(sdPD_{randomized}) \quad (1)$$

We also applied the net relatedness index (NRI) and the nearest taxon index (NTI), which are widely used to investigate the phylogenetic structure of species assemblages, i.e., clustered or overdispersed[113,114]. NRI is a measure of the standardized effect size of mean phylogenetic distance (MPD) and primarily reflects the structure at deeper nodes of the phylogeny. NTI is based on the mean nearest taxon distance (MNTD), which is the mean distance between each terminal taxon and its sister lineage and reflects shallower nodes within the phylogeny. NRI and NTI were determined as follows[113]:

$$NRI = -(MPD_{observed} - MPD_{randomized})/(sdMPD_{randomized}) \quad (2)$$

$$NTI = -(MNTD_{observed} - MNTD_{randomized})/(sdMNTD_{randomized}) \quad (3)$$

In these equations, MPD$_{observed}$ and MNTD$_{observed}$ are the observed MPD and MNTD, MPD$_{randomized}$ and MNTD$_{random}$ are the expected (i.e., average) MPD and MNTD of the randomized assemblages[115], which were calculated based on the null model uniform in the R package PhyloMeasures[112], and sdMPD$_{randomized}$ and sdMNTD$_{random}$ are the SD of the MPD and MNTD for the randomized assemblages. A positive value of NRI or NTI indicates phylogenetic clustering, whereas negative values indicate phylogenetic overdispersion.

## Species divergence time estimation

We calculated the mean divergence time (MDT) for each flora using the mean ages of its species[50] according to the dated phylogenetic tree generated in R package V.PhyloMaker2[49]. The divergence time of each species used to calculate MDT was not the absolute age, as they were extracted from the megatree generated in R package V.PhyloMaker2[49]. In this approach, divergence time of species is expected to be overestimated, as the branch of some species in a local phylogeny is usually longer than that in the global phylogeny (including all species). For instance, if a lineage became extinct, the divergence time of its existing closest relative species would be dated at the point of their last common ancestor. To assess the robustness and the effect of this sampling bias on the final results of species age structure of a mountain flora, we used four divergence time datasets and found similar MDT patterns between mountains of different landforms (Supplementary Fig. 8). This result is consistent with a study[114], which found that "in large-scale biodiversity and phylogenetic analyzes, sources of noise in divergence time estimation are to be expected, but they did not affect the reliability of the results". We believe that our dated megaphylogenetic tree was suitable for this study because our aim was to reveal the general patterns of landform influence on the formation of mountain flora rather than focusing on the age of each species.

We facilitated comparisons among landforms by mapping the 140 floras to their landform type through the integration of spatial data in ArcGIS v. 10.8. Each species within a flora was assembled at a different time[30,32,56]. Previous studies have shown that the species ages within floras are quite different and that environmental variables have better explanatory power for herbaceous species[50]. Because herbaceous species have shorter generation times than woody species and, consequently, tend to be evolutionarily younger[116]. To investigate whether old species and young species exhibit different patterns of assembly, we partitioned all species into quartiles based on their divergence times and, in addition to computing the MDT of all species, we also calculated the MDT of the oldest 25% of species (MDT$_{\cdot oldest}$) and of the youngest 25% of species (MDT$_{\cdot youngest}$) for each mountain flora following the method in Lu et al.[50].

## Predictor variables

The diversity and phylogenetic structure of mountain floras are influenced by many factors, including climate, regional geologic history, and geographic heterogeneity[13,31,117]. To take into account aspects of these factors, we obtained data representing a total of 18 predictor variables (Supplementary Table 2). For each of the 140 mountain floras, these variables included geographic characteristics, type of landform and tectonic plate, and climatic features. The 18 variables were obtained as follows:

Geographic information included longitude, which reflects the distribution pattern of rainfall on the Chinese mainland; latitude, which reflects the temperature gradient; area; median elevation (elevmid) of species; and elevational range of each mountain (elevdiff), which is related to local heterogeneity[118]. We obtained these geographic information data from local governmental reports on physical geography (Supplementary Table 1).

We assigned a landform type to each mountain based on relevant regional geological and geomorphological survey reports (Supplementary Table 1), as well as a world geological map (http://portal.

onegeology.org/OnegeologyGlobal/). In general, the developmental stage of a mountain landform depends on the stage of its rock-stratigraphic denudation[65,119,120]. Therefore, based on the sequence of stratigraphic denudation and bedrock type occurring in a mountain, we defined five types of landforms: karst, karst-granitic, granitic, desert, and Danxia (Fig. 1).

Karst develops from a high-carbonate limestone stratum with a marine sedimentary origin[121]. Granitic landforms are characterized by igneous bedrocks, which are crystalline and poorly soluble in water compared to limestone. Granitic landforms are formed when the intrusion of acidic magmatic rocks causes overlying strata to become denuded and exposed[65,72]. In cases where the bedrock of a mountain is composed of both limestone and granite, it is defined here as a 'karst-granitic' landform and represents the intermediate state of karst evolving into a granitic landform. Danxia landforms are made up of non-marine clastic rock and characterized by red walls and cliffs, which are usually Mesozoic continental sediment strata and were developed along with the Himalayan orogeny in China[65,122]. The development of desert landforms is usually closely related to aridification[75]. In China, desert landforms are mainly located in the northern and western provinces and consist of flat, arid plains and exposed, rocky mountains resulting from the strengthening of the winter monsoon on the mainland after the Neogene[123]. Therefore, 'desert' in this study represents the mountain floras located in the arid region. We used these five types of landforms to classify the 140 mountain floras included in this study, which comprised 19 karst floras, 14 karst-granitic floras, 84 granitic floras, 13 desert floras, and 10 Danxia floras (Supplementary Table 1, Fig. 1).

We distinguished the tectonic plate to which each mountain belonged by referring to the *Plate Tectonic Regionalization of China*[124]. The mountains on cratons, such as the Yangtze or North China Cratons, represent stable geological regions and were coded as 'craton'. In contrast, maintains located in orogenic belts were coded as 'orogenic'.

For each of the 140 mountain floras, we downloaded CHELSA climate data (v. 1.2, available at http://chelsa-climate.org/) at a spatial resolution of 30 arc-seconds[125], and extracted the climatic variable mean values of each mountain layer using the zonal statistics function in ArcGIS 10.8. CHELSA is a high resolution climatology dataset widely used in recent years for modeling species distributions and inferring the evolution of climatic niches[24,125]. The 19 bioclimatic variables of CHELSA, BIO1-BIO19, describe temperature, precipitation, and fluctuations in temperature and precipitation at various time scales[125]. Of the 19 variables, we excluded eight that had pairwise Pearson correlation coefficients > 0.95 to avoid collinearity. We included the remaining 11 bioclimatic variables in our analyzes (Supplementary Table 2).

### Data analysis

A total of 140 mountain floras were included in our analysis. These mountains were representative of major Chinese climate regions and showed a high degree of variation in plant species richness, climate, and geology (Fig. 2; Supplementary Table 1). For each mountain, we log-transformed geographic area[126] to account for power relationships to species diversity, phylogenetic diversity, and MDT. The other 15 numerical variables included all 11 climatic variables, longitude, latitude, elevmid, and elevdiff. We standardized these 15 variables from 0-1 according to the formula $(x - x_{min})/(x_{max} - x_{min})$, following Ricklefs & He[53].

We used generalized linear models (GLMs) to model log-transformed species richness as the response variable and landforms, tectonics, and climate variables as predictors. Initially, we modeled species richness as a function landform only (i.e., landform model) because, in this study, we focused primarily on the roles of landforms in floristic assembly. However, we also extended the landform model to include all other variables to assess the effect of climate on species diversity (i.e., full model). The determinants of species richness might change with landform type, and we therefore test for interactions between landform and other predictor variables (only significant variables are shown in the full model). We further used GLM to determine the effects of landform, tectonics, and climate on phylogenetic diversity, NRI, NTI, PDI, MDT, MDT.oldest, and MDT.youngest (Supplementary Table 2). As with species richness, we initially modeled landform as a single predicting factor (i.e., landform model) before extending to all variables within a full model. We performed all GLM analyzes in R version 4.1 (https://www.r-project.org/) using the glm function in the package MASS[127].

The step function in package MASS is used for Akaike Information Criterion (AIC) model selection to derive a minimum adequate model[128]. We further applied the 'leave one out' approach to find the best model for each full model. Under this approach, the importance of a variable is evaluated by comparing a full model that includes the variable to a reduced model that excludes it[53]. We also analyzed the standardized coefficients of predictors to compare the importance of each predictor variable based on Antonelli et al.[24].

Spatial autocorrelation is a general feature of macroecological data and may lead to erroneous interpretations[129]. We used Moran's I values to quantify residual spatial autocorrelation. These are considered the spatial equivalent of Pearson´s correlation coefficients and normally vary between 1 and −1, with values close to 0 indicating a lack of spatial autocorrelation[130]. Because of spatial autocorrelation present in our dataset, we performed a spatial error model (SEM) to account for residual spatial autocorrelation[131]. The expected Moran´s I values for the response variable, as well as for GLM and SEM residuals, are shown in the appendix (Supplementary Tables 3−9). Spatial statistics were performed with the package spdep in R version 4.1 (https://www.r-project.org/).

### Reporting summary

Further information on research design is available in the Nature Portfolio Reporting Summary linked to this article.

## Data availability

All original mountain flora data used in this study have been published and are accessible to readers from the cited sources (Supplementary Table 1). A standardized distribution dataset of the 140 Chinese mountain floras and dated phylogenetic tree are provided with the paper, available at https://doi.org/10.5061/dryad.b2rbnzsk1[132]. Climate data was downloaded from the CHELSA climate data (v. 1.2, available at http://chelsa-climate.org/). A dataset containing all the necessary predictor variables for evaluating the conclusions of this study is provided as Supplementary Data 1. Background map shapefile for Fig. 2, and Supplementary Fig. 5 is available on the DataV. GeoAtlas (http://datav.aliyun.com/portal/school/atlas/area_selector). Source data are provided with this paper and can also be found at https://doi.org/10.5061/dryad.3n5tb2rkg[133].

## Code availability

R code and related data needed to generate the figures and tables in this study is available at https://doi.org/10.5061/dryad.3n5tb2rkg[133] and https://doi.org/10.5281/zenodo.6374741[134].

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

## Acknowledgements

We thank Brach A. R., Chen G.-X., H. Zhu, Li Z., Miao S.-Y., Ye H.-G., Shui Y.-M., Zheng B.-J., Wu Z.-H., Jia F.-L., Zhan X.-H., Liu K.-M., Yang X.-B., Ding M.-Y., Lin S.-S., Lee S.-Y., Wei S.-J., Shen J.-N., Pan J.-W., Zou Y.-L., Zhang X.-J., Liu J., Zhang J.-J., Liu N.-N., Wen J.-L., Liu J.-Y., Wang Y.-Q., Zhu H.-L., Xu Y.-G., Wang L., Ai T.-L., Yang L.-H., Liu Z.-C., Wen S.-J., Shi G., Li Y.-H., Zeng X.-H., Yang S., Wang Y.-H., Hua W.-S., and Chen F. for assistance with the data collection. We thank Liu Y. and Xu W.-B. for providing photographs of the karst landform and associated species. We thank Chen Y.-Q., Ye J.-F., and Lee T.-M. for constructive comments. This research was partially supported by grants from the Basic Work Special Project of the National Ministry of Science and Technology of China (2013FY111500), the special project of Co-evolution of Vegetation and Geological Environment in Shenzhen Dapeng Peninsula National Geopark (HT-99982020-0258), the Guangzhou Project of Science and Technology Program (201903010076), Guangdong Project of Science

and Technology Program (2018B030320001) to Liao W.-B., the Natural Science Foundation of Guangdong Province (2021A1515110425) to Zhao W.-Y., the Guangdong Provincial Special Research Grant for the creation of National Parks (2021GJGY034) to Fan Q., the Guangzhou Science and Technology Project (202102021016) to Shi S., and the Zhang Hong-Da (Chang Hung-Ta) Science Foundation at Sun Yat-Sen University.

## Author contributions

Liao W.-B., Fan Q., Wang L., and Zhao W.-Y. conceived the study. Chen S.-F. and Zhao W.-Y. led and designed the molecular phylogenetic analyzes framework. Liao W.-B., Zhao W.-Y., Liu Z.-C., Chen Z.-H., Wang Y., Chen J.-R., Xu K.-W., Shi S., Yin Q.-Y., Li X.-J., Laing W.-X., and D.E. Boufford compiled the datasets of floras. Zhao W.-Y., Huang K.-Y., Wang Y.-R., and Huang C.-Y. produced the figures. Wang Y.-R. extracted climate data from CHELSA. Zhao W.-Y., Liu Z.-C., and Shi S. analyzed the data and wrote the initial draft. Liao W.-B., Fan Q., Wang L., Chen S.-F., and Zhao W.-Y. revised the manuscript. Liao W.-B., Fan Q., Wang L., Chen S.-F., Zhao W.-Y., Shi S., Liu Z.-C., Li J.-L., Sun X.-L., Wu H., Wang Z.-D., Guo W., Wang L.-Y., Meng K.-K., Zhou R.-C., Cui D.-F., Su Z.-Y., Xin G.-R., Liu W.-Q., Shu W.-S., and Jin J.-H. contributed to the workshop and commented on the final manuscript. All authors read and agreed to the final draft.

## Competing interests

The authors declare no competing interests.

## Additional information

[1]State Key Laboratory of Biocontrol and Guangdong Provincial Key Laboratory of Plant Stress Biology, School of Life Sciences, Sun Yat-sen University, Guangzhou, China. [2]College of Resource Environment and Tourism, Capital Normal University, Beijing, China. [3]Guangdong Key Laboratory for Innovative Development and Utilization of Forest Plant Germplasm, South China Agricultural University, Guangzhou, China. [4]Shenzhen Dapeng Peninsula National Geopark, Shenzhen, China. [5]Co-Innovation Center for Sustainable Forestry in Southern China, College of Biology and the Environment, Nanjing Forestry University, Nanjing, China. [6]School of Earth Science and Engineering, Sun Yat-sen University, Zhuhai, China. [7]School of Ecology, Sun Yat-sen University, Shenzhen, China. [8]School of Agriculture, Sun Yat-sen University, Shenzhen, China. [9]College of Horticulture and Landscape Architecture, Zhongkai University of Agriculture and Engineering, Guangzhou, China. [10]Harvard University Herbaria, Cambridge, USA. [11]These authors contributed equally: Wan-Yi Zhao, Zhong-Cheng Liu, Shi Shi. ✉e-mail: fanqiang@mail.sysu.edu.cn; lwang@cnu.edu.cn; chsuf@mail.sysu.edu.cn; lsslwb@mail.sysu.edu.cn

