## [Peer Review File · Nature Communications]

Landform and lithospheric development contribute the assembly of mountain floras in ChinaEditorial Note: Parts of this Peer Review File have been redacted as indicated to remove third-party material where no permission to publish could be obtained.

With regard to figure 1, all photographs published with permission according to the image rights agreement between each photographer.

REVIEWER COMMENTS

Reviewer #1 (Remarks to the Author):

This is a well structured and put together paper. I'm certain you're right that the factors you examined explain patterns of montane diversity and phylogenetic community structure in these mountains in China. I'll be happy to see this excellent work published somewhere. And I do think this is the first time something like this has been attempted, at this scale, with montane plant diversity in China. However, the overall concept and approach is not novel. Others, sometimes working on other taxa and in other locations, have taken the approach of combining contemporary environmental variables with various suites of geological variables to explain spatial variation in diversity. As one key example, I'd point to Antonelli et al. (2018) in Nature Geoscience. In that paper, Antonelli and colleagues analyze "...how erosion, relief, soil, and climate relate to the geographical distribution of terrestrial tetrapods, which include amphibians, birds and mammals. [They] find that centres of species richness correlate with areas of high temperatures, annual rainfall and topographic relief, supporting previous studies. We unveil additional links between mountain-building processes and biodiversity: species richness correlates with erosion rates and heterogeneity of soil types, with a varying response across continents. These additional links are prominent but under-explored, and probably relate to the interplay between surface uplift, climate change and atmospheric circulation through time."

Note that Antonelli et al.'s paper is global in scale and examines multiple taxa. Catherine Badgley and colleagues have also done similar work, focused on mammals in mountains.

All of that is to say that the work is solid, and I believe it. It's just not that novel and limited in taxonomic and geographic scope.

Reviewer #2 (Remarks to the Author):

This paper explored the relationship between the type of landforms and floral diversity in mountain ranges. Floral checklists from 140 natural reserves and natural parks in Chinese mountain ranges were compiled. Plant species richness, phylogenetic diversity, species relatedness and time since divergence were modeled based on landforms and geological, geographic and climatic variables, controlling for the potential area effect. All measures of floral diversity differed among landforms, with additional effects from some other variables.

Findings in this study are important for understanding biodiversity in mountain ranges which is a central topic in biodiversity and biogeographic research. In particular, the significant role of landforms in generating and maintaining mountain biodiversity across a broad spatial scale is a novel finding. I do have some concerns and suggestions listed below, but if the results are robust, I think this study has very good potential to make a fine publication which will be of great interest for the broad readership of the journal. The writing is clear and generally easy to follow.

Overall, I feel that the specific mechanisms of how landforms could shape the plant communities/floras could be explained further in depth (e.g. in the introduction), especially considering the broad readership of the journal. Comparing the importance of landforms and other variables in explaining diversity seems to be a logical step but theoretically, their effects can't really be separated. The importance of landforms in shaping floral diversity/composition thus could be further established by investigating their interactions with other environmental variables, as briefly discussed by the authors (L131-132). Either a boosted regression or a multi-level model with landforms as a group-level effect would be very illuminating. In addition, a lot of the variation among landforms seems to come from deserts vs others, based on fig. 3a-c -- do the other groups differ?

A major technical concern is about how the phylogenetic data were handled. The phrasing at L416 seems to suggest that species not occurring in your studied areas are pruned, which will affect your null expectation for those phylogenetic metrics and your calculation of

phylogenetic dispersion/clustering. I do not have a good answer to what would be the best way to construct a null model for phylogenetic diversity or dispersion, whether it should be the global plant phylogeny or a Chinese plant phylogeny, or for a slightly different question, a regional phylogeny just for the floras around the focal site. Any of these cases would be better than a phylogeny of only the species in your dataset. Please clarify this and explain your reasoning.

Related to that, the interpretation of terminal branch length as species age and all associated discussions are problematic, because the plant phylogeny presumably does not include extinct lineages. The terminal branch length thus only reflects a species' distinctiveness from extant plant species within your sampling pool. In fact, the whole section about the ages of floras is problematic and not very informative unless the formation time frame of their habitats could be provided. To some extent, these analyses could help inferring whether modern floras were shaped by radiation in-situ, but this might involve a lot of confounding factors. The outcome also heavily depends on whether you have stripped away species not included in the study, and the data coverage of higher taxa (which is not explained, unless I miss it).

The models do not seem to account for spatial auto-correlation, and no justification has been given on why that is not an issue.

Minor comments:

L67-77: this paragraph might be better for the beginning of the Intro. The current beginning completely ignores elevation and associated environmental heterogeneity, which doesn't provide a broad enough overview of the topic.

L72-77: the two sentences are a bit contradicting with each other about whether the subject has been studied or not -- better rephrase.

L83: add "to" to before "refer to"

L87: change "resolved" to "reflected" or "represented"

L89: better clarify what "phylogenetic structure" is referring to (relatedness?)

L93: change "additional" to something like "in addition to" or "along with"

L140: PDI has not been explained at this point -- given the format with Methods after everything, it would be helpful to have some technical information specified in the results

(e.g. whether area and latitude were accounted for in the models).

L244-250: environmental filtering as a potential mechanism should probably be discussed here, especially given L235-237.

L246: TAR reflects "instability", right?

L260: change "maybe" to "may be"

L398: "a total of"

L343: better use "allows" than "facilitates"

L440: is NRI driven by sample coverage, i.e. the number of artificial sister species/genera in the floras?

L461: "is different" or "differs"

L509: please clarify whether the resolution of the climatic layers matches your sampling areas -- are most floras much larger than the units of the climatic datasets?

Reviewer #3 (Remarks to the Author):

Review of the manuscript "Landform and lithosphere development drive the assembly of mountain floras" by Zhao Wan-Yi et al.

The authors investigated 140 mountain regions (with their flora and bedrocks) in China to study the relationship between mountain floras and the types of landforms in which they occur. The authors obtained a backbone phylogeny from a recently generated time-calibrated mega-tree, including species across all floras. Based on the phylogeny, they calculated phylogenetic diversity, phylogenetic structure, and mean divergence times. They constructed regression models to predict species richness, phylogenetic diversity and structure, and mean divergence times, using landform as a predictor, and using additional tectonic, climatic, and geographic explanatory variables as predictors. Based on their findings, the authors put forward what they call "the geological lithology hypothesis of assembly and differentiation of mountain floras". They claim that this hypothesis "provides a novel framework rooted in geology for future research on the origins, differentiation, and migration of angiosperm assemblages and mountain floras."

This study complements a growing body of literature on the close interrelationships

between geological diversity (geodiversity) and biological diversity (biodiversity), on their “concerted” evolution, and on the assembly of mountain floras. It stresses edaphic factors (related to bedrock, landforms) as important for ecological filtering, species immigration, and evolution, further driving differences among landforms.

I enjoyed reading this manuscript, and think it can potentially make an important contribution. However, at the same time I found that reference to highly relevant recent literature (pertaining to the 'mountain geobiodiversity hypothesis') was lacking and thus not considered, especially with regard to the hypothesis here by the authors. In addition, some methodological issues, e.g. pertaining to taxonomic name resolution (use of the now long outdated Plant List) and molecular dating (derivation of species divergence times from a phylogeny not sampled for taxonomic coverage), need consideration.

In the following, I will provide my comments which I hope will be useful for the authors. Line numbers refer to the pdf.

Title: As the study is confined to mountains in China, and the authors emphasize the special geological conditions in China in their manuscript, the title could be adjusted to reflect this. E.g. “Landform and lithosphere development drive the assembly of Chinese mountain floras”. Also, the word “drive” may be reviewed (“drivers” is a strong term).

Line 30: “The biodiversity of mountains is attributable in part to their geological activities, namely uplift and erosion, as well as the habitat heterogeneity that they provide, such as along their elevational gradients⁴⁻⁶”. It would be appropriate to cite the work by Antonelli et al. 2018 already here.

Line 31: “how mountain biodiversity is shaped on temporal and spatial scales remains poorly understood⁷⁻⁹” This ignores a growing body of literature investigating this in detail, such as literature on the 'mountain geobiodiversity hypothesis' (MGH) (Mosbrugger et al. 2018, Muellner-Riehl 2019, Muellner-Riehl et al. 2019, and references therein). There are, by now, hundreds of papers on Chinese mountain biogeography and Chinese mountain phylogeography. This needs more to receive more attention in the manuscript, and

sentences such as the one cited above (“poorly understood”) need to be rephrased to pay tribute to the growing knowledge.

Line 38-40: “the ages of the studied floras are consistent with the geological development of the different landforms with mean divergence times being the highest...” This is not clear - what exactly does it mean that ages are consistent, given that the authors later define „flora“ (line 83) as „sum of all angiosperm families, genera, and species“? Different taxonomic levels (species – genera – families) have vastly different ages (unless a genus or a family are both monospecific, or close to), and including only a very small fraction of the species of a genus or family, as done here (because only the species occurring on the 140 Chinese mountain regions are included in the analyses), will not allow to arrive at ages which are anywhere close to the “real” ages of the species (or higher taxonomic levels).

Line 42: What do the authors consider as “younger” species?

Line 42-47: “We put forward “the geological lithology hypothesis of assembly and differentiation of mountain floras”.” “Our hypothesis provides a novel framework rooted in geology for future research on the origins, differentiation, and migration of angiosperm assemblages and mountain floras.” As mentioned before, the authors seem unaware of the MGH ('mountain geobiodiversity hypothesis'). This hypothesis was originally formulated by Mosbrugger et al. 2018, later refined by Muellner-Riehl 2019, and finally tested on a global scale by Muellner-Riehl et al. 2019. The MGH is supposed to provide an overarching framework for the investigation of the concerted evolution of mountains and their biotas, and on the assembly of mountain biodiversity, through time, and also considering climatic fluctuations which act on top of orogenic processes. Together with the flickering connectivity system by Flantua et al. (the latter which is cited by the authors), the MHG thus deals with the intricate relationship between mountains and their biotas/flora. It needs to be elaborated and explained what the “geological lithology hypothesis” would add specifically, and how it differs from the other hypothesis already brought forward and tested by empirical studies in the past few years. The authors should consult the work by Mosbrugger et al. 2018, Muellner-Riehl 2019, Muellner-Riehl et al. 2019. Is it justified to formulate yet another hypothesis, or may the hypothesis introduced here be considered under the umbrella of the MGH?

Line 43/44: “Under this hypothesis, landforms develop according to their underlying bedrock, and their geological development drives both the assembly and subsequent differentiation of mountain floras.” This may be considered less a hypothesis than a fact.

General comment: Missing in this first paragraph of the text is also a general reference to “geodiversity” and “geobiodiversity”. It may be worth noting here, for the information of the authors, that the recognition of the intricate relationship between geological diversity and biodiversity has led to specific research activities in the scientific community, such as e.g. the Research Activity “Geobiodiversity and Climate” which “studies the interactions between climate, Earth surface processes and biodiversity on different time scales. It further examines the impacts of climate and Earth surface processes on the evolutionary and ecological dynamics of species and communities. An important research topic are the effects of anthropogenic climate change on biodiversity and ecosystem functions.” (<https://www.senckenberg.de/en/science/biodiversity-and-climate/geobiodiversity-and-climate/>). What I am trying to say here is that the introductory part of the manuscript leaves the impression that the authors are not aware of an important body of literature and concepts pertaining to geobiodiversity, and this should be avoided.

Line 67/68: “Historically, mountain biodiversity and, more broadly, global biodiversity, have been explained by numbers of hypotheses such as the climate stability²³, habitat heterogeneity²⁴⁻²⁵, and energy hypotheses²⁶⁻²⁷.” Again, reference to the MGH is missing here. The sentence should be rephrased to accommodate it, as it specifically refers to mountains, which is the core topic of this manuscript, and thus especially relevant. Muellner-Riehl et al. 2019 already reflected about general hypotheses explaining high biodiversity levels, and then specifically tested the MGH, which was originally proposed for the Tibet-Himalaya-Hengduan region (thus especially relevant for Chinese mountains), and then tested for global relevance.

Line 75-77: „Despite this, previous researches have focused primarily on the impacts of ecological factors on mountain biodiversity, while the contributions of geological and lithologic processes are largely unexplored^{3,9,29}.” This may have been considered true (at

least, to some extent - though the field of „geobotany“ is, of course, a very old discipline) some ten years ago, but ignores recent literature on geodiversity-biodiversity relationships. See also my comments further above. But there is also a growing body of literature of several working groups and researchers specifically addressing these issues.

Line 84/85: “growing on a specific mountain or in a well-delimited area, which is a relatively independent and self-evolving natural complex³⁰⁻³¹.” This is not correct. Mountains are usually not considered „independent“ and „self-evolving“, i.e. disconnected from surrounding areas (unlike true islands). There is quite some literature on the similarities, but more importantly differences, between island and mountain systems (e.g. see Mendez-Castro et al. 2021, Itescu 2018, Flantua et al. 2020). This statement needs to be re-phrased accordingly.

Line 114: “(median = 1,462.0)” – not entirely clear what this number represents.

Line 122-136: “Based on the full model, longitude, elevational difference between the highest and lowest points of a flora (elevdiff hereafter), and mean temperature of the coldest quarter (TCQ hereafter) had positive effects on species richness (Extended Data Table 3).” In this sentence, and the following ones, it should be made clear whether larger/smaller or higher/lower values have a positive or negative effect. This seems not evident from the text, as currently written. E.g. do the authors mean “a larger elevational difference”? Or later in line 128, high or low precipitation seasonality and high or low mean temperature? As a note aside, net primary productivity (NPP) might have been a better predictor for species richness.

Line 142 f.: What about the role of dispersal from other mountain systems? (see e.g. Jiménez-Alfaro et al. 2020, DOI: 10.1111/geb.13274; Ding et al. 2020, Science)

Line 149-150: “This is consistent with some fossil evidence, the fossil flora discovered in southwestern China indicate local karst vegetation may have existed since the early Oligocene³⁷⁻³⁸.” Is this fossil evidence from the same species or genera as the living ones, and thus specifically relevant?

Line 151 f.: “In the full model, orogenic, latitude, temperature annual range (TAR), TCQ is negatively correlated to PDI,...” Again here, as already mentioned for lines 122-136, the sentence does not provide as much information as it could: high range? Low range?

Line 153-157: “may have recently undergone higher rates of evolution related to orogeny¹⁴” „while higher TWQ may have led to higher rates of extinction in mountains of sedimentary rock“ Note: This could be tested (speciation rates, extinction rates), and might be worth testing in the future. Mountain biogeography studies often investigate speciation rates and extinction rates, and net diversification rates, respectively, from dated phylogenies, albeit with dense taxonomic sampling (unlike the study here). Investigation of floristic differences of karst versus non-karst, calcareous versus silicate, is a well-researched topic of geobotany (e.g. in the Alps).

Line 172: “Landforms should originate earlier than the floras that inhabit them due to the long times that floras require for assembly.” “earlier” – in concert, why earlier? Species can immigrate into a region (dispersal from e.g. other mountain regions or other habitats), and thus can be older than the landform they inhabit nowadays.

Line 173-174: “we measured the mean divergence times of all species“ Divergence times can only be measured reliably when (almost) all species of a genus are included in a phylogeny, otherwise, ages may be considered not reliable/meaningful. The use of single phylogenies, targeting specific groups, would be needed for that.

Line 188: “and this may be caused by the stronger influence of glacial periods“ – What was the effect of these glacial periods in the investigated mountain systems? Which were impacted by glaciations, which were affected by colder/drier conditions only? The effect of the glacial periods was much dependent on the availability of water (precipitation; monsoons) and thus regionally very different.

Line 214: “Our results are consistent with prior studies^{9,14}...” Reference to MGH papers missing here.

Line 215: “closely related to geomorphic processes” – strictly speaking, this study does not investigate processes as such, but patterns.

Line 215: “For example, most floras of China, including both mountains and lowland floras, diverged during the Miocene¹¹ when the East Asian monsoon began to prevail..:” – not clear what is meant by “floras diverged”, see my previous comments on this issue of dating “floras”. Be more specific about “when the East Asian monsoon began to prevail” – when was the presumed onset, when intensification, when were current levels reached? There is plenty of literature available on this topic (e.g. see publications by Dupont-Nivet, and others).

Line 218 following: “At the intercontinental scale, the ages of floras are also largely consistent with regional developmental processes of landforms. For example, the floras of eastern Asia are typically younger than those of South Africa and Australia, where the landform processes have been very long, but older than those of the Andes and Amazonia, where landform processes have occurred more recently⁴⁸. Thus, the relationship of floristic assembly of mountains to the type of landform and landform developmental process seems to be global in scope, at least for angiosperms. Thus, landform types may make a suitable indicator for explore floristic assembly of mountains, and such an indicator is needed by the scientific community to support biogeographic and other research on mountainous regions²⁹.” This is a rather simplified statement, needs more elaboration and detail, and also much more consideration of literature that exists for all of these regions. E.g. Do the authors here talk about mountain floras specifically, and if so, which? Different mountain systems on these continents have different ages and history, different orientation (Elsen and Tingley 2015, Nat. Clim. Change), and were also differently impacted by the Last Glacial Maximum (LGM). „explore floristic assembly of mountains“ – the floristic assembly is not explored here per se, but only the patterns. But the assembly can actually be investigated, see Ding et al. 2020 (who you also cite).

Line 230: results for desert landforms – this may not come as a surprise, as only relatively few plant genera/families are especially adapted to desert conditions.

Line 271 following: Here again, reference to the MGH is missing. “Specifically, under this

hypothesis, the ages of mountain floras are determined by the time when erosion of strata begins,...“ This disregards the effect of immigration on floristic assembly in mountains which can account for most of the species richness on some mountains (compare Ding et al. 2020).

Methods

Line 405: “we reconciled taxonomy to The Plant List v. 1.1“. The Plant List (TPL) is an outdated resource, thus other resources should be used instead.

For plants, four global authoritative taxonomic lists exist: World Checklist of Vascular Plants (WCVP), the World Flora Online (WFO, an explicit community effort to tackle the increasing wealth of taxonomic information as successor of The Plant List), the Leipzig Catalogue of Vascular Plants (LCVP), and World Plants (WP; both works of dedicated individuals). These four lists each provide a global list of plant names, but differ considerably in size and likely in completeness and accuracy across taxa and geographic regions.

The WFO Plant List (www.wfoplantlist.org) replaces the now long outdated Plant List.

Line 462: “studies have shown that the species ages within floras are quite different“. Strictly speaking, species ages themselves are not investigated here

Data availability

Line 548: “All other additional data are available from the corresponding author upon reasonable request.“ What is considered “reasonable“? Data should be made available, whenever possible, according to the FAIR principles (findability, accessibility, interoperability, and reusability).

Response to reviewers' comments

REVIEWER COMMENTS

Reviewer #1 (Remarks to the Author):

This is a well structured and put together paper. I'm certain you're right that the factors you examined explain patterns of montane diversity and phylogenetic community structure in these mountains in China. I'll be happy to see this excellent work published somewhere. And I do think this is the first time something like this has been attempted, at this scale, with montane plant diversity in China. However, the overall concept and approach is not novel. Others, sometimes working on other taxa and in other locations, have taken the approach of combining contemporary environmental variables with various suites of geological variables to explain spatial variation in diversity. As one key example, I'd point to Antonelli et al. (2018) in Nature Geoscience. In that paper, Antonelli and colleagues analyze “...how erosion, relief, soil, and climate relate to the geographical distribution of terrestrial tetrapods, which include amphibians, birds and mammals. [They] find that centres of species richness correlate with areas of high temperatures, annual rainfall and topographic relief, supporting previous studies. We unveil additional links between mountain-building processes and biodiversity: species richness correlates with erosion rates and heterogeneity of soil types, with a varying response across continents. These additional links are prominent but under-explored, and probably relate to the interplay between surface uplift, climate change and atmospheric circulation through time.”

Note that Antonelli et al.'s paper is global in scale and examines multiple taxa. Catherine Badgley and colleagues have also done similar work, focused on mammals in mountains.

All of that is to say that the work is solid, and I believe it. It's just not that novel and limited in taxonomic and geographic scope.

Response: We thank the reviewers for their affirmation of our research result. We also appreciate the previous outstanding researches on mountain biodiversity, as you mentioned those literature in the review comments (Antonelli et al., 2018, Badgley et al., 2017). In their study, they highlighted the impact of geological processes (erosion rate, soil type, etc.) and climate on mountain species diversity based on animal distribution data.

In our study, we test the landform effect on mountain plant species richness, phylogenetic diversity and structure, and mean divergence times. Because plants are more dependent on local rocks and soils, and less responsive to climate than animals, landform (result of rock erosion in mountains) effects may play important roles in the assembly of mountain floras, which maybe not detectable in related studies about animals.

To investigate the relationship between mountain floras assembly and the types of landform. In total, 140 natural mountains flora (including 17,576 species) in China were collected, involving five major geological and landform types (karst, karst-granitic, granitic, Danxia, and desert landform), covering 35 latitudes (about 3500 km) from north to south, and 50 longitudes (about 5000 km) from east to west. In such a large and relatively concentrated area, the analysis of plant floras and plant species diversity in the independent mountain regions has sufficient differences and good comparability, and it is different from the gridding or planarizing analysis of regional species diversity. Therefore, we believe that the analysis in this paper should have different meanings compared with the previous analysis results.

Our results shown that mountains of igneous bedrock have higher species richness and exhibit phylogenetic overdispersion, while mountains of sedimentary bedrock have lower species richness and clustered phylogenetic structure. We also find that the type of landform has a greater effect on floristic assembly for floras containing evolutionarily older species, while climate is a greater determinant for floras with younger species. These results indicate landform development process not only affects mountain species richness, but also determines the composition of floras. In conclusion, our results highlight bedrock in mountain and landform effects plays a key role in floristic assembly, especially considering the bedrocks on Earth's is unevenly distributed. These are new insights of the occurrence of plant diversity and bio-diversity protection in mountain areas.

Reviewer #2 (Remarks to the Author):

This paper explored the relationship between the type of landforms and floral diversity in mountain ranges. Floral checklists from 140 natural reserves and natural parks in Chinese mountain ranges were compiled. Plant species richness, phylogenetic diversity, species relatedness and time since divergence were modeled based landforms and geological, geographic and climatic variables, controlling for the potential area effect. All measures of floral diversity differed among landforms, with additional effects from some other variables.

Findings in this study are important for understanding biodiversity in mountain ranges which is a central topic in biodiversity and biogeographic research. In particular, the significant role of landforms in generating and maintaining mountain biodiversity across a broad spatial scale is a novel finding. I do have some concerns and suggestions listed below, but if the results are robust, I think this study has very good potential to make a fine publication which will be of great interest for the broad readership of the journal. The writing is clear and generally easy to follow.

Response: We thank reviewer 2 for appreciating our work and, even more, for the useful insights on how to improve it.

Overall, I feel that the specific mechanisms of how landforms could shape the plant communities/floras could be explained further in depth (e.g. in the introduction), especially considering the broad readership of the journal.

Response: Thanks for your suggestions. To our knowledge, there are no studies that specifically discuss the impact of landform (which is more or less involved in heterogeneity, lithology types, etc.) on species richness of mountains. In the discussion part (line 281-327), we added three paragraph to discussion how landforms could shape the assembly of mountain floras.

Comparing the importance of landforms and other variables in explaining diversity seems to be a logical step but theoretically, their effects can't really be separated. The importance of landforms in shaping floral diversity/composition thus could be further established by investigating their interactions with other environmental variables, as briefly discussed by the authors (L131-132). Either a boosted regression or a multi-level model with landforms as a group-level effect would be very illuminating.

Response: Very thanks for your advice. We accepted the suggestion of reviewer #2, and investigated the interaction of landforms and other variables. The results show that the landform effect is always significant. Please see L130-L132 "We further tested the landform effects when interactions with others variables. The results showed that landform combined with any other variables will significantly be improved the explanatory power of the model for species richness (Extended Data Table 10)." The details results of model fitting are shown in the are shown in Extended Data Table 10.

Variables	GLM			SAR		
	Deviance, %	AIC	Moran's I	Deviance, %	AIC	Moran's I
Landform+Tectonic	33.57	108.80	0.064 ^{ns}	34.70	109.25	0.002 ^{ns}
Landform+log(area)	34.52	106.79	0.068 ^{ns}	35.79	107.03	-0.001 ^{ns}
Landform+Longitude ^{ns}	29.17	117.77	0.886*	31.41	116.87	0.002 ^{ns}
Landform+Elevdiff	38.33	98.39	0.077 ^{ns}	40.33	97.67	0.003 ^{ns}
Landform+TWQ	34.62	106.56	0.210***	43.00	97.20	-0.011*

Landform+TCQ ^{ns}	29.64	116.85	0.077 ^{ns}	31.36	116.64	0.002 ^{ns}
Landform+P _{WM}	33.00	109.99	0.089 [*]	34.90	109.29	0.003 ^{ns}
Landform+P _{var}	35.74	104.15	0.054 ^{ns}	36.46	105.10	0.002 ^{ns}
Landform+PCQ	31.81	112.45	0.074 ^{ns}	33.19	112.55	0.003 ^{ns}
Landform+Tectonic+	55.58	60.44	0.049 ^{ns}	56.18	61.32	0.003 ^{ns}
Elevdiff+TWQ+TCQ+P _{var}						

In addition, a lot of the variation among landforms seems to come from deserts vs others, based on fig. 3a-c -- do the other groups differ?

Response: The desert indeed provides a lot of variation among landforms. While, the others landform types also provide a lot of variation. Such as, the species richness of Granitic and Karst-Granitic landform is significant higher the Danxia landform(the reference) (see “*Extended data Table 3*”). In the landform model for MDT, MDT.oldest, and NTI, we also detected the Desert, Granitic, Karst, and Karst-Granitic landform are always significant differences between Danxia landform (see “*Extended data Table 5-6, 9*”).

A major technical concern is about how the phylogenetic data were handled. The phrasing at L416 seems to suggest that species not occurring in your studied areas are pruned, which will affect your null expectation for those phylogenetic metrics and your calculation of phylogenetic dispersion/clustering. I do not have a good answer to what would be the best way to construct a null model for phylogenetic diversity or dispersion, whether it should be the global plant phylogeny or a Chinese plant phylogeny, or for a slightly different question, a regional phylogeny just for the floras around the focal site. Any of these cases would be better than a phylogeny of only the species in your dataset. Please clarify this and explain your reasoning.

Response: Thanks for your advice. In this study we pruned those species not occurring in our studied areas, as the null model demanded that the species pool should be consistent in the phylogenetic tree and distribution database to ensure all species can be sampled randomly in the NRI, NTI, and PDI null model (Webb, 2000, doi: 10.1006/jema.1996.0042; Jin & Qian, 2022, doi: 10.1016/j.pld.2022.05.005).

Here, the reviewers 2 also raised the concern about “if an incomplete phylogeny can really reflect the history of flora”. Phylogeny-based approaches are essential to understanding differences in species richness and the assemble history of floras between regions. In fact, using mega-phylogeny data to extract regional species phylogeny (such as phylogenies generated by V.PhyloMaker) is a common approaches in community ecology, macroecology and study on regional flora.

In fact, the mode of phylogenetic inference has little influence on phylogeny diversity metrics. The developer of V.PhyloMaker had specially written a paper to clarify the issue. They test the difference using 1093 angiosperm tree assemblages in North America (Qian & Jin, 2021; doi.org/10.1016/j.pld.2020.11.005; Jin & Qian, 2022;https://doi.org/10.1016/j.pld.2022.05.005). Several previous case studies have also demonstrated that equivalent results have been obtained from purpose-built and synthetic phylogenetic trees (Allen et al., 2019, doi: 10.1016/j.isci.2018.12.002; Jantzen et al., 2019, doi: 10.1002/ece3.5425; Li, Trotta, et al., 2019, doi: 10.1002/ecy.2788).

Related to that, the interpretation of terminal branch length as species age and all associated discussions are problematic, because the plant phylogeny presumably does not include extinct lineages. The terminal branch length thus only reflects a species' distinctiveness from extant plant species within your sampling pool. In fact, the whole section about the ages of floras is problematic and not very informative unless the formation time frame of their habitats could be provided. To some extent, these analyses could help inferring whether modern floras were shaped by radiation in-situ, but this might involve a lot of confounding factors. The outcome also heavily depends on whether you have stripped away species not included in the study, and the data coverage of higher taxa (which is not explained, unless I miss it).

Response: Thanks for your advice. It is inevitable that the species divergence age obtained from the regional phylogenetic data will be biased without extinct lineages. Here, we did an analysis to test the divergence time consistency between the two results of species ages extract by **pruned phylogeny** (used in our study) and the ages extract from **unpruned phylogeny** (GBOTB.extended.LCVP.tre, a global plant phylogeny). In total 17,576 species included in this study, of which 8,663 species were included in "GBOTB.extended.LCVP.tre" and 8713 species not occurs in GBOTB.extended.LCVP.tre. We reconstruct the phylogeny of 8863 species which occurs in GBOTB.extended.LCVP.tre, based on the same method used in this study. Then, extract the divergence time of these 8863 species in new gathered pruned tree and in GBOTB.extended.LCVP.tre (a global plant phylogeny, unpruned tree), respectively. Finally, we get four group of species divergence time data, which are:

"pruned.add" age (used in our study)=included 17,576 species, of which ages of 8863 species are extract from the pruned phylogeny tree, and ages of others 8713 species(add) are added by BLADJ method (Webb et al., 2011)

"unpruned.add" age =included 17,576 species, of which ages of 8863 species extract from the unpruned phylogeny, and ages of others 8713 species(add) are added by BLADJ method (Webb et al., 2011)

"pruned" age=included 8863 species, age extract from the pruned phylogeny tree

"unpruned" age=included 8863 species, age extract from the unpruned phylogeny

The results also shown the MDT patterns of mountain flora are similar of the four different species divergence time datasets (Supplemental fig. 1). The MDT of karst landform is always the highest, while the MDT of Danxia landform and desert landform is lower. The results also shown that the MDT estimate from "pruned" phylogeny is higher than the "unpruned" phylogeny (Supplemental fig. 1-5). This was could be expected, as the species branch in a "pruned" phylogeny is usually longer than which in the global plant phylogeny or a phylogeny included more species. Besides, there are 8713 species not occurs in GBOTB.extended.LCVP.tre, the age of these species was treated as their genus node branch (see method "*Phylogenetic reconstruction*").

Supplemental fig1. MDT results of mountain flora derived from these four species divergence time datasets

Further, we did the regression analyse on MDT of four different age datasets and find the close correlation of MDT estimate from “pruned” phylogeny with “unpruned” phylogeny (R^2 of MDT.all is 0.63), especially the younger species ($R^2=0.84$) (Supplemental fig 2). These could also be observed in linear regression fitting result of “unpruned.add” VS “pruned.add” (Supplemental fig 3), “unpruned” VS “pruned.add” (Supplemental fig4), “unpruned” VS “unpruned.add” (Supplemental fig 5). It is particularly to point out that the results of MDT.all and MDT.oldest is consistent (see Supplemental fig 2 and Supplemental fig 4), although “pruned.add” datasets included others 8713 species. This result suggest that ages of 8713 species not occurs in

GBOTB.extended.LCVP.tre which inferred in “pruned.add” datasets may be reasonable estimates.

Supplemental fig 2. unprunedVS pruned

Supplemental fig 3. unpruned.add VS pruned.add

Supplemental fig 4. unpruned VS pruned.add

Supplemental fig 5. unpruned VS unpruned.add

Nevertheless, the impact of limited sampling in a regional phylogeny on divergence times still remains to be assessed directly in the future. The species age of “pruned” phylogeny is biased

(usually larger) than the “unpruned” phylogeny. While, in large-scale biodiversity and phylogenetic analyses, sources of noise in divergence time estimation are to be expected, but they did not affect the reliability of the results (Lu et al. 2019). Just as the species ages inferred in our phylogeny (a pruned tree of GBOTB.extended.LCVP.tre) did not affect the MDT patterns between mountain in different landform.

Although the estimates species ages “pruned” phylogeny is larger than “unpruned” phylogeny. We believe that our dated mega-phylogeny tree is suitable for this study. Because the aim of this study is to reveal the general pattern of landform influence on the formation of mountain flora, rather than focus the time of each species occurs in a mountain. In fact, the latter is difficult to achieve. In the manuscripts, we made no further changes and discuss of the phylogeny tree in order to maintain succinctness and consistency. The test result of four different age dataset herein will be provided as “Peer review information” for the readers (when this study be published). We hope our reply will clear up your doubts.

The models do not seem to account for spatial auto-correlation, and no justification has been given on why that is not an issue.

Response: Thanks for this important and insightful comment. In the methods part we added Moran's I value to quantify the residual spatial auto-correlation under the GLM, SAR models, and they obtained similar results. Detail see Methods-“Spatial analysis” in L503-L511 *“Spatial autocorrelation is a general feature of macro-ecological data and may leading to erroneous interpretations”¹⁰⁰. We use Moran’s I values to quantify residual spatial autocorrelation, which is considered as a spatial equivalent to Pearson’s correlation coefficient and normally varies between 1 and -1, and expect Moran’s I values for lacking spatial autocorrelation is close to 0¹⁰¹. Because of spatial autocorrelation also present in our data set, we performed spatial simultaneous autoregressive model (SAR) to account for residual spatial autocorrelation¹⁰². The expect Moran’s I values for the response variable sa well as for GLM and SAR residual are shown in appendix (Extended data Table 3-9). Spatial statistics were performed with “spdep” package in the R version 4.1 (<https://www.r-project.org/>).*”

Minor comments:

L67-77: this paragraph might be better for the beginning of the Intro. The current beginning completely ignores elevation and associated environmental heterogeneity, which doesn't provide a broad enough overview of the topic.

Response: Thanks for this kindly reminder. We rewrote the Intro part. At the beginning, we summarized the general explanations for the higher diversity of mountain species. The effects of geological processes and lithology on mountain plant diversity were further introduced. Please see revised manuscript L47-L62. *“Globally, mountains play dual roles as museums and cradles in the formation of species diversity¹⁻² and, therefore, it is unsurprising that much of global biodiversity is concentrated within mountains, especially within the tropics³⁻⁵. The worldwide mountains harbor the 40% of the global diversity, and species inhabiting in mountains are double to the lowlands when taking into account the area effect⁶. How extraordinary diversity of mountains occurs is still a great challenge since the Humboldt's time⁷.*

Historically, mountain biodiversity and, more broadly, global biodiversity, have been explained by numbers of hypotheses such as the climate stability⁸, habitat heterogeneity⁹⁻¹⁰, and energy hypotheses¹¹⁻¹². More recently, a comprehensive model for biodiversity prediction is established¹³⁻¹⁷

that includes ecological processes (survival, competition, and niche differentiation), biological processes (species divergence and extinction), and geological and lithologic processes (such as orogeny and rock formation)¹⁸⁻¹⁹. Among which, a representative theory is “mountain geobiodiversity hypothesis” (MGH), which proposed to explain the biodiversity of Tibeto-Himalayan region². Within these framework, geological and lithological processes, especially uplift and erosion, are known to have strong effects on mountain biodiversity, probably through their roles in species formation, immigration, and extinction^{12,20-21}.”

L72-77: the two sentences are a bit contradicting with each other about whether the subject has been studied or not -- better rephrase.

Response: We revised the sentence. Please see revised manuscript L53-L74 (the last reply).

L83: add "to" to before "refer to"

L87: change "resolved" to "reflected" or "represented"

Response: Thanks for your help to improve the linguistics of our manuscript!

L89: better clarify what "phylogenetic structure" is referring to (relatedness?)

Response: Thank you for this suggestion. Phylogenetic structure represents the phylogeny relatedness (clustered or overdispersed) of species in a mountain flora, as you mentioned, which is an important concept in the study of mountain species assemblages. The phylogenetic structure of a flora is most commonly assessed by NRI and NTI.

L93: change "additional" to something like "in addition to" or "along with"

Response: Thanks for your help to improve the linguistics of our manuscript! We had revised it in the article accordingly.

L140: PDI has not been explained at this point -- given the format with Methods after everything, it would be helpful to have some technical information specified in the results (e.g. whether area and latitude were accounted for in the models).

Response: The phylogenetic diversity index (PDI) is a standardized PD_{Faith} using null models, allows comparisons among floras phylogenetic diversity with different underlying species richness values. As you can see in Figure 3b, we first shown the PDI difference between five landform types. We further detected the landform effects on PDI in both “landform” model and “full” model, just as the analysis performed for species richness.

To avoid confusion, we rewrote the methods section (see Methods “GLM analysis” in L486-L496) “*We performed generalized liner models (GLM) to model log-transformed species richness (SR) as the response variable with landforms and tectonic and climate variables as predictors. Initially, we modeled SR as a function of only landform (i.e., “landform” model) because, in this study, we focused primarily on the roles of landforms in floristic assembly. However, we also extended the “landform” model to include all others variables, especially to assess the climatic impact on species diversity (i.e., “full model”). Further, we used GLM to determine the effects of landform, tectonics and climate on NRI, NTI, PD, PDI, MDT, MDT_{oldest} , and $MDT_{youngest}$ (Extended Data, Table 2). As with SR, we initially modeled landform as a single predicting factor (“landform” model) before extending to all variables within a “full model”. We performed all GLM analysis in*

R version 4.1 (<https://www.r-project.org/>) using the “glm” function in the MASS package⁹⁸.”

The “full” model of PDI also included all the predictor variables listed in “Extended data Table 2”, in which area and latitude were considered.

L244-250: environmental filtering as a potential mechanism should probably be discussed here, especially given L235-237.

Response: This is a good suggestion. Because most of the granitic landforms floras is phylogenetic overdispersion, while phylogenetic clustering granitic landforms floras is usually located at high latitudes. We takeout a discussion on how landform and environmental filtering affects the flora phylogenetic structure of floras in L313-L316. *“The mountains of sedimentary bedrock have much stronger environmental filtering effect, as these mountain ecosystems are more sensitive to rainfall. The environmental filtering effect further promoted the clustering of phylogentic structures of mountain floras as predicted by the phylogenetic niche conservatism (PNC)^{46,50,64}.”*

L246: TAR reflects "instability", right?

L260: change "maybe" to "may be"

L398: "a total of"

L343: better use "allows" than "facilitates"

Response: Thanks for your help to improve the linguistics of our manuscript! We had revised this in the article accordingly.

L440: is NRI driven by sample coverage, i.e. the number of artificial sister species/genera in the floras?

Response: In general, the more species contains in a genus, the mean phylogenetic relationship of species in a mountain may be more closely (the NRI tend to be > 0). However, the NRI is derived from the null model. In our result, we did not observed the correlation between NRI and the mountain flora species/genera number. Such as, the mountain Taibaishan in Shaanxi province has 1656 species, and its NRI is 4.54. In contrast, the mountain Lushan in Jiangxi province has similar species number (1590) to Taibaishan, but its NRI is -2.43. In other words, NRI reflects the dispersion degree of phylogenetic relationships among species, and has little correlation with the number of species.

L461: "is different" or "differs"

Response: Thanks for your help to improve the linguistics of our manuscript! We had revised it in the article accordingly.

L509: please clarify whether the resolution of the climatic layers matches your sampling areas -- are most floras much larger than the units of the climatic datasets?

Response: Thanks for this important and insightful comment. We agree with this criticism, and indeed authors had discussed about these issues before the original submission. We test the collinearity relation between climatic data which extract by GPS location from CHELSA(30 arc-seconds), World Climate (2.5 m, 5 m, 10 m). The results are closely related. So that, we select CHELSA(30 arc-seconds) which is a high resolution climate data. However, we were not fully successful the problem, because the climate data of a site note welly represent the mountain (as most of the mountain site area in this study over 100 km²).

Therefore, in the revised manuscripts we gathered the .shp file layer of each mountain site. The new climate data for analysis were extracted as mean value of each mountain site layer using the zonal statistics in Arcgis. See L467-L470. *“For each of the 140 mountain floras, we downloaded climatic data from the CHELSA climate dataset (v. 1.2, available at <http://chelsa-climate.org/>) at a spatial resolution of 30 arc-seconds⁹⁶, and extracted the climatic variables mean values of each mountain layer using the zonal statistics function in ArcGIS 10.8.”*

We suspect that the climate data extracted by this method can represent the actual mountain climate data.

Reviewer #3 (Remarks to the Author):

Review of the manuscript “Landform and lithosphere development drive the assembly of mountain floras” by Zhao Wan-Yi et al.

The authors investigated 140 mountain regions (with their flora and bedrocks) in China to study the relationship between mountain floras and the types of landforms in which they occur. The authors obtained a backbone phylogeny from a recently generated time-calibrated mega-tree, including species across all floras. Based on the phylogeny, they calculated phylogenetic diversity, phylogenetic structure, and mean divergence times. They constructed regression models to predict species richness, phylogenetic diversity and structure, and mean divergence times, using landform as a predictor, and using additional tectonic, climatic, and geographic explanatory variables as predictors. Based on their findings, the authors put forward what they call “the geological lithology hypothesis of assembly and differentiation of mountain floras”. They claim that this hypothesis “provides a novel framework rooted in geology for future research on the origins, differentiation, and migration of angiosperm assemblages and mountain floras.”

This study complements a growing body of literature on the close interrelationships between geological diversity (geodiversity) and biological diversity (biodiversity), on their “concerted” evolution, and on the assembly of mountain floras. It stresses edaphic factors (related to bedrock, landforms) as important for ecological filtering, species immigration, and evolution, further driving differences among landforms.

I enjoyed reading this manuscript, and think it can potentially make an important contribution. However, at the same time I found that reference to highly relevant recent literature (pertaining to the 'mountain geobiodiversity hypothesis') was lacking and thus not considered, especially with regard to the hypothesis here by the authors. In addition, some methodological issues, e.g. pertaining to taxonomic name resolution (use of the now long outdated Plant List) and molecular dating (derivation of species divergence times from a phylogeny not sampled for taxonomic coverage), need consideration.

Response: Above three paragraph are an excellent summary of what we did and conveys very well the general aim of our work. We wish to thank reviewer #3 for appreciating our work and, even more, for the useful insights on how to improve it. We have studied these comments carefully and have made substantial changes. The corrections are indicated in red in the manuscript, and replies to comments are listed as follow.

In the following, I will provide my comments which I hope will be useful for the authors. Line numbers refer to the pdf.

Title: As the study is confined to mountains in China, and the authors emphasize the special geological conditions in China in their manuscript, the title could be adjusted to reflect this. E.g. “Landform and lithosphere development drive the assembly of Chinese mountain floras”. Also, the word “drive” may be reviewed (“drivers” is a strong term).

Response: Thanks! We accept the suggestion to change the manuscript title as “Landform and lithosphere development drive the assembly of mountain floras in China”. On the other hand, we

suspect “drive” is a suitable word to describe the effect of landform development/lithosphere cycle on mountain flora aggregation. Because once the geological layer is uplifted during orogeny. The subsequent erosion in mountain is irreversible. However, the erosion rate of different bedrock types and the interaction process with rainfall and other environmental conditions are very different. Thus will development into different landscape and mountain flora. Such as, along with the occurrence of mountain erosion process, the adjacent limestone strata and igneous rock strata will finally develop into different flora.

Furthermore, the formation of mountains on the Earth's surface is the result of the cycle of sedimentary, igneous and metamorphic rocks. We believe that the basic conclusions of this study are also applicable on a global scale. This is worthy of further research in the future.

Line 30: “The biodiversity of mountains is attributable in part to their geological activities, namely uplift and erosion, as well as the habitat heterogeneity that they provide, such as along their elevational gradients4-6 “. It would be appropriate to cite the work by Antonelli et al. 2018 already here.

Response: Yes, the research carry out by Antonelli et al. (2018) is quiet an important in mountain species diversity. We cited this literature in the introduction and other part of our manuscript, please see L56, L65, L173, L234.

Line 31: “how mountain biodiversity is shaped on temporal and spatial scales remains poorly understood7-9 “. This ignores a growing body of literature investigating this in detail, such as literature on the 'mountain geobiodiversity hypothesis' (MGH) (Mosbrugger et al. 2018, Muellner-Riehl 2019, Muellner-Riehl et al. 2019, and references therein). There are, by now, hundreds of papers on Chinese mountain biogeography and Chinese mountain phylogeography. This needs more to receive more attention in the manuscript, and sentences such as the one cited above (“poorly understood”) need to be rephrased to pay tribute to the growing knowledge.

Response: Thank you for raising this important points. We read the relevant literature about “mountain geobiodiversity hypothesis”, the monograph “*Mountains, climate and biodiversity*”, and some related Chinese mountain biogeography and Chinese mountain phylogeography references carefully. We furtherly reorganized the preface. First, we briefly summarize the comprehensive framework to explain the formation of mountain species diversity. Furthermore, we focus on the influence of lithology or bedrocks on plant diversity in mountain areas. Because the differences in the bedrock of the stratum largely dominate the final type of the mountain landform. Landform types have the unique impact on the plant species assemblages within the mountains. This is a direction that has not been well studied. Please see L47-74. “*Globally, mountains play dual roles as museums and cradles in the formation of species diversity¹⁻² and, therefore, it is unsurprising that much of global biodiversity is concentrated within.....*

Historically, mountain biodiversity and, more broadly, global biodiversity, have been explained by numbers of hypotheses such as the climate stability⁸, habitat heterogeneity⁹⁻¹⁰, and energy hypotheses¹¹⁻¹². More recently, a comprehensive model for biodiversity prediction.....

Mountains comprise cradles of species diversity largely because their formation and subsequent bedrock erosion yield topographic complexities and produce new niches for all kinds of organisms^{13,22-24}. Additionally, they also facilitate the formation of endemic species.....”

Line 38-40: “the ages of the studied floras are consistent with the geological development of the

different landforms with mean divergence times being the highest... “ This is not clear - what exactly does it mean that ages are consistent, given that the authors later define „flora “ (line 83) as „sum of all angiosperm families, genera, and species “? Different taxonomic levels (species – genera – families) have vastly different ages (unless a genus or a family are both monospecific, or close to), and including only a very small fraction of the species of a genus or family, as done here (because only the species occurring on the 140 Chinese mountain regions are included in the analyses), will not allow to arrive at ages which are anywhere close to the “real” ages of the species (or higher taxonomic levels).

Response: This comment is useful. In our study, flora age differences were compared by MDT (average age of extant species in the mountain flora, MDT). The MDT is a index to assess the relative age of modern flora (Lu et al. 2018). We revised this confused sentence, please see L47-74. “Moreover, the mean divergence times of floras (MDT) being the highest for karst, followed by karst-granitic, granitic, Danxia, and desert landforms. ”

Line 42: What do the authors consider as “younger” species?

Response: The “younger” species represent the species that diverged later in the mountain flora. To avoid potential bias between mountains, we ranked all species occurs in 140 mountains from youngest to oldest, partitioned them into quartiles based on their ages, computed MDT in each mountain for the absolute youngest 25% and the absolute oldest 25% of species (Lu et al. 2018).

Line 42-47: “We put forward “the geological lithology hypothesis of assembly and differentiation of mountain floras” .” “Our hypothesis provides a novel framework rooted in geology for future research on the origins, differentiation, and migration of angiosperm assemblages and mountain floras.” As mentioned before, the authors seem unaware of the MGH ('mountain geobiodiversity hypothesis'). This hypothesis was originally formulated by Mosbrugger et al. 2018, later refined by Muellner-Riehl 2019, and finally tested on a global scale by Muellner-Riehl et al. 2019.

The MGH is supposed to provide an overarching framework for the investigation of the concerted evolution of mountains and their biotas, and on the assembly of mountain biodiversity, through time, and also considering climatic fluctuations which act on top of orogenic processes. Together with the flickering connectivity system by Flantua et al. (the latter which is cited by the authors), the MHG thus deals with the intricate relationship between mountains and their biotas/flora. It needs to be elaborated and explained what the “geological lithology hypothesis” would add specifically, and how it differs from the other hypothesis already brought forward and tested by empirical studies in the past few years. The authors should consult the work by Mosbrugger et al. 2018, Muellner-Riehl 2019, Muellner-Riehl et al. 2019. Is it justified to formulate yet another hypothesis, or may the hypothesis introduced here be considered under the umbrella of the MGH?

Response: Thank you for point out this important literature that we previous missed. We read the MGH relevant literature carefully. The MGH has three boundary conditions 1) presence of lowland, montane and alpine zones, 2)climatic fluctuations for a “species pump” effect, and 3) high-relief terrain with environmental in a given mountain region. Thus, the diversity of species in mountains is due to high differentiation, low extinction, and migration. MGH theory is a systematic and explanatory hypothesis especially in explaining the formation of species diversity in high mountain areas experienced many climatic fluctuations.

Our “geological lithology hypothesis of flora” differ from MGH. The “geological lithology hypothesis of flora” proposed in this study highlight the bedrock in mountain and landform process plays a key role in flora assembly. To be specific, bedrocks and associated micro-landforms in mountains is the most important factor promote speciation local endemic species. Second, landform effects (as environmental filtering) restrict the free dispersal of plants between mountains of different landform types (because edaphic species cannot exist outside the original bedrock, dispersal event could occurs with phylogenetic nich evolution), and this further leads to differences among the mountain flora in genera/species. Furthermore, “geological lithology hypothesis of flora” not only useful in predict species richness in a mountain, but also provided a new view to understand which species will finally present in the mountains of different landforms. Moreover, our hypothesis also tries to explain the reason why each mountain has high or low plant diversity, or suitable plant diversity.

Line 43/44: “Under this hypothesis, landforms develop according to their underlying bedrock, and their geological development drives both the assembly and subsequent differentiation of mountain floras. “ This may be considered less a hypothesis than a fact.

Response: Here we emphasize the final landform type of mountain is determined by the bedrock. Such as, limestone stratum can't form a mountain with high elevdiff (highest minus lowest) as igneous rock strata mountain. Thus, the assemblages history in mountain flora of different landforms will be different.

General comment: Missing in this first paragraph of the text is also a general reference to “geodiversity” and “geobiodiversity” . It may be worth noting here, for the information of the authors, that the recognition of the intricate relationship between geological diversity and biodiversity has led to specific research activities in the scientific community, such as e.g. the Research Activity “Geobiodiversity and Climate” which “studies the interactions between climate, Earth surface processes and biodiversity on different time scales. It further examines the impacts of climate and Earth surface processes on the evolutionary and ecological dynamics of species and communities. An important research topic are the effects of anthropogenic climate change on biodiversity and ecosystem functions. ” (<https://www.senckenberg.de/en/science/biodiversity-and-climate/geobiodiversity-and-climate/>). What I am trying to say here is that the introductory part of the manuscript leaves the impression that the authors are not aware of an important body of literature and concepts pertaining to geobiodiversity, and this should be avoided.

Response: Thanks for the reviewer's criticism. After a long period of field investigation, we realized that the flora composition in mountains is different among different landforms. So that, in this study, we focused on the impact of landform effects on the process of mountain flora assemblage (included species richness, phylogenetic structure, and MDT) in China. To our knowledge, landform effect on mountain flora assemblage is not tested in previous geobiodiversity studies. Landform type maybe a brief indices of geodiversity, as landform is the result of comprehensive interaction of bedrocks, climate and other factors. We are grateful to the reviewer 3 for pointing out the frontiers of mountain biodiversity research here.

In carrying out this work, we have reviewed the literatures for the high abundance of mountain or regional biodiversity, as well as several important hypotheses. The reviewer

reminded us to add the MGH hypothesis, which is now revised in the new draft. Please see line 58-60 “Among which, a representative theory is “mountain geobiodiversity hypothesis” (MGH), which proposed to explain the biodiversity of Tibeto-Himalayan region2.”

also in line 291-293 “.....the local flora assemble is a comprehensive result of species evolution, landform development and climate change^{2,13,44}.”

Line 67/68: “Historically, mountain biodiversity and, more broadly, global biodiversity, have been explained by numbers of hypotheses such as the climate stability²³, habitat heterogeneity²⁴⁻²⁵, and energy hypotheses²⁶⁻²⁷. “ Again, reference to the MGH is missing here. The sentence should be rephrased to accommodate it, as it specifically refers to mountains, which is the core topic of this manuscript, and thus especially relevant. Muellner-Riehl et al. 2019 already reflected about general hypotheses explaining high biodiversity levels, and then specifically tested the MGH, which was originally proposed for the Tibet-Himalaya-Hengduan region (thus especially relevant for Chinese mountains), and then tested for global relevance.

Response: Thank you for pointing out these important literature. We reorganized the preface. Please see L47-74.

Line 75-77: „Despite this, previous researches have focused primarily on the impacts of ecological factors on mountain biodiversity, while the contributions of geological and lithologic processes are largely unexplored^{3,9,29}. “ This may have been considered true (at least, to some extent - though the field of „geobotany“ is, of course, a very old discipline) some ten years ago, but ignores recent literature on geodiversity-biodiversity relationships. See also my comments further above. But there is also a growing body of literature of several working groups and researchers specifically addressing these issues.

Response: Thank you for pointing out these important literature. We reorganized the preface. Please see L47-74.

Line 84/85: “growing on a specific mountain or in a well-delimited area, which is a relatively independent and self-evolving natural complex³⁰⁻³¹. “ This is not correct. Mountains are usually not considered „independent “ and „self-evolving “, i.e. disconnected from surrounding areas (unlike true islands). There is quite some literature on the similarities, but more importantly differences, between island and mountain systems (e.g. see Mendez-Castro et al. 2021, Itescu 2018, Flantua et al. 2020). This statement needs to be re-phrased accordingly.

Response: Thanks for this criticism. We re-phrased this sentence as “Here, we apply the term “flora” to refer to the sum of all angiosperm families, genera, and species growing on a specific mountain or in a well-delimited area³⁴⁻³⁵, which is a relatively independent biogeographical unit”.

Line 114: “(median = 1,462.0) “ – not entirely clear what this number represents.

Response: We made clear the means of this number is species richness. See L125“.....angiosperms in granitic (species number median= 1,456) and karst-granitic.....”

Line 122-136: “Based on the full model, longitude, elevational difference between the highest and lowest points of a flora (elevdiff hereafter), and mean temperature of the coldest quarter (TCQ hereafter) had positive effects on species richness (Extended Data Table 3). “ In this sentence, and

the following ones, it should be made clear whether larger/smaller or higher/lower values have a positive or negative effect. This seems not evident from the text, as currently written. E.g. do the authors mean “a larger elevational difference “? Or later in line 128, high or low precipitation seasonality and high or low mean temperature? As a note aside, net primary productivity (NPP) might have been a better predictor for species richness.

Response: Thanks for this criticism. We made a change to that, please see L138-L141 “*Based on the full model, larger longitude, larger elevation difference between the highest and lowest points of a flora (elevdiff hereafter), high precipitation of wettest month (PWM hereafter) and high mean temperature of the coldest quarter (TCQ hereafter) had positive effects on species richness (Extended Data Table 3).*” The others sentence had been revised accordingly.

Net primary productivity (NPP) is closely related to the availability of energy and water in the environment. Our model had included 19 climate factors, so that NPP is not included in our analysis.

Line 142 f.: What about the role of dispersal from other mountain systems? (see e.g. Jiménez-Alfaro et al. 2020, DOI: 10.1111/geb.13274; Ding et al. 2020, Science)

Response: Species dispersal is one of the important sources of mountain species diversity. In the discussion section, we specifically discuss the effects of dispersal. Please see L317-L327 “*Dispersal also contributes to the mountain diversity^{2,13,44}. A well-known example is climatic fluctuations during the Quaternary ice age drove changes in the distribution of species around the globe^{51,65}. The spatial configuration of mountain range affect their functional connectivity, influencing species dispersal^{66,67}. In fact, dispersal more easily occurs at initial stage of landform develop mountain, as the slope of mountain still gentle and geographical barrier still relatively low¹³. Such as the plants dispersal events in Qinghai-Tibet Plateau (QTP, low barrier) is obvious higher than that in Hengduan Mountains (high barrier)⁴⁴. Forthermore, the effects of dispersal on species diversity in mountainous areas mainly occurred at lowland, and is limited at highlands where has more local endemic species^{6,30}. This means that dispersal has weak influence on the unique composition of mountain flora of different landforms, especially considering that landform have a significant filtering effect on the species.*”

Line 149-150: “ This is consistent with some fossil evidence, the fossil flora discovered in southwestern China indicate local karst vegetation may have existed since the early Oligocene37-38. “ Is this fossil evidence from the same species or genera as the living ones, and thus specifically relevant?

Response: Yes. The fossil data evidence confirmed the Oligocene flora in southeastern Yunnan is closely similar to the current karst flora. In Dong et al. (2018), a fossil species of *Burretiodendron* is described, which is a endemic genus of karst flora. In another paper (Huang et al., 2018), a species of genus *Ficus* which discovered in Wenshan basin is very similar to living species *Ficus trivialis*. The present species *Ficus trivialis* grows only in limestone ridge scrub, this suggests that limestone shrub vegetation was present during the Oligocene.

Line 151 f.: “ In the full model, orogenic, latitude, temperature annual range (TAR), TCQ is negatively correlated to PDI,·· “ Again here, as already mentioned for lines 122-136, the sentence does not provide as much information as it could: high range? Low range?

Response: Thank you for pointing this out. We revised this sentence to avoid ambiguity, see L169-171 “*In the full model, orogenic, high latitude, high temperature annual range (TAR), and high TCQ*

is negatively correlated to PDI, and high TWQ are the only variables significantly positively correlated to PDI (Figure 4b; Extended Data Table 5)". Other parts of the revised manuscript have also been modified accordingly.

Line 153-157: "may have recently undergone higher rates of evolution related to orogeny¹⁴ " „, while higher TWQ may have led to higher rates of extinction in mountains of sedimentary rock
" Note: This could be tested (speciation rates, extinction rates), and might be worth testing in the future. Mountain biogeography studies often investigate speciation rates and extinction rates, and net diversification rates, respectively, from dated phylogenies, albeit with dense taxonomic sampling (unlike the study here). Investigation of floristic differences of karst versus non-karst, calcareous versus silicate, is a well-researched topic of geobotany (e.g. in the Alps).

Response: We appreciate the review #3's suggestion, which would make for a nice paper in the future. However, that research is different to what we attempted here. In this research, to our knowledge, the first attempt to investigate the relationship between mountain assemble of floras and the types of landforms in which they occur. Our result highlight the floristic assembly in mountains is affect by the bedrock-constrained developmental processes of landforms. As predicted in our hypothesis, nich evolution can promote the spread of species among different landform flora (such as karst landform flora to non-karst landform floras). This could be test in the future.

Line 172: "Landforms should originate earlier than the floras that inhabit them due to the long times that floras require for assembly. " "earlier" – in concert, why earlier? Species can immigrate into a region (dispersal from e.g. other mountain regions or other habitats), and thus can be older than the landform they inhabit nowadays.

Response: Thanks for the criticism. Although, most species which occurs in a mountain is always later than the landform develop process (or evolution with the landform process). Some ancient species could migrate into the mountains by chance.

We revised this inappropriate sentence. Please see line 190-191 "To estimate age pattern of floras between landforms, we measured the mean divergence times of all species (MDT), mean divergence times of the youngest 25% of species (MDT.youngest),"

Line 173-174: "we measured the mean divergence times of all species " Divergence times can only be measure reliably when (almost) all species of a genus are included in a phylogeny, otherwise, ages may be considered not reliable/meaningful. The use of single phylogenies, targeting specific groups, would be needed for that.

Response: Thanks for this advise. Because the aim of this study is to reveal the general pattern of landform influence on the formation of mountain flora, rather than focus the time of each species occurs in a mountain. Thus, the species ages is inferred from a pruned phylogeny based on GBOTB.extended.LCVP.tre (a dated mega-phylogeny). In large-scale biodiversity and phylogenetic analyses, sources of noise in divergence time estimation are to be expected, but they did not affect the reliability of the results.

*Here, we did an analysis to test the divergence time consistency between the two results of species ages eatract by **pruned phylogeny** (used in our study) and the ages eatract from **unpruned phylogeny** (GBOTB.extended.LCVP.tre, a global plant phylogeny). In total 17,576 species included*

in this study, of which 8,663 species were included in “GBOTB.extended.LCVP.tre” and 8713 species not occurs in GBOTB.extended.LCVP.tre. We reconstruct the phylogeny of 8863 species which occurs in GBOTB.extended.LCVP.tre, based on the same method used in this study. Then, extract the divergence time of these 8863 species in new gathered pruned tree and in GBOTB.extended.LCVP.tre (a global plant phylogeny, unpruned tree), respectively. Finally, we get four group of species divergence time data, which are:

“**pruned.add**” age (used in our study)=included 17,576 species, of which ages of 8863 species are extract from the pruned phylogeny tree, and ages of others 8713 species(add) are added by BLADJ method (Webb et al., 2011)

“**unpruned.add**” age =included 17,576 species, of which ages of 8863 species extract from the unpruned phylogeny, and ages of others 8713 species(add) are added by BLADJ method (Webb et al., 2011)

“**pruned**” age=included 8863 species, age extract from the pruned phylogeny tree

“**unpruned**” age=included 8863 species, age extract from the unpruned phylogeny

We did the regression analyse on MDT of four different age datasets and find the close correlation of MDT estimate from “pruned” phylogeny with “unpruned” phylogeny (R^2 of MDT.all is 0.63), especially the younger species ($R^2=0.84$) (Supplemental fig 1). These could also be observed in linear regression fitting result of “unpruned.add” VS “pruned.add” (Supplemental fig 2), “unpruned” VS “pruned.add” (Supplemental fig3), “unpruned” VS “unpruned.add” (Supplemental fig 4). It is particularly to point out that the results of MDT.all and MDT.oldest is consistent (see Supplemental fig 1 and Supplemental fig 3), although “pruned.add” datasets included others 8713 species. This result suggest that ages of 8713 species not occurs in GBOTB.extended.LCVP.tre which inferred in “pruned.add” datasets may be reasonable estimates.

Supplemental fig 1. unprunedVS pruned

Supplemental fig 2. unpruned.add VS pruned.add

Supplemental fig 3. unpruned VS pruned.add

Supplemental fig 4. unpruned VS unpruned.add

Nevertheless, the impact of limited sampling in a regional phylogeny on divergence times still remains to be assessed directly in the future. The species age of “pruned” phylogeny is biased (usually larger) than the “unpruned” phylogeny. While, in large-scale biodiversity and phylogenetic analyses, sources of noise in divergence time estimation are to be expected, but they did not affect the reliability of the results (Lu et al. 2019). Just as the species ages inferred in our phylogeny (a pruned tree of GBOTB.extended.LCVP.tre) did not affect the MDT patterns between mountain in different landform.

Although the estimates species ages “pruned” phylogeny is larger than “unpruned” phylogeny. We believe that our dated mega-phylogeny tree is suitable for this study. Because the aim of this study is to reveal the general pattern of landform influence on the formation of mountain flora, rather than focus the time of each species occurs in a mountain. In fact, the latter is difficult to achieve.

Line 188: “and this may be caused by the stronger influence of glacial periods “ – What was the effect of these glacial periods in the investigated mountain systems? Which were impacted by glaciations, which were affected by colder/drier conditions only? The effect of the glacial periods was much dependent on the availability of water (precipitation; monsoons) and thus regionally very different.

Response: This sentence is to explain the effects of glaciations on the northern mountains in China. The Quaternary glaciation had little influence on the mountain floras in southern China (south of Yangtze valley). Because there are many east-west mountain range that can reduce the impact of ice age cooling (many species surviving in mountain refuges). However, the mountains in the north of Yangtze valley were strongly influenced by the Quaternary glacial period (many species extinct).

Line 214: “Our results are consistent with prior studies^{9,14}” Reference to MGH papers missing here.

Response: We added the citation of reference to MGH (Muellner-Riehl, 2019) herein.

Line 215: “closely related to geomorphic processes” – strictly speaking, this study does not investigate processes as such, but patterns.

Response: Thanks for this comments. Here we added a fig (Please see “Extended Data Fig. 6”) of angiosperm species number of different landforms during specified geological times. The results shows the species accumulation rate increased rapidly after the Miocene (especially in karst, granitic, and karst-granitic landform). Because the Miocene was an important stage when the east Asia monsoon intensified and subsequently accelerated landform processes in China.

Extended Data Fig. 6| Number of angiosperm species of different landforms during specified geological times. a, all species; b, species occurs in granitic landform; c, species occurs in karst-granitic landform; d, species occurs in karst landform; e, species occurs in Danxia landform; f, species occurs in desert landform.

Line 215: “For example, most floras of China, including both mountains and lowland floras, diverged during the Miocene¹¹ when the East Asian monsoon began to prevail..” – not clear what is meant by “floras diverged”, see my previous comments on this issue of dating “floras”. Be more specific about “when the East Asian monsoon began to prevail” – when was the presumed onset, when intensification, when were current levels reached? There is plenty of literature available on this topic (e.g. see publications by Dupont-Nivet, and others).

Response: The formation of the East Asian monsoon is closely related to the uplift of the Qinghai-Tibet Plateau. Although, the process of lifting the Tibetan plateau and when East Asian monsoon start, remains controversial. It is widely accepted that the East Asian monsoon intensified rapidly during the Miocene (Spicer, 2017; 10.1016/j.pld.2017.09.001), Li et al., 2021(10.1126/sciadv.abc7). This is a period of rapid species divergence, such as *Rhododendron* (Xia et al., 2021;

10.1093/molbev/msab314), *Begonia* sect. *Coelocentrum*, (Chung et al., 2014; <http://www.as-botanicalstudies.com/content/55/1/1>), Chen et al. 2018 (10.1093/nsr/nwx156). The period of rapid landform process in modern China coincides with the period of rapid plant species divergence.

We rewrite this sentence, see L235-L238 *“For example, most floras of China, including both mountains and lowland floras, diverged during the Miocene when the East Asian monsoon intensified^{38,52}. Accordingly, this period was also in high rates of development of modern karst, Danxia, and granitic landforms in China⁵³”*.

Line 218 following: “At the intercontinental scale, the ages of floras are also largely consistent with regional developmental processes of landforms. For example, the floras of eastern Asia are typically younger than those of South Africa and Australia, where the landform processes have been very long, but older than those of the Andes and Amazonia, where landform processes have occurred more recently⁴⁸. Thus, the relationship of floristic assembly of mountains to the type of landform and landform developmental process seems to be global in scope, at least for angiosperms. Thus, landform types may make a suitable indicator for explore floristic assembly of mountains, and such an indicator is needed by the scientific community to support biogeographic and other research on mountainous regions²⁹. “ This is a rather simplified statement, needs more elaboration and detail, and also much more consideration of literature that exists for all of these regions. E.g. Do the authors here talk about mountain floras specifically, and if so, which? Different mountain systems on these continents have different ages and history, different orientation (Elsen and Tingley 2015, Nat. Clim. Change), and were also differently impacted by the Last Glacial Maximum (LGM). „explore floristic assembly of mountains “ – the floristic assembly is not explored here per se, but only the patterns. But the assembly can actually be investigated, see Ding et al. 2020 (who you also cite).

Response: Here, we're not focus on specific mountains. We want to point out that the age of local flora in different regions is related on how long the local landform process, even between continents. Chen et al. (2018), had did a systematic comparison of flora age between Andes, Amazonia, California, Australia, South Africa, and East Asia. The Australian and South African foras have older median ages, although the Andes, Amazonian and Californian foras have younger median ages. The formation and evolution of these foras were closely linked to local or global environmental changes. Such as, the Geological history and stratigraphic structure of Australia and South Africa is fairly stable than East Asia (which had experienced Yenshan orogeny at Late Triassic to Cretaceous, Himalayan orogeny at Cenozoic) and Andes (Alpine orogeny at late Cretaceous to Cenozoic). As our view is steady and slow geological processes developed relatively old floras. This is especially obvious in the mountain area.

The second question is how to explored the flora assemble process. The assemble of flora we mentioned is not only a unit, but also an assemble mode of different species, which is related to some lithology types and landform types. Thus, in this study we trace the process of flora assembly by comparison the flora characteristics between mountain landforms. Because the lithosphere cycle and the landform process are well documented. In this study, we drawing a view of flora differences between landform shifts. We believe that if we can grasp the patterns and causes of assemble of flora, it will be possible to further understand how different species and individuals may gather in one aggregation and one flora.

Line 230: results for desert landforms – this may not come as a surprise, as only relatively few plant genera/families are especially adapted to desert conditions.

Response: Yes, the phylogenetic relationship of species in the desert landform flora are closely related. That's mainly due to habitat filtering, as the desert landform development is mainly drought-dominated.

Line 271 following: Here again, reference to the MGH is missing. “ Specifically, under this hypothesis, the ages of mountain floras are determined by the time when erosion of strata begins,…” “ This disregards the effect of immigration on floristic assembly in mountains which can account for most of the species richness on some mountains (compare Ding et al. 2020).

Response: We agree with this criticism. In the new MS, we added a paragraph to discussing the effect of immigration on floristic assembly in mountains. Please see L317-327. “*Dispersal also contributes to the mountain diversity*^{2,13,44}. A well-known example is climatic fluctuations during the Quaternary ice age drove changes in the distribution of species around the globe^{51,65}. The spatial configuration of mountain range affect their functional connectivity, influencing species dispersal^{66,67}. In fact, dispersal more easily occurs at initial stage of landform develop mountain, as the slope of mountain still gentle and geographical barrier still relatively low¹³. Such as the plants dispersal events in Qinghai-Tibet Plateau (QTP, low barrier) is obvious higher than that in Hengduan Mountains (high barrier)⁴⁴. Forthermore, the effects of dispersal on species diversity in mountainous areas mainly occurred at lowland, and is limited at highlands where has more local endemic species^{6,30}. This means that dispersal has weak influence on the unique composition of mountain flora of different landforms, especially considering that landform have a significant filtering effect on the species.”

Methods

Line 405: “we reconciled taxonomy to The Plant List v. 1.1 “. The Plant List (TPL) is an outdated resource, thus other resources should be used instead. For plants, four global authoritative taxonomic lists exist: World Checklist of Vascular Plants (WCVP), the World Flora Online (WFO, an explicit community effort to tackle the increasing wealth of taxonomic information as successor of The Plant List), the Leipzig Catalogue of Vascular Plants (LCVP), and World Plants (WP; both works of dedicated individuals). These four lists each provide a global list of plant names, but differ considerably in size and likely in completeness and accuracy across taxa and geographic regions. The WFO Plant List (www.wfoplantlist.org) replaces the now long outdated Plant List.

Response: Thanks for this important comment. The reason fo our previous checklist reconciled to The Plant List is that the backbone phylogeny tree in V.PhyloMaker (Jin & Qian, 2019). A updated version, V.PhyloMaker2, is available in 2022 (Jin & Qian, 2022;10.1016/j.pld.2022.05.005). With V.PhyloMaker2, one can generate a phylogenetic tree for vascular plants based on one of three different botanical nomenclature systems (TPL, LCVP, and WP).

We accept reviewer’s suggestion and standardized our checklist to match the database LCVP. See L364-L367 “*From our initial checklists, we excluded all nonnative species, and we reconciled taxonomy to Leipzig Catalogue of Vascular Plants (LCVP) using the R4.1.0 (<http://www.r-project.org/>) package, *lcvplants*⁷⁴, with infraspecific taxa combined under their respective species⁴⁷.”*

Accordingly, we reconstructed the phylogeny for the species in this study. See L375-L382 “*With the recently published, dated megaphylogeny tree “GBOTB.extended.LCVP.tre”³⁷, as a backbone,*

we generated the phylogeny for the species in this study using the R package ‘V.PhyloMaker2’³⁷. Of the 2,585 genera and 17,576 species in this study, 2,349 genera and 8,663 species were included in “GBOTB.extended.LCVP.tre”. For the 236 genera missing, we treated each as sister its most closely related genus in “GBOTB.extended.LCVP.tre” based on megaphylogenies within other references⁵²⁻⁷⁶. For species in this study that were absent from “GBOTB.extended.LCVP.tre”, we added them to their respective genera using Phylomatic and generated their branch lengths with BLADJ⁷⁷ implemented in the R package V.PhyloMaker2³⁷. ”

Line 462: “ studies have shown that the species ages within floras are quite different “. Strictly speaking, species ages themselves are not investigated here

Response: Species ages in this study are fitted by using the dated megaphylogeny tree “GBOTB.extended.LCVP.tre” , which is a phylogenetic tree of fossil dating.

Data availability

Line 548: “ All other additional data are available from the corresponding author upon reasonable request.“ What is considered “reasonable“ ? Data should be made available, whenever possible, according to the FAIR principles (findability, accessibility, interoperability, and reusability).

Response: All original mountain floras data used in this study have been published and are accessible to readers from the cited sources. A standardized spatial distribution data of mountain floras are provided with paper. Please see “Data availability” in L514-518.

REVIEWER COMMENTS

Reviewer #2 (Remarks to the Author):

I am re-reviewing this manuscript and I was R2 in the last round. The manuscript is much improved but several of the responses from the authors are not very satisfying. I would particularly like the authors to re-consider the issues below -- I try to explain them better this time.

The authors have kindly pointed to their extended table 10 in response to my request for testing interactions between landforms and other environmental variables. As far as I can tell, Table 10 only contains additive models and cannot identify interactive effects. For example, rather than $Y \sim \text{landform} + \text{climate}$, it needs to be $Y \sim \text{landform} * \text{climate}$, or better yet, $Y \sim \text{climate} + (\text{climate} | \text{landform})$ in a multi-level framework.

In terms of distinguishing the other categories, I think the upper half of the extended Table 3 only shows that most categories are significantly different from Danxia, which is good to know but it still does not distinguish the other three most similar ones: Karst, Karst-Gr and Granitic. It might be best to do pair-wise comparison (t test or Wicoxon test) for all possible pairs.

In terms of the phylogenetic analyses, they authors have apparently misunderstood my point which is NOT about the effect of phylogenetic completeness. I have no problem with an incomplete tree as long as the sampling is not known to be bias in a systematic way. Rather, I am asking for a clear explanation of the reasoning behind the simulation, in terms of what it reflects when random samples are drawn from a phylogeny of only the species in the dataset rather than in the region. I do not see why the null model demanded that the species pool should be consistent in the tree and the distribution database unless it is required by the specific package you use (I do not think it is based on the package documentation). It is possible that if you use a global tree or a regional tree, the patterns remain quite similar if the effect of contingency is not too strong; strong contingency plus large difference in species richness among regions could really bias your simulations. Therefore, I suggest a check on that to ensure the robustness of your findings.

With regard to the "age" of species, I appreciate the authors' effort to provide additional information by comparing the pruned and unpruned trees. However, my point is more about the concept of species or assemblage age -- it is not the real age and can only be used for representing the patterns of age when extinction is random; in contrast, a comparative analysis can be problematic if extinction was not random with regard to the factors that are being examined. Therefore, I suggest rephrasing it in terms of species distinctiveness, so that an assemblage with low MDT contains more closely related species and infer mechanisms from there.

Reviewer #3 (Remarks to the Author):

Review of the revised manuscript "Landform and lithosphere development drive the assembly of mountain floras in China" by Zhao Wan-Yi et al.

I appreciate the efforts by the authors in their attempt to improve the quality of their manuscript. While many, if not most, of my suggestions have been followed, some aspects have not yet been fully addressed and need further attention. Among those are adequate consideration of relevant literature by other authors, importantly with respect to hypotheses and frameworks, especially in the introduction and discussion. I have provided comments and suggestions in the rebuttal and manuscript text files and hope these will be useful to the authors. In addition, I found that especially the newly drafted texts (but not only exclusively those) will require a careful read in terms of use of the English language and terminology to be deemed acceptable.

[editorial note: the mentioned rebuttal starts on the next page.]

Response to reviewers' comments

REVIEWER COMMENTS

Reviewer #1 (Remarks to the Author):

This is a well structured and put together paper. I'm certain you're right that the factors you examined explain patterns of montane diversity and phylogenetic community structure in these mountains in China. I'll be happy to see this excellent work published somewhere. And I do think this is the first time something like this has been attempted, at this scale, with montane plant diversity in China. However, the overall concept and approach is not novel. Others, sometimes working on other taxa and in other locations, have taken the approach of combining contemporary environmental variables with various suites of geological variables to explain spatial variation in diversity. As one key example, I'd point to Antonelli et al. (2018) in Nature Geoscience. In that paper, Antonelli and colleagues analyze "...how erosion, relief, soil, and climate relate to the geographical distribution of terrestrial tetrapods, which include amphibians, birds and mammals. [They] find that centres of species richness correlate with areas of high temperatures, annual rainfall and topographic relief, supporting previous studies. We unveil additional links between mountain-building processes and biodiversity: species richness correlates with erosion rates and heterogeneity of soil types, with a varying response across continents. These additional links are prominent but under-explored, and probably relate to the interplay between surface uplift, climate change and atmospheric circulation through time."

Note that Antonelli et al.'s paper is global in scale and examines multiple taxa. Catherine Badgley and colleagues have also done similar work, focused on mammals in mountains.

All of that is to say that the work is solid, and I believe it. It's just not that novel and limited in taxonomic and geographic scope.

Response: We thank the reviewers for their affirmation of our research result. We also appreciate the previous outstanding researches on mountain biodiversity, as you mentioned those literature in the review comments (Antonelli et al., 2018, Badgley et al., 2017). In their study, they highlighted the impact of geological processes (erosion rate, soil type, etc.) and climate on mountain species diversity based on animal distribution data.

In our study, we test the landform effect on mountain plant species richness, phylogenetic diversity and structure, and mean divergence times. Because plants are more dependent on local rocks and soils, and less responsive to climate than animals, landform (result of rock erosion in mountains) effects may play important roles in the assembly of mountain floras, which maybe not detectable in related studies about animals.

To investigate the relationship between mountain floras assembly and the types of landform. In total, 140 natural mountains flora (including 17,576 species) in China were collected, involving five major geological and landform types (karst, karst-granitic, granitic, Danxia, and desert landform), covering 35 latitudes (about 3500 km) from north to south, and 50 longitudes (about 5000 km) from east to west. In such a large and relatively concentrated area, the analysis of plant floras and plant species diversity in the independent mountain regions has sufficient differences

Commented [A1]: Comment: In addition to my comments included here in this document (Reviewer #1 and #3), I have also provided comments to the manuscript text directly.

I hope my comments here as well as those in the manuscript will be helpful for the authors to improve their manuscript further.

Commented [A2]: I don't agree. Plants can't move and therefore are expected to show even stronger response.

and good comparability, and it is different from the gridding or planarizing analysis of regional species diversity, [Therefore, we believe that the analysis in this paper should have different meanings compared with the previous analysis results.]

Our results shown that mountains of igneous bedrock have higher species richness and exhibit phylogenetic overdispersion, while mountains of sedimentary bedrock have lower species richness and clustered phylogenetic structure. We also find that the type of landform has a greater effect on floristic assembly for floras containing evolutionarily older species, while climate is a greater determinant for floras with younger species. These results indicate landform development process not only affects mountain species richness, but also determines the composition of floras. In conclusion, our results highlight bedrock in mountain and landform effects plays a key role in floristic assembly, especially considering the bedrocks on Earth's is unevenly distributed. These are new insights of the occurrence of plant diversity and bio-diversity protection in mountain areas.

Commented [A3]: I don't regard this as really convincing argumentation.

Reviewer #2 (Remarks to the Author):

This paper explored the relationship between the type of landforms and floral diversity in mountain ranges. Floral checklists from 140 natural reserves and natural parks in Chinese mountain ranges were compiled. Plant species richness, phylogenetic diversity, species relatedness and time since divergence were modeled based landforms and geological, geographic and climatic variables, controlling for the potential area effect. All measures of floral diversity differed among landforms, with additional effects from some other variables.

Findings in this study are important for understanding biodiversity in mountain ranges which is a central topic in biodiversity and biogeographic research. In particular, the significant role of landforms in generating and maintaining mountain biodiversity across a broad spatial scale is a

novel finding. I do have some concerns and suggestions listed below, but if the results are robust, I think this study has very good potential to make a fine publication which will be of great interest for the broad readership of the journal. The writing is clear and generally easy to follow.

Response: We thank reviewer 2 for appreciating our work and, even more, for the useful insights on how to improve it.

Overall, I feel that the specific mechanisms of how landforms could shape the plant communities/floras could be explained further in depth (e.g. in the introduction), especially considering the broad readership of the journal.

Response: Thanks for your suggestions. To our knowledge, there are no studies that specifically discuss the impact of landform (which is more or less involved in heterogeneity, lithology types, etc.) on species richness of mountains. In the discussion part (line 281-327), we added three paragraph to discussion how landforms could shape the assembly of mountain floras.

Comparing the importance of landforms and other variables in explaining diversity seems to be a logical step but theoretically, their effects can't really be separated. The importance of landforms in shaping floral diversity/composition thus could be further established by investigating their interactions with other environmental variables, as briefly discussed by the authors (L131-132). Either a boosted regression or a multi-level model with landforms as a group-level effect would be very illuminating.

Response: Very thanks for your advice. We accepted the suggestion of reviewer #2, and investigated the interaction of landforms and other variables. The results show that the landform effect is always significant. Please see L130-L132 *"We further tested the landform effects when interactions with others variables. The results showed that landform combined with any other variables will significantly be improved the explanatory power of the model for species richness (Extended Data Table 10)."* The details results of model fitting are shown in the are shown in Extended Data Table 10.

Variables	GLM			SAR		
	Deviance, %	AIC	Moran's I	Deviance, %	AIC	Moran's I
Landform+Tectonic	33.57	108.80	0.064 ^{ns}	34.70	109.25	0.002 ^{ns}
Landform+log(area)	34.52	106.79	0.068 ^{ns}	35.79	107.03	-0.001 ^{ns}
Landform+Longitude ^{ns}	29.17	117.77	0.886*	31.41	116.87	0.002 ^{ns}
Landform+Elevdiff	38.33	98.39	0.077 ^{ns}	40.33	97.67	0.003 ^{ns}
Landform+TWQ	34.62	106.56	0.210***	43.00	97.20	-0.011*
Landform+TCQ ^{ns}	29.64	116.85	0.077 ^{ns}	31.36	116.64	0.002 ^{ns}
Landform+PWM	33.00	109.99	0.089*	34.90	109.29	0.003 ^{ns}
Landform+P _{var}	35.74	104.15	0.054 ^{ns}	36.46	105.10	0.002 ^{ns}
Landform+PCQ	31.81	112.45	0.074 ^{ns}	33.19	112.55	0.003 ^{ns}
Landform+Tectonic+ Elevdiff+TWQ+TCQ+P _{var}	55.58	60.44	0.049 ^{ns}	56.18	61.32	0.003 ^{ns}

In addition, a lot of the variation among landforms seems to come from deserts vs others, based on fig. 3a-c -- do the other groups differ?

Response: The desert indeed provides a lot of variation among landforms. While, the others landform types also provide a lot of variation. Such as, the species richness of Granitic and Karst-Granitic landform is significant higher the Danxia landform(the reference) (see "Extended data Table 3"). In the landform model for MDT, MDT.oldest, and NTI, we also detected the Desert, Granitic, Karst, and Karst-Granitic landform are always significant differences between Danxia landform (see "Extended data Table 5-6, 9").

A major technical concern is about how the phylogenetic data were handled. The phrasing at L416 seems to suggest that species not occurring in your studied areas are pruned, which will affect your null expectation for those phylogenetic metrics and your calculation of phylogenetic dispersion/clustering. I do not have a good answer to what would be the best way to construct a null model for phylogenetic diversity or dispersion, whether it should be the global plant phylogeny or a Chinese plant phylogeny, or for a slightly different question, a regional phylogeny just for the floras around the focal site. Any of these cases would be better than a phylogeny of only the species in your dataset. Please clarify this and explain your reasoning.

Response: Thanks for your advice. In this study we pruned those species not occurring in our studied areas, as the null model demanded that the species pool should be consistent in the phylogenetic tree and distribution database to ensure all species can be sampled randomly in the NRI, NTI, and PDI null model (Webb, 2000, doi: 10.1006/jema.1996.0042; Jin & Qian, 2022, doi: 10.1016/j.pld.2022.05.005).

Here, the reviewers 2 also raised the concern about "if an incomplete phylogeny can really reflect the history of flora". Phylogeny-based approaches are essential to understanding differences in species richness and the assemble history of floras between regions. In fact, using mega-phylogeny data to extract regional species phylogeny (such as phylogenies generated by V.PhyloMaker) is a common approaches in community ecology, macroecology and study on regional flora.

In fact, the mode of phylogenetic inference has little influence on phylogeny diversity metrics. The developer of V.PhyloMaker had specially written a paper to clarify the issue. They test the difference using 1093 angiosperm tree assemblages in North America (Qian & Jin, 2021; doi.org/10.1016/j.pld.2020.11.005; Jin & Qian, 2022;https://doi.org/10.1016/j.pld.2022.05.005). Several previous case studies have also demonstrated that equivalent results have been obtained from purpose-built and synthetic phylogenetic trees (Allen et al., 2019, doi: 10.1016/j.isci.2018.12.002; Jantzen et al., 2019, doi: 10.1002/ece3.5425; Li, Trotta, et al., 2019, doi: 10.1002/ecy.2788).

Related to that, the interpretation of terminal branch length as species age and all associated discussions are problematic, because the plant phylogeny presumably does not include extinct lineages. The terminal branch length thus only reflects a species' distinctiveness from extant plant species within your sampling pool. In fact, the whole section about the ages of floras is problematic and not very informative unless the formation time frame of their habitats could be provided. To some extent, these analyses could help inferring whether modern floras were shaped by radiation in-situ, but this might involve a lot of confounding factors. The outcome also heavily depends on whether you have stripped away species not included in the study, and the data coverage of higher

taxa (which is not explained, unless I miss it).

Response: Thanks for your advice. It is inevitable that the species divergence age obtained from the regional phylogenetic data will be biased without extinct lineages. Here, we did an analysis to test the divergence time consistency between the two results of species ages extracted by **pruned phylogeny** (used in our study) and the ages extracted from **unpruned phylogeny** (GBOTB.extended.LCVP.tre, a global plant phylogeny). In total 17,576 species included in this study, of which 8,663 species were included in "GBOTB.extended.LCVP.tre" and 8713 species not occurs in GBOTB.extended.LCVP.tre. We reconstruct the phylogeny of 8863 species which occurs in GBOTB.extended.LCVP.tre, based on the same method used in this study. Then, extract the divergence time of these 8863 species in new gathered pruned tree and in GBOTB.extended.LCVP.tre (a global plant phylogeny, unpruned tree), respectively. Finally, we get four group of species divergence time data, which are:

"pruned.add" age (used in our study)=included 17,576 species, of which ages of 8863 species are extracted from the pruned phylogeny tree, and ages of others 8713 species(add) are added by BLADJ method (Webb et al., 2011)

"unpruned.add" age =included 17,576 species, of which ages of 8863 species extracted from the unpruned phylogeny, and ages of others 8713 species(add) are added by BLADJ method (Webb et al., 2011)

"pruned" age=included 8863 species, age extracted from the pruned phylogeny tree

"unpruned" age=included 8863 species, age extracted from the unpruned phylogeny

The results also shown the MDT patterns of mountain flora are similar of the four different species divergence time datasets (Supplemental fig. 1). The MDT of karst landform is always the highest, while the MDT of Danxia landform and desert landform is lower. The results also shown that the MDT estimate from "pruned" phylogeny is higher than the "unpruned" phylogeny (Supplemental fig. 1-5). This could be expected, as the species branch in a "pruned" phylogeny is usually longer than which in the global plant phylogeny or a phylogeny included more species. Besides, there are 8713 species not occurs in GBOTB.extended.LCVP.tre, the age of these species was treated as their genus node branch (see method "*Phylogenetic reconstruction*").

Supplemental fig1. MDT results of mountain flora derived from these four species divergence time datasets

Further, we did the regression analyse on MDT of four different age datasets and find the close correlation of MDT estimate from “pruned” phylogeny with “unpruned” phylogeny (R^2 of MDT.all is 0.63), especially the younger species ($R^2=0.84$) (Supplemental fig 2). These could also be observed in linear regression fitting result of “unpruned.add” VS “pruned.add” (Supplemental fig 3), “unpruned” VS “pruned.add” (Supplemental fig4), “unpruned” VS “unpruned.add” (Supplemental fig 5). It is particularly to point out that the results of MDT.all and MDT.oldest is consistent (see Supplemental fig 2 and Supplemental fig 4), although “pruned.add” datasets included others 8713 species. This result suggest that ages of 8713 species not occurs in

GBOTB.extended.LCVP.tre which inferred in “pruned.add” datasets may be reasonable estimates.

Supplemental fig 2. unprunedVS pruned

Supplemental fig 3. unpruned.add VS pruned.add

Supplemental fig 4. unpruned VS pruned.add

Supplemental fig 5. unpruned VS unpruned.add

Nevertheless, the impact of limited sampling in a regional phylogeny on divergence times still remains to be assessed directly in the future. The species age of “pruned” phylogeny is biased

(usually larger) than the “unpruned” phylogeny. While, in large-scale biodiversity and phylogenetic analyses, sources of noise in divergence time estimation are to be expected, but they did not affect the reliability of the results (Lu et al. 2019). Just as the species ages inferred in our phylogeny (a pruned tree of GBOTB.extended.LCVP.tre) did not affect the MDT patterns between mountain in different landform.

Although the estimates species ages “pruned” phylogeny is larger than “unpruned” phylogeny. We believe that our dated mega-phylogeny tree is suitable for this study. Because the aim of this study is to reveal the general pattern of landform influence on the formation of mountain flora, rather than focus the time of each species occurs in a mountain. In fact, the latter is difficult to achieve. In the manuscripts, we made no further changes and discuss of the phylogeny tree in order to maintain succinctness and consistency. The test result of four different age dataset herein will be provided as “Peer review information” for the readers (when this study be published). We hope our reply will clear up your doubts.

The models do not seem to account for spatial auto-correlation, and no justification has been given on why that is not an issue.

Response: Thanks for this important and insightful comment. In the methods part we added Moran's I value to quantify the residual spatial auto-correlation under the GLM, SAR models, and they obtained similar results. Detail see Methods-“Spatial analysis” in L503-L511 *“Spatial autocorrelation is a general feature of macro-ecological data and may leading to erroneous interpretations¹⁰⁰. We use Moran’s I values to quantify residual spatial autocorrelation, which is considered as a spatial equivalent to Pearson’s correlation coefficient and normally varies between 1 and -1, and expect Moran’s I values for lacking spatial autocorrelation is close to 0¹⁰¹. Because of spatial autocorrelation also present in our data set, we performed spatial simultaneous autoregressive model (SAR) to account for residual spatial autocorrelation¹⁰². The expect Moran’s I values for the response variable as well as for GLM and SAR residual are shown in appendix (Extended data Table 3-9). Spatial statistics were performed with “spdep” package in the R version 4.1 (<https://www.r-project.org/>).”*

Minor comments:

L67-77: this paragraph might be better for the beginning of the Intro. The current beginning completely ignores elevation and associated environmental heterogeneity, which doesn't provide a broad enough overview of the topic.

Response: Thanks for this kindly reminder. We rewrote the Intro part. At the beginning, we summarized the general explanations for the higher diversity of mountain species. The effects of geological processes and lithology on mountain plant diversity were further introduced. Please see revised manuscript L47-L62. *“Globally, mountains play dual roles as museums and cradles in the formation of species diversity¹⁻² and, therefore, it is unsurprising that much of global biodiversity is concentrated within mountains, especially within the tropics³⁻⁵. The worldwide mountains harbor the 40% of the global diversity, and species inhabiting in mountains are double to the lowlands when taking into account the area effect⁶. How extraordinary diversity of mountains occurs is still a great challenge since the Humboldt's time⁷.*

Historically, mountain biodiversity and, more broadly, global biodiversity, have been explained by numbers of hypotheses such as the climate stability⁸, habitat heterogeneity⁹⁻¹⁰, and energy hypotheses¹¹⁻¹². More recently, a comprehensive model for biodiversity prediction is established¹³⁻¹⁷

that includes ecological processes (survival, competition, and niche differentiation), biological processes (species divergence and extinction), and geological and lithologic processes (such as orogeny and rock formation)¹⁸⁻¹⁹. Among which, a representative theory is “mountain geobiodiversity hypothesis” (MGH), which proposed to explain the biodiversity of Tibeto-Himalayan region². Within these framework, geological and lithological processes, especially uplift and erosion, are known to have strong effects on mountain biodiversity; probably through their roles in species formation, immigration, and extinction^{12,20-21}.”

L72-77: the two sentences are a bit contradicting with each other about whether the subject has been studied or not -- better rephrase.

Response: We revised the sentence. Please see revised manuscript L53-L74 (the last reply).

L83: add "to" to before "refer to"

L87: change "resolved" to "reflected" or "represented"

Response: Thanks for your help to improve the linguistics of our manuscript!

L89: better clarify what "phylogenetic structure" is referring to (relatedness?)

Response: Thank you for this suggestion. Phylogenetic structure represents the phylogeny relatedness (clustered or overdispersed) of species in a mountain flora, as you mentioned, which is an important concept in the study of mountain species assemblages. The phylogenetic structure of a flora is most commonly assessed by NRI and NTI.

L93: change "additional" to something like "in addition to" or "along with"

Response: Thanks for your help to improve the linguistics of our manuscript! We had revised it in the article accordingly.

L140: PDI has not been explained at this point -- given the format with Methods after everything, it would be helpful to have some technical information specified in the results (e.g. whether area and latitude were accounted for in the models).

Response: The phylogenetic diversity index (PDI) is a standardized PD_{Faith} using null models, allows comparisons among floras phylogenetic diversity with different underlying species richness values. As you can see in Figure 3b, we first shown the PDI difference between five landform types. We further detected the landform effects on PDI in both “landform” model and “full” model, just as the analysis performed for species richness.

To avoid confusion, we rewrote the methods section (see Methods “GLM analysis” in L486-L496) “We performed generalized liner models (GLM) to model log-transformed species richness (SR) as the response variable with landforms and tectonic and climate variables as predictors. Initially, we modeled SR as a function of only landform (i.e., “landform” model) because, in this study, we focused primarily on the roles of landforms in floristic assembly. However, we also extended the “landform” model to include all others variables, especially to assess the climatic impact on species diversity (i.e., “full model”). Further, we used GLM to determine the effects of landform, tectonics and climate on NRI, NTI, PD, PDI, MDT, MDT_{oldest} , and MDT_{youngest} (Extended Data, Table 2). As with SR, we initially modeled landform as a single predicting factor (“landform” model) before extending to all variables within a “full model”. We performed all GLM analysis in

R version 4.1 (<https://www.r-project.org/>) using the "glm" function in the MASS package⁶⁸.

The "full" model of PDI also included all the predictor variables listed in "Extended data Table 2", in which area and latitude were considered.

L244-250: environmental filtering as a potential mechanism should probably be discussed here, especially given L235-237.

Response: This is a good suggestion. Because most of the granitic landforms floras is phylogenetic overdispersion, while phylogenetic clustering granitic landforms floras is usually located at high latitudes. We takeout a discussion on how landform and environmental filtering affects the flora phylogenetic structure of floras in L313-L316. *"The mountains of sedimentary bedrock have much stronger environmental filtering effect, as these mountain ecosystems are more sensitive to rainfall. The environmental filtering effect further promoted the clustering of phylogentic structures of mountain floras as predicted by the phylogenetic niche conservatism (PNC)^{46,50,64}."*

L246: TAR reflects "instability", right?

L260: change "maybe" to "may be"

L398: "a total of"

L343: better use "allows" than "facilitates"

Response: Thanks for your help to improve the linguistics of our manuscript! We had revised this in the article accordingly.

L440: is NRI driven by sample coverage, i.e. the number of artificial sister species/genera in the floras?

Response: In general, the more species contains in a genus, the mean phylogenetic relationship of species in a mountain may be more closely (the NRI tend to be > 0). However, the NRI is derived from the null model. In our result, we did not observed the correlation between NRI and the mountain flora species/genera number. Such as, the mountain Taibaishan in Shaanxi province has 1656 species, and its NRI is 4.54. In contrast, the mountain Lushan in Jiangxi province has similar species number (1590) to Taibaishan, but its NRI is -2.43. In other words, NRI reflects the dispersion degree of phylogenetic relationships among species, and has little correlation with the number of species.

L461: "is different" or "differs"

Response: Thanks for your help to improve the linguistics of our manuscript! We had revised it in the article accordingly.

L509: please clarify whether the resolution of the climatic layers matches your sampling areas -- are most floras much larger than the units of the climatic datasets?

Response: Thanks for this important and insightful comment. We agree with this criticism, and indeed authors had discussed about these issues before the original submission. We test the collinearity relation between climatic data which extract by GPS location from CHELSA(30 arc-seconds), World Climate (2.5 m, 5 m, 10 m). The results are closely related. So that, we select CHELSA(30 arc-seconds) which is a high resolution climate data. However, we were not fully successful the problem, because the climate data of a site note welly represent the mountain (as most of the mountain site area in this study over 100 km²).

Therefore, in the revised manuscripts we gathered the .shp file layer of each mountain site. The new climate data for analysis were extracted as mean value of each mountain site layer using the zonal statistics in Arcgis. See L467-L470. *"For each of the 140 mountain floras, we downloaded climatic data from the CHELSA climate dataset (v. 1.2, available at <http://chelsa-climate.org/>) at a spatial resolution of 30 arc-seconds⁹⁶, and extracted the climatic variables mean values of each mountain layer using the zonal statistics function in ArcGIS 10.8."*

We suspect that the climate data extracted by this method can represent the actual mountain climate data.

Reviewer #3 (Remarks to the Author):

Review of the manuscript “Landform and lithosphere development drive the assembly of mountain floras” by Zhao Wan-Yi et al.

The authors investigated 140 mountain regions (with their flora and bedrocks) in China to study the relationship between mountain floras and the types of landforms in which they occur. The authors obtained a backbone phylogeny from a recently generated time-calibrated mega-tree, including species across all floras. Based on the phylogeny, they calculated phylogenetic diversity, phylogenetic structure, and mean divergence times. They constructed regression models to predict species richness, phylogenetic diversity and structure, and mean divergence times, using landform as a predictor, and using additional tectonic, climatic, and geographic explanatory variables as predictors. Based on their findings, the authors put forward what they call "the geological lithology hypothesis of assembly and differentiation of mountain floras". They claim that this hypothesis “provides a novel framework rooted in geology for future research on the origins, differentiation, and migration of angiosperm assemblages and mountain floras.”

This study complements a growing body of literature on the close interrelationships between geological diversity (geodiversity) and biological diversity (biodiversity), on their “concerted” evolution, and on the assembly of mountain floras. It stresses edaphic factors (related to bedrock, landforms) as important for ecological filtering, species immigration, and evolution, further driving differences among landforms.

I enjoyed reading this manuscript, and think it can potentially make an important contribution. However, at the same time I found that reference to highly relevant recent literature (pertaining to the 'mountain geobiodiversity hypothesis') was lacking and thus not considered, especially with regard to the hypothesis here by the authors. In addition, some methodological issues, e.g. pertaining to taxonomic name resolution (use of the now long outdated Plant List) and molecular dating (derivation of species divergence times from a phylogeny not sampled for taxonomic coverage), need consideration.

Response: Above three paragraph are an excellent summary of what we did and conveys very well the general aim of our work. We wish to thank reviewer #3 for appreciating our work and, even more, for the useful insights on how to improve it. We have studied these comments carefully and have made substantial changes. The corrections are indicated in red in the manuscript, and replies to comments are listed as follow.

In the following, I will provide my comments which I hope will be useful for the authors. Line numbers refer to the pdf.

Title: As the study is confined to mountains in China, and the authors emphasize the special geological conditions in China in their manuscript, the title could be adjusted to reflect this. E.g. “Landform and lithosphere development drive the assembly of Chinese mountain floras”. Also, the word “drive” may be reviewed (“drivers” is a strong term).

Response: Thanks! We accept the suggestion to change the manuscript title as “Landform and lithosphere development drive the assembly of mountain floras in China”. On the other hand, we suspect “drive” is a suitable word to describe the effect of landform development/lithosphere

cycle on mountain flora aggregation. Because once the geological layer is uplifted during orogeny. The subsequent erosion in mountain is irreversible. However, the erosion rate of different bedrock types and the interaction process with rainfall and other environmental conditions are very different. Thus will development into different landscape and mountain flora. Such as, along with the occurrence of mountain erosion process, the adjacent limestone strata and igneous rock strata will finally develop into different flora.

Furthermore, the formation of mountains on the Earth's surface is the result of the cycle of sedimentary, igneous and metamorphic rocks. We believe that the basic conclusions of this study are also applicable on a global scale. This is worthy of further research in the future.

Line 30: “The biodiversity of mountains is attributable in part to their geological activities, namely uplift and erosion, as well as the habitat heterogeneity that they provide, such as along their elevational gradients⁴⁻⁶ “. It would be appropriate to cite the work by Antonelli et al. 2018 already here.

Response: Yes, the research carry out by Antonelli et al. (2018) is quiet an important in mountain species diversity. We cited this literature in the introduction and other part of our manuscript, please see L56, L65, L173, L234.

Line 31: “how mountain biodiversity is shaped on temporal and spatial scales remains poorly understood⁷⁻⁹ “ This ignores a growing body of literature investigating this in detail, such as literature on the 'mountain geobiodiversity hypothesis' (MGH) (Mosbrugger et al. 2018, Muellner-Riehl 2019, Muellner-Riehl et al. 2019, and references therein). There are, by now, hundreds of papers on Chinese mountain biogeography and Chinese mountain phylogeography. This needs more to receive more attention in the manuscript, and sentences such as the one cited above (“poorly understood”) need to be rephrased to pay tribute to the growing knowledge.

Response: Thank you for raising this important points. We read the relevant literature about “mountain geobiodiversity hypothesis”, the monograph “Mountains, climate and biodiversity”, and some related Chinese mountain biogeography and Chinese mountain phylogeography references carefully. We furtherly reorganized the preface. First, we briefly summarize the comprehensive framework to explain the formation of mountain species diversity. Furthermore, we focus on the influence of lithology or bedrocks on plant diversity in mountain areas. Because the differences in the bedrock of the stratum largely dominate the final type of the mountain landform. Landform types have the unique impact on the plant species assemblages within the mountains. This is a direction that has not been well studied. Please see L47-74. “Globally, mountains play dual roles as museums and cradles in the formation of species diversity¹⁻² and, therefore, it is unsurprising that much of global biodiversity is concentrated within.....”

Historically, mountain biodiversity and, more broadly, global biodiversity, have been explained by numbers of hypotheses such as the climate stability⁸, habitat heterogeneity⁹⁻¹⁰, and energy hypotheses¹¹⁻¹². More recently, a comprehensive model for biodiversity prediction.....

Mountains comprise cradles of species diversity largely because their formation and subsequent bedrock erosion yield topographic complexities and produce new niches for all kinds of organisms^{13,22-24}. Additionally, they also facilitate the formation of endemic species.....”

Line 38-40: “the ages of the studied floras are consistent with the geological development of the different landforms with mean divergence times being the highest... “ This is not clear - what

Commented [A4]: My comment had not addressed the citation being missing, but I ad requested an earlier mention of it. Preferably, the answer in a rebuttal letter should directly refer to what has been asked for. Here, it does not become evident from the text straightforwardly, whether this is the case.

Commented [A5]: I appreciate the improvements, while at the same time I would like to mention that the text still needs further adjustments. In its current new form, the text in the introduction still does not play out its full potential. I have left some comments in the introduction for the authors.

Commented [A6]: Geodiversity as a measure captures this, and as such, the MGH inherently includes this aspect, which should be acknowledged in the manuscript. The intro text needs to go more into some more detail concerning the similarities and differences between the MGH and the approach proposed here. It is important to carve out the further refinements of the approach here for the reader. This will not diminish the accomplishments of the authors, but rather help them to out their work in context. Otherwise, the text will imply more novelty than justified.

exactly does it mean that ages are consistent, given that the authors later define „flora “ (line 83) as „sum of all angiosperm families, genera, and species “? Different taxonomic levels (species – genera – families) have vastly different ages (unless a genus or a family are both monospecific, or close to), and including only a very small fraction of the species of a genus or family, as done here (because only the species occurring on the 140 Chinese mountain regions are included in the analyses), will not allow to arrive at ages which are anywhere close to the “real” ages of the species (or higher taxonomic levels).

Response: This comment is useful. In our study, flora age differences were compared by MDT (average age of extant species in the mountain flora, MDT). The MDT is a index to assess the relative age of modern flora (Lu et al. 2018). We revised this confused sentence, please see L47-74. “Moreover, the mean divergence times of floras (MDT) being the highest for karst, followed by karst-granitic, granitic, Danxia, and desert landforms.”

Line 42: What do the authors consider as “younger” species?

Response: The “younger” species represent the species that diverged later in the mountain flora. To avoid potential bias between mountains, we ranked all species occurs in 140 mountains from youngest to oldest, partitioned them into quartiles based on their ages, computed MDT in each mountain for the absolute youngest 25% and the absolute oldest 25% of species (Lu et al. 2018).

Line 42-47: “We put forward “the geological lithology hypothesis of assembly and differentiation of mountain floras” .” “Our hypothesis provides a novel framework rooted in geology for future research on the origins, differentiation, and migration of angiosperm assemblages and mountain floras.” As mentioned before, the authors seem unaware of the MGH ('mountain geobiodiversity hypothesis'). This hypothesis was originally formulated by Mosbrugger et al. 2018, later refined by Muellner-Riehl 2019, and finally tested on a global scale by Muellner-Riehl et al. 2019.

The MGH is supposed to provide an overarching framework for the investigation of the concerted evolution of mountains and their biotas, and on the assembly of mountain biodiversity, through time, and also considering climatic fluctuations which act on top of orogenic processes. Together with the flickering connectivity system by Flantua et al. (the latter which is cited by the authors), the MHG thus deals with the intricate relationship between mountains and their biotas/flora. It needs to be elaborated and explained what the “geological lithology hypothesis” would add specifically, and how it differs from the other hypothesis already brought forward and tested by empirical studies in the past few years. The authors should consult the work by Mosbrugger et al. 2018, Muellner-Riehl 2019, Muellner-Riehl et al. 2019. Is it justified to formulate yet another hypothesis, or may the hypothesis introduced here be considered under the umbrella of the MGH?

Response: Thank you for point out this important literature that we previous missed. We read the MGH relevant literature carefully. The MGH has three boundary conditions 1) presence of lowland, montane and alpine zones, 2)climatic fluctuations for a “species pump” effect, and 3) high-relief terrain with environmental in a given mountain region. Thus, the diversity of species in mountains is due to high differentiation, low extinction, and migration. MGH theory is a systematic and explanatory hypothesis especially in explaining the formation of species diversity in high mountain areas experienced many climatic fluctuations.

Our “geological lithology hypothesis of flora” differ from MGH. The “geological lithology

Commented [A7]: As also mentioned in my comments to the manuscript, the latter needs to include some statement about the limitations and shortcomings of this approach.

Commented [A8]: This does not answer my question as intended. I was interested in absolute ages.

hypothesis of flora” proposed in this study highlight the bedrock in mountain and landform process plays a key role in flora assembly. To be specific, bedrocks and associated micro-landforms in mountains is the most important factor promote speciation local endemic species. Second, landform effects (as environmental filtering) restrict the free dispersal of plants between mountains of different landform types (because edaphic species cannot exist outside the original bedrock, dispersal event could occurs with phylogenetic nich evolution), and this further leads to differences among the mountain flora in genera/species. Furthermore, “geological lithology hypothesis of flora” not only useful in predict species richness in a mountain, but also provided a new view to understand which species will finally present in the mountains of different landforms. Moreover, our hypothesis also tries to explain the reason why each mountain has high or low plant diversity, or suitable plant diversity.

Line 43/44: “Under this hypothesis, landforms develop according to their underlying bedrock, and their geological development drives both the assembly and subsequent differentiation of mountain floras. “ This may be considered less a hypothesis than a fact.

Response: Here we emphasize the final landform type of mountain is determined by the bedrock. Such as, limestone stratum can't form a mountain with high elevdiff (highest minus lowest) as igneous rock strata mountain. Thus, the assemblages history in mountain flora of different landforms will be different.

General comment: Missing in this first paragraph of the text is also a general reference to “geodiversity” and “geobiodiversity”. It may be worth noting here, for the information of the authors, that the recognition of the intricate relationship between geological diversity and biodiversity has led to specific research activities in the scientific community, such as e.g. the Research Activity “Geobiodiversity and Climate” which “studies the interactions between climate, Earth surface processes and biodiversity on different time scales. It further examines the impacts of climate and Earth surface processes on the evolutionary and ecological dynamics of species and communities. An important research topic are the effects of anthropogenic climate change on biodiversity and ecosystem functions.” (<https://www.senckenberg.de/en/science/biodiversity-and-climate/geobiodiversity-and-climate/>). What I am trying to say here is that the introductory part of the manuscript leaves the impression that the authors are not aware of an important body of literature and concepts pertaining to geobiodiversity, and this should be avoided.

Response: Thanks for the reviewer's criticism. After a long period of field investigation, we realized that the flora composition in mountains is different among different landforms. So that, in this study, we focused on the impact of landform effects on the process of mountain flora assemblage (included species richness, phylogenetic structure, and MDT) in China. To our knowledge, landform effect on mountain flora assemblage is not tested in previous geobiodiversity studies. Landform type maybe a brief indices of geodiversity, as landform is the result of comprehensive interaction of bedrocks, climate and other factors. We are grateful to the reviewer 3 for pointing out the frontiers of mountain biodiversity research here.

In carrying out this work, we have reviewed the literatures for the high abundance of mountain or regional biodiversity, as well as several important hypotheses. The reviewer reminded us to add the MGH hypothesis, which is now revised in the new draft. Please see line 58-

Commented [A9]: It may be argued that this is not correct, i.e. it may be disputed that bedrock and micro-landforms are THE MOST IMPORTANT factor to promote the evolution of local endemics. Mountain orientation plays a key role. Comparing mountains of different orientation (W-E versus N-S) shows that those oriented N-S harbor a higher no. of species that were able to survive during times of climatic change and that were able to form endemic lineages than those oriented W-E. This is because a W-E orientation prevents populations moving into more favourable latitudes during times of climate change (barrier effect) that has happened in the past. A N-S orientation, in contrast, as e.g. found in the Hengduan Mountains or the Andes, enabled populations to move into more favourable latitudes.

Commented [A10]: I guess the authors here mean “establishment”, not “dispersal” per se may be restricted.

Commented [A11]: The meaning of this sentence it not entirely clear to me. I assume what is meant here is that if individuals of a species which are adapted to a specific soil type happen to be dispersed to areas of another soil type, in order to survive, they would have to adapt, and as a result, may become evolutionary independent lineages, potentially new species? This is actually less likely than if their propagules dispersed to areas of the same soil type, which may foster a simpler type of allopatric speciation.

Commented [A12]: This is, in the core, not different from the MGH (compare Muellner-Riehl et al. 2019), but it adds an additional component.

Commented [A13]: I would have appreciated an answer which more directly reflects what was actually done in the new version of the manuscript to satisfy the criticism. This is also true for the answers to the other questions/remarks by reviewer 3. What was asked here was that the intro does not reflect the body of literature sufficiently. Just adding two sentences, as indicated further below at the end of the answer does not do justice to what is known about

Commented [A14]: Was this studied here? I need to re-check whether the (tax.) composition of the flora was investigated.

Commented [A15]: But geodiversity as a measure (geodiversity index) is an integral part of investigations under the MGH.

60 “Among which, a representative theory is “mountain geobiodiversity hypothesis” (MGH), which proposed to explain the biodiversity of Tibeto-Himalayan region².”
also in line 291-293 “.....[the local flora assemble is a comprehensive result of species evolution, landform development and climate change^{3,13,44}.”

Commented [A16]: But it applies not only there. While it was originally developed for the THR, later work by Muellner-Riehl et al. 2019 in JBI tested its global validity.

Line 67/68: “Historically, mountain biodiversity and, more broadly, global biodiversity, have been explained by numbers of hypotheses such as the climate stability²³, habitat heterogeneity²⁴⁻²⁵, and energy hypotheses²⁶⁻²⁷. “ Again, reference to the MGH is missing here. The sentence should be rephrased to accommodate it, as it specifically refers to mountains, which is the core topic of this manuscript, and thus especially relevant. Muellner-Riehl et al. 2019 already reflected about general hypotheses explaining high biodiversity levels, and then specifically tested the MGH, which was originally proposed for the Tibet-Himalaya-Hengduan region (thus especially relevant for Chinese mountains), and then tested for global relevance.

Commented [A17]: See the MGH – dispersal of pre-adapted lineages and extinction are equally important. Also, the order “*species evolution, landform development and climate change*” should be reconsidered. See also my comments in the manuscript.

Response: Thank you for pointing out these important literature. We reorganized the preface. Please see L47-74.

Line 75-77: „Despite this, previous researches have focused primarily on the impacts of ecological factors on mountain biodiversity, while the contributions of geological and lithologic processes are largely unexplored^{3,9,29}. “ This may have been considered true (at least, to some extent - though the field of „geobotany“ is, of course, a very old discipline) some ten years ago, but ignores recent literature on geodiversity-biodiversity relationships. See also my comments further above. But there is also a growing body of literature of several working groups and researchers specifically addressing these issues.

Response: Thank you for pointing out these important literature. We reorganized the preface. Please see L47-74.

Line 84/85: “growing on a specific mountain or in a well-delimited area, which is a relatively independent and self-evolving natural complex³⁰⁻³¹. “ This is not correct. Mountains are usually not considered „independent “ and „self-evolving “, i.e. disconnected from surrounding areas (unlike true islands). There is quite some literature on the similarities, but more importantly differences, between island and mountain systems (e.g. see Mendez-Castro et al. 2021, Itescu 2018, Flantua et al. 2020). This statement needs to be re-phrased accordingly.

Response: Thanks for this criticism. We re-phrased this sentence as “Here, we apply the term “flora” to refer to the sum of all angiosperm families, genera, and species growing on a specific mountain or in a well-delimited area^{34,35}, which is a relatively independent biogeographical unit”.

Commented [A18]: Unclear what this is supposed to mean exactly. Delimited by researchers for their analyses, or by natural barriers?

Line 114: “(median = 1,462.0) “ – not entirely clear what this number represents.

Response: We made clear the means of this number is species richness. See L125“.....angiosperms in granitic (species number median= 1,456) and karst-granitic.....”

Also, again, I would argue against the flora of a mountain being a relatively independent biogeographical unit. First, because related lineages are shared between mountains, and second, because most biogeographers, in terms of terminology, would not consider the flora of a mountain as a “biogeographical unit”.

Line 122-136: “Based on the full model, longitude, elevational difference between the highest and lowest points of a flora (elevdiff hereafter), and mean temperature of the coldest quarter (TCQ hereafter) had positive effects on species richness (Extended Data Table 3). “ In this sentence, and the following ones, it should be made clear whether larger/smaller or higher/lower values have a

positive or negative effect. This seems not evident from the text, as currently written. E.g. do the authors mean “a larger elevational difference “? Or later in line 128, high or low precipitation seasonality and high or low mean temperature? As a note aside, net primary productivity (NPP) might have been a better predictor for species richness.

Response: Thanks for this criticism. We made a change to that, please see L138-L141 “Based on the full model, larger longitude, larger elevation difference between the highest and lowest points of a flora (elevation hereafter), high precipitation of wettest month (PWM hereafter) and high mean temperature of the coldest quarter (TCQ hereafter) had positive effects on species richness (Extended Data Table 3).” The others sentence had been revised accordingly.

Net primary productivity (NPP) is closely related to the availability of energy and water in the environment. Our model had included 19 climate factors, so that NPP is not included in our analysis.

Line 142 f.: What about the role of dispersal from other mountain systems? (see e.g. Jiménez-Alfaro et al. 2020, DOI: 10.1111/geb.13274; Ding et al. 2020, Science)

Response: Species dispersal is one of the important sources of mountain species diversity. In the discussion section, we specifically discuss the effects of dispersal. Please see L317-L327 “Dispersal also contributes to the mountain diversity^{2,13,44}. A well-known example is climatic fluctuations during the Quaternary ice age drove changes in the distribution of species around the globe^{51,65}. The spatial configuration of mountain range affect their functional connectivity, influencing species dispersal^{66,67}. In fact, dispersal more easily occurs at initial stage of landform develop mountain, as the slope of mountain still gentle and geographical barrier still relatively low¹³. Such as the plants dispersal events in Qinghai-Tibet Plateau (QTP, low barrier) is obvious higher than that in Hengduan Mountains (high barrier)⁴⁴. Furthermore, the effects of dispersal on species diversity in mountainous areas mainly occurred at lowland, and is limited at highlands where has more local endemic species^{6,30}. This means that dispersal has weak influence on the unique composition of mountain flora of different landforms, especially considering that landform have a significant filtering effect on the species.”

Commented [A19]: Avoid this term.

Commented [A20]: I don't understand this sentence. As outlined previously, it is exactly the other way round. Dispersal and establishment was easier in the Hengduan mountains at times of climate change as they have a N-S orientation, enabling plants to change their latitudinal distribution.

Commented [A21]: Reading this, I am not sure the authors have read the papers I had suggested to consult, and which they cited in the paper already before (e.g. Ding et al.).

Line 149-150: “This is consistent with some fossil evidence, the fossil flora discovered in southwestern China indicate local karst vegetation may have existed since the early Oligocene37-38. “ Is this fossil evidence from the same species or genera as the living ones, and thus specifically relevant?

Response: Yes. The fossil data evidence confirmed the Oligocene flora in southeastern Yunnan is closely similar to the current karst flora. In Dong et al. (2018), a fossil species of *Burretiodendron* is described, which is a endemic genus of karst flora. In another paper (Huang et al., 2018), a species of genus *Ficus* which discovered in Wenshan basin is very similar to living species *Ficus trivialis*. The present species *Ficus trivialis* grows only in limestone ridge scrub, this suggests that limestone shrub vegetation was present during the Oligocene.

Line 151 f.: “In the full model, orogenic, latitude, temperature annual range (TAR), TCQ is negatively correlated to PDI,…” Again here, as already mentioned for lines 122-136, the sentence does not provide as much information as it could: high range? Low range?

Response: Thank you for pointing this out. We revised this sentence to avoid ambiguity, see L169-171 “In the full model, orogenic, high latitude, high temperature annual range (TAR), and high TCQ is negatively correlated to PDI, and high TWQ are the only variables significantly positively

correlated to PDI (Figure 4b; Extended Data Table 5)". Other parts of the revised manuscript have also been modified accordingly.

Line 153-157: "may have recently undergone higher rates of evolution related to orogeny" " " , while higher TWQ may have led to higher rates of extinction in mountains of sedimentary rock
" Note: This could be tested (speciation rates, extinction rates), and might be worth testing in the future. Mountain biogeography studies often investigate speciation rates and extinction rates, and net diversification rates, respectively, from dated phylogenies, albeit with dense taxonomic sampling (unlike the study here). Investigation of floristic differences of karst versus non-karst, calcareous versus silicate, is a well-researched topic of geobotany (e.g. in the Alps).

Response: We appreciate the review #3's suggestion, which would make for a nice paper in the future. However, that research is different to what we attempted here. In this research, to our knowledge, the first attempt to investigate the relationship between mountain assemble of floras and the types of landforms in which they occur. Our result highlight the floristic assembly in mountains is affect by the bedrock-constrained developmental processes of landforms. As predicted in our hypothesis, nich evolution can promote the spread of species among different landform flora (such as karst landform flora to non-karst landform floras). This could be test in the future.

Line 172: "Landforms should originate earlier than the floras that inhabit them due to the long times that floras require for assembly. " "earlier" - in concert, why earlier? Species can immigrate into a region (dispersal from e.g. other mountain regions or other habitats), and thus can be older than the landform they inhabit nowadays.

Response: Thanks for the criticism. Although, most species which occurs in a mountain is always later than the landform develop process (or evolution with the landform process). Some ancient species could migrate into the mountains by chance.

We revised this inappropriate sentence. Please see line 190-191 "To estimate age pattern of floras between landforms, we measured the mean divergence times of all species (MDT), mean divergence times of the youngest 25% of species (MDT.youngest),"

Line 173-174: "we measured the mean divergence times of all species " Divergence times can only be measure reliably when (almost) all species of a genus are included in a phylogeny, otherwise, ages may be considered not reliable/meaningful. The use of single phylogenies, targeting specific groups, would be needed for that.

Response: Thanks for this advise. Because the aim of this study is to reveal the general pattern of landform influence on the formation of mountain flora, rather than focus the time of each species occurs in a mountain. Thus, the species ages is inferred from a pruned phylogeny based on GBOTB.extended.LCVP.tre (a dated mega-phylogeny). In large-scale biodiversity and phylogenetic analyses, sources of noise in divergence time estimation are to be expected, but they did not affect the reliability of the results.

Here, we did an analysis to test the divergence time consistency between the two results of species ages eactract by **pruned phylogeny** (used in our study) and the ages eactract from **unpruned phylogeny** (GBOTB.extended.LCVP.tre, a global plant phylogeny). In total 17,576 species included in this study, of which 8,663 species were included in "GBOTB.extended.LCVP.tre" and 8713 species

Commented [A22]: I don't understand this sentence. How could "niche evolution" possibly promote "the spread" (dispersal?) of species? Which niche evolution do the authors mean here, on which taxonomic level? This needs to be clarified. As currently phrased, this sentence does not make sense and I don't get the meaning.

not occurs in GBOTB.extended.LCVP.tre. We reconstruct the phylogeny of 8863 species which occurs in GBOTB.extended.LCVP.tre, based on the same method used in this study. Then, extract the divergence time of these 8863 species in new gathered pruned tree and in GBOTB.extended.LCVP.tre (a global plant phylogeny, unpruned tree), respectively. Finally, we get four group of species divergence time data, which are:

“pruned.add” age (used in our study)=included 17,576 species, of which ages of 8863 species are extract from the pruned phylogeny tree, and ages of others 8713 species(add) are added by BLADJ method (Webb et al., 2011)

“unpruned.add” age =included 17,576 species, of which ages of 8863 species extract from the unpruned phylogeny, and ages of others 8713 species(add) are added by BLADJ method (Webb et al., 2011)

“pruned” age=included 8863 species, age extract from the pruned phylogeny tree

“unpruned” age=included 8863 species, age extract from the unpruned phylogeny

We did the regression analyse on MDT of four different age datasets and find the close correlation of MDT estimate from “pruned” phylogeny with “unpruned” phylogeny (R^2 of MDT.all is 0.63), especially the younger species ($R^2=0.84$) (Supplemental fig 1). These could also be observed in linear regression fitting result of “unpruned.add” VS “pruned.add” (Supplemental fig 2), “unpruned” VS “pruned.add” (Supplemental fig3), “unpruned” VS “unpruned.add” (Supplemental fig 4). It is particularly to point out that the results of MDT.all and MDT.oldest is consistent (see Supplemental fig 1 and Supplemental fig 3), although “pruned.add” datasets included others 8713 species. This result suggest that ages of 8713 species not occurs in GBOTB.extended.LCVP.tre which inferred in “pruned.add” datasets may be reasonable estimates.

Supplemental fig 1. unprunedVS pruned

Supplemental fig 2. unpruned.add VS pruned.add

Supplemental fig 3. unpruned VS pruned.add

Supplemental fig 4. unpruned VS unpruned.add

Nevertheless, the impact of limited sampling in a regional phylogeny on divergence times still remains to be assessed directly in the future. The species age of “pruned” phylogeny is biased (usually larger) than the “unpruned” phylogeny. While, in large-scale biodiversity and phylogenetic analyses, sources of noise in divergence time estimation are to be expected, but they did not affect the reliability of the results (Lu et al. 2019). Just as the species ages inferred in our phylogeny (a pruned tree of GBOTB.extended.LCVP.tre) did not affect the MDT patterns between mountain in different landform.

Commented [A23]: Repetition of the above.

Although the estimates species ages “pruned” phylogeny is larger than “unpruned” phylogeny. We believe that our dated mega-phylogeny tree is suitable for this study. Because the aim of this study is to reveal the general pattern of landform influence on the formation of mountain flora, rather than focus the time of each species occurs in a mountain. In fact, the latter is difficult to achieve.

Line 188: “and this may be caused by the stronger influence of glacial periods “ – What was the effect of these glacial periods in the investigated mountain systems? Which were impacted by glaciations, which were affected by colder/drier conditions only? The effect of the glacial periods was much dependent on the availability of water (precipitation; monsoons) and thus regionally very different.

Response: This sentence is to explain the effects of glaciations on the northern mountains in China. The Quaternary glaciation had little influence on the mountain floras in southern China (south of Yangtze valley). Because there are many east-west mountain range that can reduce the impact of ice age cooling (many species surviving in mountain refuges). However, the mountains in the north of Yangtze valley were strongly influenced by the Quaternary glacial period (many species extinct).

Commented [A24]: As I pointed out previously, the opposite would be expected. Survival on mountains with N-S orientation is easier during glacials than on those of W-E orientation (e.g. compare Himalayas versus Hengduan Mts.).

Line 214: “Our results are consistent with prior studies9,14…” Reference to MGH papers missing here.

Response: We added the citation of reference to MGH (Muellner-Riehl, 2019) herein.

Line 215: “closely related to geomorphic processes “ – strictly speaking, this study does not investigate processes as such, but patterns.

Response: Thanks for this comments. Here we added a fig (Please see “Extended Data Fig. 6”) of angiosperm species number of different landforms during specified geological times. The results shows the species accumulation rate increased rapidly after the Miocene (especially in karst, granitic, and karst-granitic landform). Because the Miocene was an important stage when the east Asia monsoon intensified and subsequently accelerated landform processes in China.

Extended Data Fig. 6| Number of angiosperm species of different landforms during specified geological times. a, all species; b, species occurs in granitic landform; c, species occurs in karst-granitic landform; d, species occurs in karst landform; e, species occurs in Danxia landform; f, species occurs in desert landform.

Line 215: “For example, most floras of China, including both mountains and lowland floras, diverged during the Miocene11 when the East Asian monsoon began to prevail..: “ – not clear what is meant by “floras diverged “, see my previous comments on this issue of dating “floras “. Be more specific about “when the East Asian monsoon began to prevail “ – when was the presumed onset, when intensification, when were current levels reached? There is plenty of literature available on this topic (e.g. see publications by Dupont-Nivet, and others).

Response: The formation of the East Asian monsoon is closely related to the uplift of the Qinghai-Tibet Plateau. Although, the process of lifting the Tibetan plateau and when East Asian monsoon start, remains controversial. It is widely accepted that the East Asian monsoon intensified rapidly during the Miocene (Spicer, 2017; 10.1016/j.pld.2017.09.001),Li et al., 2021(10.1126/sciadv.abc7). This is a period of rapid species divergence, such as *Rhododendron* (Xia et al., 2021;

10.1093/molbev/msab314), *Begonia* sect. *Coelocentrum*, (Chung et al., 2014; <http://www.as-botanicalstudies.com/content/55/1/1>), Chen et al. 2018 (10.1093/nsr/nwx156). The period of rapid landform process in modern China coincides with the period of rapid plant species divergence.

We rewrite this sentence, see L235-L238 “For example, most floras of China, including both mountains and lowland floras, diverged during the Miocene when the East Asian monsoon intensified^{38,52}. Accordingly, this period was also in high rates of development of modern karst, Danxia, and granitic landforms in China⁵³”.

Line 218 following: “At the intercontinental scale, the ages of floras are also largely consistent with regional developmental processes of landforms. For example, the floras of eastern Asia are typically younger than those of South Africa and Australia, where the landform processes have been very long, but older than those of the Andes and Amazonia, where landform processes have occurred more recently⁴⁸. Thus, the relationship of floristic assembly of mountains to the type of landform and landform developmental process seems to be global in scope, at least for angiosperms. Thus, landform types may make a suitable indicator for explore floristic assembly of mountains, and such an indicator is needed by the scientific community to support biogeographic and other research on mountainous regions²⁹. “ This is a rather simplified statement, needs more elaboration and detail, and also much more consideration of literature that exists for all of these regions. E.g. Do the authors here talk about mountain floras specifically, and if so, which? Different mountain systems on these continents have different ages and history, different orientation (Elsen and Tingley 2015, Nat. Clim. Change), and were also differently impacted by the Last Glacial Maximum (LGM). „explore floristic assembly of mountains “ – the floristic assembly is not explored here per se, but only the patterns. But the assembly can actually be investigated, see Ding et al. 2020 (who you also cite).

Response: Here, we're not focus on specific mountains. We want to point out that the age of local flora in different regions is related on how long the local landform process, even between continents. Chen et al. (2018), had did a systematic comparison of flora age between Andes, Amazonia, California, Australia, South Africa, and East Asia. The Australian and South African foras have older median ages, although the Andes, Amazonian and Californian foras have younger median ages. The formation and evolution of these foras were closely linked to local or global environmental changes. Such as, the Geological history and stratigraphic structure of Australia and South Africa is fairly stable than East Asia (which had experienced Yenshan orogeny at Late Triassic to Cretaceous, Himalayan orogeny at Cenozoic) and Andes (Alpine orogeny at late Cretaceous to Cenozoic). As our view is steady and slow geological processes developed relatively old floras. This is especially obvious in the mountain area.

The second question is how to explored the flora assemble process. The assemble of flora we mentioned is not only a unit, but also an assemble mode of different species, which is related to some lithology types and landform types. Thus, in this study we trace the process of flora assembly by comparison the flora characteristics between mountain landforms. Because the lithosphere cycle and the landform process are well documented. In this study, we drawing a view of flora differences between landform shifts. We believe that if we can grasp the patterns and causes of assemble of flora, it will be possible to further understand how different species and individuals may gather in one aggregation and one flora.

Commented [A25]: But has this been actually rigorously tested, or there is just temporal coincidence?

Commented [A26]: “floras” don’t “diverge”. I would argue against mixing vegetational terms and evolutionary terms. Divergence in an evolutionary sense would not apply to “floras”.

The sections also needs some general language improvement.

Commented [A27]: Some of what is explained by the authors in the following text may be viewed useful for readers to better understand the reasoning behind this study. If possible, I suggest to incorporate some of the aspects in the intro and discussion.

Commented [A28]: Why “although”? I don’t understand.

Commented [A29]: Different parts of the Andes are of vastly different age, “median ages” may not be a meaningful measure for all mountains.

Commented [A30]: I am not sure how “unit” here would refer to the biogeographic units mentioned in the ms. This may be worth of clarification.

Line 230: results for desert landforms – this may not come as a surprise, as only relatively few plant genera/families are especially adapted to desert conditions.

Response: Yes, the phylogenetic relationship of species in the desert landform flora are closely related. That's mainly due to habitat filtering, as the desert landform development is mainly drought-dominated.

Line 271 following: Here again, reference to the MGH is missing. “Specifically, under this hypothesis, the ages of mountain floras are determined by the time when erosion of strata begins,...” “This disregards the effect of immigration on floristic assembly in mountains which can account for most of the species richness on some mountains (compare Ding et al. 2020).

Response: We agree with this criticism. In the new MS, we added a paragraph to discussing the effect of immigration on floristic assembly in mountains. Please see L317-327. “Dispersal also contributes to the mountain diversity^{2,13,44}. A well-known example is climatic fluctuations during the Quaternary ice age drove changes in the distribution of species around the globe^{51,65}. The spatial configuration of mountain range affect their functional connectivity, influencing species dispersal^{66,67}. In fact, dispersal more easily occurs at initial stage of landform develop mountain, as the slope of mountain still gentle and geographical barrier still relatively low¹³. [Such as the plants dispersal events in Qinghai-Tibet Plateau (QTP, low barrier) is obvious higher than that in Hengduan Mountains (high barrier)⁴⁴. Furthermore, the effects of dispersal on species diversity in mountainous areas mainly occurred at lowland, and is limited at highlands where has more local endemic species^{6,30}. This means that dispersal has weak influence on the unique composition of mountain flora of different landforms, especially considering that landform have a significant filtering effect on the species.”]

Commented [A31]: See my other comments on this text in the manuscript.

Methods

Line 405: “we reconciled taxonomy to The Plant List v. 1.1 “. The Plant List (TPL) is an outdated resource, thus other resources should be used instead. For plants, four global authoritative taxonomic lists exist: World Checklist of Vascular Plants (WCVP), the World Flora Online (WFO, an explicit community effort to tackle the increasing wealth of taxonomic information as successor of The Plant List), the Leipzig Catalogue of Vascular Plants (LCVP), and World Plants (WP; both works of dedicated individuals). These four lists each provide a global list of plant names, but differ considerably in size and likely in completeness and accuracy across taxa and geographic regions. The WFO Plant List (www.wfoplantlist.org) replaces the now long outdated Plant List.

Response: Thanks for this important comment. The reason fo our previous checklist reconciled to The Plant List is that the backbone phylogeny tree in V.PhyloMaker (Jin & Qian, 2019). A updated version, V.PhyloMaker2, is available in 2022 (Jin & Qian, 2022;10.1016/j.pld.2022.05.005). With V.PhyloMaker2, one can generate a phylogenetic tree for vascular plants based on one of three different botanical nomenclature systems (TPL, LCVP, and WP).

We accept reviewer’s suggestion and standardized our checklist to match the database LCVP. See L364-L367 “From our initial checklists, we excluded all nonnative species, and we reconciled taxonomy to Leipzig Catalogue of Vascular Plants (LCVP) using the R4.1.0 (<http://www.r-project.org/>) package, *lcvplants*⁷⁴, with infraspecific taxa combined under their respective species⁴⁷.”

Accordingly, we reconstructed the phylogeny for the species in this study. See L375-L382 “With the recently published, dated megaphylogeny tree “*GBOTB.extended.LCVP.tre*”³⁷, as a backbone,

we generated the phylogeny for the species in this study using the R package 'V.PhyloMaker2'³⁷. Of the 2,585 genera and 17,576 species in this study, 2,349 genera and 8,663 species were included in "GBOTB.extended.LCVP.tre". For the 236 genera missing, we treated each as sister its most closely related genus in "GBOTB.extended.LCVP.tre" based on megaphylogenies within other references⁵²⁻⁷⁶. For species in this study that were absent from "GBOTB.extended.LCVP.tre", we added them to their respective genera using Phylomatic and generated their branch lengths with BLADJ⁷⁷ implemented in the R package V.PhyloMaker2³⁷. "

Line 462: "studies have shown that the species ages within floras are quite different ". Strictly speaking, species ages themselves are not investigated here

Response: Species ages in this study are fitted by using the dated megaphylogeny tree "GBOTB.extended.LCVP.tre" , which is a phylogenetic tree of fossil dating.

Commented [A32]: See my comments on this matter above and in the ms text.

Data availability

Line 548: "All other additional data are available from the corresponding author upon reasonable request." What is considered "reasonable"? Data should be made available, whenever possible, according to the FAIR principles (findability, accessibility, interoperability, and reusability).

Response: All original mountain floras data used in this study have been published and are accessible to readers from the cited sources. A standardized spatial distribution data of mountain floras are provided with paper. Please see "Data availability" in L514-518.

Response to reviewers' comments

Thank you for taking the time to review our manuscript entitled “Landform and lithosphere development drive the assembly of mountain floras”. We appreciate your insightful comments and suggestions revised by Reviewer #2 (mainly data analysis) and Reviewer #3 (mainly review and comparison of previous studies), which have been immensely helpful. We have carefully studied the reviewers' comments and made corresponding modifications. The manuscript has also been carefully grammatically revised. The corrections are indicated in red in the manuscript (clean version), and we also submitted a version to track all the revisions. The replies to reviewers' comments are listed as follows.

In order to make it easier for reviewers to read, we will sort the questions mentioned in the main text and responses in order of Q (Q1, Q2...), and then reply separately.

REVIEWER COMMENTS AND OUR RESPONSE

Reviewer #2 (Remarks to the Author):

Q1:

I am re-reviewing this manuscript and I was R2 in the last round. The manuscript is much improved but several of the responses from the authors are not very satisfying. I would particularly like the authors to re-consider the issues below -- I try to explain them better this time.

The authors have kindly pointed to their extended table 10 in response to my request for testing interactions between landforms and other environmental variables. As far as I can tell, Table 10 only contains additive models and cannot identify interactive effects. For example, rather than $Y \sim \text{landform} + \text{climate}$, it needs to be $Y \sim \text{landform} * \text{climate}$, or better yet, $Y \sim \text{climate} + (\text{climate} | \text{landform})$ in a multi-level framework.

Response: We are sorry that some of our responses did not address your concerns, and thank you for your kind clarification here. Following the excellent suggestion by the reviewer, we investigated the interaction of landforms and other variables based on $Y \sim \text{climate} + (\text{climate} | \text{landform})$ in a multi-level framework. Our revisions in the manuscript method part as lines 569-573: *“However, we also extended the “landform model” to include all others variables to assess the effect of climate on species diversity (i.e., “full model”). Determinants of SR might change with landform type, and we therefore test for interactions between landform and others predictor variables (only shown significant variables in full model).”*

The results showed that the species richness (SR) affects the interactions of landform and TCQ, Annual Precipitation (PREC), and PCQ (see “Extended Data Table 10”). Accordingly, we add landform: TCQ, landform: PREC, and landform: P_{var} as variables to the full model. The best full model is shown in the manuscript, see “Extended Data Table 3”. Based on the new full model results, when considering the interaction effects between landforms and environmental variables, the high precipitation of wettest month (PWM), and high precipitation of coldest quarter (PCQ) are no longer detected as significant predictor. This could be expected, as the most important outcome of the interactions between landforms and environmental variables is about water availability in mountain systems. The models that essentially accounted for the interaction effects did not differ from the previous one, and both models explained more than 60% (63.7% vs 62%) of the variation of species

richness. We also provided the “*Extended Data Fig. 9*” to show the influence of interactions between landforms and other environmental variables on species diversity in montane areas.

In the SR model result part, we add lines 144-150 “*The full model explains 62.8% of the observed deviance of SR in the GLM and 63.7% in the SAR, and strong interaction effects between landform and mean temperature of the coldest quarter (TCQ) were detected (Extended Data Table 3). We also found weak interactions between the landform effects and annual precipitation (PREC), as well as precipitation of coldest quarter (PCQ) (Extended Data Table 10).*”, also lines 165-168 “*Interestingly, TWQ is not a significant predictor and explains only 0.9% of the variation of SR (Extended Data Fig. 9g). The observed significant negative correlation between SR and high TWQ (Extended Data Table 3) may be the result of incorporating landform effects into the regression models (Extended Data Fig. 9i)*”. For details, please see the result section of revised manuscript “***Landforms effects on species richness and phylogenetic diversity***” in lines 138-172.

We also find the phylogenetic structure and MDT are affected by interactive effects. For example, the MDT is affected by the interactions between landforms and log(area), long, elevmid, bio2, bio8, bio10, and bio11. However, if all these interaction effects are included in the full model, the prediction model becomes too complex for interpretation. As such, in order to maintain the simplicity of our prediction model, we do not consider the interaction effects in the full model of NRI, NTI, PDI, MDT, MDT_{.oldest}, and MDT_{.youngest} in the main text. Nevertheless, as supplementary results, we provided figures to show the impact of interactions between landforms and other environmental variables (which were detected as significant predictors in the full models) on NRI, NTI, PDI, MDT, MDT_{.oldest}, and MDT_{.youngest}. See “*Extended Data Fig. 10-15*”.

Extended Data Table 10|Results of the interactions between landform and others variables on the species richness (SR).

Variables	GLM			SAR		
	Coefficient	SE	t	Coefficient	SE	z
TWQ	0.206	0.187	1.106ns	-1.072	0.229	-4.691***
TCQ	0.760	0.153	4.956***	0.668	0.179	3.733***
PREC	0.957	0.132	7.269***	0.968	0.151	6.428***
PCQ	0.498	0.117	4.275***	0.466	0.150	3.102***
Landform*TCQ (Intercept)	7.747	1.171	6.616***	8.141	1.130	7.2***
Desert	-0.513	1.183	-0.434ns	-0.859	1.142	-0.752ns
Granitic	-0.935	1.179	-0.793ns	-1.325	1.139	-1.163ns
Karst	-0.212	1.251	-0.169ns	-0.634	1.206	-0.526ns
Karst-Gr	1.020	1.324	0.771ns	0.399	1.283	0.311ns
TCQ	-1.113	1.620	-0.687ns	-1.724	1.573	-1.096ns
Desert:TCQ	-5.291	2.025	-2.613ns	-4.812	1.920	-2.506*
Granitic:TCQ	1.767	1.633	1.082*	2.367	1.587	1.492ns
Karst:TCQ	0.488	1.736	0.281ns	1.152	1.683	0.685ns
Karst-Gr:TCQ	-1.384	1.925	-0.719ns	-0.329	1.879	-0.175ns
Landform*PREC (Intercept)	7.765	0.997	6.784***	6.320	0.907	6.948***
Desert	-0.702	1.018	-0.689ns	-0.584	0.942	-0.620ns
Granitic	0.0153	1.006	0.015ns	0.523	0.918	0.570ns
Karst	-0.607	1.195	-0.509ns	-0.449	1.084	-0.414ns

Karst-Gr	0.517	1.088	0.475ns	1.008	0.980	1.029ns
PREC	0.274	1.505	0.182ns	0.888	1.371	0.648ns
Desert:PREC	3.950	2.368	1.668ns	6.115	2.372	2.578**
Granitic:PREC	0.408	1.516	0.269ns	-0.268	1.380	-0.194ns
Karst:PREC	1.497	1.941	0.771ns	1.381	1.766	0.782ns
Karst-Gr:PREC	-0.240	1.706	-0.140ns	-0.942	1.538	-0.612ns
Landform*PCQ (Intercept)	6.821	0.485	14.071***	6.643	0.474	14.005***
Desert	-0.765	0.511	-1.495ns	-0.563	0.501	-1.124ns
Granitic	0.231	0.491	0.471ns	0.391	0.480	0.816ns
Karst	0.148	0.572	0.259ns	0.292	0.554	0.527ns
Karst-Gr	0.690	0.516	1.338ns	0.899	0.499	1.801ns
PCQ	0.192	0.730	0.263ns	0.430	0.711	0.605ns
Desert:PCQ	13.196	4.178	3.158**	14.191	4.023	3.528***
Granitic:PCQ	0.159	0.742	0.214ns	-0.040	0.718	-0.055ns
Karst:PCQ	0.261	1.248	0.209ns	0.057	1.195	0.048ns
Karst-Gr:PCQ	-0.968	0.918	-1.055ns	-1.298	0.879	-1.480ns

Extended Data Fig. 9 Interaction effects between landform and environmental variables on species richness (SR).

Q2:

In terms of distinguishing the other categories, I think the upper half of the extended Table 3 only shows that most categories are significantly different from Danxia, which is good to know but it

still does not distinguish the other three most similar ones: Karst, Karst-Gr and Granitic. It might be best to do pair-wise comparison (t test or Wicoxon test) for all possible pairs.

Response: We accept this advice and did the pair-wise comparison (t-test) for all possible pairs. The result is shown in “Fig. 3”. The differences in species richness, PDI, NRI, and MDT between the landforms can be clearly seen in “Fig. 3” below.

The result showed that, in general, the Danxia and desert landform are significantly different from Granitic, Karst, and Karst-Gr landform. The confidence intervals between the latter three landforms usually overlap. We suspect this may be influenced by spatio-temporal correlation of landform development, as Karst-Gr is an intermediate stage of Karst and Granitic.

Fig. 3 | The differences in species richness, phylogenetic structure, and age of floras among types of landforms. P value of T-test result between each pairs of landforms were showed above the black line: **p<.001; ***p<.001; **p<.01; *p<.05; ns = not significant.**

Q3:

In terms of the phylogenetic analyses, they authors have apparently misunderstood my point which is NOT about the effect of phylogenetic completeness. I have no problem with an incomplete tree as long as the sampling is not known to be bias in a systematic way. Rather, I am asking for a clear explanation of the reasoning behind the simulation, in terms of what it reflects when random samples are drawn from a phylogeny of only the species in the dataset rather than in the region. I

do not see why the null model demanded that the species pool should be consistent in the tree and the distribution database unless it is required by the specific package you use (I do not think it is based on the package documentation). It is possible that if you use a global tree or a regional tree, the patterns remain quite similar if the effect of contingency is not too strong; strong contingency plus large difference in species richness among regions could really bias your simulations. Therefore, I suggest a check on that to ensure the robustness of your findings.

Response: Thanks for further clarifying this question. The reviewer mentioned “I do not see why the null model demanded that the species pool should be consistent in the tree and the distribution database?” And why doesn't the author use a global database or a more complete regional database? While the reviewer raised interesting points to examine, we will highlight the main reasons for taking our approach by providing a more detailed explanation below.

At first, we would like to explain the null model used in our study. As supplementary, we provide a brief introduction of the packages and the null model used to calculate the PDI, NRI and NTI in the methods.

Please see lines 466-467 “*To calculate PDI, we used ‘PhyloMeasures’^{Error! Reference source not found.} in R, in which the null model was set as “uniform” and the following typical algorithm was implemented: ”,*

and line 482-485 “*In these equations, $MPD_{observed}$ and $MNTD_{observed}$ are the observed MPD and MNTD, $MPD_{randomized}$ and $MNTD_{random}$ are the expected (i.e., average) MPD and MNTD of the randomized assemblages^{Error! Reference source not found.}, which were calculated based on the null model “uniform” in the R packages ‘PhyloMeasures’^{Error! Reference source not found.}, ”.*

Here, the null model "uniform" considers samples with equal (uniform) probability among all possible tip samples of the same richness. This was based on the assumption “...species have been able to disperse (possibly over many generations) anywhere...”, which was proposed by Webb in 2000 (DOI:10.1086/303378). The null model in this study fully takes into account the contingency of species which occurs in a mountain flora. Species richness is only related to phylogenetic diversity (such as PD), but the species richness effects on phylogenetic structure (this study’s focus) can be excluded. In the null model, 100 or 1000 species will be randomly extracted from the phylogenetic tree to calculate the $MPD_{randomized}$ and $MNTD_{random}$ if a mountain contains only 100 or 1000 species (MPD, and MNTD). In other words, phylogenetic structure (PDI, NRI, and NTI) is mainly related to the distance of the phylogenetic relationship of species in the mountain, rather than the species richness per se so the pool size likely does not matter as much (see “*Extended Data Fig. 3 | Regression analyses between species richness and PD, PDI, NRI, NTI.*”).

Second, based on the technical requirements (this is based on the null model assumption) of the software package, flora data is extracted from the mountain species database to reflect the phylogenetic structure of the flora. Thus, a global or regional tree is not needed here. In fact, as we know there isn’t currently an approach that is calculates a regional phylogenetic structure based on a global tree.

In our study, the calculation of phylogenetic diversity and phylogenetic structure were performed by R packages “PhyloMeasures” (Tsirogianis & Sandel, 2016; doi: 10.1111/ecog.01814). This package is widely used in studying the community phylogenetic structure of floras. The “PhyloMeasures” requires the species pool to be consistent with the phylogenetic tree and distribution database. Otherwise the functions (“mpd.query”, “mntd.query”

and “pd.query”) in “PhyloMeasures” would not work. See the screenshot below for the warning in R.

```
> mpd.std<-mpd.query(tree,data, null.model="uniform", T)

Warning: the input matrix has fewer columns than the number of species in the tree.

Error in mpd.query.uniform(tree, matrix, standardize) :
  One of the species names in input the matrix was not found in the tree (Staurogyne_rivularis)
> |
```

Q4:

With regard to the "age" of species, I appreciate the authors' effort to provide additional information by comparing the pruned and unpruned trees.

However, my point is more about the concept of species or assemblage age -- it is a not the real age and can only be used for representing the patterns of age when extinction is random; in contrast, a comparative analysis can be problematic if extinction was not random with regard to the factors that are being examined. Therefore, I suggest rephrasing it in terms of species distinctiveness, so that an assemblage with low MDT contains more closely related species and infer mechanisms from there.

Response: Thank you for clarifying your concerns, and we appreciate your suggestion to use the term “species distinctiveness” to replace “flora age of species age”. Here, we're unsure the meaning of the term “species distinctiveness”. According to our understanding, we assumed the reviewer's proposed ‘species distinctiveness’ to be the evolutionary distinctiveness of species as proposed by Isaac et al., (2007, doi:10.1371/journal.pone.0000296). We suspect there are no differences between “species distinctiveness” and species age, because both of them are based on the in-clade in the phylogenetic tree. Besides, species age (or species divergence time) is more well known than “species distinctiveness” in evolutionary biology and biogeography.

On the other hand, we accept that the concept of “flora age” is problematic. The complexity assembly history of flora makes it impossible to interpret the real age of flora. In contrast, the age of the species, which estimate based on the molecular clock hypothesis (Ho, S., 2008; The Molecular Clock and Estimating Species Divergence) and fossil calibrated, is more reliable. In fact, the inference of floristic history is often based on the estimated divergence time (or stem age) of species which occurs in the flora (such as Dagallier et al., 2020, doi: 10.1111/nph.16293; Qian & Deng, 2022, doi: 10.1111/jse.12856; and Chen et al., 2018, DOI: 10.1093/nsr/nwx156).

In our study, we used the mean divergence times (MDT) proposed by Lu et al.(2018) to represent the “flora age”. As such, the MDT reflected the age composition of species within a flora. A flora with larger MDT has more ancient species and hence is expected to have older floristic ages. Overall, we believe that our use of MDT to measure the species age structure of flora is consistent with the existing literature and therefore is appropriate.

Nevertheless, in the revised manuscript we removed the use of “flora age”, and replaced it with “species age structure”. See lines 208-211 *“Landform effects on species age structure of floras The flora assembly history differs between mountain flora, which can be reflected in both species composition and species age structure. To estimate such species distinctiveness between landforms,.....”*

Reviewer #3 (Remarks to the Author):

Review of the revised manuscript “Landform and lithosphere development drive the assembly of mountain floras in China” by Zhao Wan-Yi et al.

Q1:

I appreciate the efforts by the authors in their attempt to improve the quality of their manuscript. While many, if not most, of my suggestions have been followed, some aspects have not yet been fully addressed and need further attention. Among those are adequate consideration of relevant literature by other authors, importantly with respect to hypotheses and frameworks, especially in the introduction and discussion. I have provided comments and suggestions in the rebuttal and manuscript text files and hope these will be useful to the authors. In addition, I found that especially the newly drafted texts (but not only exclusively those) will require a careful read in terms of use of the English language and terminology to be deemed acceptable.

Response: We accept the reviewer#3's suggestion to further consider the relevant literature for our introduction and discussion, particularly for the section on MGH, geodiversity, and adaptive radiation. In response to the questions raised by the reviewer, we have made careful changes to the entire manuscript. The manuscript has also been carefully grammatically revised. We listed the major responses below, and for more detailed edits, please see the revised manuscript (all changes are marked). The comments on the manuscript are shown as “**Line ??**”, and the comments raised in previous point-by-point response document are mark by “**(Pre-response)**”. The related comments are put together to avoid repeated response.

Q2:

Comment: In addition to my comments included here in this document, I have also provided comments to the rebuttal letter.

I hope my comments here as well as those in the rebuttal will be helpful for the authors to improve their manuscript further.

Response: Thanks for these helpful comments.

Q3:

Line48: I suggest to add studies that have a global scope. E.g. your current reference 16 for plants.

Response: We accept the reviewer’s comment to include reference 16, and also Rahbek et al.(2019) “Building mountain biodiversity: Geological and evolutionary processes”, in which the authors provided an overview of the proposed biogeographical roles of mountains (cradles, barriers, reservoirs, museums, graves, etc.).

Q4:

Line53:

This paragraph needs improvement. The facts are not presented clearly enough, and the content does not yet sufficiently pay justice to the state-of-the-art in the field.

Response: We have further revised this paragraph. Please refer to the following paragraph.

A number of hypotheses have been proposed to explain both montane and global biodiversity, such as those pertaining to climate stability^{Error! Reference source not found.}, habitat heterogeneity^{Error!}

Reference source not found., Error! Reference source not found., and energetics Error! Reference source not found., Error! Reference source not found. . Along latitudinal gradients, current evidence suggests that biodiversity is affected by environmental energetics, in particular potential evapotranspiration (PET) and average annual temperature Error! Reference source not found. . At finer scale, plant alpha diversity in some extratropical mountain regions (such as Cape, East Australia) does not substantially differ from that in the tropics Error! Reference source not found. . Contemporary climatic regimes are also insufficient to explain the pantropical diversity disparity in Neotropical and Indomalayan moist forests Error! Reference source not found. . These results strongly suggest that montane species diversity is largely affected by habitat heterogeneity Error! Reference source not found. , or the so-called geodiversity Error! Reference source not found. Error! Reference source not found. . Furthermore, the evolutionary history of plant species also affects biodiversity Error! Reference source not found. Error! Reference source not found. . It is clear that an integrated framework for the prediction of montane biodiversity is necessary Error! Reference source not found. Error! Reference source not found. Error! Reference source not found. , should include includes ecological processes (e.g., survival, competition, and niche differentiation) Error! Reference source not found. , biological processes (e.g., species divergence and extinction) Error! Reference source not found. Error! Reference source not found. , and geological and lithologic processes (e.g., orogeny and rock formation) Error! Reference source not found. Error! Reference source not found. . A recent attempt at such a framework, the “mountain geobiodiversity hypothesis” (MGH), was first proposed to explain the biodiversity of the Tibeto-Himalayan region Error! Reference source not found. Error! Reference source not found. and then extended to explain the origin of montane plant diversity at a global scale Error! Reference source not found. . The MGH proposes that the evolution of montane biodiversity results from a combination of mountain-uplift, geodiversity evolution, and Neogene and Pleistocene climate changes Error! Reference source not found. Error! Reference source not found. .

Q5:

Line55-56: Unclear: which kind of “model” ? These are different papers of different scope and with important findings, but what and where in these can one find the “model”? As currently written, this remains dubious. I assume “model” is not the right word here. The sentence should be re-written.

Response: Yes. The model we are writing here does need to be modified. It should refer to the distribution pattern of biological diversity, the theories, hypotheses, or processes that measure this pattern. Here, the entire paragraph has been revised, referring to the reply in the previous sentence.

Q6:

Line59-60: A “theory” is not exactly the same as a “hypothesis”, I suggest to re-write this. A hypothesis can be tested. The MGH was tested for world-wide applicability by Muellner-Riehl et al. 2019 JBI. This should be mentioned here. I suggest to be more specific here – which aspect of “biodiversity”? The MGH refers not only to standing biodiversity levels, but also how biota and geology and climate evolved in concert, incl. speciation, extinction and dispersal, which all contributed to yield the current biotic assemblages. See my previous comment. The MGH was originally developed to explain high levels of biodiversity in the THR, and thus it is of relevance to the study here. But in addition, it was later tested on global mountain data. This does not become evident from the text here yet and needs to be added.

Response: We appreciate the reviewer's comments to the third paragraph of the previous manuscript.

We had carefully considered these comments and rewrote the entire paragraph. Please see lines 54-71:

A number of hypotheses have been proposed to explain both montane and global biodiversity, such as those pertaining to climate stability^{Error! Reference source not found.}, habitat heterogeneity^{Error! Reference source not found.}, and energetics^{Error! Reference source not found.}. Along latitudinal gradients, current evidence suggests that biodiversity is affected by environmental energetics, in particular potential evapotranspiration (PET) and average annual temperature^{Error! Reference source not found.}. At finer scale, plant alpha diversity in some extratropical mountain regions (such as Cape, East Australia) does not substantially differ from that in the tropics^{Error! Reference source not found.}. Contemporary climatic regimes are also insufficient to explain the pantropical diversity disparity in Neotropical and Indomalayan moist forests^{Error! Reference source not found.}. These results strongly suggest that montane species diversity is largely affected by habitat heterogeneity^{Error! Reference source not found.}, or the so-called geodiversity^{Error! Reference source not found.}. Furthermore, the evolutionary history of plant species also affects biodiversity^{Error! Reference source not found.}. It is clear that an integrated framework for the prediction of montane biodiversity is necessary^{Error! Reference source not found.}, should include includes ecological processes (e.g., survival, competition, and niche differentiation)^{Error! Reference source not found.}, biological processes (e.g., species divergence and extinction)^{Error! Reference source not found.}, and geological and lithologic processes (e.g., orogeny and rock formation)^{Error! Reference source not found.}. A recent attempt at such a framework, the "mountain geobiodiversity hypothesis" (MGH), was first proposed to explain the biodiversity of the Tibeto-Himalayan region^{Error! Reference source not found.} and then extended to explain the origin of montane plant diversity at a global scale^{Error! Reference source not found.}. The MGH proposes that the evolution of montane biodiversity results from a combination of mountain-uplift, geodiversity evolution, and Neogene and Pleistocene climate changes^{Error! Reference source not found.}.

Q7:

Line60-62: This sentence is unclear. Which "framework"? Before, the text talks about the MGH, then it says "Within these framework", which probably is supposed to link to the MGH, but the citations 12, 20-21 are not related to the MGH. The sentence needs adjustment.

Response: This refers to the theory proposed by MGH and the framework for calculating biodiversity, and literature is also annotated here. As following:

It is clear that an integrated framework for the prediction of montane biodiversity is necessary^{Error! Reference source not found.}, should include includes ecological processes (e.g., survival, competition, and niche differentiation)^{Error! Reference source not found.}, biological processes (e.g., species divergence and extinction)^{Error! Reference source not found.}, and geological and lithologic processes (e.g., orogeny and rock formation)^{Error! Reference source not found.}. A recent attempt at such a framework, the "mountain geobiodiversity hypothesis" (MGH), was first proposed to explain the biodiversity of the Tibeto-Himalayan region^{Error! Reference source not found.} and then extended to explain the origin of montane plant diversity at a global scale^{Error! Reference source not found.}. The MGH

proposes that the evolution of montane biodiversity results from a combination of mountain-uplift, geodiversity evolution, and Neogene and Pleistocene climate changes^{Error! Reference source not found.}^{Error! Reference source not found.}

Q8:

Line69:

“Although climate change force plant species migration is well known, those specific species”

I regard my comments to the original draft of the manuscript not sufficiently considered here. If the text is written in a general manner as here, i.e. not specifically addressing Chinese mountains, then mountain literature with topics of relevance to the study here, but which were done in other regions of the globe, need representative consideration to reflect the state of knowledge.

Response: Many thanks for the feedback, we now listed several other relevant studies from other regions, which have reported that the bedrocks also have significant impact on the diversity of mountain plants.

Please see lines 82-88. *“Although it is well known that climate change forces plant species migration, those species which are adapted to local bedrocks are constrained in their ability to migrate*^{Error! Reference source not found.}*. Bedrock geochemistry is on par with climate as a regulator of vegetation in granitic mountains*^{Error! Reference source not found.}*. Some studies also suggest that local species diversification processes are consistent with edaphic rather than climatic filtration, such in the Cape flora*^{Error! Reference source not found.}*, Teesdale flora*^{Error! Reference source not found.}*, and New Caledonian flora, in which ca. 50% of the endemic floristic elements are ultramafic-obligate species*^{Error! Reference source not found.}*.”*

Q9: Line90:

“Here, we apply the term “flora” to refer to the sum of all angiosperm families, genera, and species growing on a specific mountain or in a well-delimited area^{Error! Reference source not found.}^{Error! Reference source not found.}*, which is a relatively independent biogeographical unit*^{Error! Reference source not found.}*.”*

I would argue against the flora of a mountain being *a relatively independent biogeographical unit*₂ if this is what is meant here (but I may have misunderstood the sentence). First, because related lineages are shared between mountains, and second, because most biogeographers, in terms of terminology, would not consider the flora of a mountain as a “biogeographical unit”.

Also in **Pre-response**: *“we mentioned is not only a unit, but als”* I am not sure how “unit” here would refer to the biogeographic units mentioned in the ms. This may be worth of clarification.

Response: We appreciate the key question raised by the reviewer. In generally, the term “biogeographic unit” is used in the context of biogeographic regionalization.

Here we consider that the flora of mountain and biogeographic zones is equivalent at small scale. First, mountain is geographical barrier to plants diffusion. Second, a mountain usually shows sky island effects and contain their own unique clades or several endemic species.

There are also some literature to suggest that mountain region could be viewed as a biogeographical unit. For instance, Rahbek et al. (2019, Science 365, 1108-1113) wrote that “Like an island, a mountain region may be viewed as a biogeographical unit in itself, with in situ speciation

and extinction playing a key role in building the regional species assemblage”.

Q10: Line95:

“Based on the phylogeny, we calculated the phylogenetic diversity, phylogenetic structure (relatedness clustered or overdispersed), and mean divergence times for the 140 floras.”

I would like to see some justification for this approach mentioned in the manuscript. I raised the issue of calculating species ages, as done here, in my review of the first manuscript draft. At the very least, a note of caution needs to be added at the appropriate space in the manuscript, explaining the shortcomings and limitations.

And also in **Pre-response**: *“In our study, flora age differences were compared by MDT (average age of extant species in the mountain flora, MDT). The MDT is a index to assess the relative age of modern flora (Lu et al. 2018)..... ”*

As also mentioned in my comments to the manuscript, the latter needs to include some statement about the limitations and shortcomings of this approach.

Response: We accept this suggestion by the reviewer. The mean divergence times (MDT), mentioned in the text, is used to measure the relative age of a flora which is proposed by Lu et al. (2018). We have now added a brief explanation about the shortcomings and limitations of MDT in the methods.

Please see lines 490-498 *“The divergence time of each species used to calculate MDT is not absolute age, as they were extracted from the megatree generated in V.PhyloMaker2⁴⁹. We tested the robustness of this method by using four divergence time datasets and found similar MDT patterns between mountain of different landforms (Extended Data Fig. 8). This result is consistent with¹¹⁰, who found that “in large-scale biodiversity and phylogenetic analyses, sources of noise in divergence time estimation are to be expected, but they did not affect the reliability of the results”. We believe that our dated megaphylogenetic tree was suitable for this study because our aim was to reveal the general patterns of landform influence on the formation of mountain flora, rather than focusing on the age of each species.”*

Q11: Line291:

“development process is associated with local bedrocks and regional climate^{Error! Reference source not found.}, thus, the local flora assemble is a comprehensive result of species evolution, landform development and climate change^{Error! Reference source not found.},^{Error! Reference source not found.},^{Error! Reference source not found.}. However,”

This list contains various aspects, biotic and abiotic, which are interrelated. According to the MGH, dispersal of pre-adapted lineages and extinction are equally important. Also, the order “species evolution, landform development and climate change” may be reconsidered.

Response: We have added the following content in lines 311-313 *“Because landforms developmental processes are associated with local bedrocks and regional climate^{17,59}, the assembly of local flora results from a combination of geological and climatic processes, and biological processes.”*

Q12: Line295:

“The underlying reason is that water cycle process and rock erosion rate differs between igneous and sedimentary mountains ecosystem. As is well known water in limestone mountains can easily

be lost through underground river systems, while more overland runoff water is more available for plants in mountains of igneous bedrock. In extreme case,”

As for the introduction, here in the discussion previous work by other authors (citations) should receive appropriate consideration.

Response: We accept this advice and supplemented corresponding literature in the manuscripts. See lines 316-320 *“The underlying reason is that water cycle processes and rock erosion rates differ between igneous and sedimentary mountain ecosystem⁷². As is well known, water in limestone mountains can easily be lost through underground river systems⁷³, while a greater quantity of overland runoff is available for plants in mountains of igneous bedrock.”*

Q13: Line302:

“In adaption to mountain phylogenetic structure of flora and mean divergence times (MDT) of flora,”

Meaning of sentence not clear, needs re-phrasing.

Response: Thank you for your assessment. We revised this sentence in lines 323-324 *“Based on comparisons between mountain flora phylogenetic structure and MDT, we found that landform effects partially determine the final floristic composition.”*

Q14: Line313:

“The mountains of sedimentary bedrock have much stronger environmental filtering effect, as these mountain ecosystems are more sensitive to rainfall. The environmental filtering effect further promoted the clustering of phylogentic structures of mountain floras as predicted by the phylogenetic niche conservatism (PNC)^{Error! Reference source not found.,Error! Reference source not found.,Error! Reference source not found.}”

It would be interesting to discuss here radiations more generally, e.g., how do the studies of plant radiations included in the global study by Muellner-Riehl et al. 2019 (JBI) compare to these statements/findings? Are the findings of these studies (and other studies published since then, i.e. after 2019) comparable to this? I suggest looking at the mountain systems (and their bedrocks) where these studies were undertaken.

And **Pre-response:** *“Second, landform effects (as environmental filtering) restrict the free dispersal of plants between mountains of different landform types (because edaphic species cannot exist outside the original bedrock, dispersal event could occurs with phylogenetic nich evolution).....”*

The meaning of this sentence it not entirely clear to me. I assume what is meant here is that if individuals of a species which are adapted to a specific soil type happen to be dispersed to areas of another soil type, in order to survive, they would have to adapt, and as a result, may become evolutionary independent lineages, potentially new species? This is actually less likely than if their propagules dispersed to areas of the same soil type, which may foster a simpler type of allopatric speciation.

And **Pre-response:** I don't understand this sentence. How could “niche evolution” possibly promote “the spread” (dispersal?) of species? Which niche evolution do the authors mean here, on which taxonomic level? This needs to be clarified. As currently phrased, this sentence does not make sense and I don't get the meaning.

Response: These are really good advice. The potential radiation speciation occurs when plants colonize from one landform to another which maybe an important path of plant diversification. Here,

we document several cases (not all) on adaptative radiation that occur on the bedrocks. But the relations between radiations and bedrocks/landform are still poorly known. We suspect that the effect of landform and bedrock, which promote species differentiation, is more or less neglected under the shadow of “climate change”. Nevertheless, we have reorganized this paragraph carefully.

Please see line 334-348. “Mountains composed of sedimentary bedrock have much stronger environmental filtering effects, as these mountain ecosystems are more sensitive to rainfall. Such effects further promoted the clustering of the phylogenetic structures of mountain floras, as predicted by phylogenetic niche conservatism (PNC)^{Error! Reference source not found.,Error! Reference source not found.,Error! Reference source not found.}. The transformation of landforms effectively promotes the plant radiation, such as the Old World gesneriads^{Error! Reference source not found.} and North American deserts rock daisies (Compositae tribe Perityleae)^{Error! Reference source not found.}. Similar patterns have been observed in insects. For example, two radiation clades (clade nodes 14 to 16 and nodes 31 to 33) of *Exocelina* resulted from the transition from uplifted Australian Plate rock to ultramafic/ophiolite^{Error! Reference source not found.}. Although, many previous studies also document climate change as an important driver of species radiations^{Error! Reference source not found.,Error! Reference source not found.,Error! Reference source not found.,Error! Reference source not found.}. The adaptative radiations of plant genera are more or less associated with bedrock type. For example, the key innovation (lime-secreting hydathodes) of *Saxifraga* sect. *Porphyron* is clearly an adaptation to the limestone bedrock^{Error! Reference source not found.}, and the low specific leaf area (SLA) exhibited by *Erica* is an adaptation to oligotrophic habitats (Quartzite/sandstone) in Cape^{Error! Reference source not found.}. Thus, we suggest that landform and bedrock effects strongly promote species differentiation between different regions, even if such an effect has not been mentioned in previous studies.”

Q15: Line 317-327:

“Dispersal also contributes to the mountain diversity^{Error! Reference source not found.,Error! Reference source not found.,Error! Reference source not found.}. A well-known example is climatic fluctuations during the Quaternary ice age drove changes in the distribution of species around the globe^{Error! Reference source not found.,Error! Reference source not found.}. The spatial configuration of mountain range affect their functional connectivity, influencing species dispersal^{Error! Reference source not found.,Error! Reference source not found.}. In fact, dispersal more easily occurs at initial stage of landform develop mountain, as the slope of mountain still gentle and geographical barrier still relatively low^{Error! Reference source not found.}. Such as the plants dispersal events in Qinghai-Tibet Plateau (QTP, low barrier) is obvious higher than that in Hengduan Mountains (high barrier)^{Error! Reference source not found.}. Forthermore, the effects of dispersal on species diversity in mountainous areas mainly occurred at lowland, and is limited at highlands where has more local endemic species^{Error! Reference source not found.,Error! Reference source not found.}. This means that dispersal has weak influence on the unique composition of mountain flora of different landforms, especially considering that landform have a significant filtering effect on the species.”

The entire paragraph needs re-writing. Dispersal between mountains is not limited to times of climate change, which somehow is suggested here as currently written. Dispersal and establishment was easier in the Hengduan mountains at times of climate change as they have a N-S orientation, enabling plants to change their latitudinal distribution range more easily.

And **Pre-response**: “Such as the plants dispersal events in Qinghai-Tibet Plateau (QTP, low barrier) is obvious higher than that in Hengduan Mountains (high barrier)⁴⁴.”

I don’t understand this sentence. As outlined previously, it is exactly the other way round. Dispersal and establishment was easier in the Hengduan mountains at times of climate change as

they have a N-S orientation, enabling plants to change their latitudinal distribution.

And **Pre-response**: As I pointed out previously, the opposite would be expected. Survival on mountains with N-S orientation is easier during glacials than on those of W-E orientation (e.g. compare Himalayas versus Hengduan Mts.).

And **Pre-response**: “This means that dispersal has weak influence on the unique composition of mountain flora of different landforms, especially considering that landform have a significant filtering effect on the species.”

Reading this, I am not sure the authors have read the papers I had suggested to consult, and which they cited in the paper already before (e.g. Ding et al.).

Response: We accept this advice and wrote this paragraph. We would like to clarify the questions about dispersal events and diffusion barrier. We had carefully readed the paper by Ding et al (2020). It is not difficult to deduce the conclusion “*dispersal events in Qinghai-Tibet Plateau (QTP, low barrier) is obvious higher than that in Hengduan Mountains (high barrier)*” (see Ding et al, 2020, fugure 2). On the other hand, we thought dispersal would reduce the β -diversity between different floras. Such as the proportion of endemic species in Hengduan Mountains is the highest is largely affect by its low rate of colonization (<0.05), and high rate of *in situ* speciation and local and recruitment. In contrast, the low proportion of endemic species in Qinghai-Tibet Plateau has the highest rate of colonization scince early Miocene. It is clearly the dispersal is negative correlation to the proportion of endemic species in flora.

The re-wrote paragraph please see lines 349-367. “*Dispersal also contributes to montane floristic diversity, although dispersal is only effective if it is followed by successful establishment. Dispersal occurs more freely during the initial stages of landform development in mountains, when slopes are still gentle and geographical barriers are minimal. For example, the plant dispersal events across the Qinghai-Tibet Plateau (low barrier) have been considerably greater than that in Hengduan Mountains (high barrier). The role of local recruitment was most important during the early stages of mountain flora assembly, a role which was subsequently replaced by local adaptations or in situ speciation. Mountains in different landforms will recruit different plant species, because each species is differently adapted to specific types of bedrock. The spatial configuration of mountain ranges alters their functional connectivity, thus influencing the species dispersal process. Furthermore, during the middle stages of landform development, the effects of dispersal on species diversity in mountainous areas are primarily restricted to lowlands, and are limited in highlands where there are more local endemic species. Taken together, this suggests that dispersal only weakly influences the unique composition of mountain flora associated with different landforms, especially considering that landforms exhibit a significant filtering effect. Variation in the species composition of mountain floras between different landforms will increase under the significant, combined landform environment filtering effect, and local endemic speciation. Therefore, we propose that the patchy spatial distribution of different mountain landforms is an important factor in shaping surface biogeographical zoning.*”

Q16: Line332:

“Thus, landforms are suitable indicators of geodiversity for tracking changes in mountain floras along geological time scales.”

I am missing more elaboration on measure of geodiversity. E.g., how does this compare to the geodiversity indices used in previous mountain studies (such as the MGH global study by Muellner-Riehl et al. 2019 in JBI)?

Again, the lack of discussion of work by other authors is misleading and suggests more novelty here than is actually inherent in this present study. Acknowledging previous work by other authors does not diminish the achievements of this study here, but rather empowers readers to compare this and previous studies.

And **Pre-response:** But geodiversity as a measure (geodiversity index) is an integral part of investigations under the MGH.

Response: Thanks for pointing out the shortcomings. There is no denying that linking geological diversity to biodiversity is an important step. However, there are still some problems in the application of geodiversity. In previous studies, geodiversity is usually treated as the environment variables integrated into the compound geodiversity index (GD, or GDCs). But the power of compound geodiversity index was not good enough in predicting species diversity. As the case you mentioned (Muellner-Riehl et al. 2019 in JBI, wrote “*The GD index and Elevational range were strongly correlated ($r = .70$) and given that Elevational range showed better performance in single predictor models...*”), the geodiversity index (GD) was found to be less effective than a single topography variable (elevational range) for predicting mountain species diversity. In contrast, the landform type (characteristic variable) is more concise and easier to understand. For a detail discussion, please see line 372-388.

“Geodiversity, the variability of Earth’s surface materials, forms, and physical processes, is an integral part of nature and crucial for sustaining ecosystems and their services. Several studies proposed to use geodiversity to model biodiversity based on the hypotheses that specific geo-sites should support unique biota and that high geodiversity is coupled with high biodiversity. However, the predictive power of geodiversity has not been high. For example, the geodiversity index (GD) was found to be no better than elevational range for predicting mountain species diversity. Another recent study reported that combinations of environmental variables better explained tropical species diversity than GD. In fact, the contribution of geodiversity component (GDC) showed similar patterns to topography, and usually effects at smaller extents and finer grain size. The GD is a poor predictor of mountain diversity in large part because interactions between mountains and biodiversity are complex, and thus geodiversity could not be treated as an index. In contrast, landform type is an objective description of the present state of erosion of a mountain or bedrocks. When we stand in front of a karst mountain, we can see quite clearly that the plants are different from those in a granite mountain. We propose that landform identity rather than GD, is a more suitable indicators of geodiversity for tracking changes in mountain floras along geological time scales.”

Q17: Line334:

“Based on our results, we put forward the “geological lithology hypothesis of flora” to explain the assemble and differentiation of mountain floras. In this theoretical framework, floristic assembly in

mountains is driven by the lithospheric cycle, which refers to the bedrock-constrained developmental processes of landforms. Specifically, under this hypothesis, the mountain species differentiation closely related to the type of bedrock and degree of erosion, species richness and species composition in mountain flora are interaction result of landform and environment, and phylogenetic niche evolution^{Error! Reference source not found.} can promote the spread of species among different landform floras. Overall, our study provides a novel framework and approach for determining the mechanisms of species diversity within mountains and the distributional patterns of some of the world's richest floras.”

This section needs to be improved and provide a more balanced view on previous work by other authors and suggestions put forward here. As I had already suggested in my review of the first manuscript draft, the manuscript here needs to go into some more depth concerning previous hypotheses put forward, importantly the MGH, and I still don't see this has been done. While only briefly mentioned in the into, the MGH does not show up here in the discussion. I would like to see some of what the authors have answered in their rebuttal be actually also included here in the manuscript text.

Response: This is a great suggestion. We rewrote this section. Please see line 389-413.

“Based on our results, we propose the "geological lithology hypothesis of flora" to explain the assembly and differentiation of mountain floras. In this theoretical framework, floristic assembly in mountains is driven by the lithospheric cycle, which refers to the bedrock-constrained developmental processes of landforms. Specifically, under this hypothesis, montane species differentiation is closely related to the type of bedrock and degree of erosion. Both SR and species composition in mountain flora result from interactions between the landform and the environment. In addition, the dispersal of plants between different landform types is more restricted than within the same landform type. Successful diffusion across a landform is often accompanied by the emergence or radiation of adaptive traits. This is called phylogenetic niche evolution^{Error! Reference source not found.} and can promote the spread of species among different landform floras. This differs from the MGH, which assumes that the montane biodiversity hotspots require three key boundary conditions: 1) the presence of lowland, montane and alpine zones, 2) climatic fluctuations to produce a “species pump” effect, and 3) high-relief terrain with environmental in a given mountain region^{Error! Reference source not found.,Error! Reference source not found.}. In contrast, our hypothesis suggests that montane bedrocks and landform processes determine the geographic distribution of plants. The MGH effectively explains the high biodiversity of mountains characterized by large elevational differences (such as the Himalayan and Andes mountains), but such restrictive boundary conditions constrain the applicability of the hypothesis. Biodiversity hotspots also occur in regions with stable climates, such as the Namib desert^{Error! Reference source not found.}, or with with minimal elevational gradients, such as the Southeast Asia karst landforms^{Error! Reference source not found.}. We argue that the novel "geological lithology hypothesis of flora" could serve as a general explanation for global diversity patterns, as the formation of mountains on the Earth's surface is the result of the cycling of sedimentary, igneous and metamorphic rocks. In conclusion, our study has highlighted the floristic patterns of different landforms and provided a novel framework for studying the mechanisms of plant species diversification within mountains and the distributional patterns of mountain floras of the world.”

Q18: Line378:

“For the 236 genera missing, we treated each as sister its most closely related genus in “GBOTB.extended.LCVP.tre” based on megaphylogenies within other references^{Error! Reference source not found.}.”

I suggest to provide at least some information about the limitations of this approach.

Response: Thank you for your assessment. A brief introduction about the limitations of this approach is provided in methods.

Please see lines 444-450 *“Of the 2,585 genera and 17,576 species studies here, 2,349 genera and 8,663 species were included in “GBOTB.extended.LCVP.tre”. Based on previously-published megaphylogenies^{50,101}, we treated each of the 236 missing genera as sister to their most closely related genus in “GBOTB.extended.LCVP.tre” using V.PhyloMaker2⁴⁹. Although this method resulted in more robust phylogenetic relationships than Phylocom¹⁰², the ultimate phylogenetic relationships should still be considered relative, as complete phylogenetic data are still lacking for many families and genera.”*

Q19: Line417: Divergence time estimation

I suggest to provide at least some information about the limitations of this approach.

See also my previous comment further above: I would like to see some justification for this approach mentioned in the manuscript. I raised the issue of calculating species ages, as done here, in my review of the first manuscript draft. At the very least, a note of caution needs to be added at the appropriate space in the manuscript, explaining the shortcomings and limitations.

Response: We accept this comment and we have provided justification for the approach in lines 490-498 *“The divergence time of each species used to calculate MDT are not absolute age, as they were extracted from the megatree generated in V.PhyloMaker2⁴⁹. We tested the robustness of this method by using four divergence time datasets and found similar MDT patterns between mountain of different landforms (Extended Data Fig. 8). This result is consistent with¹¹⁰, who found that “in large-scale biodiversity and phylogenetic analyses, sources of noise in divergence time estimation are to be expected, but they did not affect the reliability of the results”. We believe that our dated megaphylogenetic tree was suitable for this study because our aim was to reveal the general patterns of landform influence on the formation of mountain flora, rather than focusing on the age of each species.”*

The response to other comments raised in previous response letter is below.

Q20:

Pre-response: *“Because plants are more dependent on local rocks and soils, and less responsive to climate than animals...”*

I don't agree. Plants can't move and therefore are expected to show even stronger response.

Response: We agree with your point. In fact, we don't think our “less responsive” and “show even stronger response” are in conflict. Because the former is a timely response to climate change, while the latter is a relatively delayed result. Anyway, plants are more dependent on the regional environment (especially the bedrocks and soil) than animals.

Q21:

Pre-response: Geodiversity as a measure captures this, and as such, the MGH inherently includes

this aspect, which should be acknowledged in the manuscript. The intro text needs to go more into some more detail concerning the similarities and differences between the MGH and the approach proposed here. It is important to carve out the further refinements of the approach here for the reader. This will not diminish the accomplishments of the authors, but rather help them to put their work in context. Otherwise, the text will imply more novelty than justified.

Pre-response: I would have appreciated an answer which more directly reflects what was actually done in the new version of the manuscript to satisfy the criticism. This is also true for the answers to the other questions/remarks by reviewer 3. What was asked here was that the intro does not reflect the body of literature sufficiently. Just adding two sentences, as indicated further below at the end of the answer does not do justice to what is known about geobiodiversity.

Pre-response: *“Among which, a representative theory is “mountain geobiodiversity hypothesis” (MGH), which proposed to explain the biodiversity of Tibeto-Himalayan region2.”*

But it applies not only there. While it was originally developed for the THR, later work by Muellner-Riehl et al. 2019 in JBI tested its global validity.

Response: These comments are about geodiversity and associated MGH. Revision associated with this comments in new manuscript is in line 62-63 “These results strongly suggest that montane species diversity is largely affected by habitat heterogeneity¹³, or the so-called geodiversity¹⁸⁻¹⁹. ”

Line 68-72 “A recent attempt at such a framework, the “mountain geobiodiversity hypothesis” (MGH), was first proposed to explain the biodiversity of the Tibeto-Himalayan region^{2,29} and then extended to explain the origin of montane plant diversity at a global scale³. The MGH proposes that the evolution of montane biodiversity results from a combination of mountain-uplift, geodiversity evolution, and Neogene and Pleistocene climate changes^{3,29}.”

And line 373-384 *“Geodiversity, the variability of Earth’s surface materials, forms, and physical processes, is an integral part of nature and crucial for sustaining ecosystems and their services¹⁸. Several studies proposed to use geodiversity to model biodiversity based on the hypotheses that specific geo-sites should support unique biota and that high geodiversity is coupled with high biodiversity^{2,3,18,19}. However, the predictive power of geodiversity has not been high. For example, the geodiversity index (GD) was not found to be more effective than elevational range for predicting mountain species diversity³. Another recent study reported that combinations of environmental variables better explained tropical species diversity than GD⁹². In fact, the contribution of geodiversity component (GDC) showed similar patterns to topography, and usually effects at smaller extents and finer grain size⁹³. The GD is a poor predictor of mountain diversity in large part because interactions between mountains and biodiversity are complex²⁴, and thus geodiversity could not be treated as an index.”*

Q22:

Pre-response: *“Therefore, we believe that the analysis in this paper should have different meanings compared with the previous analysis results.”*

I don’t regard this as really convincing argumentation.

Response: Thanks for this insightful criticism. The key topic of mountain biodiversity is to reveal the reason of unusually rich beyond latitudinal diversity gradient. We speculate that the landform pattern is differs from latitudinal diversity gradient, and landform identity represents a new approach to understand the assembly and differentiation of mountain floras. We further clarified our views in

the new manuscript, please see line 369-414.

Q23:

Pre-response: *“Yes, the research carry out by Antonelli et al. (2018) is quiet an important in mountain species diversity. We cited this literature in the introduction and other part of our manuscript, please see L56, L65, L173, L234.”*

My comment had not addressed the citation being missing, but I ad requested an earlier mention of it. Preferably, the answer in a rebuttal letter should directly refer to what has been asked for. Here, it does not become evident from the text straight forwardedly, whether this is the case.

Response: Thanks for this comments. The manuscript has been extensively revised. In the currently manuscript, the research carry out by Antonelli et al. (2018) is references 24.

Q24:

Pre-response: *“Thanks for the reviewer's criticism. After a long period of field investigation, we realized that the flora composition in mountains is different among different landforms.”*

Was this studied here? I need to re-check whether the (tax.) composition of the flora was investigated.

Response: The basic data of this study mainly based on published flora checklist (see Extended Data Table 1, and Additional information), in which 12 mountains (reference 129, 144, 161, 163, 165, 171, 173, 201, 203, 209, 219, 264) are based on our previous field investigation.

Q25:

Pre-response: What do the authors consider as “younger” species?

“The “younger” species represent the species that diverged later in the mountain flora. To avoid potential bias between mountains, we ranked all species occurs in 140 mountains from youngest to oldest, partitioned them into quartiles based on their ages, computed MDT in each mountain for the absolute youngest 25% and the absolute oldest 25% of species (Lu et al. 2018).”

This does not answer my question as intended. I was interested in absolute ages.

Response: Thank you for your concerns. We think this question is similiar to the third question raised by reviewer#2. Unfortunately, as far as we know, there is unable to determine the “absolute age” of a species. In general, the age of a species is based on the branch in the phylogenetic tree, which estimate based on the molecular clock hypothesis and fossil calibrated (but still not “absolute ages”). The “younger” species represents the species with shorter branches in phylogenetic tree.

Q26:

Pre-response: *“To be specific, bedrocks and associated micro-landforms in mountains is the most important factor promote speciation local endemic species. ”*

It may be argued that this is not correct, i.e. it may be disputed that bedrock and micro-landforms are THE MOST IMPORTANT factor to promote the evolution of local endemics. Mountain orientation plays a key role. Comparing mountains of different orientation (W-E versus N-S) shows that those oriented N-S harbor a higher no. of species that were able to survive during times of climatic change and that were able to form endemic lineages than those oriented W-E. This is because a W-E orientation prevents populations moving into more favourable latitudes during

times of climate change (barrier effect) that has happened in the past. A N-S orientation, in contrast, as e.g. found in the Hengduan Mountains or the Andes, enabled populations to move into more favourable latitudes.

Response: That's a good advice which further promotes our thinking on the role of diffusion processes in the assemble of endemic element in mountain biodiversity. The reviewers point out the endemic species in mountain may mainly derived from migration during periods of climate fluctuation. Endemic plants in this situation can also be called climate relict species. We do not deny the contribution of climate relict species to local endemic. While, dispersal is only effective if it is followed by successful establishment. In our view, successful migration events more easily in mountain with similar landform type (as the Hengduan Mountains) than different landform type. We are more interested in the differences in mountain flora of different landform types. According to your opinion, the section dispersal and conclusion has been rewritten (please see line 350-414).

Q27:

Pre-response: “*Second, landform effects (as environmental filtering) restrict the free dispersal of plants between mountains of different landform types (because edaphic species cannot exist outside the original bedrock, dispersal event could occurs with phylogenetic nich evolution).....*”

The meaning of this sentence it not entirely clear to me. I assume what is meant here is that if individuals of a species which are adapted to a specific soil type happen to be dispersed to areas of another soil type, in order to survive, they would have to adapt, and as a result, may become evolutionary independent lineages, potentially new species? This is actually less likely than if their propagules dispersed to areas of the same soil type, which may foster a simpler type of allopatric speciation.

Response: Thank you for your concerns. We think this is more or less a neglected research topic in previous studies. Differences in soil and bedrock have been suggested in some studies to promote species differentiation, such as Savolainen et al.(2006: doi:10.1038/nature04566). For more details, please see line 336-349 “*The transformation of landforms effectively promotes the plant radiation, such as the Old World gesneriads⁷⁹ and North American deserts rock daisies (Compositae tribe Perityleae)⁸⁰. Similar patterns have been observed in insects. For example, two radiation clades (clade nodes 14 to 16 and nodes 31 to 33) of Exocelina resulted from the transition from uplifted Australian Plate rock to ultramafic/ophiolite⁸¹. Although, many previous studies also document climate change as an important driver of species radiations^{3,62,82-83}. The adaptative radiations of plant genera are more or less associated with bedrock type. For example, the key innovation (lime-secreting hydathodes) of Saxifraga sect. Porphyron is clearly an adaptation to the limestone bedrock⁸⁴, and the low specific leaf area (SLA) exhibited by Erica is an adaptation to oligotrophic habitats (Quartzite/sandstone) in Cape⁶⁷. Thus, we suggest that landform and bedrock effects strongly promote species differentiation between different regions, even if such an effect has not been mentioned in previous studies.*”

Q28:

Pre-response: “*Moreover, our hypothesis also tries to explain the reason why each mountain has high or low plant diversity, or suitable plant diversity.*”

Second, This is, in the core, not different from the MGH (compare Muellner-Riehl et al. 2019), but it adds an additional component.

Response: Thanks for this comments. Our hypothesis “geological lithology hypothesis of flora” more focus in the species composition differences of mountain flora. We emphasize the dominant role of geological processes in the formation of species diversity.

Q29:

Pre-response: “*The period of rapid landform process in modern China coincides with the period of rapid plant species divergence.*”

But has this been actually rigorously tested, or there is just temporal coincidence?

Response: To our knowledge, there are no such a test study. However, the correlation between species divergence and Cenozoic plate tectonic change in East Asia is a well reported pattern. This is a valuable research direction and may test in future.

Q30:

Pre-response: “*For example, most floras of China, including both mountains and lowland floras, diverged during the Miocene when the East Asian monsoon intensified*^{38,52}.”

“floras” don’t “diverge”. I would argue against mixing vegetational terms and evolutionary terms. Divergence in an evolutionary sense would not apply to “floras”. The sections also needs some general language improvement.

Response: We accept this advice. The revised sentence in line 255-256 “*For example, most Chinese floras, including both montane and lowland floras, differentiated during the Miocene when the East Asian monsoon climate intensified*^{50,51}.”

Q31:

Pre-response: “Here, we’re not focus on specific mountains. We want to point out that the age of local flora in different regions is related on how long the local landform process, even between continents. Chen et al. (2018),”

Some of what is explained by the authors in the following text may be viewed useful for readers to better understand the reasoning behind this study. If possible, I suggest to incorporate some of the aspects in the intro and discussion.

Response: We revised this section, Please see line 258-265 “*At the intercontinental scale, the ages of floras were largely consistent with regional landform developmental processes. For example, the floras of eastern Asia are typically younger than those of South Africa and Australia, where the landform processes are fairly old, but older than those of the Andes and Amazonia, where landform processes have occurred more recently*^{1,57,66}. Thus, the relationship between the floristic assembly of mountains and the type of landform and landform developmental process seems to be global in scope, at least for angiosperms. Therefore, landform identity may make a suitable indicator for exploring the floristic assembly of mountains.”

REVIEWER COMMENTS

Reviewer #2 (Remarks to the Author):

The manuscript is much improved from the last round and reads well generally. I also very appreciate the responses and revisions with regard to my comments. In particular, the interaction terms added interesting perspectives to the data and actually strengthen the paper. It's a lot of results to report here but the overall message is strong: landforms are important factors of floral diversity. The other issues with phylogenetic analyses seem more difficult to resolve and acknowledging the limitation there would be sufficient.

Minor comments:

A lot of the citations are not showing properly, but as "Error! Reference source not found"

Is the first paragraph supposed to be an abstract? It looks quite out of place. The mention of "landform development" is a bit sudden

L31: remove "analysing" to make the rest of the sentence about a "topic"; and move "processes" to after geological to improve clarity.

L63: do you mean animal diversity?

L64: add "and" to before "should include" and delete "includes"

L105: change "sum" to "collection" or "community" and remove "families, genera, and" unless you mean there are taxa that are not identified to species but also included here.

L107: I think I get your point but it needs to be more explicit. How about mentioning species diversity only one characteristic of a flora and can be quantified by species and phylogenetic diversity?

Figure 1: please add the names of the landforms to the figure to improve clarity for the

broad audience of the journal. It was very hard for me to matching things up using the caption.

L152-153: not sure what the last sentence means. They all still look significant in the full model.

L154: add "(towards eastern China)" to help guide non-expert audience.

L183: the point about higher rates of extinction needs further explanation and should probably be in Discussion - this happens at several places in the Results section (e.g. the paragraphs starting at L250 and L281)

Figure 3-4: It's very easy to lose track of all the abbreviations but I see NTI is not in Figure 3 and SR is not in Figure 4. It would improve clarity if Figure 3 and 4 have consistent panels, though showing the landform and full models respectively.

L303: "recognised as positively correlated to..." ("significantly" is usually unnecessary as we can't say "correlated" if it's not significant).

Reviewer #3 (Remarks to the Author):

For convenience, I have provided my detailed comments in the "response to comments" file.

[editorial note: see the following page for the start of this file]

Reviewer 3 comments are inserted in GREEN colour below (with parts of the authors' text occasionally marked in BLUE, if comments refer specifically to some statements.

Response to reviewers' comments

Thank you for taking the time to review our manuscript entitled "Landform and lithosphere development drive the assembly of mountain floras". We appreciate your insightful comments and suggestions revised by Reviewer #2 (mainly data analysis) and Reviewer #3 (mainly review and comparison of previous studies), which have been immensely helpful. We have carefully studied the reviewers' comments and made corresponding modifications. The manuscript has also been carefully grammatically revised.

REV: According to my observations this is not correct, at least not for all parts of the manuscript. For example, revised M&M texts are of higher quality than other parts of the manuscript which are poorly written and where writing needs to be substantially improved to be deemed acceptable – and understandable to the readers. There are many sentences which are neither understandable in terms of language nor scientifically correct (in content and use of terminology).

Some single errors I have marked using magenta, but for other larger sections that need re-writing I mention this in my text.

The corrections are indicated in red in the manuscript (clean version), and we also submitted a version to track all the revisions. The replies to reviewers' comments are listed as follows.

In order to make it easier for reviewers to read, we will sort the questions mentioned in the main text and responses in order of Q (Q1, Q2...), and then reply separately.

REVIEWER COMMENTS AND OUR RESPONSE

Reviewer #2 (Remarks to the Author):

Q1:

I am re-reviewing this manuscript and I was R2 in the last round. The manuscript is much improved but several of the responses from the authors are not very satisfying. I would particularly like the authors to re-consider the issues below – I try to explain them better this time.

The authors have kindly pointed to their extended table 10 in response to my request for testing interactions between landforms and other environmental variables. As far as I can tell, Table 10 only contains additive models and cannot identify interactive effects. For example, rather than $Y \sim \text{landform} + \text{climate}$, it needs to be $Y \sim \text{landform} * \text{climate}$, or better yet, $Y \sim \text{climate} + (\text{climate} | \text{landform})$ in a multi-level framework.

Response: We are sorry that some of our responses did not address your concerns, and thank you for your kind clarification here. Following the excellent suggestion by the reviewer, we investigated the interaction of landforms and other variables based on $Y \sim \text{climate} + (\text{climate} | \text{landform})$ in a multi-level framework. Our revisions in the manuscript method part as lines 569-573: "However, we also extended the "landform model" to include all others variables to assess the effect of climate

on species diversity (i.e., “full model”). Determinants of SR might change with landform type, and we therefore test for interactions between landform and other predictor variables (only shown significant variables in full model).”

The results showed that the species richness (SR) affects the interactions of landform and TCQ, Annual Precipitation (PREC), and PCQ (see “Extended Data Table 10”). Accordingly, we add landform: TCQ, landform: PREC, and landform: P_{var} as variables to the full model. The best full model is shown in the manuscript, see “Extended Data Table 3”. Based on the new full model results, when considering the interaction effects between landforms and environmental variables, the high precipitation of wettest month (PWM), and high precipitation of coldest quarter (PCQ) are no longer detected as significant predictor. This could be expected, as the most important outcome of the interactions between landforms and environmental variables is about water availability in mountain systems. The models that essentially accounted for the interaction effects did not differ from the previous one, and both models explained more than 60% (63.7% vs 62%) of the variation of species richness. We also provided the “Extended Data Fig. 9” to show the influence of interactions between landforms and other environmental variables on species diversity in montane areas.

In the SR model result part, we add lines 144-150 “The full model explains 62.8% of the observed deviance of SR in the GLM and 63.7% in the SAR, and strong interaction effects between landform and mean temperature of the coldest quarter (TCQ) were detected (Extended Data Table 3). We also found weak interactions between the landform effects and annual precipitation (PREC), as well as precipitation of coldest quarter (PCQ) (Extended Data Table 10).”, also lines 165-168 “Interestingly, TWQ is not a significant predictor and explains only 0.9% of the variation of SR (Extended Data Fig. 9g). The observed significant negative correlation between SR and high TWQ (Extended Data Table 3) may be the result of incorporating landform effects into the regression models (Extended Data Fig. 9i)”. For details, please see the result section of revised manuscript “Landforms effects on species richness and phylogenetic diversity” in lines 138-172.

We also find the phylogenetic structure and MDT are affected by interactive effects. For example, the MDT is affected by the interactions between landforms and log(area), long, elevmid, bio2, bio8, bio10, and bio11. However, if all these interaction effects are included in the full model, the prediction model becomes too complex for interpretation. As such, in order to maintain the simplicity of our prediction model, we do not consider the interaction effects in the full model of NRI, NTI, PDI, MDT, MDT_{oldest}, and MDT_{youngest} in the main text. Nevertheless, as supplementary results, we provided figures to show the impact of interactions between landforms and other environmental variables (which were detected as significant predictors in the full models) on NRI, NTI, PDI, MDT, MDT_{oldest}, and MDT_{youngest}. See “Extended Data Fig. 10-15”.

Extended Data Table 10|Results of the interactions between landform and others variables on the species richness (SR).

Variables	GLM			SAR		
	Coefficient	SE	t	Coefficient	SE	z
TWQ	0.206	0.187	1.106ns	-1.072	0.229	-4.691***
TCQ	0.760	0.153	4.956***	0.668	0.179	3.733***
PREC	0.957	0.132	7.269***	0.968	0.151	6.428***
PCQ	0.498	0.117	4.275***	0.466	0.150	3.102***
Landform*TCQ (Intercept)	7.747	1.171	6.616***	8.141	1.130	7.2***

Desert	-0.513	1.183	-0.434ns	-0.859	1.142	-0.752ns
Granitic	-0.935	1.179	-0.793ns	-1.325	1.139	-1.163ns
Karst	-0.212	1.251	-0.169ns	-0.634	1.206	-0.526ns
Karst-Gr	1.020	1.324	0.771ns	0.399	1.283	0.311ns
TCQ	-1.113	1.620	-0.687ns	-1.724	1.573	-1.096ns
Desert:TCQ	-5.291	2.025	-2.613ns	-4.812	1.920	-2.506*
Granitic:TCQ	1.767	1.633	1.082*	2.367	1.587	1.492ns
Karst:TCQ	0.488	1.736	0.281ns	1.152	1.683	0.685ns
Karst-Gr:TCQ	-1.384	1.925	-0.719ns	-0.329	1.879	-0.175ns
Landform*PREC (Intercept)	7.765	0.997	6.784***	6.320	0.907	6.948***
Desert	-0.702	1.018	-0.689ns	-0.584	0.942	-0.620ns
Granitic	0.0153	1.006	0.015ns	0.523	0.918	0.570ns
Karst	-0.607	1.195	-0.509ns	-0.449	1.084	-0.414ns
Karst-Gr	0.517	1.088	0.475ns	1.008	0.980	1.029ns
PREC	0.274	1.505	0.182ns	0.888	1.371	0.648ns
Desert:PREC	3.950	2.368	1.668ns	6.115	2.372	2.578**
Granitic:PREC	0.408	1.516	0.269ns	-0.268	1.380	-0.194ns
Karst:PREC	1.497	1.941	0.771ns	1.381	1.766	0.782ns
Karst-Gr:PREC	-0.240	1.706	-0.140ns	-0.942	1.538	-0.612ns
Landform*PCQ (Intercept)	6.821	0.485	14.071***	6.643	0.474	14.005***
Desert	-0.765	0.511	-1.495ns	-0.563	0.501	-1.124ns
Granitic	0.231	0.491	0.471ns	0.391	0.480	0.816ns
Karst	0.148	0.572	0.259ns	0.292	0.554	0.527ns
Karst-Gr	0.690	0.516	1.338ns	0.899	0.499	1.801ns
PCQ	0.192	0.730	0.263ns	0.430	0.711	0.605ns
Desert:PCQ	13.196	4.178	3.158**	14.191	4.023	3.528***
Granitic:PCQ	0.159	0.742	0.214ns	-0.040	0.718	-0.055ns
Karst:PCQ	0.261	1.248	0.209ns	0.057	1.195	0.048ns
Karst-Gr:PCQ	-0.968	0.918	-1.055ns	-1.298	0.879	-1.480ns

Extended Data Fig. 9 Interaction effects between landform and environmental variables on species richness (SR).

Q2:

In terms of distinguishing the other categories, I think the upper half of the extended Table 3 only shows that most categories are significantly different from Danxia, which is good to know but it still does not distinguish the other three most similar ones: Karst, Karst-Gr and Granitic. It might be best to do pair-wise comparison (t test or Wicoxon test) for all possible pairs.

Response: We accept this advice and did the pair-wise comparison (t-test) for all possible pairs. The result is shown in “Fig. 3”. The differences in species richness, PDI, NRI, and MDT between the landforms can be clearly seen in “Fig. 3” below.

The result showed that, in general, the Danxia and desert landform are significantly different from Granitic, Karst, and Karst-Gr landform. The confidence intervals between the latter three landforms usually overlap. We suspect this may be influenced by spatio-temporal correlation of landform development, as Karst-Gr is an intermediate stage of Karst and Granitic.

Fig. 3 | The differences in species richness, phylogenetic structure, and age of floras among types of landforms. P value of T-test result between each pairs of landforms were shown above the black line: **** $p < .001$; *** $p < .001$; ** $p < .01$; * $p < .05$; ns = not significant.

Q3:

In terms of the phylogenetic analyses, they authors have apparently misunderstood my point which is NOT about the effect of phylogenetic completeness. I have no problem with an incomplete tree as long as the sampling is not known to be bias in a systematic way. Rather, I am asking for a clear explanation of the reasoning behind the simulation, in terms of what it reflects when random samples are drawn from a phylogeny of only the species in the dataset rather than in the region. I do not see why the null model demanded that the species pool should be consistent in the tree and the distribution database unless it is required by the specific package you use (I do not think it is based on the package documentation). It is possible that if you use a global tree or a regional tree, the patterns remain quite similar if the effect of contingency is not too strong; strong contingency plus large difference in species richness among regions could really bias your simulations. Therefore, I suggest a check on that to ensure the robustness of your findings.

Response: Thanks for further clarifying this question. The reviewer mentioned “I do not see why the null model demanded that the species pool should be consistent in the tree and the distribution database?” And why doesn't the author use a global database or a more complete regional database? While the reviewer raised interesting points to examine, we will highlight the main reasons for

taking our approach by providing a more detailed explanation below.

At first, we would like to explain the null model used in our study. As supplementary, we provide a brief introduction of the packages and the null model used to calculate the PDI, NRI and NTI in the methods.

Please see lines 466-467 “To calculate PDI, we used ‘PhyloMeasures’^{Error! Reference source not found.} in R, in which the null model was set as “uniform” and the following typical algorithm was implemented: ”,

and line 482-485 “In these equations, $MPD_{observed}$ and $MNTD_{observed}$ are the observed MPD and MNTD, $MPD_{randomized}$ and $MNTD_{random}$ are the expected (i.e., average) MPD and MNTD of the randomized assemblages^{Error! Reference source not found.}, which were calculated based on the null model “uniform” in the R packages ‘PhyloMeasures’^{Error! Reference source not found.} ”,

Here, the null model “uniform” considers samples with equal (uniform) probability among all possible tip samples of the same richness. This was based on the assumption “...species have been able to disperse (possibly over many generations) anywhere...”, which was proposed by Webb in 2000 (DOI:10.1086/303378). The null model in this study fully takes into account the contingency of species which occurs in a mountain flora. Species richness is only related to phylogenetic diversity (such as PD), but the species richness effects on phylogenetic structure (this study’s focus) can be excluded. In the null model, 100 or 1000 species will be randomly extracted from the phylogenetic tree to calculate the $MPD_{randomized}$ and $MNTD_{random}$ if a mountain contains only 100 or 1000 species (MPD, and MNTD). In other words, phylogenetic structure (PDI, NRI, and NTI) is mainly related to the distance of the phylogenetic relationship of species in the mountain, rather than the species richness per se so the pool size likely does not matter as much (see “Extended Data Fig. 3 | Regression analyses between species richness and PD, PDI, NRI, NTI.”).

Second, based on the technical requirements (this is based on the null model assumption) of the software package, flora data is extracted from the mountain species database to reflect the phylogenetic structure of the flora. Thus, a global or regional tree is not needed here. In fact, as we know there isn’t currently an approach that is calculates a regional phylogenetic structure based on a global tree.

REV: The meaning of this sentence is not clear to me. Of course, there exist studies that have been using e.g. large angiosperm-wide trees as source for creating much smaller trees with only regional species samples representation. This needs clarification.

In our study, the calculation of phylogenetic diversity and phylogenetic structure were performed by R packages “PhyloMeasures” (Tsirogianis & Sandel, 2016; doi: 10.1111/ecog.01814). This package is widely used in studying the community phylogenetic structure of floras. The “PhyloMeasures” requires the species pool to be consistent with the phylogenetic tree and distribution database. Otherwise the functions (“mpd.query”, “mntd.query” and “pd.query”) in “PhyloMeasures” would not work. See the screenshot below for the warning in R.

```
> mpd.std<-mpd.query(tree,data, null.model="uniform", T)

Warning: the input matrix has fewer columns than the number of species in the tree.

Error in mpd.query.uniform(tree, matrix, standardize) :
  One of the species names in input the matrix was not found in the tree (Staurogyne_rivularis)
> |
```

REV: This is not clear to me and needs explanation. *Staurogyne rivularis* is a synonym for *Staurogyne spatulata*, only the latter name which is accepted. Not finding “*Staurogyne rivularis*” could mean that the dataset uses the correct name instead, *St. spatulata*, rather than the synonym. Which taxonomic name resolution approach was used by the authors and how was it guaranteed that species were not omitted from the analysis due to synonyms or spelling errors? This could have introduced substantial error.

Q4:

With regard to the "age" of species, I appreciate the authors' effort to provide additional information by comparing the pruned and unpruned trees.

However, my point is more about the concept of species or assemblage age -- it is not the real age and can only be used for representing the patterns of age when extinction is random; in contrast, a comparative analysis can be problematic if extinction was not random with regard to the factors that are being examined. Therefore, I suggest rephrasing it in terms of species distinctiveness, so that an assemblage with low MDT contains more closely related species and infer mechanisms from there.

Response: Thank you for clarifying your concerns, and we appreciate your suggestion to use the term “species distinctiveness” to replace “flora age of species age”. Here, we're unsure the meaning of the term “species distinctiveness”. According to our understanding, we assumed the reviewer's proposed ‘species distinctiveness’ to be the evolutionary distinctiveness of species as proposed by Isaac et al., (2007, doi:10.1371/journal.pone.0000296). We suspect there are no differences between “species distinctiveness” and species age, because both of them are based on the **in-clade** in the phylogenetic tree. Besides, species age (or species divergence time) is more well known than “species distinctiveness” in evolutionary biology and biogeography.

REV: What do the authors mean by “in-clade”?

On the other hand, we accept that the concept of “flora age” is problematic. The complexity assembly history of flora makes it impossible to interpret the real age of flora. In contrast, the age of the species, which estimate based on the molecular clock hypothesis (Ho, S., 2008; The Molecular Clock and Estimating Species Divergence) and fossil calibrated, is more reliable. In fact, the inference of floristic history is often based on the estimated divergence time (or stem age) of species which occurs in the flora (such as Dagallier et al., 2020, doi: 10.1111/nph.16293; Qian & Deng, 2022, doi: 10.1111/jse.12856; and Chen et al., 2018, DOI: 10.1093/nsr/nwx156).

In our study, we used the mean divergence times (MDT) proposed by Lu et al.(2018) to represent the “flora age”. As such, the MDT reflected the age composition of species within a flora. A flora with larger MDT has more ancient species and hence is expected to have older floristic ages.

Overall, we believe that our use of MDT to measure the species age structure of flora is consistent with the existing literature and therefore is appropriate.

REV: Not clear to me. Only because many people have been doing something does not mean it is correct or the most appropriate way to do it.

Nevertheless, in the revised manuscript we removed the use of “flora age”, and replaced it with “species age structure”. See lines 208-211 “*Landform effects on species age structure of floras*”

The flora assembly history differs between mountain flora, which can be reflected in both species composition and species age structure. To estimate such species distinctiveness between landforms,.....”

Reviewer #3 (Remarks to the Author):

Review of the revised manuscript “Landform and lithosphere development drive the assembly of mountain floras in China” by Zhao Wan-Yi et al.

Q1:

I appreciate the efforts by the authors in their attempt to improve the quality of their manuscript. While many, if not most, of my suggestions have been followed, some aspects have not yet been fully addressed and need further attention. Among those are adequate consideration of relevant literature by other authors, importantly with respect to hypotheses and frameworks, especially in the introduction and discussion. I have provided comments and suggestions in the rebuttal and manuscript text files and hope these will be useful to the authors. In addition, I found that especially the newly drafted texts (but not only exclusively those) will require a careful read in terms of use of the English language and terminology to be deemed acceptable.

Response: We accept the reviewer#3's suggestion to further consider the relevant literature for our introduction and discussion, particularly for the section on MGH, geodiversity, and adaptive radiation. In response to the questions raised by the reviewer, we have made careful changes to the entire manuscript. The manuscript has also been carefully grammatically revised. We listed the major responses below, and for more detailed edits, please see the revised manuscript (all changes are marked). The comments on the manuscript are shown as “**Line ??**”, and the comments raised in previous point-by-point response document are mark by “**(Pre-response)**”. The related comments are put together to avoid repeated response.

Q2:

Comment: In addition to my comments included here in this document, I have also provided comments to the rebuttal letter.

I hope my comments here as well as those in the rebuttal will be helpful for the authors to improve their manuscript further.

Response: Thanks for these helpful comments.

Q3:

Line48: I suggest to add studies that have a global scope. E.g. your current reference 16 for plants.

Response: We accept the reviewer's comment to include reference 16, and also Rahbek et al.(2019) “Building mountain biodiversity: Geological and evolutionary processes”, in which the authors provided an overview of the proposed biogeographical roles of mountains (cradles, barriers, reservoirs, museums, graves, etc.).

Q4:

Line53:

This paragraph needs improvement. The facts are not presented clearly enough, and the content does not yet sufficiently pay justice to the state-of-the-art in the field.

Response: We have further revised this paragraph. Please refer to the following paragraph.

A number of hypotheses have been proposed to explain both montane and global biodiversity, such as those pertaining to climate stability, habitat heterogeneity, and energetics. Along latitudinal gradients, current evidence suggests that biodiversity is affected by environmental energetics, in particular potential evapotranspiration (PET) and average annual temperature. At finer scale, plant alpha diversity in some extratropical mountain regions (such as Cape, East Australia) does not substantially differ from that in the tropics. Contemporary climatic regimes are also insufficient to explain the pantropical diversity disparity in Neotropical and Indomalayan moist forests. These results strongly suggest that montane species diversity is largely affected by habitat heterogeneity, or the so-called geodiversity. Furthermore, the evolutionary history of plant species also affects biodiversity. It is clear that an integrated framework for the prediction of montane biodiversity is necessary. **should include includes** ecological processes (e.g., survival, competition, and niche differentiation), biological processes (e.g., species divergence and extinction), and geological and lithologic processes (e.g., orogeny and rock formation). A recent attempt at such a framework, the “mountain geobiodiversity hypothesis” (MGH), was first proposed to explain the biodiversity of the Tibeto-Himalayan region and then extended to explain the origin of montane plant diversity at a global scale. The MGH proposes that the evolution of montane biodiversity results from a combination of mountain-uplift, geodiversity evolution, and Neogene and Pleistocene climate changes.

Q5:

Line55-56: Unclear: which kind of “model” ? These are different papers of different scope and with important findings, but what and where in these can one find the “model”? As currently written, this remains dubious. I assume “model” is not the right word here. The sentence should be re-written.

Response: Yes. The model we are writing here does need to be modified. It should refer to the distribution pattern of biological diversity, the theories, hypotheses, or processes that measure this pattern. Here, the entire paragraph has been revised, referring to the reply in the previous sentence.

Q6:

Line59-60: A “theory” is not exactly the same as a “hypothesis”, I suggest to re-write this. A hypothesis can be tested. The MGH was tested for world-wide applicability by Muellner-Riehl et al. 2019 JBI. This should be mentioned here. I suggest to be more specific here – which aspect of “biodiversity”? The MGH refers not only to standing biodiversity levels, but also how biota and geology and climate evolved in concert, incl. speciation, extinction and

dispersal, which all contributed to yield the current biotic assemblages. See my previous comment. The MGH was originally developed to explain high levels of biodiversity in the THR, and thus it is of relevance to the study here. But in addition, it was later tested on global mountain data. This does not become evident from the text here yet and needs to be added.

Response: We appreciate the reviewer's comments to the third paragraph of the previous manuscript.

We had carefully considered these comments and rewrote the entire paragraph. Please see lines 54-71:

A number of hypotheses have been proposed to explain both montane and global biodiversity, such as those pertaining to climate stability^{Error! Reference source not found.}, habitat heterogeneity^{Error! Reference source not found.}, and energetics^{Error! Reference source not found.}. Along latitudinal gradients, current evidence suggests that biodiversity is affected by environmental energetics, in particular potential evapotranspiration (PET) and average annual temperature^{Error! Reference source not found.}. At finer scale, plant alpha diversity in some extratropical mountain regions (such as Cape, East Australia) does not substantially differ from that in the tropics^{Error! Reference source not found.}. Contemporary climatic regimes are also insufficient to explain the pantropical diversity disparity in Neotropical and Indomalayan moist forests^{Error! Reference source not found.}. These results strongly suggest that montane species diversity is largely affected by habitat heterogeneity^{Error! Reference source not found.}, or the so-called geodiversity^{Error! Reference source not found.}. Furthermore, the evolutionary history of plant species also affects biodiversity^{Error! Reference source not found.}. It is clear that an integrated framework for the prediction of montane biodiversity is necessary^{Error! Reference source not found.}, should include ecological processes (e.g., survival, competition, and niche differentiation)^{Error! Reference source not found.}, biological processes (e.g., species divergence and extinction)^{Error! Reference source not found.}, and geological and lithologic processes (e.g., orogeny and rock formation)^{Error! Reference source not found.}. A recent attempt at such a framework, the "mountain geobiodiversity hypothesis" (MGH), was first proposed to explain the biodiversity of the Tibeto-Himalayan region^{Error! Reference source not found.} and then extended to explain the origin of montane plant diversity at a global scale^{Error! Reference source not found.}. The MGH proposes that the evolution of montane biodiversity results from a combination of mountain-uplift, geodiversity evolution, and Neogene and Pleistocene climate changes^{Error! Reference source not found.}.

Q7:

Line60-62: This sentence is unclear. Which "framework"? Before, the text talks about the MGH, then it says "Within these framework", which probably is supposed to link to the MGH, but the citations 12, 20-21 are not related to the MGH. The sentence needs adjustment.

Response: This refers to the theory proposed by MGH and the framework for calculating biodiversity, and literature is also annotated here. As following:

It is clear that an integrated framework for the prediction of montane biodiversity is necessary^{Error! Reference source not found.}, should include^{Error! Reference source not found.} ecological processes (e.g., survival, competition, and niche differentiation)^{Error! Reference}

source not found, **biological processes** (e.g., species divergence and extinction)^{Error! Reference source not found.}
^{Error! Reference source not found.}, and geological and lithologic processes (e.g., orogeny and rock formation)^{Error! Reference source not found.}^{Error! Reference source not found.}. A recent attempt at such a framework, the “mountain geobiodiversity hypothesis” (MGH), was first proposed to explain the biodiversity of the Tibeto-Himalayan region^{Error! Reference source not found.}^{Error! Reference source not found.} and then extended to explain the origin of montane plant diversity at a global scale^{Error! Reference source not found.}. The MGH proposes that the evolution of montane biodiversity results from a combination of mountain-uplift, geodiversity evolution, and Neogene and Pleistocene climate changes^{Error! Reference source not found.}^{Error! Reference source not found.}
^{Error! Reference source not found.}

REV: Ecology is a branch of biology, authors may rather use “evolutionary processes” than “biological processes”.

Q8:

Line69:

“Although climate change force plant species migration is well known, those specific species

I regard my comments to the original draft of the manuscript not sufficiently considered here. If the text is written in a general manner as here, i.e. not specifically addressing Chinese mountains, then mountain literature with topics of relevance to the study here, but which were done in other regions of the globe, need representative consideration to reflect the state of knowledge.

Response: Many thanks for the feedback, we now listed several other relevant studies from other regions, which have reported that the bedrocks also have significant impact on the diversity of mountain plants.

Please see lines 82-88. “Although it is well known that climate change forces plant species migration, those species which are adapted to local bedrocks are constrained in their ability to migrate^{Error! Reference source not found.}. **Bedrock geochemistry is on par with climate as a regulator of vegetation in granitic mountains**^{Error! Reference source not found.}. Some studies also suggest that local species diversification processes are consistent with edaphic rather than climatic filtration, **such in the Cape flora**^{Error! Reference source not found.}, **Teesdale flora**^{Error! Reference source not found.}, and **New Caledonian flora**, in which ca. 50% of the endemic floristic elements are ultramafic-obligate species^{Error! Reference source not found.}.”

REV: I am unhappy with the statement that geochemistry, which is a field of research, is a regulator – authors needs to take care with precision in their statements.

Q9: Line90:

“Here, we apply the term “flora” to refer to the sum of all angiosperm families, genera, and species growing on a specific mountain or in a well-delimited area^{Error! Reference source not found.}^{Error! Reference source not found.}, which is a relatively independent biogeographical unit^{Error! Reference source not found.}.”

I would argue against the flora of a mountain being a *relatively independent biogeographical unit*, if this is what is meant here (but I may have misunderstood the sentence). First, because related lineages are shared between mountains, and second, because most biogeographers, in terms of terminology, would not consider the flora of a mountain as a “biogeographical unit”.

Also in **Pre-response**: “we mentioned is not only a unit, but als” I am not sure how “unit” here

would refer to the biogeographic units mentioned in the ms. This may be worth of clarification.

Response: We appreciate the key question raised by the reviewer. In generally, the term “biogeographic unit” is used in the context of biogeographic regionalization.

Here we consider that the flora of mountain and **biogeographic zones** is equivalent at small scale.

REV: This is unclear. Which biogeographic zones are meant here?

First, **mountain is geographical barrier to plants diffusion.**

REV: This is not correct (as I had stated in the previous rounds of review already), phrased in this general manner. It depends on the orientation of the mountain. Only if a mountain has W-E orientation, it can act as barrier (e.g. Himalayas, Alps). For N-S-oriented mountains (such as the Andes, or the Hengduan mountains), this is not correct.

Second, **a mountain usually shows sky island effects** and contain their own unique clades or several endemic species.

REV: This is not correct. First, there is no such thing like a “sky island effect” (phrased this way). Second, only a small fraction of the world’s mountains qualify as sky islands – only, if their vegetation is drastically different from the surrounding lowlands. This may only be the case if plants living on the mountains are different from the lowlands and were therefore not likely recruited/evolved from lowlands in which case the nearest relatives would come from other mountains (example: some mountains located in drylands, e.g. Africa).

There are also some literature to suggest that mountain region could be viewed as a biogeographical unit. **For instance, Rahbek et al. (2019, Science 365, 1108-1113) wrote that “Like an island, a mountain region may be viewed as a biogeographical unit in itself, with in situ speciation and extinction playing a key role in building the regional species assemblage”.**

REV: Is anyone else except this cited paper claiming this as well? I could imagine that most mountain bioeographers would disagree with this oversimplification.

Q10: Line95:

“Based on the phylogeny, we calculated the phylogenetic diversity, phylogenetic structure (relatedness clustered or overdispersed), and mean divergence times for the 140 floras.”

I would like to see some justification for this approach mentioned in the manuscript. I raised the issue of calculating species ages, as done here, in my review of the first manuscript draft. At the very least, a note of caution needs to be added at the appropriate space in the manuscript, explaining the shortcomings and limitations.

And also in **Pre-response:** *“In our study, flora age differences were compared by MDT (average age of extant species in the mountain flora, MDT). The MDT is a index to assess the relative age of modern flora (Lu et al. 2018).....”*

As also mentioned in my comments to the manuscript, the latter needs to include some statement about the limitations and shortcomings of this approach.

Response: We accept this suggestion by the reviewer. The mean divergence times (MDT), mentioned in the text, is used to measure the relative age of a flora which is proposed by Lu et al.

(2018). We have now added a brief explanation about the shortcomings and limitations of MDT in the methods.

Please see lines 490-498 “The divergence time of each species used to calculate MDT is not absolute age, as they were extracted from the megatree generated in V.PhyloMaker2⁴⁹. We tested the robustness of this method by using four divergence time datasets and found similar MDT patterns between mountain of different landforms (Extended Data Fig. 8). This result is consistent with¹¹⁰, who found that “in large-scale biodiversity and phylogenetic analyses, sources of noise in divergence time estimation are to be expected, but they did not affect the reliability of the results”. We believe that our dated megaphylogenetic tree was suitable for this study because our aim was to reveal the general patterns of landform influence on the formation of mountain flora, rather than focusing on the age of each species.”

Q11: Line291:

“development process is associated with local bedrocks and regional climate^{Error! Reference source not found.}, thus, the local flora assemble is a comprehensive result of species evolution, landform development and climate change^{Error! Reference source not found..Error! Reference source not found..Error! Reference source not found.}. However;”

This list contains various aspects, biotic and abiotic, which are interrelated. According to the MGH, dispersal of pre-adapted lineages and extinction are equally important. Also, the order “species evolution, landform development and climate change” may be reconsidered.

Response: We have added the following content in lines 311-313 “~~Landform, developmental processes are associated with local bedrocks and regional climate^{17,59}.~~ ~~The assembly of the local flora is impacted by a combination of geological and climatic processes, and biological processes.”~~

REV: This sentence is devoid of logic; the first matters mentioned do not automatically cause the second. The sentence as such is therefore not correct. I suggest to re-phrase this – as corrected above (using tracked changes).

Q12: Line295:

“The underlying reason is that water cycle process and rock erosion rate differs between igneous and sedimentary mountains ecosystem. As is well known water in limestone mountains can easily be lost through underground river systems, while more overland runoff water is more available for plants in mountains of igneous bedrock. In extreme case,”

As for the introduction, here in the discussion previous work by other authors (citations) should receive appropriate consideration.

Response: We accept this advice and supplemented corresponding literature in the manuscripts. See lines 316-320 “The underlying reason is that water cycle processes and rock erosion rates differ between igneous and sedimentary mountain ecosystem⁷². As is well known, water in limestone mountains can easily be lost through underground river systems⁷³, while a greater quantity of overland runoff is available for plants in mountains of igneous bedrock.”

Q13: Line302:

“In adaption to mountain phylogenetic structure of flora and mean divergence times (MDT) of flora,”
Meaning of sentence not clear, needs re-phrasing.

Deleted: Because l

Deleted: s

Deleted: ,

Deleted: t

Deleted: results

Deleted: from

Response: Thank you for your assessment. We revised this sentence in lines 323-324 “Based on comparisons between mountain flora phylogenetic structure and MDT, we found that landform effects partially determine the final floristic composition.”

Q14: Line313:

“The mountains of sedimentary bedrock have much stronger environmental filtering effect, as these mountain ecosystems are more sensitive to rainfall. The environmental filtering effect further promoted the clustering of phylogenetic structures of mountain floras as predicted by the phylogenetic niche conservatism (PNC)^{Error! Reference source not found..Error! Reference source not found..Error! Reference source not found.}”

It would be interesting to discuss here radiations more generally, e.g., how do the studies of plant radiations included in the global study by Muellner-Riehl et al. 2019 (JBI) compare to these statements/findings? Are the findings of these studies (and other studies published since then, i.e. after 2019) comparable to this? I suggest looking at the mountain systems (and their bedrocks) where these studies were undertaken.

And **Pre-response:** “Second, landform effects (as environmental filtering) restrict the free dispersal of plants between mountains of different landform types (because edaphic species cannot exist outside the original bedrock, dispersal event could occurs with phylogenetic nich evolution).....”

The meaning of this sentence it not entirely clear to me. I assume what is meant here is that if individuals of a species which are adapted to a specific soil type happen to be dispersed to areas of another soil type, in order to survive, they would have to adapt, and as a result, may become evolutionary independent lineages, potentially new species? This is actually less likely than if their propagules dispersed to areas of the same soil type, which may foster a simpler type of allopatric speciation.

And **Pre-response:** I don’t understand this sentence. How could “niche evolution” possibly promote “the spread” (dispersal?) of species? Which niche evolution do the authors mean here, on which taxonomic level? This needs to be clarified. As currently phrased, this sentence does not make sense and I don’t get the meaning.

Response: These are really good advice. The potential radiation speciation occurs when plants colonize from one landform to another which maybe an important path of plant diversification. Here, we document several cases (not all) on adaptative radiation that occur on the bedrocks. But the relations between radiations and bedrocks/landform are still poorly known. We suspect that the effect of landform and bedrock, which promote species differentiation, is more or less neglected under the shadow of “climate change”. Nevertheless, we have reorganized this paragraph carefully. **REV:** I don’t understand the reasoning here, as currently phrased. It is unclear what the authors mean (potential speciation radiation?). There exist different kinds of radiations, not all of them are “adaptive”.

Please see line 334-348. “Mountains composed of sedimentary bedrock have much stronger environmental filtering effects, as these mountain ecosystems are more sensitive to rainfall. Such effects further promoted the clustering of the phylogenetic structures of mountain floras, as predicted by phylogenetic niche conservatism (PNC)^{Error! Reference source not found..Error! Reference source not found..Error! Reference source not found.}. The transformation of landforms effectively promotes the plant

radiation, such as the Old World gesneriads and North American deserts rock daisies (Compositae tribe Perityleae). Similar patterns have been observed in insects. For example, two radiation clades (clade nodes 14 to 16 and nodes 31 to 33) of Exocelina resulted from the transition from uplifted Australian Plate rock to ultramafic/ophiolite. Although, many previous studies also document climate change as an important driver of species radiations.

REV: This text needs re-writing, major parts are not understandable. “clustering of the phylogenetic structures”? Authors don’t explain what the connection is between PNC and phylogenetic clustering. “transition”? formulation and meaning unclear.

The adaptive radiations of plant genera are more or less associated with bedrock type. For example, the key innovation (lime-secreting hydathodes) of Saxifraga sect. Porphyron is clearly an adaptation to the limestone bedrock, and the low specific leaf area (SLA) exhibited by Erica is an adaptation to oligotrophic habitats (Quartzite/sandstone) in Cape. Thus, we suggest that landform and bedrock effects strongly promote species differentiation between different regions, even if such an effect has not been mentioned in previous studies.

REV: “The adaptive radiations of plant genera are more or less associated with bedrock type.” If at all, adaptive radiations can, among others factors, be more or less associated with bedrock type...

The entire text section needs an English language check (grammar, incomplete sentences,...).

different regions, even if such an effect has not been mentioned in previous studies.” – this is not correct. There is a long history of investigations in the European Alps on this matter, dating back to the mid of the 20th century.

“the key innovation (lime-secreting hydathodes) of Saxifraga sect. Porphyron is clearly an adaptation to the limestone bedrock.” Authors needs to carefully read and describe findings in literature, not overstating findings. For example, to my knowledge, some Saxifraga species with lime-secreting hydathodes do actually not grow on limestone – therefore, the sentence as currently phrased does not hold true for all lineages. Care needs to be taken with re-phrasing.

Q15: Line 317-327:

“Dispersal also contributes to the mountain diversity. A well-known example is climatic fluctuations during the Quaternary ice age drove changes in the distribution of species around the globe. The spatial configuration of mountain range affect their functional connectivity, influencing species dispersal. In fact, dispersal more easily occurs at initial stage of landform develop mountain, as the slope of mountain still gentle and geographical barrier still relatively low. Such as the plants dispersal events in Qinghai-Tibet Plateau (QTP, low barrier) is obvious higher than that in Hengduan Mountains (high barrier). Furthermore, the effects of dispersal on species diversity in mountainous areas mainly occurred at lowland, and is limited at highlands where has more local endemic species. This means that dispersal has weak influence on the unique composition of mountain flora of different landforms,

especially considering that landform have a significant filtering effect on the species.”

The entire paragraph needs re-writing. Dispersal between mountains is not limited to times of climate change, which somehow is suggested here as currently written. Dispersal and establishment was easier in the Hengduan mountains at times of climate change as they have a N-S orientation, enabling plants to change their latitudinal distribution range more easily.

And Pre-response: “*Such as the plants dispersal events in Qinghai-Tibet Plateau (QTP, low barrier) is obvious higher than that in Hengduan Mountains (high barrier)*”⁴⁴.

I don’t understand this sentence. As outlined previously, it is exactly the other way round. Dispersal and establishment was easier in the Hengduan mountains at times of climate change as they have a N-S orientation, enabling plants to change their latitudinal distribution.

And Pre-response: As I pointed out previously, the opposite would be expected. Survival on mountains with N-S orientation is easier during glacials than on those of W-E orientation (e.g. compare Himalayas versus Hengduan Mts.).

And Pre-response: “*This means that dispersal has weak influence on the unique composition of mountain flora of different landforms, especially considering that landform have a significant filtering effect on the species.”*

Reading this, I am not sure the authors have read the papers I had suggested to consult, and which they cited in the paper already before (e.g. Ding et al.).

Response: We accept this advice and wrote this paragraph. We would like to clarify the questions about dispersal events and diffusion barrier. We had carefully readed the paper by Ding et al (2020). It is not difficult to deduce the conclusion “*dispersal events in Qinghai-Tibet Plateau (QTP, low barrier) is obvious higher than that in Hengduan Mountains (high barrier)*” (see Ding et al, 2020, figure 2). On the other hand, we thought dispersal would reduce the β -diversity between different floras. Such as the proportion of endemic species in Hengduan Mountains is the highest is largely affect by its low rate of colonization (<0.05), and **high rate of *in situ* speciation and local and recruitment**.

REV: This could also/additionally be the effect of higher levels of extinction on the QTP, in contrast to the Hengduan mountains.

In contrast, the low proportion of endemic species in Qinghai-Tibet Plateau has the highest rate of colonization since early Miocene. It is clearly the dispersal is negative correlation to the proportion of endemic species in flora.

REV: I am not pleased with this entire section of text (incl. the one further above), as the main points of criticism still remain to be addressed.

The re-wrote paragraph please see lines 349-367. “*Dispersal also contributes to montane floristic diversity*,^{Error! Reference source not found.} *although dispersal is only effective if it is followed by successful establishment*^{Error! Reference source not found.} *. Dispersal occurs more freely during the initial stages of landform development in mountains, when slopes are still gentle and geographical barriers are minimal*^{Error! Reference source not found.} *. For example, the plant dispersal events across the Qinghai-Tibet Plateau (low barrier) have been considerably greater than that in Hengduan Mountains (high barrier)*^{Error! Reference source not found.} *. The role of local recruitment was most important during the early stages of mountain flora assembly, a role which was subsequently replaced by local adaptations or in situ speciation*^{Error! Reference source not found.}

found. Error! Reference source not found. Mountains in different landforms will recruit different plant species, because each species is differently adapted to specific types of bedrock *Error! Reference source not found., Error! Reference source not found.* The **spatial configuration** of mountain ranges alters their functional connectivity, thus influencing the species dispersal process *Error! Reference source not found.-Error! Reference source not found.* Furthermore, during the middle stages of landform development, the effects of dispersal on species diversity in mountainous areas are primarily restricted to lowlands, and are limited in highlands where there are more local endemic species *Error! Reference source not found., Error! Reference source not found.* Taken together, this suggests that dispersal only weakly influences the unique composition of mountain flora associated with different landforms, especially considering that landforms exhibit a significant filtering effect. Variation in the species composition of mountain floras between different landforms will increase under the significant, combined landform environment filtering effect *Error! Reference source not found.-Error! Reference source not found.*, and local endemic speciation *Error! Reference source not found.* Therefore, we propose that the patchy spatial distribution of different mountain landforms is an important factor in shaping surface biogeographical zoning.”

REV: Again, this text’s meaning is still largely unclear.

Qinghai-Tibet Plateau (low barrier) have been considerably greater than that in Hengduan Mountains (high barrier) – which kind of barrier do the authors means – and barrier to what?

Role of local recruitment: In times of climate change, local recruitment from lowlands may play a potentially important role.

“Spatial configuration” – meaning unclear.

“Furthermore,” – but this disregards climatic fluctuations acting on mountains at various times.

Q16: Line332:

“Thus, landforms are suitable indicators of geodiversity *Error! Reference source not found.-Error! Reference source not found.* for tracking changes in mountain floras along geological time scales.”

I am missing more elaboration on measure of geodiversity. E.g., how does this compare to the geodiversity indices used in previous mountain studies (such as the MGH global study by Muellner-Riehl et al. 2019 in JBI)?

Again, the lack of discussion of work by other authors is misleading and suggests more novelty here than is actually inherent in this present study. Acknowledging previous work by other authors does not diminish the achievements of this study here, but rather empowers readers to compare this and previous studies.

And **Pre-response:** But geodiversity as a measure (geodiversity index) is an integral part of investigations under the MGH.

Response: Thanks for pointing out the shortcomings. There is no denying that linking geological diversity to biodiversity is an important step. However, there are still some problems in the application of geodiversity. In previous studies, geodiversity is usually treated as the environment variables integrated into the compound geodiversity index (GD, or GDCs). But the power of compound geodiversity index was not good enough in predicting species diversity. As the case you mentioned (Muellner-Riehl et al. 2019 in JBI, wrote “*The GD index and Elevational range were strongly correlated ($r = .70$) and given that Elevational range showed better performance in single predictor models...*”), the geodiversity index (GD) was found to be less effective than a single topography variable (elevational range) for predicting mountain species diversity. In contrast, the landform type (characteristic variable) is more concise and easier to understand. For a detail

discussion, please see line 372-388.

“Geodiversity, the variability of Earth’s surface materials, forms, and physical processes, is an integral part of nature and crucial for sustaining ecosystems and their services. Several studies proposed to use geodiversity to model biodiversity based on the hypotheses that specific geo-sites should support unique biota and that high geodiversity is coupled with high biodiversity. However, the predictive power of geodiversity has not been high. For example, the geodiversity index (GD) was found to be no better than elevational range for predicting mountain species diversity. Another recent study reported that combinations of environmental variables better explained tropical species diversity than GD. In fact, the contribution of geodiversity component (GDC) showed similar patterns to topography, and usually effects at smaller extents and finer grain size. The GD is a poor predictor of mountain diversity in large part because interactions between mountains and biodiversity are complex, and thus geodiversity could not be treated as an index. In contrast, landform type is an objective description of the present state of erosion of a mountain or bedrocks. When we stand in front of a karst mountain, we can see quite clearly that the plants are different from those in a granite mountain. We propose that landform identity rather than GD, is a more suitable indicators of geodiversity for tracking changes in mountain floras along geological time scales.”

REV: I see a lot of problems with this text, for various reasons:

- Basically none of the sentences is clearly and understandably formulated
- scientific terms are not properly used
- the text, in parts, is misleading and, in parts, contains wrong claims

To illustrate this, I have above left some comments – many more would have been possible. I hope the authors will see what I mean when commenting about the necessity to improve this text considerable.

Q17: Line334:

“Based on our results, we put forward the “geological lithology hypothesis of flora” to explain the assemble and differentiation of mountain floras. In this theoretical framework, floristic assembly in mountains is driven by the lithospheric cycle, which refers to the bedrock-constrained developmental processes of landforms. Specifically, under this hypothesis, the mountain species differentiation closely related to the type of bedrock and degree of erosion, species richness and species composition in mountain flora are interaction result of landform and environment, and phylogenetic niche evolution can promote the spread of species among different landform floras. Overall, our study provides a novel framework and approach for determining the mechanisms of species diversity within mountains and the distributional patterns of some of the world’s richest floras.”

This section needs to be improved and provide a more balanced view on previous work by other authors and suggestions put forward here. As I had already suggested in my review of the first manuscript draft, the manuscript here needs to go into some more depth concerning previous hypotheses put forward, importantly the MGH, and I still don’t see this has been done. While only briefly mentioned in the into, the MGH does not show up here in the discussion. I would like to see some of what the authors have answered in their rebuttal be actually also included here in the manuscript text.

Commented [A1]: wrong

Commented [A2]: ?

Commented [A3]: This statement is wrong. There are plenty of studies in the literature showing the opposite. This does not pay justice to the state-of-the-art knowledge.

Commented [A4]: There is not “the” one GD, but there are different methods used in literature for estimating geodiversity (incl. remote sensing approaches more lately)

Commented [A5]: This is a strongly misleading formulation. Both perform equally good.

Commented [A6]: Meaning unclear – re-formulate. GD is usually a combination of environmental variables.

Commented [A7]: Not clear – which pattern?

Commented [A8]: No, it is not, as has been shown in previous studies.

Commented [A9]: Geodiversity measures are supposed to capture a considerable proportion of this complexity.

Commented [A10]: It can and has been treated as an index in some studies, and in other works, single parameters were used. Thus, the sentence, as formulated, is wrong.

Commented [A11]: Why in contrast?

Commented [A12]: Geodiversity is also “objective”

Commented [A13]: This is not new and is basic textbook knowledge. E.g. compare works in the Alps from the mid 20th century and thereafter.

Commented [A14]: Part 1 of the sentence – refers to patterns, current;

Commented [A15]: This refers to timing – but what is the connection to the first part of the sentence? This is not clear.

Response: This is a great suggestion. We rewrote this section. Please see line 389-413.

“Based on our results, we propose the “geological lithology hypothesis of flora” to explain the assembly and differentiation of mountain floras. In this theoretical framework, floristic assembly in mountains is driven by the lithospheric cycle, which refers to the bedrock-constrained developmental processes of landforms. Specifically, under this hypothesis, montane species differentiation is closely related to the type of bedrock and degree of erosion. Both SR and species composition in mountain flora result from interactions between the landform and the environment. In addition, the dispersal of plants between different landform types is more restricted than within the same landform type. Successful diffusion across a landform is often accompanied by the emergence or radiation of adaptive traits. This is called phylogenetic niche evolution^{Error! Reference source not found.} and can promote the spread of species among different landform floras. This differs from the MGH, which assumes that the montane biodiversity hotspots require three key boundary conditions: 1) the presence of lowland, montane and alpine zones, 2) climatic fluctuations to produce a “species pump” effect, and 3) high-relief terrain with environmental in a given mountain region^{Error! Reference source not found.}. In contrast, our hypothesis suggests that montane bedrocks and landform processes determine the geographic distribution of plants. The MGH effectively explains the high biodiversity of mountains characterized by large elevational differences (such as the Himalayan and Andes mountains), but such restrictive boundary conditions constrain the applicability of the hypothesis. Biodiversity hotspots also occur in regions with stable climates, such as the Namib desert^{Error! Reference source not found.}, or with with minimal elevational gradients, such as the Southeast Asia karst landforms^{Error! Reference source not found.}. We argue that the novel “geological lithology hypothesis of flora” could serve as a general explanation for global diversity patterns, as the formation of mountains on the Earth’s surface is the result of the cycling of sedimentary, igneous and metamorphic rocks. In conclusion, our study has highlighted the floristic patterns of different landforms and provided a novel framework for studying the mechanisms of plant species diversification within mountains and the distributional patterns of mountain floras of the world.”

REV: Similar to what I mentioned for the text above does also apply here. The authors need to pay attention to content correctness (e.g., when referring to the MGH, and comparing to their hypothesis), precision in formulation and use of terminology (e.g. concerning PNC) and claims. In addition, it appears to me that especially in the last third of the text, different topics are being confused/intermixed. The MGH refers to mountains, but the authors talk about the “Namib desert” and “SE Asian karst landforms” for which the MGH would not apply anyway. They also talk about “global diversity patterns” when referring to their hypothesis. The MGH specifically deals with mountains, the hypothesis by the authors is supposed now to refer to global phenomena? If so, do the hypotheses actually deal with different matters? I strongly encourage the authors to carefully revise the text, and stick to what the hypotheses were developed for. Also, point in time geographic distribution of plants, and processes acting in concert, appear to be compared despite being different. As such, the text is not clear for the reader.

Q18: Line378:

“For the 236 genera missing, we treated each as sister its most closely related genus in “GBOTB.extended.LCVP.tre” based on megaphylogenies within other references^{Error! Reference source not found.}”

I suggest to provide at least some information about the limitations of this approach.

Response: Thank you for your assessment. A brief introduction about the limitations of this approach is provided in methods.

Please see lines 444-450 “*Of the 2,585 genera and 17,576 species studies here, 2,349 genera and 8,663 species were included in “GBOTB.extended.LCVP.tre”. Based on previously-published megaphylogenies^{50,101}, we treated each of the 236 missing genera as sister to their most closely related genus in “GBOTB.extended.LCVP.tre” using V.PhyloMaker2⁴⁹. Although this method resulted in more robust phylogenetic relationships than Phylocom¹⁰², the ultimate phylogenetic relationships should still be considered relative, as complete phylogenetic data are still lacking for many families and genera.*”

Q19: Line417: Divergence time estimation

I suggest to provide at least some information about the limitations of this approach.

See also my previous comment further above: I would like to see some justification for this approach mentioned in the manuscript. I raised the issue of calculating species ages, as done here, in my review of the first manuscript draft. At the very least, a note of caution needs to be added at the appropriate space in the manuscript, explaining the shortcomings and limitations.

Response: We accept this comment and we have provided justification for the approach in lines 490-498 “*The divergence time of each species used to calculate MDT are not absolute age, as they were extracted from the megatree generated in V.PhyloMaker2⁴⁹. We tested the robustness of this method by using four divergence time datasets and found similar MDT patterns between mountain of different landforms (Extended Data Fig. 8). This result is consistent with¹¹⁰, who found that “in large-scale biodiversity and phylogenetic analyses, sources of noise in divergence time estimation are to be expected, but they did not affect the reliability of the results”. We believe that our dated megaphylogenetic tree was suitable for this study because our aim was to reveal the general patterns of landform influence on the formation of mountain flora, rather than focusing on the age of each species.*”

The response to other comments raised in previous response letter is below.

Q20:

Pre-response: “*Because plants are more dependent on local rocks and soils, and less responsive to climate than animals....*”

I don't agree. Plants can't move and therefore are expected to show even stronger response.

Response: We agree with your point. In fact, we don't think our “less responsive” and “show even stronger response” are in conflict. Because the former is a timely response to climate change, while the latter is a relatively delayed result. Anyway, plants are more dependent on the regional environment (especially the bedrocks and soil) than animals.

Q21:

Pre-response: Geodiversity as a measure captures this, and as such, the MGH inherently includes

this aspect, which should be acknowledged in the manuscript. The intro text needs to go more into some more detail concerning the similarities and differences between the MGH and the approach proposed here. It is important to carve out the further refinements of the approach here for the reader. This will not diminish the accomplishments of the authors, but rather help them to put their work in context. Otherwise, the text will imply more novelty than justified.

Pre-response: I would have appreciated an answer which more directly reflects what was actually done in the new version of the manuscript to satisfy the criticism. This is also true for the answers to the other questions/remarks by reviewer 3. What was asked here was that the intro does not reflect the body of literature sufficiently. Just adding two sentences, as indicated further below at the end of the answer does not do justice to what is known about geobiodiversity.

Pre-response: *“Among which, a representative theory is “mountain geobiodiversity hypothesis” (MGH), which proposed to explain the biodiversity of Tibeto-Himalayan region2.”*

But it applies not only there. While it was originally developed for the THR, later work by Muellner-Riehl et al. 2019 in JBI tested its global validity.

Response: These comments are about geodiversity and associated MGH. Revision associated with this comments in new manuscript is in line 62-63 *“These results strongly suggest that montane species diversity is largely affected by habitat heterogeneity¹³, or the so-called geodiversity¹⁸⁻¹⁹. ”*

Line 68-72 *“A recent attempt at such a framework, the “mountain geobiodiversity hypothesis” (MGH), was first proposed to explain the biodiversity of the Tibeto-Himalayan region^{2,29} and then extended to explain the origin of montane plant diversity at a global scale³. The MGH proposes that the evolution of montane biodiversity results from a combination of mountain-uplift, geodiversity evolution, and Neogene and Pleistocene climate changes^{3,29}.”*

And line 373-384 *“Geodiversity, the variability of Earth’s surface materials, forms, and physical processes, is an integral part of nature and crucial for sustaining ecosystems and their services¹⁸. Several studies proposed to use geodiversity to model biodiversity based on the hypotheses that specific geo-sites should support unique biota and that high geodiversity is coupled with high biodiversity^{2,3,18,19}. However, the predictive power of geodiversity has not been high. For example, the geodiversity index (GD) was not found to be more effective than elevational range for predicting mountain species diversity³. Another recent study reported that combinations of environmental variables better explained tropical species diversity than GD⁹². In fact, the contribution of geodiversity component (GDC) showed similar patterns to topography, and usually effects at smaller extents and finer grain size⁹³. The GD is a poor predictor of mountain diversity in large part because interactions between mountains and biodiversity are complex²⁴, and thus geodiversity could not be treated as an index.”*

Q22:

Pre-response: *“Therefore, we believe that the analysis in this paper should have different meanings compared with the previous analysis results.”*

I don't regard this as really convincing argumentation.

Response: Thanks for this insightful criticism. The key topic of mountain biodiversity is to reveal the reason of unusually rich beyond latitudinal diversity gradient. We speculate that the landform pattern is differs from latitudinal diversity gradient, and landform identity represents a new approach to understand the assembly and differentiation of mountain floras. We further clarified our views in

the new manuscript, please see line 369-414.

Q23:

Pre-response: “Yes, the research carry out by Antonelli et al. (2018) is quiet an important in mountain species diversity. We cited this literature in the introduction and other part of our manuscript, please see L56, L65, L173, L234.”

My comment had not addressed the citation being missing, but I ad requested an earlier mention of it. Preferably, the answer in a rebuttal letter should directly refer to what has been asked for. Here, it does not become evident from the text straight forwardedly, whether this is the case.

Response: Thanks for this comments. The manuscript has been extensively revised. In the currently manuscript, the research carry out by Antonelli et al. (2018) is references 24.

Q24:

Pre-response: “Thanks for the reviewer's criticism. After a long period of field investigation, we realized that the flora composition in mountains is different among different landforms.”

Was this studied here? I need to re-check whether the (tax.) composition of the flora was investigated.

Response: The basic data of this study mainly based on published flora checklist (see Extended Data Table 1, and Additional information), in which 12 mountains (reference 129, 144, 161, 163, 165, 171, 173, 201, 203, 209, 219, 264) are based on our previous field investigation.

Q25:

Pre-response: What do the authors consider as “younger” species?

“The “younger” species represent the species that diverged later in the mountain flora. To avoid potential bias between mountains, we ranked all species occurs in 140 mountains from youngest to oldest, partitioned them into quartiles based on their ages, computed MDT in each mountain for the absolute youngest 25% and the absolute oldest 25% of species (Lu et al. 2018).”

This does not answer my question as intended. I was interested in absolute ages.

Response: Thank you for your concerns. We think this question is similiar to the third question raised by reviewer#2. Unfortunately, as far as we know, there is unable to determine the “absolute age” of a species. In general, the age of a species is based on the branch in the phylogenetic tree, which estimate based on the molecular clock hypothesis and fossil calibrated (but still not “absolute ages”). The “younger” species represents the species with shorter branches in phylogenetic tree.

Q26:

Pre-response: “To be specific, bedrocks and associated micro-landforms in mountains is the most important factor promote speciation local endemic species. ”

It may be argued that this is not correct, i.e. it may be disputed that bedrock and micro-landforms are THE MOST IMPORTANT factor to promote the evolution of local endemics. Mountain orientation plays a key role. Comparing mountains of different orientation (W-E versus N-S) shows that those oriented N-S harbor a higher no. of species that were able to survive during times of climatic change and that were able to form endemic lineages than those oriented W-E. This is because a W-E orientation prevents populations moving into more favourable latitudes during

times of climate change (barrier effect) that has happened in the past. A N-S orientation, in contrast, as e.g. found in the Hengduan Mountains or the Andes, enabled populations to move into more favourable latitudes.

Response: That's a good advice which further promotes our thinking on the role of diffusion processes in the assemble of endemic element in mountain biodiversity. The reviewers point out the endemic species in mountain may mainly derived from migration during periods of climate fluctuation. Endemic plants in this situation can also be called climate relict species. We do not deny the contribution of climate relict species to local endemic. While, dispersal is only effective if it is followed by successful establishment. In our view, successful migration events more easily in mountain with similar landform type (as the Hengduan Mountains) than different landform type. We are more interested in the differences in mountain flora of different landform types. According to your opinion, the section dispersal and conclusion has been rewritten (please see line 350-414).

Q27:

Pre-response: *“Second, landform effects (as environmental filtering) restrict the free dispersal of plants between mountains of different landform types (because edaphic species cannot exist outside the original bedrock, dispersal event could occurs with phylogenetic nich evolution).....”*

The meaning of this sentence it not entirely clear to me. I assume what is meant here is that if individuals of a species which are adapted to a specific soil type happen to be dispersed to areas of another soil type, in order to survive, they would have to adapt, and as a result, may become evolutionary independent lineages, potentially new species? This is actually less likely than if their propagules dispersed to areas of the same soil type, which may foster a simpler type of allopatric speciation.

Response: Thank you for your concerns. We think this is more or less a neglected research topic in previous studies. Differences in soil and bedrock have been suggested in some studies to promote species differentiation, such as Savolainen et al.(2006: doi:10.1038/nature04566). For more details, please see line 336-349 *“The transformation of landforms effectively promotes the plant radiation, such as the Old World gesneriads⁷⁹ and North American deserts rock daisies (Compositae tribe Perityleae)⁸⁰. Similar patterns have been observed in insects. For example, two radiation clades (clade nodes 14 to 16 and nodes 31 to 33) of Exocelina resulted from the transition from uplifted Australian Plate rock to ultramafic/ophiolite⁸¹. Although, many previous studies also document climate change as an important driver of species radiations^{3,62,82,83}. The adaptative radiations of plant genera are more or less associated with bedrock type. For example, the key innovation (lime-secreting hydathodes) of Saxifraga sect. Porphyron is clearly an adaptation to the limestone bedrock⁸⁴, and the low specific leaf area (SLA) exhibited by Erica is an adaptation to oligotrophic habitats (Quartzite/sandstone) in Cape⁶⁷. Thus, we suggest that landform and bedrock effects strongly promote species differentiation between different regions, even if such an effect has not been mentioned in previous studies.”*

Q28:

Pre-response: *“Moreover, our hypothesis also tries to explain the reason why each mountain has high or low plant diversity, or suitable plant diversity.”*

Second, This is, in the core, not different from the MGH (compare Muellner-Riehl et al. 2019), but it adds an additional component.

Response: Thanks for this comments. Our hypothesis “geological lithology hypothesis of flora” more focus in the species composition differences of mountain flora. We emphasize the dominant role of geological processes in the formation of species diversity.

Q29:

Pre-response: “*The period of rapid landform process in modern China coincides with the period of rapid plant species divergence.*”

But has this been actually rigorously tested, or there is just temporal coincidence?

Response: To our knowledge, there are no such a test study. However, the correlation between species divergence and Cenozoic plate tectonic change in East Asia is a well reported pattern. This is a valuable research direction and may test in future.

Q30:

Pre-response: “*For example, most floras of China, including both mountains and lowland floras, diverged during the Miocene when the East Asian monsoon intensified*^{38,52}. ”

“floras” don’t “diverge”. I would argue against mixing vegetational terms and evolutionary terms. Divergence in an evolutionary sense would not apply to “floras”. The sections also needs some general language improvement.

Response: We accept this advice. The revised sentence in line 255-256 “*For example, most Chinese floras, including both montane and lowland floras, differentiated during the Miocene when the East Asian monsoon climate intensified*^{50,51}. ”

Q31:

Pre-response: “Here, we’re not focus on specific mountains. We want to point out that the age of local flora in different regions is related on how long the local landform process, even between continents. Chen et al. (2018),”

Some of what is explained by the authors in the following text may be viewed useful for readers to better understand the reasoning behind this study. If possible, I suggest to incorporate some of the aspects in the intro and discussion.

Response: We revised this section, Please see line 258-265 “*At the intercontinental scale, the ages of floras were largely consistent with regional landform developmental processes. For example, the floras of eastern Asia are typically younger than those of South Africa and Australia, where the landform processes are fairly old, but older than those of the Andes and Amazonia, where landform processes have occurred more recently*^{1,57,66}. Thus, the relationship between the floristic assembly of mountains and the type of landform and landform developmental process seems to be global in scope, at least for angiosperms. Therefore, landform identity may make a suitable indicator for exploring the floristic assembly of mountains.”

Thank you very much for your effort reviewing our manuscript and for your positive comments. We have revised the manuscript following your suggestions, which has led to a substantial improvement of the manuscript. The new manuscript has also been carefully grammatically revised by the Springer Nature Author Services. We provided point-by-point responses to your comments as following. Our new response is in red colour "RESPONSE: ".

Reviewer #2 (Remarks to the Author):

The manuscript is much improved from the last round and reads well generally. I also very appreciate the responses and revisions with regard to my comments. In particular, the interaction terms added interesting perspectives to the data and actually strengthen the paper. It's a lot of results to report here but the overall message is strong: landforms are important factors of floral diversity. The other issues with phylogenetic analyses seem more difficult to resolve and acknowledging the limitation there would be sufficient.

Minor comments:

A lot of the citations are not showing properly, but as "Error! Reference source not found"
RESPONSE:

Thank you for the reminder. We checked the MS and suspect this situation may be caused by software incompatibility. Because our manuscript was written in WPS. Anyway, we re-examined the citations carefully and provided new manuscript which written in Microsoft Office.

Is the first paragraph supposed to be an abstract? It looks quite out of place. The mention of "landform development" is a bit sudden

RESPONSE:

Yeh, the first paragraph is abstract. We've taken your advice and rewritten this section to made it more concise. The revised abstract is below:

“Although it is well documented that mountains tend to exhibit high biodiversity, whether and how geological processes affect the assemblage of montane floras remains unclear. Here, we address this knowledge gap by exploring landform-specific differences among mountain floras based on a dataset of 17,576 angiosperm species representing 140 well-studied Chinese mountain floras. Our results show that igneous bedrock (granitic and karst-granitic landforms) is correlated with higher species richness and phylogenetic overdispersion, while sedimentary bedrock (karst, Danxia, and desert landforms) is associated with opposite, i.e., phylogenetic clustering. Furthermore, landform type was the primary determinant of the assembly of evolutionarily older species within floras, while climate was a greater determinant for younger species. Our study indicates that landform type not only affects montane species richness, but also determines the composition of montane floras. To explain the assembly and differentiation of mountain floras, we propose the “floristic geo-lithology hypothesis”, which highlights the role of bedrock and landform processes in the assemblage of mountain floras and provides novel insight for future research on speciation, migration, and biodiversity in montane regions.”

L31: remove "analysing" to make the rest of the sentence about a "topic"; and move "processes" to after geological to improve clarity.

RESPONSE:

Thanks for the kindly advice, and also the previous response. It seems this sentence is a bit redundant, so that we deleted this sentence.

The new statement in abstract is “*Although it is well documented that mountains tend to exhibit high biodiversity, whether and how geological processes affect the assemblage of montane floras remains unclear. Here, we address this knowledge gap by exploring.....*”. Please refer to the previous reply.

L63: do you mean animal diversity?

RESPONSE:

“*Furthermore, the evolutionary history of plant species also affects biodiversity.*” The literature cited here is an example of animal studies, which may leads to misunderstandings. In fact, evolutionary history has an effect on whole biota. We revised this sentence to improve clarity. See line 57-58: “*Furthermore, the evolutionary history of each biological taxa also affects local biodiversity*^{Error! Reference source not found.}*..*”

L64: add "and" to before "should include" and delete "includes"

RESPONSE:

Done. See line 59: “*.....It is clear that an integrated framework is needed for the prediction of montane biodiversity*^{Error! Reference source not found.}*..*^{Error! Reference source not found.}*..*^{Error! Reference source not found.}*..* and it should include ecological processes (e.g.,”

L105: change "sum" to "collection" or "community" and remove "families, genera, and" unless you mean there are taxa that are not identified to species but also included here.

RESPONSE:

Done. This sentence has been revised as “*Here, we used the term “flora” to refer to the collection of all angiosperm species growing on a specific mountain or in a well-delimited area*^{Error! Reference source not found.}*.....*”

L107: I think I get your point but it needs to be more explicit. How about mentioning species diversity only one characteristic of a flora and can be quantified by species and phylogenetic diversity?

RESPONSE:

Thank you for your valuable advice. This section has been revised to made it more concise. See line 102-104:

“*Then, the species richness (SR), phylogenetic diversity (PD=Faith’s PD), phylogenetic structure indices (PDI, NRI, NTI), and mean divergence times (MDT) were calculated for each of the 140 floras*^{Error! Reference source not found.}*..*^{Error! Reference source not found.} (Extended Data Table 2, see Methods). Finally, we constructed regression models (1) using.....”

Figure 1: please add the names of the landforms to the figure to improve clarity for the broad audience of the journal. It was very hard for me to matching things up using the caption.

RESPONSE:

We accept this suggestion and add the names of the landforms to the figure 1. Additionally, in order to more intuitively reflect the impact of the five landform and lithological types on the floras, we have made made appropriate modifications to the original Figure 1. The Danxia series has been placed under the other four landform types. This modification does not affect the principle of the figure, so it is appropriate. Please see the modified figure 1 below:

L152-153: not sure what the last sentence means. They all still look significant in the full model.

RESPONSE:

The last sentence is “*However, the partial landform effects on SR are often superimposed with environment variables and are difficult to separate.*” This sentence emphasizes the interactions effect between landform effects with climate. It seems a bit repetitive to previous result description, we deleted it in the new MS.

L154: add "(towards eastern China)" to help guide non-expert audience.

RESPONSE:

Done. This sentence has been revised as line 147-148 “*The mountain floras with higher SR were mainly located in the monsoon climatic zone of eastern China (Fig. 2).*”

L183: the point about higher rates of extinction needs further explanation and should probably be in Discussion - this happens at several places in the Results section (e.g. the paragraphs starting at L250 and L281)

RESPONSE:

The original paragraph mentioned by the reviewer is:

“*Thus, species with the deepest phylogenetic divergences occur in karst landforms, while species with the shallowest divergences occur in desert landforms. This is consistent with some fossil evidence. For example, the fossil flora discovered in southwestern China indicate that local karst vegetation may have existed since the early Oligocene*”
Error! Reference source not found.,Error! Reference source not found.,”

Here, we would like to indicated the species in the limestone mountains are generally had earlier diverge age and could survive for a long time. Especially those species can adapted to the arid habitat of the limestone mountain top. We present an example, the Oligocene fossil flora Wenshan basin which located in Yunnan, China. The vegetation (infer from fossil data) and genera composition of this fossil

flora in is very similar to the current karst flora in Wenshan, China.

For the second question, “Discussion” occurs in Result section. The reason we did this is we want to give a brief explanation of the results or compared to previous study. We thought this would help readers understand the meaning of our results more quickly, and also avoid repetition before and after the article. We notes this approach is generally be accept in the articles published in Nature Communications. In the later “Discussion” section (line 286), we further discussed the relationship between landform process, species richness, local speciation, dispersal in mountain flora.

Figure 3-4: It's very easy to lose track of all the abbreviations but I see NTI is not in Figure 3 and SR is not in Figure 4. It would improve clarity if Figure 3 and 4 have consistent panels, though showing the landform and full models respectively.

RESPONSE:

Thanks for the suggestion. In the previous Figure 3-4, we have only presented the the most representative result for the sake of brevity. While, keep the variables in Figure 3 and 4 have consistent panels sounds better. Following your suggestion, we revised the Figure 3 and Figure 4. Please see below:

Figure 3. Differences in species richness, phylogenetic diversity, phylogenetic structure, and age of floras among different landforms.

Figure 4. Standardized coefficients of determination for species richness, phylogenetic diversity, phylogenetic structures and divergence times of mountain floras.

L303: "recognised as positively correlated to..." ("significantly" is usually unnecessary as we can't say "correlated" if it's not significant).

RESPONSE:

We have incorporated this suggestion. All of the similar questions were revised in the manuscript.

Reviewer #3 (Remarks to the Author):

For convenience, I have provided my detailed comments in the "response to comments" file.

RESPONSE:

As many comments put forward by reviewers#3 in our previous response letter were marked as annotations. We have retained the original appearance of the parts that the reviewers still have questions. Our new response is in red colour "RESPONSE:". Hereafter are details peer-to-peer responses.

REV: According to my observations this is not correct, at least not for all parts of the manuscript. For example, revised M&M texts are of higher quality than other parts of the manuscript which are poorly written and where writing needs to be substantially improved to be deemed acceptable – and understandable to the readers. There are many sentences which are neither understandable in terms of language nor scientifically correct (in content and use of terminology).

Some single errors I have marked using magenta, but for other larger sections that need re-writing I mention this in my text.

RESPONSE:

Thanks for this suggestion. The new manuscript has been improved base on the Springer Nature Author Services. We suspect the new manuscript could meet the requirements of the reviewers and also of the journal.

This document certifies that the manuscript

Landform and lithosphere development drive the assembly of mountain floras in China

prepared by the authors

Wan-Yi Zhao, Zhong-Cheng Liu, Shi Shi, Jie-Lan Li, Ke-Wang Xu, Kang-You Huang, Zhi-Hui Chen, Ya-Rong Wang, Cui-Ying Huang, Yan Wang, Jing-Rui Chen, Xian-Ling Sun, Wen-Xing Liang, Wei Guo, Long-Yuan Wang, Kai-Kai Meng, Xu-Jie Li, Qian-Yi Yin, Ren-Chao Zhou, Zhao-Dong Wang, Hao Wu, Da-Fang Cui, Zhi-Yao Su, Guo-Rong Xin, Wei-Qiu Liu, Wen-Sheng Shu, Jian-Hua Jin, E. Boufford David, Qiang Fan, Lei Wang, Su-Fang Chen, Wen-Bo Liao

was edited for proper English language, grammar, punctuation, spelling, and overall style by one or more of the highly qualified native English speaking editors at SNAS.

L566-567: “Determinants of SR might change with landform type, and we therefore test for interactions between landform and others predictor variables (only shown significant variables in full model).”

RESPONSE:

Revised.

L203-204: “P value of T-test result between each pairs of landforms were showed above the black line: *** $p < .001$; *** $p < .001$; ** $p < .01$; * $p < .05$; ns = not significant.”

RESPONSE:

Done. Revised “were showed” as “are shown”.

Question about previous response.

Second, based on the technical requirements (this is based on the null model assumption) of the software package, flora data is extracted from the mountain species database to reflect the phylogenetic structure of the flora. Thus, a global or regional tree is not needed here. In fact, as we know there isn't currently an approach that is calculates a regional phylogenetic structure based on a global tree.

REV: The meaning of this sentence is not clear to me. Of course, there exist studies that have been using e.g. large angiosperm-wide trees as source for creating much smaller trees with only regional species samples representation. This needs clarification.

RESPONSE:

“Using e.g. large angiosperm-wide trees as source for creating much smaller trees with only regional species samples representation”, which is also the method used in our study.

However, the meaning of “In fact, as we know there isn't currently an approach that is calculates a regional phylogenetic structure based on a global tree.” is we can't use a large phylogenetic tree (included about 30000 species) to calculate the PDI, NRI and NTI of floristic distribution dataset included less species (such as 10 floras in total included 3000 species). The species in the phylogenetic tree should be consistent to the floristic distribution dataset, otherwise would violate the assumption of the null model.

In our study, the calculation of phylogenetic diversity and phylogenetic structure were performed by R packages “PhyloMeasures” (Tsirogianis & Sandel, 2016; doi:

10.1111/ecog.01814). This package is widely used in studying the community phylogenetic structure of floras. The “PhyloMeasures” requires the species pool to be consistent with the phylogenetic tree and distribution database. Otherwise the functions (“mpd.query”, “mntd.query” and “pd.query”) in “PhyloMeasures” would not work. See the screenshot below for the warning in R.

```
> mpd.std<-mpd.query(tree,data, null.model="uniform", T)

Warning: the input matrix has fewer columns than the number of species in the tree.

Error in mpd.query.uniform(tree, matrix, standardize) :
  One of the species names in input the matrix was not found in the tree (Staurogyne_rivularis)
> |
```

REV: This is not clear to me and needs explanation. *Staurogyne rivularis* is a synonym for *Staurogyne spatulata*, only the latter name which is accepted. Not finding “*Staurogyne rivularis*” could mean that the dataset uses the correct name instead, *St. spatulata*, rather than the synonym. Which taxonomic name resolution approach was used by the authors and how was it guaranteed that species were not omitted from the analysis due to synonyms or spelling errors? This could have introduced substantial error.

RESPONSE:

Reviewer raised concerns about errors that may result from inconsistent scientific names. In fact, the taxonomic name included in our mountain dataset of floras had been standardized to Leipzig Catalogue of Vascular Plants (LCVP), which it mentioned and proposed by Reviewer #2 in secondly review (details see **Methods** section in line 403; “*Dataset generation and reconciliation*”). Furthermore, the mega-phylogenies used in our study is “GBOTB.extended.LCVP.tre”, which has also standardize the plant names to LCVP database (Jin & Qian, 2022) (details see **Methods** section in line 414; “*Phylogenetic reconstruction*”). Thus, the species name in phylogenetic tree is consistent to our floristic distribution dataset.

So that the software warning reason here is not the inconsistencies of taxonomic name. The true reason is species number in the phylogenetic tree not consistent to the floristic distribution dataset (more species included in phylogenetic tree than floristic distribution dataset, as the case we shown above, less species in floristic dataset than in phylogenetic tree).

```
.. Warning: the input matrix has fewer columns than the number of species in the tree. ..
```

Please also refer to the previous reply. We hope our response made it clear.

Q4: With regard to the "age" of species, I appreciate the authors' effort to provide additional information by comparing the pruned and unpruned trees.

However, my point is more about the concept of species or assemblage age -- it is a not the real age and can only be used for representing the patterns of age when extinction is random; in contrast, a comparative analysis can be problematic if extinction was not random with regard to the factors that are being examined. Therefore, I suggest rephrasing it in terms of species distinctiveness, so that an assemblage with low MDT contains more closely related species and infer mechanisms from there.

Response: Thank you for clarifying your concerns, and we appreciate your suggestion to use the term “species distinctiveness” to replace “flora age of species age”. Here, we're unsure the meaning of the term “species distinctiveness”. According to our understanding, we assumed the reviewer’s proposed ‘species distinctiveness’ to be the evolutionary distinctiveness of species as proposed by Isaac et al., (2007, doi:10.1371/journal.pone.0000296). We suspect there are no differences between

“species distinctiveness” and species age, because both of them are based on the **in-clade** in the phylogenetic tree. Besides, species age (or species divergence time) is more well known than “species distinctiveness” in evolutionary biology and biogeography.

REV: What do the authors mean by “in-clade” ?

RESPONSE:

Here, “in-clade” just represents the branches of each species contained in the phylogenetic tree.

On the other hand, we accept that the concept of “flora age” is problematic. The complexity assembly history of flora makes it impossible to interpret the real age of flora. In contrast, the age of the species, which estimate based on the molecular clock hypothesis (Ho, S., 2008; The Molecular Clock and Estimating Species Divergence) and fossil calibrated, is more reliable. In fact, the inference of floristic history is often based on the estimated divergence time (or stem age) of species which occurs in the flora (such as Dagallier et al., 2020, doi: 10.1111/nph.16293; Qian & Deng, 2022, doi: 10.1111/jse.12856; and Chen et al., 2018, DOI: 10.1093/nsr/nwx156).

In our study, we used the mean divergence times (MDT) proposed by Lu et al.(2018) to represent the “flora age”. As such, the MDT reflected the age composition of species within a flora. A flora with larger MDT has more ancient species and hence is expected to have older floristic ages.

Overall, we believe that our use of MDT to measure the species age structure of flora is consistent with the existing literature and therefore is appropriate.

REV: Not clear to me. Only because many people have been doing something does not mean it is correct or the most appropriate way to do it.

RESPONSE:

Reviewer # 3 mentioned that “many people are doing the same thing, which does not necessarily mean that this approach is correct”. We cannot fully agree with the comments of Reviewer #3. Many people have used this method to carry out a lot of work, although it cannot be considered right, it cannot be considered wrong, and it cannot be considered that several journals currently publish the wrong method? Especially, if there is no better update method available, traditional methods can only be used.

However, frankly, it is difficulty of develop a better or appropriate method to measure the species age structure of flora. Because species age occurs in a flora is still can't be well defined. At present, we can only study the floristic assemble process on a macroscale of statistical significance. So that in our study we used the MDT method proposed by Lu et al. (2018), which is not so bad. Development new methods to understand the species assemble process of flora over time scale is an important topic for future research.

In addition, in this article, we only hope to calculate the aggregation and divergence of plant flora at a macro scale. Therefore, we used the MDT method proposed by Lu et al. (2018). The limitations of this method have been explained and revised as previous reviewer comments (such as MS Line 461-470). Developing new methods to reveal the species composition of plant flora on a temporal scale is an important topic for future research. Of course, we would greatly appreciate it if the reviewer could recommend a better method to us.

L64: “It is clear that an integrated framework for the prediction of montane biodiversity is necessary. **should include** **includes ecological processes** (e.g., survival, competition, and niche differentiation), **biological processes** (e.g., species divergence and extinction), and geological and lithologic processes (e.g., orogeny and rock

formation)^{Error! Reference source not found.,Error! Reference source not found.}”

REV: Ecology is a branch of biology, authors may rather use “evolutionary processes” than “biological processes”.

RESPONSE: We accept this suggestion. Revised as line 58-61 “*It is clear that an integrated framework is needed for the prediction of montane biodiversity*^{Error! Reference source not found.,Error! Reference source not found.,Error! Reference source not found.}, and it should include ecological processes (e.g., survival, competition, and niche differentiation)^{Error! Reference source not found.}, evolutionary processes (e.g., species divergence and extinction)^{Error! Reference source not found.,Error! Reference source not found.}, and geological processes (e.g., orogeny and lithosphere cycling)^{Error! Reference source not found.,Error! Reference source not found.}”

L82-87: “*Although it is well known that climate change forces plant species migration, those species which are adapted to local bedrocks are constrained in their ability to migrate*^{Error! Reference source not found.} *Bedrock geochemistry is on par with climate as a regulator of vegetation in granitic mountains*^{Error! Reference source not found.} *Some studies also suggest that local species diversification processes are consistent with edaphic rather than climatic filtration, such in the Cape flora*^{Error! Reference source not found.}, *Teesdale flora*^{Error! Reference source not found.}, and *New Caledonian flora, in which ca. 50% of the endemic floristic elements are ultramafic-obligate species*^{Error! Reference source not found.}”

REV: I am unhappy with the statement that geochemistry, which is a field of research, is a regulator – authors needs to take care with precision in their statements.

RESPONSE:

Reviewer # 3 mentioned our statement “*Bedrock geochemistry is on par with climate as a regulator...*” is incorrect. This statement is cited from previous literatures “*These results are important because they demonstrate that bedrock geochemistry is on par with climate as a regulator of vegetation in the Sierra Nevada and likely in other granitic mountain ranges around the world.*”, which occurs in Hahm et al. (2014, <https://doi.org/10.1073/pnas.1315667111>). In fact, when mention bedrock geochemistry, it is usually understood as the chemical elements in bedrocks (eg. “*Bedrock geochemistry influences vegetation growth by regulating the regolith water holding capacity*”).

Anyway, to avoid ambiguity. We revised this sentence as line 79-80 “*The geochemical characteristics of bedrock is on par with climate as a regulator of vegetation in granitic mountains*^{Error! Reference source not found.} *Some studies also suggest that local species diversification processes are consistent with edaphic rather than climatic filtration, such as in the Cape flora*^{Error! Reference source not found.}, *Teesdale flora*^{Error! Reference source not found.}, and *New Caledonian flora, in which approximately 50% of the endemic floristic elements are ultramafic-obligate species*^{Error! Reference source not found.}”

Q9: Line90:

“*Here, we apply the term “flora” to refer to the sum of all angiosperm families, genera, and species growing on a specific mountain or in a well-delimited area*^{Error! Reference source not found.,Error! Reference source not found.,Error! Reference source not found.}, which is a relatively independent biogeographical unit^{Error! Reference source not found.}”

I would argue against the flora of a mountain being a *relatively independent biogeographical unit*, if this is what is meant here (but I may have misunderstood the sentence). First, because related lineages are shared between mountains, and second, because most biogeographers, in terms of terminology, would not consider the flora of a mountain as a “biogeographical unit”.

Also in **Pre-response**: “*we mentioned is not only a unit, but als*” I am not sure how “unit” here would refer to the biogeographic units mentioned in the ms. This may be worth of clarification.

Response: We appreciate the key question raised by the reviewer. In generally, the term “biogeographic unit” is used in the context of biogeographic regionalization.

Here we consider that the flora of mountain and **biogeographic zones** is equivalent at small scale.

REV: This is unclear. Which biogeographic zones are meant here?

RESPONSE:

(1) Here, the mountains wasn't refer to a single peak, but rather to a mountainous area or a certain nature reserve, including many peaks and adjacent areas. Correspondingly, all the plants growing in this area is consist of many different species, genera, and families, whose was called on a mountain flora, it is a relatively independent natural geographical area. Due to differences in longitude, latitude, altitude, and climate factors, biological factor (or floristic compose) among different mountainous regions, these geographical regions are also bound to have differences.

(2) The biogeographical units mentioned in the main text cannot be equated with biogeographical divisions. Biogeographic zoning often divides any geographical areas into several level or grades, such as the global flora, which is often divided into kingdom, region, province, and county. These are different biogeographical units (hierarchical units). The unit itself does not have hierarchical significance, and only after studying all species in the region, namely Flora, its hierarchical level or grade be determined, such as the local flora, the geographical elements of the families, genera, or species (tropical, temperate elements), historical elements (antiquity), originative elements (endemic families, genera, species etc.), ecological elements, migration elements, etc., Only through floristic phyto-geography analysis, a detain units such as a kingdom, region, province, and county be divided.

A certain biogeographical unit, such as a mountain area, may also be divided into different geographical regions, such as the Hengduan Mountains in China and the Himalayan region, which are divided into the East Asian Flora and the China-Himalayan Forest Subregion. In the east, it is divided into the East Asian Flora and the China-Japan Forest Subregion; Correspondingly, the latter can be divided into South China Province, Central China Province, etc. (Wu Zhengyi et al., 1996). A certain mountain area, depending on its nature geography and biogeography property, they may divided into a hierarchical unit (division) or a subunit. Or they could be belonged a province, or a county.

Therefore, when we mention a mountain region, or a mountain flora, similar to a nature geographic or biogeographical area or unit with a certain grade or no grade, it is not a wrong concept.

(3) This article studied 140 mountainous areas, listed the natural geographic information of each mountainous area, and correspondingly formed 140 local mountainous floras, thereby showcasing their differences in natural geography and floristic phyto-geography.

Furthermore, this article successfully classified 140 mountains into five geomorphic types (by searching for detailed geological survey data, and special investigation report on Floras), revealing and analyzing the relationship and possible reasons between these local floras and mountainous lithology. We believe that this is a progress. On this basis, it will be possible to further study endemism, geographical elements of those floras, and diffusion, migration of floristic elements between different geomorphic types and different bedrock, floras.

(4) Of course, in our response to the reviewer's question, we wrote biogeographic zones, which are indeed not strict enough and have a larger scope. We just wanted to explain that each mountain has its unique characteristics. Wu Zhengyi et al. (1996) considered that an area of every relative independent mountainous flora covered should not be less than 100 square kilometers. Due to the large amount of content and limited space, these concepts and data were not included in the main text.

First, **mountain is geographical barrier to plants diffusion.**

REV: This is not correct (as I had stated in the previous rounds of review already), phrased in this general manner. It depends on the orientation of the mountain. Only if a mountain has W-E orientation, it can act as barrier (e.g. Himalayas, Alps). For N-S-oriented mountains (such as the Andes, or the Hengduan mountains), this is not correct.

RESPONSE:

We believe that it is unnecessary for the reviewers to oppose the above views. The author's viewpoint is simply that mountains can affect the diffusion of species or serve as a barrier of species diffusion. In fact, mountains are both a barrier for species diffusion and may also play a promoting role in species diffusion. For example, the north-south direction mountains play a promoting role in the north-south migration of species, while serving as a barrier for the east-west migration of species; Due to the existence of Hengduan Mountains, many genera and species in Chinese Mainland have formed China-Himalaya distribution subtypes and China-Japan distribution subtypes, with Hengduan Mountains as the boundary. Similarly, due to the barrier effect of the Nanling Mountain in China, many tropical genera and species cannot exceed the Nanling Mountain, such as *Pandanus*, *Endospermum*, etc.

The reviewer mentioned or emphasized the local migration, endemism, and differentiation of species within a mountainous area, especially in high mountains. Our focus is to emphasize that the migration of species between different mountainous areas will be hindered, especially between Danxia landforms, karst landforms, and granite landforms, where species diffusion is not easy.

Second, **a mountain usually shows sky island effects** and contain their own unique clades or several endemic species.

REV: This is not correct. First, there is no such thing like a “sky island effect” (phrased this way). Second, only a small fraction of the world’s mountains qualify as sky islands – only, if their vegetation is drastically different from the surrounding lowlands. This may only be the case if plants living on the mountains are different from the lowlands and were therefore not likely recruited/evolved from lowlands in which case the nearest relatives would come from other mountains (example: some mountains located in drylands, e.g. Africa).

RESPONSE:

Here is just a brief discussion with the reviewer. The “sky island effect” of mountain flora is a question worth further study and discussing. As our understand, in essence the sky islands mountains is qualified by its flora showed discontinuous distribution. In generally, in mountains vertical zonation of vegetation on the elevation gradient is widespread observed. We suspect the plants or community occurs on the top of the mountain should be treated as a result of sky islands effects? These community differs from the low land, and could represents the early stage of species assemble of mountain flora.

There are also some literature to suggest that mountain region could be viewed as a biogeographical unit. **For instance, Rahbek et al. (2019, Science 365, 1108-1113) wrote that “Like an island, a mountain region may be viewed as a biogeographical unit in itself, with in situ speciation and extinction playing a key role in building the regional species assemblage”.**

REV: Is anyone else except this cited paper claiming this as well? I could imagine that most mountain biogeographers would disagree with this oversimplification.

RESPONSE:

We cannot agree with the reviewer's opinion, as stated in the previous response.

Line 311-313 “*Landform developmental processes are associated with local bedrocks and regional*

climate^{17,59}. The assembly of the local flora is impacted by a combination of geological and climatic processes, and biological processes.”

REV: This sentence is devoid of logic; the first matters mentioned do not automatically cause the second. The sentence as such is therefore not correct. I suggest to re-phrase this – as corrected above (using tracked changes).

RESPONSE:

Thanks for the kindly advice. We accept this suggestion and revised as line 297-298 “*The assembly of a local flora is impacted by a combination of geological and climatic processes, as well as biological processes*”

Q14: Line313: “*The mountains of sedimentary bedrock have much stronger environmental filtering effect, as these mountain ecosystems are more sensitive to rainfall. The environmental filtering effect further promoted the clustering of phylogenetic structures of mountain floras as predicted by the phylogenetic niche conservatism (PNC)*”

It would be interesting to discuss here radiations more generally, e.g., how do the studies of plant radiations included in the global study by Muellner-Riehl et al. 2019 (JBI) compare to these statements/findings? Are the findings of these studies (and other studies published since then, i.e. after 2019) comparable to this? I suggest looking at the mountain systems (and their bedrocks) where these studies were undertaken.

Response: These are really good advice. The potential radiation speciation occurs when plants colonize from one landform to another which maybe an important path of plant diversification. Here, we document several cases (not all) on adaptative radiation that occur on the bedrocks. But the relations between radiations and bedrocks/landform are still poorly known. We suspect that the effect of landform and bedrock, which promote species differentiation, is more or less neglected under the shadow of “climate change”. Nevertheless, we have reorganized this paragraph carefully. REV: I don’t understand the reasoning here, as currently phrased. It is unclear what the authors mean (potential speciation radiation?). There exist different kinds of radiations, not all of them are “adaptive”.

RESPONSE:

Here we would like to shown adaptive evolution (associate with morphological traits and physiological traits) maybe the most important mechanism for plant to radiation in a new environment. Although the mechanisms of radiative evolution are diverse, plants ultimately need to adapt to their local environment in order to survive. Furthermore, evolutionary radiations underpinned by variation in physiological or behavioral traits can more easily be perceived as non-adaptive, compared to those involving more conspicuous morphological traits, causing a bias in our understanding of the extent and distribution of adaptive radiations in nature. That is reason a large number of traits of convergence (such as Alpine flora, Arid flora, and mangrove plants) and adaptive evolution have been broadly observed (Nevado et al. 2019, doi: 10.1016/j.cub.2019.07.059; Xia et al. 2021, doi: 10.1093/molbev/msab314).

Please see line 334-348. “*Mountains composed of sedimentary bedrock have much stronger environmental filtering effects, as these mountain ecosystems are more sensitive to rainfall. Such effects further promoted the clustering of the phylogenetic structures of mountain floras, as predicted by phylogenetic niche conservatism (PNC)*”

radiation, such as the Old World gesneriads and North American deserts rock daisies (Compositae tribe Perityleae). Similar patterns have been observed in insects. For example, two radiation clades (clade nodes 14 to 16 and nodes 31 to 33) of Exocelina resulted from the transition from uplifted Australian Plate rock to ultramafic/ophiolite. Although, many previous studies also document climate change as an important driver of species radiations.

REV: This text needs re-writing, major parts are not understandable. “clustering of the phylogenetic structures”? Authors don’t explain what the connection is between PNC and phylogenetic clustering. “transition”? formulation and meaning unclear.

RESPONSE:

This section is re-written to make it more clear. Please see below.

Line 323-330 “Such effects further promote the phylogenetic relatedness (clustering) of mountain floras because new lineages tend to maintain their ancestral ecological niche.”

In plants, radiative evolution often accompanies habitat and landform shifts, as seen in Old World gesneriads and North American deserts rock daisies (Compositae tribe Perityleae). Similar patterns can also be observed in insects. For example, two radiated clades (nodes 14 to 16 and nodes 31 to 33) of Exocelina resulted from the transition from uplifted Australian Plate bedrock to ultramafic/ophiolite.”

The adaptive radiations of plant genera are more or less associated with bedrock type. For example, the key innovation (lime-secreting hydathodes) of Saxifraga sect. Porphyrium is clearly an adaptation to the limestone bedrock, and the low specific leaf area (SLA) exhibited by Erica is an adaptation to oligotrophic habitats (Quartzite/sandstone) in Cape. Thus, we suggest that landform and bedrock effects strongly promote species differentiation between different regions, even if such an effect has not been mentioned in previous studies.”

REV: “The adaptive radiations of plant genera are more or less associated with bedrock type.” If at all, adaptive radiations can, among others factors, be more or less associated with bedrock type...

The entire text section needs an English language check (grammar, incomplete sentences,...). “different regions, even if such an effect has not been mentioned in previous studies.” – this is not correct. There is a long history of investigations in the European Alps on this matter, dating back to the mid of the 20th century.

“the key innovation (lime-secreting hydathodes) of Saxifraga sect. Porphyrium is clearly an adaptation to the limestone bedrock.” Authors need to carefully read and describe findings in literature, not overstating findings. For example, to my knowledge, some Saxifraga species with lime-secreting hydathodes do actually not grow on limestone – therefore, the sentence as currently phrased does not hold true for all lineages. Care needs to be taken with re-phrasing.

RESPONSE:

Thanks for your criticism. We revised this section. Please see line 331-336: “The adaptive radiation of plants is more or less associated with bedrock type. For example, the development of a key innovation (lime-secreting hydathodes) may have made Saxifraga sect. Porphyrium better suited to limestone

habitats^{Error! Reference source not found.}, and the low specific leaf area (SLA) exhibited by *Erica* is an adaptation to oligotrophic habitats (quartzite/sandstone) in Cape^{Error! Reference source not found.}. These obvious landform and bedrock effects could strongly promote both species and floristic differentiation between different regions^{Error! Reference source not found.,Error! Reference source not found.,Error! Reference source not found.}”

Q15: Line 317-327:

“Dispersal also contributes to **the** mountain diversity^{Error! Reference source not found.,Error! Reference source not found.}. A well-known example is climatic fluctuations during the Quaternary ice age drove changes in the distribution of species around the globe^{Error! Reference source not found.,Error! Reference source not found.}. The spatial configuration of mountain range affect their functional connectivity, influencing species dispersal^{Error! Reference source not found.,Error! Reference source not found.}. In fact, dispersal more easily occurs at initial stage of landform develop mountain, as the slope of mountain still gentle and geographical barrier still relatively low^{Error! Reference source not found.}. Such as the plants dispersal events in Qinghai-Tibet Plateau (QTP, low barrier) is obvious higher than that in Hengduan Mountains (high barrier)^{Error! Reference source not found.}. Furthermore, the effects of dispersal on species diversity in mountainous areas mainly occurred at lowland, and is limited at highlands where has more local endemic species^{Error! Reference source not found.,Error! Reference source not found.}. This means that dispersal has weak influence on the unique composition of mountain flora of different landforms, especially considering that landform have a significant filtering effect on the species.”

The entire paragraph needs re-writing. Dispersal between mountains is not limited to times of climate change, which somehow is suggested here as currently written. Dispersal and establishment was easier in the Hengduan mountains at times of climate change as they have a N-S orientation, enabling plants to change their latitudinal distribution range more easily.

And Pre-response: “Such as the plants dispersal events in Qinghai-Tibet Plateau (QTP, low barrier) is obvious higher than that in Hengduan Mountains (high barrier)⁴⁴.”

I don’t understand this sentence. As outlined previously, it is exactly the other way round. Dispersal and establishment was easier in the Hengduan mountains at times of climate change as they have a N-S orientation, enabling plants to change their latitudinal distribution.

And Pre-response: As I pointed out previously, the opposite would be expected. Survival on mountains with N-S orientation is easier during glacials than on those of W-E orientation (e.g. compare Himalayas versus Hengduan Mts.).

And Pre-response: “This means that dispersal has weak influence on the unique composition of mountain flora of different landforms, especially considering that landform have a significant filtering effect on the species.”

Reading this, I am not sure the authors have read the papers I had suggested to consult, and which they cited in the paper already before (e.g. Ding et al.).

Response: We accept this advice and wrote this paragraph. We would like to clarify the questions about dispersal events and diffusion barrier. We had carefully readed the paper by Ding et al (2020). It is not difficult to deduce the conclusion “dispersal events in Qinghai-Tibet Plateau (QTP, low barrier) is obvious higher than that in Hengduan Mountains (high barrier)” (see Ding et al, 2020, figure 2). On the other hand, we thought dispersal would reduce the β -diversity between different floras. Such as the proportion of endemic species in Hengduan Mountains is the highest is largely affect by its low rate of colonization (<0.05), and **high rate of *in situ* speciation and local and recruitment.**

REV: This could also/additionally be the effect of higher levels of extinction on the QTP, in contrast to the Hengduan mountains.

RESPONSE:

This may be right as the QTP has lower habitat heterogeneity than the Hengduan Mountains. The role of extinction rate QTP and Hengduan Mountains are not mentioned in the paper by Ding et al (2020). A region with low habitat consistency are associated with lower biodiversity and also more species with strong ecological adaptation. Distinguishing the roles of history, speciation and extinction in the mountain flora assemble still represents a major challenge to future research.

In contrast, the low proportion of endemic species in Qinghai-Tibet Plateau has the highest rate of colonization since early Miocene. It is clearly the dispersal is negative correlation to the proportion of endemic species in flora.

REV: I am not pleased with this entire section of text (incl. the one further above), as the main points of criticism still remain to be addressed.

RESPONSE:

We reorganized the section to make our point clear. Our main idea is that differences in landform type have a constraining effect on species dispersal between mountains. For example, species could not spread freely between limestone and non-limestone mountains. Such restrictive effect caused by landform are general associated with the differs of bedrock, soil, water cycle processes.

The revised section is please see line338-359: ***Restricted dispersal between landforms as the result of environment filtering*** “For many biogeographers, mountains are regarded as both barriers and bridges of species dispersal. The role of mountains as corridors has been documented in several mountains oriented north–south, such as the Andes and Hengduan Mountains. However, the contribution of dispersal to montane floristic diversity largely depends on the ecological and physiological requirements of the species, as well as their dispersal ability. Our research demonstrates the role of landform constraints on the interaction of different landform floras, which is shown in their SRs, phylogenetic structures and species age structures (Fig. 3-4; Extended Data Fig. 9-15). Dispersal occurs more freely during the initial stages of landform development in mountains, when slopes are still gentle and geographical barriers are minimal. The role of local species recruitment is most important during the early stages of mountain floristic assembly, the importance of which is subsequently replaced by local adaptations or in situ speciation. Mountains in different landforms will recruit different plant species as a result of environmental filtering caused by differences in bedrock. For example, mountains composed of limestone bedrock contain more species which are physiologically tolerant of drought and high calcium stress than mountains composed of metamorphic rocks and granites. The landform restriction effect on species diffusion gradually strengthens when more bedrock is exposed and the connectivity between mountains of different landforms is greatly reduced. Variation in the species composition of mountain floras between different landforms increases under the combined effects of landform, environmental filtering, and local endemic speciation. Therefore, we propose that the patchy spatial distribution of different mountain landforms is an important factor in shaping biogeographical zoning.”

Although we have made significant changes to this section. We responded to the reviewer's concerns accordingly thereafter.

The re-wrote paragraph please see lines 349-367. “Dispersal also contributes to montane floristic diversity, although dispersal is only effective if it is followed by successful establishment”

found. Dispersal occurs more freely during the initial stages of landform development in mountains, when slopes are still gentle and geographical barriers are minimal^{Error! Reference source not found.}. For example, the plant dispersal events across the Qinghai-Tibet Plateau (low barrier) have been considerably greater than that in Hengduan Mountains (high barrier)^{Error! Reference source not found.}. The role of local recruitment was most important during the early stages of mountain flora assembly, a role which was subsequently replaced by local adaptations or in situ speciation^{Error! Reference source not found.}. Mountains in different landforms will recruit different plant species, because each species is differently adapted to specific types of bedrock^{Error! Reference source not found.}. The spatial configuration of mountain ranges alters their functional connectivity, thus influencing the species dispersal process^{Error! Reference source not found.}. Furthermore, during the middle stages of landform development, the effects of dispersal on species diversity in mountainous areas are primarily restricted to lowlands, and are limited in highlands where there are more local endemic species^{Error! Reference source not found.}. Taken together, this suggests that dispersal only weakly influences the unique composition of mountain flora associated with different landforms, especially considering that landforms exhibit a significant filtering effect. Variation in the species composition of mountain floras between different landforms will increase under the significant, combined landform environment filtering effect^{Error! Reference source not found.}, and local endemic speciation^{Error! Reference source not found.}. Therefore, we propose that the patchy spatial distribution of different mountain landforms is an important factor in shaping surface biogeographical zoning.”

REV: Again, this text’s meaning is still largely unclear.

Qinghai-Tibet Plateau (low barrier) have been considerably greater than that in Hengduan Mountains (high barrier) – which kind of barrier do the authors means – and barrier to what?

RESPONSE:

“For example, the plant dispersal events across the Qinghai-Tibet Plateau (low barrier) have been considerably greater than that in Hengduan Mountains (high barrier)⁵⁷.”

– which kind of barrier do the authors means – and barrier to what?”.

The “barrier” here means the change of elevation gradient in Qinghai-Tibet Plateau (QTP) is relatively small than Hengduan Mountains (HDM) (see the fig below, left, cited from Ding et al., 2020). Such low-barrier of QTP is clearly more conducive to species dispersal, which is tested from the assembly of alpine biotas. Since the Miocene, the colonization rate QTP (0.06-0.25) is always large than the QHM (<0.05) (in Ding et al., 2020, Fig.2). In addition, there is no significant increase in species dispersal rate was detected in the Hengduan Mountains (although N-S-oriented) during the Quaternary in Ding’s result (see the fig below, right).

[editorial note: two figures redacted]

REV: Role of local recruitment: In times of climate change, local recruitment from lowlands may play a potentially important role.

RESPONSE:

“The role of local recruitment was most important during the early stages of mountain flora assembly, a role which was subsequently replaced by local adaptations or in situ speciation^{24,57}.”

We understand the point “local recruitment in climate change” made by the reviewers. The range of species is expect shrinks during colding climate and expands during interglacial periods. But that doesn’t contradict our viewpoint. Because no matter how the climate changes, species are always more

easily dispersal in low geographical barriers. In the early stages of mountain flora assembly, the topographic fluctuation in the mountain is always relatively small.

REV: "Spatial configuration" – meaning unclear.

RESPONSE:

"The spatial configuration of mountain ranges alters their functional connectivity, thus influencing the species dispersal process⁸⁷⁻⁸⁸."

Here, "Spatial configuration" means topographical configuration, which is associated with the available ecological niches of a mountain along elevation gradient.

REV: "Furthermore," – but this disregards climatic fluctuations acting on mountains at various times.

RESPONSE:

We're not sure about this question. Do you mean climatic fluctuations facilitated the species dispersal both in lowlands and highlands? We think it's almost impossible that the higher elevations have the same migration rate as the lower elevations between mountains. There may be a misunderstanding gap between us and the reviewers. We are focus on the dispersal of species between mountains, rather than within it. We have deleted this sentence to make it clearer.

The new statement see Line 349-355 "*Mountains in different landforms will recruit different plant species as a result of environmental filtering caused by differences in bedrock*^{Error! Reference source not found., Error! Reference source not found.}. For example, mountains composed of limestone bedrock contain more species which are physiologically tolerant of drought and high calcium stress than mountains composed of metamorphic rocks and granites^{Error! Reference source not found., Error! Reference source not found., Error! Reference source not found.}. The landform restriction effect on species diffusion gradually strengthens when more bedrock is exposed and the connectivity between mountains of different landforms is greatly reduced^{Error! Reference source not found., Error! Reference source not found., Error! Reference source not found.}."

Q16: Line332:

"*Thus, landforms are suitable indicators of geodiversity*^{Error! Reference source not found., Error! Reference source not found.} for tracking changes in mountain floras along geological time scales."

I am missing more elaboration on measure of geodiversity. E.g., how does this compare to the geodiversity indices used in previous mountain studies (such as the MGH global study by Muellner-Riehl et al. 2019 in JBI)?

Again, the lack of discussion of work by other authors is misleading and suggests more novelty here than is actually inherent in this present study. Acknowledging previous work by other authors does not diminish the achievements of this study here, but rather empowers readers to compare this and previous studies.

And **Pre-response:** But geodiversity as a measure (geodiversity index) is an integral part of investigations under the MGH.

Response: Thanks for pointing out the shortcomings. There is no denying that linking geological diversity to biodiversity is an important step. However, there are still some problems in the application of geodiversity. In previous studies, geodiversity is usually treated as the environment variables integrated into the compound geodiversity index (GD, or GDCs). But the power of compound geodiversity index was not good enough in predicting species diversity. As the case you mentioned (Muellner-Riehl et al. 2019 in JBI, wrote "*The GD index and Elevational range were strongly correlated ($r = .70$) and given that Elevational range showed better performance in single*

predictor models...”), the geodiversity index (GD) was found to be less effective than a single topography variable (elevational range) for predicting mountain species diversity. In contrast, the landform type (characteristic variable) is more concise and easier to understand. For a detail discussion, please see line 372-388.

“Geodiversity, the variability of Earth’s surface materials, forms, and physical processes, is an integral part of nature and crucial for sustaining ecosystems and their services^{Error! Reference source not found.}. Several studies proposed to use geodiversity to model biodiversity based on the hypotheses that specific geo-sites should support unique biota and that high geodiversity is coupled with high biodiversity^{Error! Reference source not found.}. However, the predictive power of geodiversity has not been high. For example, the geodiversity index (GD) was found to be no better than elevational range for predicting mountain species diversity^{Error! Reference source not found.}. Another recent study reported that combinations of environmental variables better explained tropical species diversity than GD^{Error! Reference source not found.}. In fact, the contribution of geodiversity component (GDC) showed similar patterns to topography, and usually effects at smaller extents and finer grain size^{Error! Reference source not found.}. The GD is a poor predictor of mountain diversity in large part because interactions between mountains and biodiversity are complex^{Error! Reference source not found.}, and thus geodiversity could not be treated as an index. In contrast, landform type is an objective description of the present state of erosion of a mountain or bedrocks^{Error! Reference source not found.}. When we stand in front of a karst mountain, we can see quite clearly that the plants are different from those in a granite mountain. We propose that landform identity rather than GD, is a more suitable indicators of geodiversity^{Error! Reference source not found.} for tracking changes in mountain floras along geological time scales.”

REV: I see a lot of problems with this text, for various reasons:

- Basically none of the sentences is clearly and understandably formulated
- scientific terms are not properly used
- the text, in parts, is misleading and, in parts, contains wrong claims

To illustrate this, I have above left some comments – many more would have been possible. I hope the authors will see what I mean when commenting about the necessity to improve this text considerable.

RESPONSE:

In this section, the reviewer #3 raised many questions in the form of annotation. We have carefully considered these review comments. In the new manuscript, we have decided to delete this part. First, delete this section does not affect the conclusion of our paper and makes the manuscript more brief and intelligible. Second, the definition, measurement and practical application of geodiversity remain confusion. Third, some questions raised by the reviewers here are ambiguous.

Hereafter are the point to point replies to the reviewer's questions raised in this section.

“Several studies proposed to use geodiversity to model biodiversity based on the hypotheses that specific geo-sites should support unique biota and that high geodiversity is coupled with high biodiversity^{Error! Reference source not found.}.”

RESPONSE:

The review 3 pointout “use geodiversity to model biodiversity” is wrong, and marked “hypotheses that specific geo-sites should support unique biota” with a “?”. These questions are quite not clear.

These statements were cited from previous literature. See:

In the article (Muellner-Riehl et al., 2019) which mentioned many times by reviewer 3, geodiversity is used as variable to model species diversity (“We used generalized linear models to test to what extent

Commented [A1]: wrong

Commented [A2]: ?

Commented [A3]: This statement is wrong. There are plenty of studies in the literature showing the opposite. This does not pay justice to the state-of-the-art knowledge.

Commented [A4]: There is not “the” one GD, but there are different methods used in literature for estimating geodiversity (incl. remote sensing approaches more lately)

Commented [A5]: This is a strongly misleading formulation. Both perform equally good.

Commented [A6]: Meaning unclear – re-formulate. GD is usually a combination of environmental variables.

Commented [A7]: Not clear – which pattern?

Commented [A8]: No, it is not, as has been shown in previous studies.

Commented [A9]: Geodiversity measures are supposed to capture a considerable proportion of this complexity.

Commented [A10]: It can and has been treated as an index in some studies, and in other works, single parameters were used. Thus, the sentence, as formulated, is wrong.

Commented [A11]: Why in contrast?

Commented [A12]: Geodiversity is also “objective”

Commented [A13]: This is not new and is basic textbook knowledge. E.g. compare works in the Alps from the mid 20th century and thereafter.

Commented [A14]: Part 1 of the sentence – refers to patterns, current;

Commented [A15]: This refers to timing – but what is the connection to the first part of the sentence? This is not clear.

Commented [A16]: wrong

Commented [A17]: ?

vascular plant species diversity in mountains is explained by net primary productivity (NPP), geodiversity and Pleistocene climate fluctuations...” DOI: 10.1111/jbi.13715, p2868).

In Hjort' paper (DOI: 10.1111/cobi.12510, p630) “We found that geosites are important to biodiversity because they often support rare or unique biota adapted to distinctive environmental conditions or create a diversity of microenvironments that enhance species richness.”

However, *the predictive power of geodiversity has not been high*.

RESPONSE:

This sentence is followed by several examples to illustrate the limitations of geodiversity. To our knowledge, the relationship between geodiversity and biodiversity remains being further explored. As Alahuhta et al. wrote “*Although theoretical foundations for the geodiversity-biodiversity relationship and its conservation implications are well-established, only a handful of empirical studies have actually tested this relationship*” (see Alahuhta et al., 2020, <https://doi.org/10.1038/s41559-019-1051-7>).

For example, *the geodiversity index (GD) was found to be no better than elevational range for predicting mountain species diversity*^{Error! Reference source not found.}.

RESPONSE:

Here is just an example, which the review #3 mentioned in previous round. We think this is the method that reviewer #3 accepted to quantified geodiversity.

The review indicated “*to be no better than*” is a strongly misleading formulation. However, that is what we get from the paper of Muellner-Riehl et al. (2019). We can't agree with a model with higher residual deviance (18281 in GD model) is equally good to a lower one (13802 in Elevation model).

Here after are the statement cited from references 3. “*Elevational range were strongly correlated ($r = .70$) and given that Elevational range showed better performance in single predictor models (Table SI.2), GD was not considered in the multivariate regression models.*” (in Muellner-Riehl et al., 2019, p2830).

Table S1.2. Evaluation of generalized linear regression models to predict plant species richness Information Criterion; NPP = net primary productivity; GD = geodiversity index (GD; see *Mate* to the present.

	AIC	Δ AIC	Null deviance	Residual deviance	df (residual)	McFadden's pseudo-R ²
Single predictor models						
NPP (mode)	17290	10252.9	18390	17130	14	0.069
NPP (mean)	13860	6822.9	18390	13700	14	0.255
Elevation (range)	13963	6925.9	18388	13802	14	0.249
GD (range)	18442	11404.9	18388	18281	14	0.006
ΔT (mode)	10141	3103.9	18388.4	9979.4	14	0.457
ΔT (mean)	11144	4106.9	18388	10982	14	0.403

Another recent study reported that *combinations of environmental variables better explained tropical species diversity than GD*^{Error! Reference source not found.}.

RESPONSE:

Of course GD is a combination of environmental variables. However, its explanatory power is really not

Commented [A18]: This statement is wrong. There are plenty of studies in the literature showing the opposite. This does not pay justice to the state-of-the-art knowledge.

Commented [A19]: There is not “the” one GD, but there are different methods used in literature for estimating geodiversity (incl. remote sensing approaches more lately)

Commented [A20]: This is a strongly misleading formulation. Both perform equally good.

Commented [A21]: Meaning unclear – re-formulate. GD is usually a combination of environmental variables.

as good as the combination of separated environmental factors.

As the case of reference 92, GD is summed spatial diversity of climate (3 variable), habitat (6 variable), and soil (3 variable). See “*The geodiversity index, in contrast, is computed as the summed spatial diversity of the same three environmental factors (included climate, habitat, and soil) measured within each plot and its surrounding and follows the recent plea to consistently use diversity indices commonly applied in biodiversity studies*” (see Wallis et al. 2021, p7: doi.org/10.1038/s41598-021-03488-1). Their result shown that combinations environmental variables better explain species diversity and ecosystem functions than a geodiversity index.

In fact, the contribution of geodiversity component (GDC) showed *similar patterns* to topography, and usually effects at smaller extents and finer grain size^{Error! Reference source not found.}.

RESPONSE:

The “similar patterns” means geodiversity component (GDC) has similar effects on biodiversity as topography.

The GD is a poor predictor of mountain diversity in large part because interactions between mountains and biodiversity are complex^{Error! Reference source not found.}, and thus *geodiversity could not be treated as a index*.

RESPONSE:

As a summary of the above. Here we just criticize the practice to treated the geographical diversity as index. The review #3 also indicated “*Geodiversity measures are supposed to capture a considerable proportion of this complexity.*” and “*It can and has been treated as an index in some studies, and in other works, single parameters were used. Thus, the sentence, as formulated, is wrong.*”.

In our opinion, the meaning of review #3 is “There are a thousand geodiversity in a thousand people's eyes” !!! On the other hand, since many parameters (such as elevation, soil type, annual mean temperature, etc.) can be used as substitutes of geodiversity. Why not we bypass the vague notion of geodiversity. Why not just explore the effects of combinations of parameters on biodiversity. There are still many questions about the application of geodiversity in current research.

In contrast, landform type is an objective description of the present state of erosion of a mountain or bedrocks^{Error! Reference source not found.}.

RESPONSE:

“*Why in contrast?*”. Here is a comparison of landform types and GD. As a characteristic variable, landform type is easy to be understand. However, how to use geodiversity in actual research is confusing. Please also refer to the upper response.

“*Geodiversity is also “objective”.*” Geodiversity is still a research direction to be discussed. Please refer to the previous reply.

When we stand in front of a karst mountain, we can see quite clearly that the plants are different from those in a granite mountain.

RESPONSE:

So what's wrong with whis statement?

We propose that landform identity rather than GD, is a more suitable indicators of geodiversity^{Error! Reference source not found.}, *for tracking changes in mountain floras along geological time scales.*”

RESPONSE:

The erosion rates between different types of rocks are differs. More knowledge about species

Commented [A22]: Not clear – which pattern?

Commented [A23]: No, it is not, as has been shown in previous studies.

Commented [A24]: Geodiversity measures are supposed to capture a considerable proportion of this complexity.

Commented [A25]: It can and has been treated as an index in some studies, and in other works, single parameters were used. Thus, the sentence, as formulated, is wrong.

Commented [A26]: Why in contrast?

Commented [A27]: Geodiversity is also “objective”

Commented [A28]: This is not new and is basic textbook knowledge. E.g. compare works in the Alps from the mid 20th century and thereafter.

Commented [A29]: Part 1 of the sentence – refers to patterns, current;

Commented [A30]: This refers to timing – but what is the connection to the first part of the sentence? This is not clear.

assemblages in mountain can be gained by comparing differences of floristic composition of mountain flora in neighboring regions (but differ in landform type). In addition, there are also different evolutionary stages within a single landform types (indicated by classical Davis' geographical cycle). By comparing the synergies between species assemblages and landform development stage, the temporal changes of mountain biodiversity can be traced.

Q17: Line334:

“Based on our results, we put forward the "geological lithology hypothesis of flora" to explain the assemble and differentiation of mountain floras. In this theoretical framework, floristic assembly in mountains is driven by the lithospheric cycle, which refers to the bedrock-constrained developmental processes of landforms. Specifically, under this hypothesis, the mountain species differentiation closely related to the type of bedrock and degree of erosion, species richness and species composition in mountain flora are interaction result of landform and environment, and phylogenetic niche evolution^{Error! Reference source not found.} can promote the spread of species among different landform floras. Overall, our study provides a novel framework and approach for determining the mechanisms of species diversity within mountains and the distributional patterns of some of the world's richest floras.”

This section needs to be improved and provide a more balanced view on previous work by other authors and suggestions put forward here. As I had already suggested in my review of the first manuscript draft, the manuscript here needs to go into some more depth concerning previous hypotheses put forward, importantly the MGH, and I still don't see this has been done. While only briefly mentioned in the into, the MGH does not show up here in the discussion. I would like to see some of what the authors have answered in their rebuttal be actually also included here in the manuscript text.

Response: This is a great suggestion. We rewrote this section. Please see line 389-413.

“Based on our results, we propose the "geological lithology hypothesis of flora" to explain the assembly and differentiation of mountain floras. In this theoretical framework, floristic assembly in mountains is driven by the lithospheric cycle, which refers to the bedrock-constrained developmental processes of landforms. Specifically, under this hypothesis, montane species differentiation is closely related to the type of bedrock and degree of erosion. Both SR and species composition in mountain flora result from interactions between the landform and the environment. In addition, the dispersal of plants between different landform types is more restricted than within the same landform type. Successful diffusion across a landform is often accompanied by the emergence or radiation of adaptive traits. This is called phylogenetic niche evolution^{Error! Reference source not found.} and can promote the spread of species among different landform floras. This differs from the MGH, which assumes that the montane biodiversity hotspots require three key boundary conditions: 1) the presence of lowland, montane and alpine zones, 2) climatic fluctuations to produce a “species pump” effect, and 3) high-relief terrain with environmental in a given mountain region)^{Error! Reference source not found.}^{Error! Reference source not found.}. In contrast, our hypothesis suggests that montane bedrocks and landform processes determine the geographic distribution of plants. The MGH effectively explains the high biodiversity of mountains characterized by large elevational differences (such as the Himalayan and Andes mountains), but such restrictive boundary conditions constrain the applicability of the hypothesis. Biodiversity hotspots also occur in regions with stable climates, such as the Namib desert^{Error! Reference source not found.}, or with with minimal elevational gradients, such as the Southeast Asia karst landforms^{Error! Reference source not found.}. We argue that the novel "geological lithology hypothesis of flora" could serve as a general explanation for global diversity patterns, as the formation of mountains on the Earth's surface is the result of the cycling

of sedimentary, igneous and metamorphic rocks. In conclusion, our study has highlighted the floristic patterns of different landforms and provided a novel framework for studying the mechanisms of plant species diversification within mountains and the distributional patterns of mountain floras of the world.”

REV: Similar to what I mentioned for the text above does also apply here. The authors need to pay attention to content correctness (e.g., when referring to the MGH, and comparing to their hypothesis), precision in formulation and use of terminology (e.g. concerning PNC) and claims. In addition, it appears to me that especially in the last third of the text, different topics are being confused/intermixed. The MGH refers to mountains, but the authors talk about the “Namib desert” and “SE Asian karst landforms” for which the MGH would not apply anyway. They also talk about “global diversity patterns” when referring to their hypothesis. The MGH specifically deals with mountains, the hypothesis by the authors is supposed now to refer to global phenomena? If so, do the hypotheses actually deal with different matters? I strongly encourage the authors to carefully revise the text, and stick to what the hypotheses were developed for. Also, point in time geographic distribution of plants, and processes acting in concert, appear to be compared despite being different. As such, the text is not clear for the reader.

RESPONSE:

Thanks for the questions raised by the reviewers. We consider the differences between our new hypothesis and MGH in the spatial scale. In terms of scale differences. The MGH focus the biodiversity of a mountain which usually the presence of lowland, montane and alpine zones. This is a hypothesis of alpha diversity. In contrast our theory is more concerned with the process of mountain flora differentiation, which is a hypothesis of beta diversity. In this study, we investigate the flora assemble differences between mountains with different landform type.

We made further revisions to the manuscript to mad it clear. Please see line 376-381 *“To explain montane species diversity, this hypothesis differs from previous assumptions, such as MGH^{Error! Reference source not found.}, which is more focus on the biodiversity hotspots in high-altitude mountainous areas and the cause of diversity. Our hypothesis focuses more on mountainous areas with different altitudes and geological and lithological types, as well as their floristic compositions and differences, which is important for explaining differences in biodiversity hotspots from both low- to high-altitude area^{Error! Reference source not found.}.”*

REVIEWER COMMENTS

Reviewer #3 (Remarks to the Author):

Review of revised manuscript version

I have left my comments in both the rebuttal as well as in the manuscript file.

The main manuscript file only contains those few annotations that are in addition to what I commented on / corrected in the rebuttal (the latter corrections/comments need to be implemented in the main manuscript revision as well).

Some of the authors' answers could not be understood due to language, and I have indicated those. In many cases, the intention of my previous comments had been that the text of the manuscript to be adjusted in language, terminology, and precision, often only requiring minor changes. However, some of the authors' answers in the rebuttal indicated to me that this was interpreted differently by the authors. I hope that my new explanations may now be clearer and more easily understood.

Altogether, the manuscript seems on a good way.

[editorial note: see the following pages for the mentioned commented on rebuttal file]

Thank you very much for your effort reviewing our manuscript and for your positive comments. We have revised the manuscript following your suggestions, which has led to a substantial improvement of the manuscript. The new manuscript has also been carefully grammatically revised by the Springer Nature Author Services. We provided point-by-point responses to your comments as following. Our new response is in red colour "RESPONSE: ".

Reviewer #2 (Remarks to the Author):

The manuscript is much improved from the last round and reads well generally. I also very appreciate the responses and revisions with regard to my comments. In particular, the interaction terms added interesting perspectives to the data and actually strengthen the paper. It's a lot of results to report here but the overall message is strong: landforms are important factors of floral diversity. The other issues with phylogenetic analyses seem more difficult to resolve and acknowledging the limitation there would be sufficient.

Minor comments:

A lot of the citations are not showing properly, but as "Error! Reference source not found"
RESPONSE:

Thank you for the reminder. We checked the MS and suspect this situation may be caused by software incompatibility. Because our manuscript was written in WPS. Anyway, we re-examined the citations carefully and provided new manuscript which written in Microsoft Office.

Is the first paragraph supposed to be an abstract? It looks quite out of place. The mention of "landform development" is a bit sudden

RESPONSE:

Yeh, the first paragraph is abstract. We've taken your advice and rewritten this section to made it more concise. The revised abstract is below:

"Although it is well documented that mountains tend to exhibit high biodiversity, whether and how geological processes affect the assemblage of montane floras remains unclear. Here, we address this knowledge gap by exploring landform-specific differences among mountain floras based on a dataset of 17,576 angiosperm species representing 140 well-studied Chinese mountain floras. Our results show that igneous bedrock (granitic and karst-granitic landforms) is correlated with higher species richness and phylogenetic overdispersion, while sedimentary bedrock (karst, Danxia, and desert landforms) is associated with opposite, i.e., phylogenetic clustering. Furthermore, landform type was the primary determinant of the assembly of evolutionarily older species within floras, while climate was a greater determinant for younger species. Our study indicates that landform type not only affects montane species richness, but also determines the composition of montane floras. To explain the assembly and differentiation of mountain floras, we propose the "floristic geo-lithology hypothesis", which highlights the role of bedrock and landform processes in the assemblage of mountain floras and provides novel insight for future research on speciation, migration, and biodiversity in montane regions."

L31: remove "analysing" to make the rest of the sentence about a "topic"; and move "processes" to after geological to improve clarity.

RESPONSE:

Commented [A1]: This is good to know.

However, I found the replies to the reviewer comments being of less good quality and in parts difficult to follow or understand.

In the following, I only make comments where I consider things to be noteworthy that should be examined further by the authors.

I have marked my new comments in yellow to avoid confusion between previous and new annotations.

Commented [A2]: Delete "whether and"

"remains unclear"  "is a matter of ongoing research"

Thanks for the kindly advice, and also the previous response. It seems this sentence is a bit redundant, so that we deleted this sentence.

The new statement in abstract is “*Although it is well documented that mountains tend to exhibit high biodiversity, whether and how geological processes affect the assemblage of montane floras remains unclear. Here, we address this knowledge gap by exploring.....*”. Please refer to the previous reply.

L63: do you mean animal diversity?

RESPONSE:

“*Furthermore, the evolutionary history of plant species also affects biodiversity.*” The literature cited here is an example of animal studies, which may leads to misunderstandings. In fact, evolutionary history has an effect on whole biota. We revised this sentence to improve clarity. See line 57-58: “*Furthermore, the evolutionary history of each biological taxa also affects local biodiversity.*”

Commented [A3]: taxon, not taxa

Commented [A4]: This is a rather awkwardly formulated sentence, I suggest to re-formulate this.

L64: add "and" to before "should include" and delete "includes"

RESPONSE:

Done. See line 59: “*.....It is clear that an integrated framework is needed for the prediction of montane biodiversity, and it should include ecological processes (e.g.,)*”

L105: change "sum" to "collection" or "community" and remove "families, genera, and" unless you mean there are taxa that are not identified to species but also included here.

RESPONSE:

Done. This sentence has been revised as “*Here, we used the term “flora” to refer to the collection of all angiosperm species growing on a specific mountain or in a well-delimited area.*”

L107: I think I get your point but it needs to be more explicit. How about mentioning species diversity only one characteristic of a flora and can be quantified by species and phylogenetic diversity?

RESPONSE:

Thank you for your valuable advice. This section has been revised to made it more concise. See line 102-104:

“*Then, the species richness (SR), phylogenetic diversity (PD=Faith's PD), phylogenetic structure indices (PDI, NRI, NTI), and mean divergence times (MDT) were calculated for each of the 140 floras. (Extended Data Table 2, see Methods). Finally, we constructed regression models (1) using.....*”

Commented [A5]: delete “the”

Figure 1: please add the names of the landforms to the figure to improve clarity for the broad audience of the journal. It was very hard for me to matching things up using the caption.

RESPONSE:

We accept this suggestion and add the names of the landforms to the figure 1. Additionally, in order to more intuitively reflect the impact of the five landform and lithological types on the floras, we have made made appropriate modifications to the original Figure 1. The Danxia series has been placed under the other four landform types. This modification does not affect the principle of the figure, so it is appropriate. Please see the modified figure 1 below:

L152-153: not sure what the last sentence means. They all still look significant in the full model.

RESPONSE:

The last sentence is “*However, the partial landform effects on SR are often superimposed with environment variables and are difficult to separate.*” This sentence emphasizes the interactions effect between landform effects with climate. It seems a bit repetitive to previous result description, we deleted it in the new MS.

L154: add "(towards eastern China)" to help guide non-expert audience.

RESPONSE:

Done. This sentence has been revised as line 147-148 “*The mountain floras with higher SR were mainly located in the monsoon climatic zone of eastern China (Fig. 2).*”

L183: the point about higher rates of extinction needs further explanation and should probably be in Discussion - this happens at several places in the Results section (e.g. the paragraphs starting at L250 and L281)

RESPONSE:

The original paragraph mentioned by the reviewer is:

“*Thus, species with the deepest phylogenetic divergences occur in karst landforms, while species with the shallowest divergences occur in desert landforms. This is consistent with some fossil evidence. For example, the fossil flora discovered in southwestern China indicate that local karst vegetation may have existed since the early Oligocene*” Error! Reference source not found. Error! Reference source not found.

Here, we would like to indicated the species in the limestone mountains are generally had earlier diverge age and could survive for a long time. Especially those species can adapted to the arid habitat of the limestone mountain top. We present an example, the Oligocene fossil flora Wenshan basin which located in Yunnan, China. The vegetation (infer from fossil data) and genera composition of this fossil

Commented [A6]: Poor language (use of tenses), up to a degree which makes understanding of the meaning difficult for readers.

flora in is very similar to the current karst flora in Wenshan, China.

For the second question, “Discussion” occurs in Result section. The reason we did this is we want to give a brief explanation of the results or compared to previous study. We thought this would help readers understand the meaning of our results more quickly, and also avoid repetition before and after the article. We notes this approach is generally be accept in the articles published in Nature Communications. In the later “Discussion” section (line 286), we further discussed the relationship between landform process, species richness, local speciation, dispersal in mountain flora.

Commented [A7]: Due to poor language really hard to understand.

Figure 3-4: It's very easy to lose track of all the abbreviations but I see NTI is not in Figure 3 and SR is not in Figure 4. It would improve clarity if Figure 3 and 4 have consistent panels, though showing the landform and full models respectively.

RESPONSE:

Thanks for the suggestion. In the previous Figure 3-4, we have only presented the the most representative result for the sake of brevity. While, keep the variables in Figure 3 and 4 have consistent panels sounds better. Following your suggestion, we revised the Figure 3 and Figure 4. Please see below:

Figure 3. Differences in species richness, phylogenetic diversity, phylogenetic structure, and age of floras among different landforms.

Figure 4. Standardized coefficients of determination for species richness, phylogenetic diversity, phylogenetic structures and divergence times of mountain floras.

L303: "recognised as positively correlated to..." ("significantly" is usually unnecessary as we can't say "correlated" if it's not significant).

RESPONSE:

We have incorporated this suggestion. All of the similar questions were revised in the manuscript.

Reviewer #3 (Remarks to the Author):

For convenience, I have provided my detailed comments in the "response to comments" file.

RESPONSE:

As many comments put forward by reviewers#3 in our previous response letter were marked as annotations. We have retained the original appearance of the parts that the reviewers still have questions. Our new response is in red colour "RESPONSE:". Hereafter are details peer-to-peer responses.

REV: According to my observations this is not correct, at least not for all parts of the manuscript. For example, revised M&M texts are of higher quality than other parts of the manuscript which are poorly written and where writing needs to be substantially improved to be deemed acceptable – and understandable to the readers. There are many sentences which are neither understandable in terms of language nor scientifically correct (in content and use of terminology).

Some single errors I have marked using magenta, but for other larger sections that need re-writing I mention this in my text.

RESPONSE:

Thanks for this suggestion. The new manuscript has been improved base on the Springer Nature Author Services. We suspect the new manuscript could meet the requirements of the reviewers and also of the journal.

This document certifies that the manuscript

Landform and lithosphere development drive the assembly of mountain floras in China

prepared by the authors

Wan-Yi Zhao, Zhong-Cheng Liu, Shi Shi, Jie-Lan Li, Ke-Wang Xu, Kang-You Huang, Zhi-Hui Chen, Ya-Rong Wang, Cui-Ying Huang, Yan Wang, Jing-Rui Chen, Xian-Ling Sun, Wen-Xing Liang, Wei Guo, Long-Yuan Wang, Kai-Kai Meng, Xu-Jie Li, Qian-Yi Yin, Ren-Chao Zhou, Zhao-Dong Wang, Hao Wu, Da-Fang Cui, Zhi-Yao Su, Guo-Rong Xin, Wei-Qiu Liu, Wen-Sheng Shu, Jian-Hua Jin, E. Boufford David, Qiang Fan, Lei Wang, Su-Fang Chen, Wen-Bo Liao

was edited for proper English language, grammar, punctuation, spelling, and overall style by one or more of the highly qualified native English speaking editors at SNAS.

L566-567: “Determinants of SR might change with landform type, and we therefore test for interactions between landform and others predictor variables (only shown significant variables in full model).”

RESPONSE:

Revised.

L203-204: “P value of T-test result between each pairs of landforms were showed above the black line: **** $p < .001$; *** $p < .001$; ** $p < .01$; * $p < .05$; ns = not significant.”

RESPONSE:

Done. Revised “were showed” as “are shown”.

Question about previous response.

Second, based on the technical requirements (this is based on the null model assumption) of the software package, flora data is extracted from the mountain species database to reflect the phylogenetic structure of the flora. Thus, a global or regional tree is not needed here. In fact, as we know there isn't currently an approach that is calculates a regional phylogenetic structure based on a global tree.

REV: The meaning of this sentence is not clear to me. Of course, there exist studies that have been using e.g. large angiosperm-wide trees as source for creating much smaller trees with only regional species samples representation. This needs clarification.

RESPONSE:

“Using e.g. large angiosperm-wide trees as source for creating much smaller trees with only regional species samples representation”, which is also the method used in our study.

However, the meaning of “In fact, as we know there isn't currently an approach that is calculates a regional phylogenetic structure based on a global tree.” is we can't use a large phylogenetic tree (included about 30000 species) to calculate the PDI, NRI and NTI of floristic distribution dataset included less species (such as 10 floras in total included 3000 species). The species in the phylogenetic tree should be consistent to the floristic distribution dataset, otherwise would violate the assumption of the null model.

In our study, the calculation of phylogenetic diversity and phylogenetic structure were performed by R packages “PhyloMeasures” (Tsirogiannis & Sandel, 2016; doi:

Commented [A8]: Unfortunately, I don't understand the reply (due to poor quality of English language).

10.1111/ecog.01814). This package is widely used in studying the community phylogenetic structure of floras. The “PhyloMeasures” requires the species pool to be consistent with the phylogenetic tree and distribution database. Otherwise the functions (“mpd.query”, “mntd.query” and “pd.query”) in “PhyloMeasures” would not work. See the screenshot below for the warning in R.

```
> mpd.std<-mpd.query(tree,data, null.model="uniform", T)

Warning: the input matrix has fewer columns than the number of species in the tree.

Error in mpd.query.uniform(tree, matrix, standardize) :
  One of the species names in input the matrix was not found in the tree (Staurogyne_rivularis)
> |
```

REV: This is not clear to me and needs explanation. *Staurogyne rivularis* is a synonym for *Staurogyne spatulata*, only the latter name which is accepted. Not finding “*Staurogyne rivularis*” could mean that the dataset uses the correct name instead, *St. spatulata*, rather than the synonym. Which taxonomic name resolution approach was used by the authors and how was it guaranteed that species were not omitted from the analysis due to synonyms or spelling errors? This could have introduced substantial error.

RESPONSE:

Reviewer raised concerns about errors that may result from inconsistent scientific names. In fact, the taxonomic name included in our mountain dataset of floras had been standardized to Leipzig Catalogue of Vascular Plants (LCVP), which it mentioned and proposed by Reviewer #2 in secondly review (details see **Methods** section in line 403; “*Dataset generation and reconciliation*”). Furthermore, the mega-phylogenies used in our study is “GBOTB.extended.LCVP.tre”, which has also standardize the plant names to LCVP database (Jin & Qian, 2022) (details see **Methods** section in line 414; “*Phylogenetic reconstruction*”). Thus, the species name in phylogenetic tree is consistent to our floristic distribution dataset.

So that the software warning reason here is not the inconsistencies of taxonomic name. The true reason is species number in the phylogenetic tree not consistent to the floristic distribution dataset (more species included in phylogenetic tree than floristic distribution dataset, as the case we shown above, less species in floristic dataset than in phylogenetic tree).

```
Warning: the input matrix has fewer columns than the number of species in the tree. »
```

Please also refer to the previous reply. We hope our response made it clear.

Q4: With regard to the "age" of species, I appreciate the authors' effort to provide additional information by comparing the pruned and unpruned trees.

However, my point is more about the concept of species or assemblage age -- it is a not the real age and can only be used for representing the patterns of age when extinction is random; in contrast, a comparative analysis can be problematic if extinction was not random with regard to the factors that are being examined. Therefore, I suggest rephrasing it in terms of species distinctiveness, so that an assemblage with low MDT contains more closely related species and infer mechanisms from there.

Response: Thank you for clarifying your concerns, and we appreciate your suggestion to use the term “species distinctiveness” to replace “flora age of species age”. Here, we're unsure the meaning of the term “species distinctiveness”. According to our understanding, we assumed the reviewer's proposed ‘species distinctiveness’ to be the evolutionary distinctiveness of species as proposed by Isaac et al., (2007, doi:10.1371/journal.pone.0000296). We suspect there are no differences between

Commented [A9]: I still don't fully understand from this reply why the error message would complain about the missing “species names” if the names would have been resolved prior or during analysis, and – I assume - measures been taken to include missing data? I am missing the context between the previous and the new replies. I still don't fully understand the connections from the reply here.

“species distinctiveness” and species age, because both of them are based on the **in-clade** in the phylogenetic tree. Besides, species age (or species divergence time) is more well known than “species distinctiveness” in evolutionary biology and biogeography.

REV: What do the authors mean by “in-clade” ?

RESPONSE:

Here, “in-clade” just represents the branches of each species contained in the phylogenetic tree.

On the other hand, we accept that the concept of “flora age” is problematic. The complexity assembly history of flora makes it impossible to interpret the real age of flora. In contrast, the age of the species, which estimate based on the molecular clock hypothesis (Ho, S., 2008; The Molecular Clock and Estimating Species Divergence) and fossil calibrated, is more reliable. In fact, the inference of floristic history is often based on the estimated divergence time (or stem age) of species which occurs in the flora (such as Dagallier et al., 2020, doi: 10.1111/nph.16293; Qian & Deng, 2022, doi: 10.1111/jse.12856; and Chen et al., 2018, DOI: 10.1093/nsr/nwx156).

In our study, we used the mean divergence times (MDT) proposed by Lu et al.(2018) to represent the “flora age”. As such, the MDT reflected the age composition of species within a flora. A flora with larger MDT has more ancient species and hence is expected to have older floristic ages. **Overall, we believe that our use of MDT to measure the species age structure of flora is consistent with the existing literature and therefore is appropriate.**

REV: Not clear to me. Only because many people have been doing something does not mean it is correct or the most appropriate way to do it.

RESPONSE:

Reviewer # 3 mentioned that “many people are doing the same thing, which does not necessarily mean that this approach is correct”. We cannot fully agree with the comments of Reviewer #3. Many people have used this method to carry out a lot of work, although it cannot be considered right, it cannot be considered wrong, and it cannot be considered that several journals currently publish the wrong method? Especially, if there is no better update method available, traditional methods can only be used.

However, frankly, it is difficulty of develop a better or appropriate method to measure the species age structure of flora. Because species age occurs in a flora is still can't be well defined. At present, we can only study the floristic assemble process on a macroscale of statistical significance. So that in our study we used the MDT method proposed by Lu et al. (2018), which is not so bad. Development new methods to understand the species assemble process of flora over time scale is an important topic for future research.

In addition, in this article, we only hope to calculate the aggregation and divergence of plant flora at a macro scale. Therefore, we used the MDT method proposed by Lu et al. (2018). The limitations of this method have been explained and revised as previous reviewer comments (such as MS Line 461-470). Developing new methods to reveal the species composition of plant flora on a temporal scale is an important topic for future research. Of course, we would greatly appreciate it if the reviewer could recommend a better method to us.

L64: “It is clear that an integrated framework for the prediction of montane biodiversity is necessary. *Error! Reference source not found.*, *Error! Reference source not found.*, *Error! Reference source not found.*, **should include** **includes** **ecological processes** (e.g., survival, competition, and niche differentiation) *Error! Reference source not found.*, **biological processes** (e.g., species divergence and extinction) *Error! Reference source not found.*, *Error! Reference source not found.*, and geological and lithologic processes (e.g., orogeny and rock

Commented [A10]: I still don't understand from this reply what the authors mean by this.

Commented [A11]: Unfortunately, I don't fully understand all parts of this reply (due to poor quality of English language).

However, from what I understand the authors claim here, I want to assure the authors that I was actually never asking to “develop a new method”. Rather, I was pointing out already previously the limitations of the approach, that when using a local geographical species pool for divergence time estimations, “true” ages of species cannot possibly be inferred. Why? Because the closest relatives of those species (which may occur elsewhere and are therefore not included in the tree) would be required to be added in such a dating analysis, to infer better estimates of the species' ages. Likewise, extinction will likely be different in different taxa/clades (e.g. due to taxon age, and/or geological/climatic events in different mountain regions), and a tree which is missing closest relatives will then be even more prone to overlooking this effect. This was the reason I was referring to “random” distribution of extinction.

The authors previously replied that they do not intend to present the “true” ages, but just the relative ages. Nevertheless, even then, because of uneven taxonomic sampling, bias will naturally be introduced. This limitation should be mentioned.

The authors claim that other studies have done similar as they do. However, at the very least, the limitations of this approach should be mentioned in the manuscript. The answer by the authors does not invalidate my calls for caution.

formation)^{Error! Reference source not found.,Error! Reference source not found.}”

REV: Ecology is a branch of biology, authors may rather use “evolutionary processes” than “biological processes”.

RESPONSE: We accept this suggestion. Revised as line 58-61 “*It is clear that an integrated framework is needed for the prediction of montane biodiversity*^{Error! Reference source not found.,Error! Reference source not found.,Error! Reference source not found.}, and it should include ecological processes (e.g., survival, competition, and niche differentiation)^{Error! Reference source not found.}, evolutionary processes (e.g., species divergence and extinction)^{Error! Reference source not found.,Error! Reference source not found.}, and geological processes (e.g., orogeny and lithosphere cycling)^{Error! Reference source not found.,Error! Reference source not found.}”

L82-87: “*Although it is well known that climate change forces plant species migration, those species which are adapted to local bedrocks are constrained in their ability to migrate*^{Error! Reference source not found.} *Bedrock geochemistry is on par with climate as a regulator of vegetation in granitic mountains*^{Error! Reference source not found.} *Some studies also suggest that local species diversification processes are consistent with edaphic rather than climatic filtration, such in the Cape flora*^{Error! Reference source not found.}, *Teesdale flora*^{Error! Reference source not found.}, and *New Caledonian flora, in which ca. 50% of the endemic floristic elements are ultramafic-obligate species*^{Error! Reference source not found.}”

REV: I am unhappy with the statement that geochemistry, which is a field of research, is a regulator – authors needs to take care with precision in their statements.

RESPONSE:

Reviewer # 3 mentioned our statement “*Bedrock geochemistry is on par with climate as a regulator...*” is incorrect. This statement is cited from previous literatures “*These results are important because they demonstrate that bedrock geochemistry is on par with climate as a regulator of vegetation in the Sierra Nevada and likely in other granitic mountain ranges around the world.*”, which occurs in Hahm et al. (2014, <https://doi.org/10.1073/pnas.1315667111>). In fact, when mention bedrock geochemistry, it is usually understood as the chemical elements in bedrocks (eg. “*Bedrock geochemistry influences vegetation growth by regulating the regolith water holding capacity*”).

Anyway, to avoid ambiguity. We revised this sentence as line 79-80 “*The geochemical characteristics of bedrock is on par with climate as a regulator of vegetation in granitic mountains*^{Error! Reference source not found.} *Some studies also suggest that local species diversification processes are consistent with edaphic rather than climatic filtration, such as in the Cape flora*^{Error! Reference source not found.}, *Teesdale flora*^{Error! Reference source not found.}, and *New Caledonian flora, in which approximately 50% of the endemic floristic elements are ultramafic-obligate species*^{Error! Reference source not found.}”

Q9: Line90:

“*Here, we apply the term “flora” to refer to the sum of all angiosperm families, genera, and species growing on a specific mountain or in a well-delimited area*^{Error! Reference source not found.,Error! Reference source not found.,Error! Reference source not found.}, which is a relatively independent biogeographical unit^{Error! Reference source not found.}”

I would argue against the flora of a mountain being a *relatively independent biogeographical unit*, if this is what is meant here (but I may have misunderstood the sentence). First, because related lineages are shared between mountains, and second, because most biogeographers, in terms of terminology, would not consider the flora of a mountain as a “biogeographical unit”.

Also in **Pre-response**: “*we mentioned is not only a unit, but als*” I am not sure how “unit” here would refer to the biogeographic units mentioned in the ms. This may be worth of clarification.

Response: We appreciate the key question raised by the reviewer. In generally, the term “biogeographic unit” is used in the context of biogeographic regionalization.

Here we consider that the flora of mountain and **biogeographic zones** is equivalent at small scale.

REV: This is unclear. Which biogeographic zones are meant here?

RESPONSE:

(1) Here, the mountains wasn't refer to a single peak, but rather to a mountainous area or a certain nature reserve, including many peaks and adjacent areas. Correspondingly, all the plants growing in this area is consist of many different species, genera, and families, whose was called on a mountain flora, it is a relatively independent natural geographical area. Due to differences in longitude, latitude, altitude, and climate factors, biological factor (or floristic compose) among different mountainous regions, these geographical regions are also bound to have differences.

(2) The biogeographical units mentioned in the main text cannot be equated with biogeographical divisions. Biogeographic zoning often divides any geographical areas into several level or grades, such as the global flora, which is often divided into kingdom, region, province, and county. These are different biogeographical units (hierarchical units). The unit itself does not have hierarchical significance, and only after studying all species in the region, namely Flora, its hierarchical level or grade be determined, such as the local flora, the geographical elements of the families, genera, or species (tropical, temperate elements), historical elements (antiquity), originative elements (endemic families, genera, species etc.), ecological elements, migration elements, etc., Only through floristic phyto-geography analysis, a detain units such as a kingdom, region, province, and county be divided.

A certain biogeographical unit, such as a mountain area, may also be divided into different geographical regions, such as the Hengduan Mountains in China and the Himalayan region, which are divided into the East Asian Flora and the China-Himalayan Forest Subregion. In the east, it is divided into the East Asian Flora and the China-Japan Forest Subregion; Correspondingly, the latter can be divided into South China Province, Central China Province, etc. (Wu Zhengyi et al., 1996). A certain mountain area, depending on its nature geography and biogeography property, they may divided into a hierarchical unit (division) or a subunit. Or they could be belonged a province, or a county.

Therefore, when we mention a mountain region, or a mountain flora, similar to a nature geographic or biogeographical area or unit with a certain grade or no grade, it is not a wrong concept.

(3) This article studied 140 mountainous areas, listed the natural geographic information of each mountainous area, and correspondingly formed 140 local mountainous floras, thereby showcasing their differences in natural geography and floristic phyto-geography.

Furthermore, this article successfully classified 140 mountains into five geomorphic types (by searching for detailed geological survey data, and special investigation report on Floras), revealing and analyzing the relationship and possible reasons between these local floras and mountainous lithology. We believe that this is a progress. On this basis, it will be possible to further study endemism, geographical elements of those floras, and diffusion, migration of floristic elements between different geomorphic types and different bedrock, floras.

(4) Of course, in our response to the reviewer's question, we wrote biogeographic zones, which are indeed not strict enough and have a larger scope. We just wanted to explain that each mountain has its unique characteristics. Wu Zhengyi et al. (1996) considered that an area of every relative independent mountainous flora covered should not be less than 100 square kilometers. Due to the large amount of content and limited space, these concepts and data were not included in the main text.

Commented [A12]: The authors present some general textbook biogeographic contents here. I assume many readers of *Nature* will be familiar with these.

However, I could not find a clear connection to my previous claim that I would not generally agree to “the flora of a mountain being a *relatively independent biogeographical unit*”.

My point was to call for caution when using terms such as “*independent biogeographical unit*”, or “*biogeographic zone*”.

This needs to be taken care of as it might lead to confusion among readers. As the authors point out themselves in their reply, biogeography has its own terminology and concepts.

First, mountain is geographical barrier to plants diffusion.

REV: This is not correct (as I had stated in the previous rounds of review already), phrased in this general manner. It depends on the orientation of the mountain. Only if a mountain has W-E orientation, it can act as barrier (e.g. Himalayas, Alps). For N-S-oriented mountains (such as the Andes, or the Hengduan mountains), this is not correct.

RESPONSE:

We believe that it is unnecessary for the reviewers to oppose the above views. The author's viewpoint is simply that mountains can affect the diffusion of species or serve as a barrier of species diffusion. In fact, mountains are both a barrier for species diffusion and may also play a promoting role in species diffusion. For example, the north-south direction mountains play a promoting role in the north-south migration of species, while serving as a barrier for the east-west migration of species; Due to the existence of Hengduan Mountains, many genera and species in Chinese Mainland have formed China-Himalaya distribution subtypes and China-Japan distribution subtypes, with Hengduan Mountains as the boundary. Similarly, due to the barrier effect of the Nanling Mountain in China, many tropical genera and species cannot exceed the Nanling Mountain, such as *Pandanus*, *Endospermum*, etc.

The reviewer mentioned or emphasized the local migration, endemism, and differentiation of species within a mountainous area, especially in high mountains. Our focus is to emphasize that the migration of species between different mountainous areas will be hindered, especially between Danxia landforms, karst landforms, and granite landforms, where species diffusion is not easy.

Second, a mountain usually shows sky island effects and contain their own unique clades or several endemic species.

REV: This is not correct. First, there is no such thing like a “sky island effect” (phrased this way). Second, only a small fraction of the world’s mountains qualify as sky islands – only, if their vegetation is drastically different from the surrounding lowlands. This may only be the case if plants living on the mountains are different from the lowlands and were therefore not likely recruited/evolved from lowlands in which case the nearest relatives would come from other mountains (example: some mountains located in drylands, e.g. Africa).

RESPONSE:

Here is just a brief discussion with the reviewer. The “sky island effect” of mountain flora is a question worth further study and discussing. As our understand, in essence the sky islands mountains is qualified by its flora showed discontinuous distribution. In generally, in mountains vertical zonality of vegetation on the elevation gradient is widespread observed. We suspect the plants or community occurs on the top of the mountain should be treated as a result of sky islands effects? These community differs from the low land, and could represents the early stage of species assemble of mountain flora.

There are also some literature to suggest that mountain region could be viewed as a biogeographical unit. For instance, Rahbek et al. (2019, Science 365, 1108-1113) wrote that “Like an island, a mountain region may be viewed as a biogeographical unit in itself, with in situ speciation and extinction playing a key role in building the regional species assemblage”.

REV: Is anyone else except this cited paper claiming this as well? I could imagine that most mountain bioeographers would disagree with this oversimplification.

RESPONSE:

We cannot agree with the reviewer's opinion, as stated in the previous response.

Line 311-313 “Landform developmental processes are associated with local bedrocks and regional

Commented [A13]: My point was simply that the statement is not correct, when phrased in this general manner. This is still valid, and thus I suggest, now more concretely, to rephrase this:

... mountains can act as barriers to plant diffusion ...

Commented [A14]: What I was intending to suggest to the authors by my previous comment was to tune down “a mountain usually shows sky island effects”. First, because “usually” is not correct, as this would imply this statement holds true for the majority of mountains, which is not the case. Secondly, “sky island effects” is unclear to me. What is a sky island effect? Has this even been defined, and by whom?

In short, what I had intended with my previous comment was to re-formulate this.

Commented [A15]: See my answers before. My point was to call for caution when using terms such as “independent biogeographical unit”, or “biogeographic zone”.

The citation from Rahbek et al. “may be viewed as a biogeographical unit in itself” seems like an agreeable cautionary phrasing.

It is different from the authors’ statement “independent biogeographical unit”, or “biogeographic zone”.

climate^{17,59}. The assembly of the local flora is impacted by a combination of geological and climatic processes, and biological processes.”

REV: This sentence is devoid of logic; the first matters mentioned do not automatically cause the second. The sentence as such is therefore not correct. I suggest to re-phrase this – as corrected above (using tracked changes).

RESPONSE:

Thanks for the kindly advice. We accept this suggestion and revised as line 297-298 “*The assembly of a local flora is impacted by a combination of geological and climatic processes, as well as biological processes*”^{Error! Reference source not found.,Error! Reference source not found.,Error! Reference source not found.}”

Q14: Line313: “*The mountains of sedimentary bedrock have much stronger environmental filtering effect, as these mountain ecosystems are more sensitive to rainfall. The environmental filtering effect further promoted the clustering of phylogenetic structures of mountain floras as predicted by the phylogenetic niche conservatism (PNC)*”^{Error! Reference source not found.,Error! Reference source not found.,Error! Reference source not found.}”

It would be interesting to discuss here radiations more generally, e.g., how do the studies of plant radiations included in the global study by Muellner-Riehl et al. 2019 (JBI) compare to these statements/findings? Are the findings of these studies (and other studies published since then, i.e. after 2019) comparable to this? I suggest looking at the mountain systems (and their bedrocks) where these studies were undertaken.

Response: These are really good advice. The potential radiation speciation occurs when plants colonize from one landform to another which maybe an important path of plant diversification. Here, we document several cases (not all) on adaptative radiation that occur on the bedrocks. But the relations between radiations and bedrocks/landform are still poorly known. We suspect that the effect of landform and bedrock, which promote species differentiation, is more or less neglected under the shadow of “climate change”. Nevertheless, we have reorganized this paragraph carefully. REV: I don’t understand the reasoning here, as currently phrased. It is unclear what the authors mean (potential speciation radiation?). There exist different kinds of radiations, not all of them are “adaptive”.

RESPONSE:

Here we would like to shown adaptive evolution (associate with morphological traits and physiological traits) maybe the most important mechanism for plant to radiation in a new environment. Although the mechanisms of radiative evolution are diverse, plants ultimately need to adapt to their local environment in order to survive. Furthermore, evolutionary radiations underpinned by variation in physiological or behavioral traits can more easily be perceived as non-adaptive, compared to those involving more conspicuous morphological traits, causing a bias in our understanding of the extent and distribution of adaptive radiations in nature. That is reason a large number of traits of convergence (such as Alpine flora, Arid flora, and mangrove plants) and adaptive evolution have been broadly observed (Nevado et al. 2019, doi: 10.1016/j.cub.2019.07.059; Xia et al. 2021, doi: 10.1093/molbev/msab314).

Please see line 334-348. “*Mountains composed of sedimentary bedrock have much stronger environmental filtering effects, as these mountain ecosystems are more sensitive to rainfall. Such effects further promoted the clustering of the phylogenetic structures of mountain floras, as predicted by phylogenetic niche conservatism (PNC)*”^{Error! Reference source not found.,Error! Reference source not found.,Error! Reference source not found.} The transformation of landforms effectively promotes the plant

Commented [A16]: From this response I cannot not see how my previous request (for discussing this in the ms) is now actually tackled in the manuscript.

radiation, such as the Old World gesneriads and North American deserts rock daisies (Compositae tribe Perityleae). Similar patterns have been observed in insects. For example, two radiation clades (clade nodes 14 to 16 and nodes 31 to 33) of Exocelina resulted from the transition from uplifted Australian Plate rock to ultramafic/ophiolite. Although, many previous studies also document climate change as an important driver of species radiations.

REV: This text needs re-writing, major parts are not understandable. “clustering of the phylogenetic structures”? Authors don’t explain what the connection is between PNC and phylogenetic clustering. “transition”? formulation and meaning unclear.

RESPONSE:

This section is re-written to make it more clear. Please see below.

Line 323-330 “Such effects further promote the phylogenetic relatedness (clustering) of mountain floras because new lineages tend to maintain their ancestral ecological niche.”

In plants, radiative evolution often accompanies habitat and landform shifts, as seen in Old World gesneriads and North American deserts rock daisies (Compositae tribe Perityleae). Similar patterns can also be observed in insects. For example, two radiated clades (nodes 14 to 16 and nodes 31 to 33) of Exocelina resulted from the transition from uplifted Australian Plate bedrock to ultramafic/ophiolite.”

The adaptive radiations of plant genera are more or less associated with bedrock type. For example, the key innovation (lime-secreting hydathodes) of Saxifraga sect. Porphyrium is clearly an adaptation to the limestone bedrock, and the low specific leaf area (SLA) exhibited by Erica is an adaptation to oligotrophic habitats (Quartzite/sandstone) in Cape. Thus, we suggest that landform and bedrock effects strongly promote species differentiation between different regions, even if such an effect has not been mentioned in previous studies.”

REV: “The adaptive radiations of plant genera are more or less associated with bedrock type.” If at all, adaptive radiations can, among others factors, be more or less associated with bedrock type...

The entire text section needs an English language check (grammar, incomplete sentences,...). “different regions, even if such an effect has not been mentioned in previous studies.” – this is not correct. There is a long history of investigations in the European Alps on this matter, dating back to the mid of the 20th century.

“the key innovation (lime-secreting hydathodes) of Saxifraga sect. Porphyrium is clearly an adaptation to the limestone bedrock.” Authors need to carefully read and describe findings in literature, not overstating findings. For example, to my knowledge, some Saxifraga species with lime-secreting hydathodes do actually not grow on limestone – therefore, the sentence as currently phrased does not hold true for all lineages. Care needs to be taken with re-phrasing.

RESPONSE:

Thanks for your criticism. We revised this section. Please see line 331-336: “The adaptive radiation of plants is more or less associated with bedrock type. For example, the development of a key innovation (lime-secreting hydathodes) may have made Saxifraga sect. Porphyrium better suited to limestone

habitats^{Error! Reference source not found.}, and the low specific leaf area (SLA) exhibited by *Erica* is an adaptation to oligotrophic habitats (quartzite/sandstone) in Cape^{Error! Reference source not found.}. These obvious landform and bedrock effects could strongly promote both species and floristic differentiation between different regions^{Error! Reference source not found.,Error! Reference source not found.,Error! Reference source not found.}”

Q15: Line 317-327:

“Dispersal also contributes to **the** mountain diversity^{Error! Reference source not found.,Error! Reference source not found.}. A well-known example is climatic fluctuations during the Quaternary ice age drove changes in the distribution of species around the globe^{Error! Reference source not found.,Error! Reference source not found.}. The spatial configuration of mountain range affect their functional connectivity, influencing species dispersal^{Error! Reference source not found.,Error! Reference source not found.}. In fact, dispersal more easily occurs at initial stage of landform develop mountain, as the slope of mountain still gentle and geographical barrier still relatively low^{Error! Reference source not found.}. Such as the plants dispersal events in Qinghai-Tibet Plateau (QTP, low barrier) is obvious higher than that in Hengduan Mountains (high barrier)^{Error! Reference source not found.}. Furthermore, the effects of dispersal on species diversity in mountainous areas mainly occurred at lowland, and is limited at highlands where has more local endemic species^{Error! Reference source not found.,Error! Reference source not found.}. This means that dispersal has weak influence on the unique composition of mountain flora of different landforms, especially considering that landform have a significant filtering effect on the species.”

The entire paragraph needs re-writing. Dispersal between mountains is not limited to times of climate change, which somehow is suggested here as currently written. Dispersal and establishment was easier in the Hengduan mountains at times of climate change as they have a N-S orientation, enabling plants to change their latitudinal distribution range more easily.

And Pre-response: “Such as the plants dispersal events in Qinghai-Tibet Plateau (QTP, low barrier) is obvious higher than that in Hengduan Mountains (high barrier)⁴⁴.”

I don’t understand this sentence. As outlined previously, it is exactly the other way round. Dispersal and establishment was easier in the Hengduan mountains at times of climate change as they have a N-S orientation, enabling plants to change their latitudinal distribution.

And Pre-response: As I pointed out previously, the opposite would be expected. Survival on mountains with N-S orientation is easier during glacials than on those of W-E orientation (e.g. compare Himalayas versus Hengduan Mts.).

And Pre-response: “This means that dispersal has weak influence on the unique composition of mountain flora of different landforms, especially considering that landform have a significant filtering effect on the species.”

Reading this, I am not sure the authors have read the papers I had suggested to consult, and which they cited in the paper already before (e.g. Ding et al.).

Response: We accept this advice and wrote this paragraph. We would like to clarify the questions about dispersal events and diffusion barrier. We had carefully readed the paper by Ding et al (2020). It is not difficult to deduce the conclusion “dispersal events in Qinghai-Tibet Plateau (QTP, low barrier) is obvious higher than that in Hengduan Mountains (high barrier)” (see Ding et al, 2020, figure 2). On the other hand, we thought dispersal would reduce the β -diversity between different floras. Such as the proportion of endemic species in Hengduan Mountains is the highest is largely affect by its low rate of colonization (<0.05), and **high rate of *in situ* speciation and local and recruitment.**

REV: This could also/additionally be the effect of higher levels of extinction on the QTP, in contrast to the Hengduan mountains.

RESPONSE:

This may be right as the QTP has lower habitat heterogeneity than the Hengduan Mountains. The role of extinction rate QTP and Hengduan Mountains are not mentioned in the paper by Ding et al (2020). A region with low habitat consistency are associated with lower biodiversity and also more species with strong ecological adaptation. Distinguishing the roles of history, speciation and extinction in the mountain flora assemble still represents a major challenge to future research.

In contrast, the low proportion of endemic species in Qinghai-Tibet Plateau has the highest rate of colonization since early Miocene. It is clearly the dispersal is negative correlation to the proportion of endemic species in flora.

REV: I am not pleased with this entire section of text (incl. the one further above), as the main points of criticism still remain to be addressed.

RESPONSE:

We reorganized the section to make our point clear. Our main idea is that differences in landform type have a constraining effect on species dispersal between mountains. For example, species could not spread freely between limestone and non-limestone mountains. Such restrictive effect caused by landform are general associated with the differs of bedrock, soil, water cycle processes.

The revised section is please see line338-359: **Restricted dispersal between landforms as the result of environment filtering** “For many biogeographers, mountains are regarded as both barriers and bridges of species dispersal. The role of mountains as corridors has been documented in several mountains oriented north-south, such as the Andes and Hengduan Mountains. However, the contribution of dispersal to montane floristic diversity largely depends on the ecological and physiological requirements of the species, as well as their dispersal ability. Our research demonstrates the role of landform constraints on the interaction of different landform floras, which is shown in their SRs, phylogenetic structures and species age structures (Fig. 3-4; Extended Data Fig. 9-15). Dispersal occurs more freely during the initial stages of landform development in mountains, when slopes are still gentle and geographical barriers are minimal. The role of local species recruitment is most important during the early stages of mountain floristic assembly, the importance of which is subsequently replaced by local adaptations or in situ speciation. Mountains in different landforms will recruit different plant species as a result of environmental filtering caused by differences in bedrock. For example, mountains composed of limestone bedrock contain more species which are physiologically tolerant of drought and high calcium stress than mountains composed of metamorphic rocks and granites. The landform restriction effect on species diffusion gradually strengthens when more bedrock is exposed and the connectivity between mountains of different landforms is greatly reduced. Variation in the species composition of mountain floras between different landforms increases under the combined effects of landform, environmental filtering, and local endemic speciation. Therefore, we propose that the patchy spatial distribution of different mountain landforms is an important factor in shaping biogeographical zoning.”

Although we have made significant changes to this section. We responded to the reviewer's concerns accordingly thereafter.

The re-wrote paragraph please see lines 349-367. “Dispersal also contributes to montane floristic diversity, although dispersal is only effective if it is followed by successful establishment

Commented [A17]: I don't fully understand* this answer (due to use of English language and terminology), but will need to check the manuscript text whether any relevant amendments have been made to the actual text.

*e.g. which region are the authors referring to as being “of low habitat consistency ...”? Roles of – which/what kind of? – history?

Commented [A18]: They can “spread” (better: disperse), but may not become established if they are not generalist, but specialist species adapted to specific bedrock. Authors need to take care of precise wording (importantly, in the manuscript text itself, for which this is even more important than in the reply text here).

Commented [A19]: See my previous comment. Dispersal and establishment are different matters.

Better/more precise:

Restricted dispersal with establishment ...

Restricted range expansion ...

found. Dispersal occurs more freely during the initial stages of landform development in mountains, when slopes are still gentle and geographical barriers are minimal^{Error! Reference source not found.}. For example, the plant dispersal events across the Qinghai-Tibet Plateau (low barrier) have been considerably greater than that in Hengduan Mountains (high barrier)^{Error! Reference source not found.}. The role of local recruitment was most important during the early stages of mountain flora assembly, a role which was subsequently replaced by local adaptations or in situ speciation^{Error! Reference source not found.}. Mountains in different landforms will recruit different plant species, because each species is differently adapted to specific types of bedrock^{Error! Reference source not found.}. The spatial configuration of mountain ranges alters their functional connectivity, thus influencing the species dispersal process^{Error! Reference source not found.}. Furthermore, during the middle stages of landform development, the effects of dispersal on species diversity in mountainous areas are primarily restricted to lowlands, and are limited in highlands where there are more local endemic species^{Error! Reference source not found.}. Taken together, this suggests that dispersal only weakly influences the unique composition of mountain flora associated with different landforms, especially considering that landforms exhibit a significant filtering effect. Variation in the species composition of mountain floras between different landforms will increase under the significant, combined landform environment filtering effect^{Error! Reference source not found.}, and local endemic speciation^{Error! Reference source not found.}. Therefore, we propose that the patchy spatial distribution of different mountain landforms is an important factor in shaping surface biogeographical zoning.”

REV: Again, this text’s meaning is still largely unclear.

Qinghai-Tibet Plateau (low barrier) have been considerably greater than that in Hengduan Mountains (high barrier) – which kind of barrier do the authors means – and barrier to what?

RESPONSE:

“For example, the plant dispersal events across the Qinghai-Tibet Plateau (low barrier) have been considerably greater than that in Hengduan Mountains (high barrier)⁵⁷.”

– which kind of barrier do the authors means – and barrier to what?”.

The “barrier” here means the change of elevation gradient in Qinghai-Tibet Plateau (QTP) is relatively small than Hengduan Mountains (HDM) (see the fig below, left, cited from Ding et al., 2020). Such low-barrier of QTP is clearly more conducive to species dispersal, which is tested from the assembly of alpine biotas. Since the Miocene, the colonization rate QTP (0.06-0.25) is always large than the QHM (<0.05) (in Ding et al., 2020, Fig.2). In addition, there is no significant increase in species dispersal rate was detected in the Hengduan Mountains (although N-S-oriented) during the Quaternary in Ding’s result (see the fig below, right).

[editorial note: figures redacted]

REV: Role of local recruitment: In times of climate change, local recruitment from lowlands may play a potentially important role.

RESPONSE:

“The role of local recruitment was most important during the early stages of mountain flora assembly, a role which was subsequently replaced by local adaptations or in situ speciation^{24,57}.”

Commented [A20]: When I pose these questions, what I inherently mean is that those points are unclear to the reader from the text as currently presented in the manuscript. Thus, my recommendation in these cases is that the text in the manuscript itself to be clarified by re-phrasing by the authors.

Commented [A21]: As mentioned by me further above, the meaning needs to get through to the reader, by the way this is explained in the actual text.

I appreciate this is explained by the authors here in their reply, but it also – and importantly – needs to be understood from the actual text as presented in the manuscript.

We understand the point “local recruitment in climate change” made by the reviewers. The range of species is expected to shrink during cooling climate and expand during interglacial periods. But that doesn't contradict our viewpoint. Because no matter how the climate changes, species are always more easily dispersed in low geographical barriers. In the early stages of mountain flora assembly, the topographic fluctuation in the mountain is always relatively small.

REV: “Spatial configuration” – meaning unclear.

RESPONSE:

“The spatial configuration of mountain ranges alters their functional connectivity, thus influencing the species dispersal process⁸⁷⁻⁸⁸.”

Here, “Spatial configuration” means topographical configuration, which is associated with the available ecological niches of a mountain along elevation gradient.

REV: “Furthermore,” – but this disregards climatic fluctuations acting on mountains at various times.

RESPONSE:

We're not sure about this question. Do you mean climatic fluctuations facilitated the species dispersal both in lowlands and highlands? We think it's almost impossible that the higher elevations have the same migration rate as the lower elevations between mountains. There may be a misunderstanding gap between us and the reviewers. We are focusing on the dispersal of species between mountains, rather than within it. We have deleted this sentence to make it clearer.

The new statement see Line 349-355 “Mountains in different landforms will recruit different plant species as a result of environmental filtering caused by differences in bedrock^{Error! Reference source not found., Error! Reference source not found.}. For example, mountains composed of limestone bedrock contain more species which are physiologically tolerant of drought and high calcium stress than mountains composed of metamorphic rocks and granites^{Error! Reference source not found., Error! Reference source not found., Error! Reference source not found.}. The landform restriction effect on species diffusion gradually strengthens when more bedrock is exposed and the connectivity between mountains of different landforms is greatly reduced^{Error! Reference source not found., Error! Reference source not found., Error! Reference source not found.}.”

Q16: Line332:

“Thus, landforms are suitable indicators of geodiversity^{Error! Reference source not found., Error! Reference source not found.} for tracking changes in mountain floras along geological time scales.”

I am missing more elaboration on measure of geodiversity. E.g., how does this compare to the geodiversity indices used in previous mountain studies (such as the MGH global study by Mueller-Riehl et al. 2019 in JBI)?

Again, the lack of discussion of work by other authors is misleading and suggests more novelty here than is actually inherent in this present study. Acknowledging previous work by other authors does not diminish the achievements of this study here, but rather empowers readers to compare this and previous studies.

And **Pre-response**: But geodiversity as a measure (geodiversity index) is an integral part of investigations under the MGH.

Response: Thanks for pointing out the shortcomings. There is no denying that linking geological diversity to biodiversity is an important step. However, there are still some problems in the application of geodiversity. In previous studies, geodiversity is usually treated as the environment variables integrated into the compound geodiversity index (GD, or GDCs). But the power of compound geodiversity index was not good enough in predicting species diversity. As the case you

Commented [A22]: This is not generally correct, it depends on whether in a given region/mountain, the climate change will lead to either an increase or decrease of suitable habitat area. This is dependent on mountain morphology (hypsography class).

Again, when I marked something as unclear or in need to more clarity in presentation, what I meant was that the text in the manuscript should be adjusted to make it easily understandable by readers.

Commented [A23]: This sentence is hard to follow.

Commented [A24]:

Here again, when I marked something as unclear or in need to more clarity in presentation, what I meant was that the text in the manuscript should be adjusted to make it easily understandable for readers.

“Topographical” bears more information than “spatial”, but I still don't fully understand the meaning of the sentence if “Topographical” is exchanged against “spatial”.

Just as a note aside: An ecological niche (sensu Hutchinson) is a characteristic of a species, not of the abiotic world. So “available ecological niches of a mountain” is, strictly speaking, not very precise.

mentioned (Muellner-Riehl et al. 2019 in JBI, wrote “The GD index and Elevational range were strongly correlated ($r = .70$) and given that Elevational range showed better performance in single predictor models...”), the geodiversity index (GD) was found to be less effective than a single topography variable (elevational range) for predicting mountain species diversity. In contrast, the landform type (characteristic variable) is more concise and easier to understand. For a detail discussion, please see line 372-388.

“Geodiversity, the variability of Earth’s surface materials, forms, and physical processes, is an integral part of nature and crucial for sustaining ecosystems and their services. Several studies proposed to use geodiversity to model biodiversity based on the hypotheses that specific geo-sites should support unique biota and that high geodiversity is coupled with high biodiversity. However, the predictive power of geodiversity has not been high. For example, the geodiversity index (GD) was found to be no better than elevational range for predicting mountain species diversity. Another recent study reported that combinations of environmental variables better explained tropical species diversity than GD. In fact, the contribution of geodiversity component (GDC) showed similar patterns to topography, and usually effects at smaller extents and finer grain size. The GD is a poor predictor of mountain diversity in large part because interactions between mountains and biodiversity are complex, and thus geodiversity could not be treated as an index. In contrast, landform type is an objective description of the present state of erosion of a mountain or bedrocks. When we stand in front of a karst mountain, we can see quite clearly that the plants are different from those in a granite mountain. We propose that landform identity rather than GD, is a more suitable indicators of geodiversity for tracking changes in mountain floras along geological time scales.”

REV: I see a lot of problems with this text, for various reasons:

- Basically none of the sentences is clearly and understandably formulated
- scientific terms are not properly used
- the text, in parts, is misleading and, in parts, contains wrong claims

To illustrate this, I have above left some comments – many more would have been possible. I hope the authors will see what I mean when commenting about the necessity to improve this text considerable.

RESPONSE:

In this section, the reviewer #3 raised many questions in the form of annotation. We have carefully considered these review comments. In the new manuscript, we have decided to delete this part. First, delete this section does not affect the conclusion of our paper and makes the manuscript more brief and intelligible. Second, the definition, measurement and practical application of geodiversity remain confusion. Third, some questions raised by the reviewers here are ambiguous.

Hereafter are the point to point replies to the reviewer's questions raised in this section.

“Several studies proposed to use geodiversity to model biodiversity based on the hypotheses that specific geo-sites should support unique biota and that high geodiversity is coupled with high biodiversity.”

RESPONSE:

The review 3 pointout “use geodiversity to model biodiversity” is wrong, and marked “hypotheses that specific geo-sites should support unique biota” with a “?”. These questions are quite not clear.

These statements were cited from previous literature. See:

Commented [A25]: wrong

Commented [A26]: ?

Commented [A27]: This statement is wrong. There are plenty of studies in the literature showing the opposite. This does not pay justice to the state-of-the-art knowledge.

Commented [A28]: There is not “the” one GD, but there are different methods used in literature for estimating geodiversity (incl. remote sensing approaches more lately)

Commented [A29]: This is a strongly misleading formulation. Both perform equally good.

Commented [A30]: Meaning unclear – re-formulate. GD is usually a combination of environmental variables.

Commented [A31]: Not clear – which pattern?

Commented [A32]: No, it is not, as has been shown in previous studies.

Commented [A33]: Geodiversity measures are supposed to capture a considerable proportion of this complexity.

Commented [A34]: It can and has been treated as an index in some studies, and in other works, single parameters were used. Thus, the sentence, as formulated, is wrong.

Commented [A35]: Why in contrast?

Commented [A36]: Geodiversity is also “objective”

Commented [A37]: This is not new and is basic textbook knowledge. E.g. compare works in the Alps from the mid 20th century and thereafter.

Commented [A38]: Part 1 of the sentence – refers to patterns, current;

Commented [A39]: This refers to timing – but what is the connection to the first part of the sentence? This is not clear.

Commented [A40]: I have looked at the comments by the authors below and have inserted my answers. However, since the authors say that they deleted the text altogether, this does not seem to have any impact on the ms anyway. Nevertheless, I rate that some of my answers may be useful for the authors to improve their future work and hopefully understand my comments and the reasoning behind a little...

Commented [A41]: wrong

Commented [A42]: ?

In the article (Muellner-Riehl et al., 2019) which mentioned many times by reviewer 3, geodiversity is used as variable to model species diversity (“We used generalized linear models to test to what extent vascular plant species diversity in mountains is explained by net primary productivity (NPP), geodiversity and Pleistocene climate fluctuations...” DOI: 10.1111/jbi.13715, p2868).

In Hjort’ paper (DOI: 10.1111/cobi.12510, p630) “We found that geosites are important to biodiversity because they often support rare or unique biota adapted to distinctive environmental conditions or create a diversity of microenvironments that enhance species richness.”

However, the predictive power of geodiversity has not been high.

RESPONSE:

This sentence is followed by several examples to illustrate the limitations of geodiversity. To our knowledge, the relationship between geodiversity and biodiversity remains being further explored. As Alahuhta et al. wrote “Although theoretical foundations for the geodiversity-biodiversity relationship and its conservation implications are well-established, only a handful of empirical studies have actually tested this relationship” (see Alahuhta et al., 2020, <https://doi.org/10.1038/s41559-019-1051-7>).

For example, the geodiversity index (GD) was found to be no better than elevational range for predicting mountain species diversity. Error! Reference source not found.

RESPONSE:

Here is just an example, which the review #3 mentioned in previous round. We think this is the method that reviewer #3 accepted to quantified geodiversity.

The review indicated “to be no better than” is a strongly misleading formulation. However, that is what we get from the paper of Muellner-Riehl et al. (2019). We can’t agree with a model with higher residual deviance (18281 in GD model) is equally good to a lower one (13802 in Elevation model).

Here after are the statement cited from references 3. “Elevational range were strongly correlated ($r = .70$) and given that Elevational range showed better performance in single predictor models (Table SI.2), GD was not considered in the multivariate regression models.” (in Muellner-Riehl et al., 2019, p2830).

Table SI.2. Evaluation of generalized linear regression models to predict plant species richness Information Criterion; NPP = net primary productivity; GD = geodiversity index (GD; see *Mate* to the present.

	AIC	Δ AIC	Null deviance	Residual deviance	df (residual)	McFadden's pseudo-R ²
Single predictor models						
NPP (mode)	17290	10252.9	18390	17130	14	0.069
NPP (mean)	13860	6822.9	18390	13700	14	0.255
Elevation (range)	13963	6925.9	18388	13802	14	0.249
GD (range)	18442	11404.9	18388	18281	14	0.006
ΔT (mode)	10141	3103.9	18388.4	9979.4	14	0.457
ΔT (mean)	11144	4106.9	18388	10982	14	0.403

Another recent study reported that combinations of environmental variables better explained tropical species diversity than GD. Error! Reference source not found.

Commented [A43]: This seems to be a misunderstanding by the authors.

My point in highlighting the sentence was that geodiversity is not used to “model” biodiversity.

Commented [A44]: Again, this seems to be a misunderstanding by the authors. The point I was hinting at was “hypotheses” when putting a question mark to the authors’ sentence.

Commented [A45]: This statement is wrong. There are plenty of studies in the literature showing the opposite. This does not pay justice to the state-of-the-art knowledge.

Commented [A46]: Unfortunately, I don’t see the point the authors are trying to make here by citing this one study back from 2020.

Commented [A47]: There is not “the” one GD, but there are different methods used in literature for estimating geodiversity (incl. remote sensing approaches more lately)

Commented [A48]: This is a strongly misleading formulation. Both perform equally good.

Commented [A49]: Again, authors seem to have overlooked the main point I was trying to make here. My point was that the formulation(s) as such is/are misleading, which means that the sentence should be re-formulated to make it more generally acceptable/applicable.

Commented [A50]: Meaning unclear – re-formulate. GD is usually a combination of environmental variables.

RESPONSE:

Of course GD is a combination of environmental variables. However, its explanatory power is really not as good as the combination of separated environmental factors.

As the case of reference 92, GD is summed spatial diversity of climate (3 variable), habitat (6 variable), and soil (3 variable). See “*The geodiversity index, in contrast, is computed as the summed spatial diversity of the same three environmental factors (included climate, habitat, and soil) measured within each plot and its surrounding and follows the recent plea to consistently use diversity indices commonly applied in biodiversity studies*” (see Wallis et al. 2021, p7: doi.org/10.1038/s41598-021-03488-1). Their result shown that combinations environmental variables better explain species diversity and ecosystem functions than a geodiversity index.

In fact, the contribution of geodiversity component (GDC) showed similar patterns to topography, and usually effects at smaller extents and finer grain size.^{Error! Reference source not found.}

RESPONSE:

The “similar patterns” means geodiversity component (GDC) has similar effects on biodiversity as topography.

The GD is a poor predictor of mountain diversity in large part because interactions between mountains and biodiversity are complex.^{Error! Reference source not found.}, and thus geodiversity could not be treated as a index.

RESPONSE:

As a summary of the above. Here we just criticize the practice to treated the geographical diversity as index. The review #3 also indicated “Geodiversity measures are supposed to capture a considerable proportion of this complexity.” and “It can and has been treated as an index in some studies, and in other works, single parameters were used. Thus, the sentence, as formulated, is wrong.”.

In our opinion, the meaning of review #3 is “There are a thousand geodiversity in a thousand people's eyes” !!! On the other hand, since many parameters (such as elevation, soil type, annual mean temperature, etc.) can be used as substitutes of geodiversity. Why not we bypass the vague notion of geodiversity. Why not just explore the effects of combinations of parameters on biodiversity. There are still many questions about the application of geodiversity in current research.

In contrast, landform type is an objective description of the present state of erosion of a mountain or bedrocks.^{Error! Reference source not found.}

RESPONSE:

“Why in contrast?”. Here is a comparison of landform types and GD. As a characteristic variable, landform type is easy to be understand. However, how to use geodiversity in actual research is confusing. Please also refer to the upper response.

“Geodiversity is also “objective”.” Geodiversity is still a research direction to be discussed. Please refer to the previous reply.

When we stand in front of a karst mountain, we can see quite clearly that the plants are different from those in a granite mountain.

RESPONSE:

So what's wrong with whis statement?

We propose that landform identity rather than GD, is a more suitable indicators of geodiversity.^{Error! Reference source not found.} for tracking changes in mountain floras along geological time scales.”

Commented [A51]: This is exactly the point – the original sentence, as formulated, is potentially misleading readers, and is different from this answer.

Authors need to take care of precise formulation in their manuscript.

Commented [A52]: Not clear – which pattern?

Commented [A53]: No, it is not, as has been shown in previous studies.

Commented [A54]: Geodiversity measures are supposed to capture a considerable proportion of this complexity.

Commented [A55]: It can and has been treated as an index in some studies, and in other works, single parameters were used. Thus, the sentence, as formulated, is wrong.

Commented [A56]: My “opinion” is not really relevant here.

My comments were, among other things, referring to the fact that there exists literature (as early as 2017, and potentially before), which has looked into definitions of geodiversity in a systematic way, and which the authors may find useful for their future studies and considerations. E.g. see Fig. 1 in:

Bailey, J. J., Boyd, D. S., Hjort, J., Lavers, C. P., & Field, R. (2017). Modelling native and alien vascular plant species richness: At which scales is geodiversity most relevant? *Global Ecology and Biogeography*, 26, ...

Commented [A57]: Why in contrast?

Commented [A58]: Geodiversity is also “objective”

Commented [A59]: I consider this to be an unjustified claim based on the available literature.

Commented [A60]: A discussion about a research topic does not mean that the matter of the discussion, namely ...

Commented [A61]: This is not new and is basic textbook knowledge. E.g. compare works in the Alps from the mid 20th ...

Commented [A62]: The point I intended to make here was that as written, the content is presented as if it was ...

Commented [A63]: Part 1 of the sentence – refers to patterns, current;

Commented [A64]: This refers to timing – but what is the connection to the first part of the sentence? This is not clear.

RESPONSE:

The erosion rates between different types of rocks are differs. More knowledge about species assemblages in mountain can be gained by comparing differences of floristic composition of mountain flora in neighboring regions (but differ in landform type). In addition, there are also different evolutionary stages within a single landform types (indicated by classical Davis' geographical cycle). By comparing the synergies between species assemblages and landform development stage, the temporal changes of mountain biodiversity can be tracted.

Q17: Line334:

"Based on our results, we put forward the "geological lithology hypothesis of flora" to explain the assemble and differentiation of mountain floras. In this theoretical framework, floristic assembly in mountains is driven by the lithospheric cycle, which refers to the bedrock-constrained developmental processes of landforms. Specifically, under this hypothesis, the mountain species differentiation closely related to the type of bedrock and degree of erosion, species richness and species composition in mountain flora are interaction result of landform and environment, and phylogenetic niche evolution^{Error! Reference source not found.} can promote the spread of species among different landform floras. Overall, our study provides a novel framework and approach for determining the mechanisms of species diversity within mountains and the distributional patterns of some of the world's richest floras."

This section needs to be improved and provide a more balanced view on previous work by other authors and suggestions put forward here. As I had already suggested in my review of the first manuscript draft, the manuscript here needs to go into some more depth concerning previous hypotheses put forward, importantly the MGH, and I still don't see this has been done. While only briefly mentioned in the into, the MGH does not show up here in the discussion. I would like to see some of what the authors have answered in their rebuttal be actually also included here in the manuscript text.

Response: This is a great suggestion. We rewrote this section. Please see line 389-413.

"Based on our results, we propose the "geological lithology hypothesis of flora" to explain the assembly and differentiation of mountain floras. In this theoretical framework, floristic assembly in mountains is driven by the lithospheric cycle, which refers to the bedrock-constrained developmental processes of landforms. Specifically, under this hypothesis, montane species differentiation is closely related to the type of bedrock and degree of erosion. Both SR and species composition in mountain flora result from interactions between the landform and the environment. In addition, the dispersal of plants between different landform types is more restricted than within the same landform type. Successful diffusion across a landform is often accompanied by the emergence or radiation of adaptive traits. This is called phylogenetic niche evolution^{Error! Reference source not found.} and can promote the spread of species among different landform floras. This differs from the MGH, which assumes that the montane biodiversity hotspots require three key boundary conditions: 1) the presence of lowland, montane and alpine zones, 2) climatic fluctuations to produce a "species pump" effect, and 3) high-relief terrain with environmental in a given mountain region)^{Error! Reference source not found.}. In contrast, our hypothesis suggests that montane bedrocks and landform processes determine the geographic distribution of plants. The MGH effectively explains the high biodiversity of mountains characterized by large elevational differences (such as the Himalayan and Andes mountains), but such restrictive boundary conditions constrain the applicability of the hypothesis. Biodiversity hotspots also occur in regions with stable climates, such as the Namib desert^{Error! Reference source not found.}, or with with minimal elevational gradients, such as the Southeast Asia karst landforms^{Error! Reference source not found.}. We argue that the

Commented [A65]: Thanks for this explanation which I now understand in contrast to the previous sentence. It seems a matter of language that some statements simply can't be understood by readers.

It is good to see that the authors have employed a language revision which seemed necessary to enable readers to follow the content.

novel "geological lithology hypothesis of flora" could serve as a general explanation for global diversity patterns, as the formation of mountains on the Earth's surface is the result of the cycling of sedimentary, igneous and metamorphic rocks. In conclusion, our study has highlighted the floristic patterns of different landforms and provided a novel framework for studying the mechanisms of plant species diversification within mountains and the distributional patterns of mountain floras of the world."

REV: Similar to what I mentioned for the text above does also apply here. The authors need to pay attention to content correctness (e.g., when referring to the MGH, and comparing to their hypothesis), precision in formulation and use of terminology (e.g. concerning PNC) and claims. In addition, it appears to me that especially in the last third of the text, different topics are being confused/intermixed. The MGH refers to mountains, but the authors talk about the "Namib desert" and "SE Asian karst landforms" for which the MGH would not apply anyway. They also talk about "global diversity patterns" when referring to their hypothesis. The MGH specifically deals with mountains, the hypothesis by the authors is supposed now to refer to global phenomena? If so, do the hypotheses actually deal with different matters? I strongly encourage the authors to carefully revise the text, and stick to what the hypotheses were developed for. Also, point in time geographic distribution of plants, and processes acting in concert, appear to be compared despite being different. As such, the text is not clear for the reader.

RESPONSE:

Thanks for the questions raised by the reviewers. We consider the differences between our new hypothesis and MGH in the spatial scale. In terms of scale differences. The MGH focus the biodiversity of a mountain which usually the presence of lowland, montane and alpine zones. This is a hypothesis of alpha diversity. In contrast our theory is more concerned with the process of mountain flora differentiation, which is a hypothesis of beta diversity. In this study, we investigate the flora assemble differences between mountains with different landform type.

We made further revisions to the manuscript to mad it clear. Please see line 376-381 "*To explain montane species diversity, this hypothesis differs from previous assumptions, [such as MGH^{Error! Reference source not found.}, Error! Reference source not found., which is more focus on the biodiversity hotspots in high-altitude mountainous areas and the cause of diversity. Our hypothesis focuses more on mountainous areas with different altitudes and geological and lithological types, as well as their floristic compositions and differences, which is important for explaining differences in biodiversity hotspots from both low- to high-altitude area^{Error! Reference source not found.}, Error! Reference source not found.."*

Commented [A66]: Thanks for this clarification. I suggest this should then be explained as such in the ms.

The text following below would not fully meet this (apart from language issues), and introduces some further error. I include my comments below.

Commented [A67]: Care needs to be taken when using the term "biodiversity hotspots". The MGH was originally proposed for the Tibet-Himalaya-Hengduan region, for which not all mountain systems qualify as biodiversity hotspots. This should be deleted, and instead could read something like:

.. such as the MGH, which focuses more on the origination of high levels of biodiversity found in mountain systems.

Commented [A68]: Why not formulate this the way it was explained before, in terms of alpha and beta diversity?

In my view, this would make the text's content more accessible to the readers.

Thanks for taking the time to review our manuscript. The new comments and suggestions from Reviewer #3 are helpful, and we appreciate them a lot. The manuscript has been furtherly revised following your comments marked in yellow (in both response letter and ms). The language in the manuscript and response has also been revised to make our point clearer. We provided point-by-point responses to your comments as following.

Our latest response is in red colour which began with “**RESPONSE:**”.

Reviewer #3 (Remarks to the Author):

“Although it is well documented that mountains tend to exhibit high biodiversity, whether and how geological processes affect the assemblage of montane floras remains unclear. Here,”

RESPONSE:

Done. This sentence has been revised as “Although it is well documented that mountains tend to exhibit high biodiversity, how geological processes affect the assemblage of montane floras is a matter of ongoing research.”

Commented [A1]: Delete “whether and”
“remains unclear”  “is a matter of ongoing research”

“Furthermore, the evolutionary history of each biological taxa also affects local biodiversity.”
Reference source not found.-Error! Reference source not found.

RESPONSE:

Done. This sentence has been revised as “Moreover, the unique evolutionary history of each biological taxon in mountainous regions could exert a profound influence on local biodiversity.”
Reference source not found.-Error! Reference source not found.

Commented [A2]: taxon, not taxa

Commented [A3]: This is a rather awkwardly formulated sentence, I suggest to re-formulate this.

“Then, the species richness (SR), phylogenetic diversity (PD=Faith’s PD), phylogenetic structure indices (PDI, NRI, NTI)”

RESPONSE:

Done. “Then, species richness (SR), phylogenetic diversity (PD=Faith’s PD), phylogenetic structure indices (PDI, NRI, NTI) ..”

Commented [A4]: delete “the”

“Thus, species with the deepest phylogenetic divergences occur in karst landforms, while species with the shallowest divergences occur in desert landforms. This is consistent with some fossil evidence. For example, the fossil flora discovered in southwestern China indicate that local karst vegetation may have existed since the early Oligocene.”
Error! Reference source not found.,Error! Reference source not found.

Here, we would like to indicated the species in the limestone mountains are generally had earlier diverge age and could survive for a long time. Especially those species can adapted to the arid habitat of the limestone mountain top. We present an example, the Oligocene fossil flora Wenshan basin which located in Yunnan, China. The vegetation (infer from fossil data) and genera composition of this fossil flora in is very similar to the current karst flora in Wenshan, China.

For the second question, “Discussion” occurs in Result section. The reason we did this is we want to give a brief explanation of the results or compared to previous study. We thought this would

Commented [A5]: Poor language (use of tenses), up to a degree which makes understanding of the meaning difficult for readers.

help readers understand the meaning of our results more quickly, and also avoid repetition before and after the article. We notes this approach is generally be accept in the articles published in Nature Communications. In the later “Discussion” section (line 286), we further discussed the relationship between landform process, species richness, local speciation, dispersal in mountain flora.

RESPONSE:

Here, we would like to indicate that the higher phylogenetic diversity index (PDI) of karst flora may be attributed to the long-term ecological stability. To be brief, karst flora exhibits long-term stability. As an example, we present the Oligocene fossil flora from Wenshan basin located in Yunnan, China. The vegetation (inferred from fossil data) and genera composition of this fossil flora closely resemble the current karst flora in Wenshan, China.

The following revision has been made to this part. See the Line 175-180 : “*The PDI results serve as an indicator of the level of floristic stability. Here, we would like to indicate that the species inhabiting the arid limestone mountains generally exhibit an earlier divergence age and possess a remarkable capacity for long-term survival. Take for example, an Oligocene fossil flora discovered in Wenshan basin located in Yunnan, China, revealed a fossil assemblage (eg. Burretiodendron, Ficus microtriviva), of which clearly indicated current local karst vegetation, may have existed since the early Oligocene*”

The second question pertains to the paper writing format. In our manuscript, we have incorporated several concise discussions within the “Results” section. This approach has been adopted to provide a succinct explanation of our findings and compare them with previous studies, aiming to enhance readers' comprehension without unnecessary repetition throughout the article. Such an writing format is accepted by Nature Communications.

Question about previous response.

Second, based on the technical requirements (this is based on the null model assumption) of the software package, flora data is extracted from the mountain species database to reflect the phylogenetic structure of the flora. Thus, a global or regional tree is not needed here. In fact, as we know there isn't currently an approach that is calculates a regional phylogenetic structure based on a global tree.

REV: The meaning of this sentence is not clear to me. Of course, there exist studies that have been using e.g. large angiosperm-wide trees as source for creating much smaller trees with only regional species samples representation. This needs clarification.

RESPONSE:

“Using e.g. large angiosperm-wide trees as source for creating much smaller trees with only regional species samples representation”, which is also the method used in our study.

However, the meaning of “In fact, as we know there isn't currently an approach that is calculates a regional phylogenetic structure based on a global tree.” is we can't use a large phylogenetic tree (included about 30000 species) to calculate the PDI, NRI and NTI of floristic distribution dataset included less species (such as 10 floras in total included 3000 species). The species in the phylogenetic tree should be consistent to the floristic distribution dataset, otherwise would violate the assumption of the null model.

So that the software warning reason here is not the inconsistencies of taxonomic name. The true reason is species number in the phylogenetic tree not consistent to the floristic distribution dataset (more species included in phylogenetic tree than floristic distribution dataset, as the case we

Commented [A6]: Due to poor language really hard to understand.

Commented [A7]: Unfortunately, I don't understand the reply (due to poor quality of English language).

shown above, less species in floristic dataset than in phylogenetic tree).

Warning: the input matrix has fewer columns than the number of species in the tree.

Please also refer to the previous reply. We hope our response made it clear.

RESPONSE:

Here, the reviewer expressed concerns regarding the phylogenetic tree employed in this research. It seems like the reviewer is not familiar with phylogenetic or community phylogenetic approaches, which has led to some misunderstandings. Further elaboration will be provided below. The key point seems to be “why not use large angiosperm-wide trees” and “why R packages warning”. To illustrate this, we have presented a brief case below, accompanied by detailed instructions.

The phylogenetic structure indices (PDI, NRI and NTI) is calculated by R packages 'PhyloMeasures' and 'picante' based on phylogenetic tree and species distribution data. The consistency between species occurs in the phylogenetic tree and those in the distribution dataset is a fundamental prerequisite for ensuring the operational functionality of R packages 'PhyloMeasures' (otherwise, it would violate the null model).

In the following case, species *Pertya corymbosa* (in red frame,) not occurs in phylogenetic tree2 and distribution data2. When utilizing the combination of phylogenetic tree2 and distribution data1, or phylogenetic tree1 and distribution data2 for calculating phylogenetic structure indices (PDI, NRI, and NTI), the 'PhyloMeasures' package will generate a 'warning'.

The relevant research methods have been extensively accepted, and the details have been elaborated upon in our manuscript (please see manuscript line 448-475). Therefore, we did not make further changes on methods in the new manuscript (to maintain the manuscript's simplicity). If you want to know more details about phylogenies and ecology, several literature is suggested here (Webb et al. 2002, doi: 10.1146/annurev.ecolsys.33.010802.150448; Tsirogiannis, C. & Sandel, B. 2016, doi: 10.1111/ecog.01814; Kembel et al., 2010, doi: 10.1093/bioinformatics/btq166).

We hope our response will answer your concerns.

We suspect there are no differences between “species distinctiveness” and species age, because both

Commented [A8]: I still don't fully understand from this reply why the error message would complain about the missing “species names” if the names would have been resolved prior or during analysis, and – I assume - measures been taken to include missing data? I am missing the context between the previous and the new replies. I still don't fully understand the connections from the reply here.

of them are based on the in-clade in the phylogenetic tree. Besides, species age (or species divergence time) is more well known than “species distinctiveness” in evolutionary biology and biogeography.

REV: What do the authors mean by “in-clade”?

RESPONSE:

Here, “in-clade” just represents the branches of each species contained in the phylogenetic tree.

RESPONSE:

We are here for further clarification. The so called “in-clade” proposed by us is used to explain the term “species distinctiveness” (which raised by review#2). The meaning of “in-clade” is a evolution branch of a phylogenetic tree. And, the branch length is closely associated with species age. The words “in-clade” does not occurs in our MS. We are sorry for any confusion have caused for you.

In our study, we used the mean divergence times (MDT) proposed by Lu et al.(2018) to represent the “flora age”. As such, the MDT reflected the age composition of species within a flora. A flora with larger MDT has more ancient species and hence is expected to have older floristic ages. Overall, we believe that our use of MDT to measure the species age structure of flora is consistent with the existing literature and therefore is appropriate.

REV: Not clear to me. Only because many people have been doing something does not mean it is correct or the most appropriate way to do it.

RESPONSE:

Reviewer # 3 mentioned that “many people are doing the same thing, which does not necessarily mean that this approach is correct”. We cannot fully agree with the comments of Reviewer #3. Many people have used this method to carry out a lot of work, although it cannot be considered right, it cannot be considered wrong, and it cannot be considered that several journals currently publish the wrong method? Especially, if there is no better update method available, traditional methods can only be used.

However, frankly, it is difficulty of develop a better or appropriate method to measure the species age structure of flora. Because species age occurs in a flora is still can’t be well defined. At present, we can only study the floristic assemble process on a macroscale of statistical significance. So that in our study we used the MDT method proposed by Lu et al. (2018), which is not so bad. Development new methods to understand the species assemble process of flora over time scale is an important topic for future research.

In addition, in this article, we only hope to calculate the aggregation and divergence of plant flora at a macro scale. Therefore, we used the MDT method proposed by Lu et al. (2018). The limitations of this method have been explained and revised as previous reviewer comments (such as MS Line 461-470). Developing new methods to reveal the species composition of plant flora on a temporal scale is an important topic for future research. Of course, we would greatly appreciate it if the reviewer could recommend a better method to us.

RESPONSE:

Thank you for further clarification of your concerns. We acknowledge that it is still impossible to determine the true age of each species in the flora based on current research approach. The limitations of MDT method have been explained and mentioned in previous manuscript (section “Divergence time estimation”), and also Extended Data Fig. 8.

In the new revised manuscript, we further clarify this issue. Please see Line 481-492:

Commented [A9]: I still don't understand from this reply what the authors mean by this.

Commented [A10]: Unfortunately, I don't fully understand all parts of this reply (due to poor quality of English language).

However, from what I understand the authors claim here, I want to assure the authors that I was actually never asking to “develop a new method”. Rather, I was pointing out already previously the limitations of the approach, that when using a local geographical species pool for divergence time estimations, “true” ages of species cannot possibly be inferred. Why? Because the closest relatives of those species (which may occur elsewhere and are therefore not included in the tree) would be required to be added in such a dating analysis, to infer better estimates of the species’ ages. Likewise, extinction will likely be different in different taxa/clades (e.g. due to taxon age, and/or geological/climatic events in different mountain regions), and a tree which is missing closest relatives will then be even more prone to overlooking this effect. This was the reason I was referring to “random” distribution of extinction.

The authors previously replied that they do not intend to present the “true” ages, but just the relative ages. Nevertheless, even then, because of uneven taxonomic sampling, bias will naturally be introduced. This limitation should be mentioned.

The authors claim that other studies have done similar as they do. However, at the very least, the limitations of this approach should be mentioned in the manuscript. The answer by the authors does not invalidate my calls for caution.

“In this approach, divergence time of species is expected to be overestimated, as the branch of some species in a local phylogeny is usually longer than that in global phylogeny (included all species). For instance, if a lineage became extinct, the divergence time of its existing closest relative species would be dated at the point of their last common ancestor. To assess the robustness and the effect of this sampling bias on the final results of species age structure of a mountain flora, we used four divergence time datasets and found similar MDT patterns between mountains of different landforms (Extended Data Fig. 8). This result is consistent with a study^{Error! Reference source not found.}, which found that “in large-scale biodiversity and phylogenetic analyses, sources of noise in divergence time estimation are to be expected, but they did not affect the reliability of the results”. We believe that our dated megaphylogenetic tree was suitable for this study because our aim was to reveal the general patterns of landform influence on the formation of mountain flora rather than focusing on the age of each species.”

“Here, we apply the term “flora” to refer to the sum of all angiosperm families, genera, and species growing on a specific mountain or in a well-delimited area^{Error! Reference source not found.}, which is a relatively independent biogeographical unit^{Error! Reference source not found.}”

REV: I would argue against the flora of a mountain being a *relatively independent biogeographical unit*, if this is what is meant here (but I may have misunderstood the sentence). First, because related lineages are shared between mountains, and second, because most biogeographers, in terms of terminology, would not consider the flora of a mountain as a “biogeographical unit”.

Also in **Pre-response**: “*we mentioned is not only a unit, but als*” I am not sure how “unit” here would refer to the biogeographic units mentioned in the ms. This may be worth of clarification.

Response: We appreciate the key question raised by the reviewer. In generally, the term “biogeographic unit” is used in the context of biogeographic regionalization.

Here we consider that the flora of mountain and biogeographic zones is equivalent at small scale.

REV: This is unclear. Which biogeographic zones are meant here?

RESPONSE:

(1) Here, the mountains wasn't refer to a single peak, but rather to a mountainous area or a certain nature reserve, including many peaks and adjacent areas. Correspondingly, all the plants growing in this area is consist of many different species, genera, and families, whose was called on a mountain flora, it is a relatively independent natural geographical area. Due to differences in longitude, latitude, altitude, and climate factors, biological factor (or floristic compose) among different mountainous regions, these geographical regions are also bound to have differences.

(2) The biogeographical units mentioned in the main text cannot be equated with biogeographical divisions. Biogeographic zoning often divides any geographical areas into several level or grades, such as the global flora, which is often divided into kingdom, region, province, and county. These are different biogeographical units (hierarchical units). The unit itself does not have hierarchical significance, and only after studying all species in the region, namely Flora, its hierarchical level or grade be determined, such as the local flora, the geographical elements of the families, genera, or species (tropical, temperate elements), historical elements (antiquity), originative elements (endemic families, genera, species etc.), ecological elements, migration elements, etc., Only through

floristic phyto-geography analysis, a certain units such as a kingdom, region, province, and county be divided.

A certain biogeographical unit, such as a mountain area, may also be divided into different geographical regions, such as the Hengduan Mountains in China and the Himalayan region, which are divided into the East Asian Flora and the China-Himalayan Forest Subregion. In the east, it is divided into the East Asian Flora and the China-Japan Forest Subregion; Correspondingly, the latter can be divided into South China Province, Central China Province, etc. (Wu Zhengyi et al., 1996). A certain mountain area, depending on its nature geography and biogeography property, they may divided into a hierarchical unit (division) or a subunit. Or they could be belonged a province, or a county.

Therefore, when we mention a mountain region, or a mountain flora, similar to a nature geographic or biogeographical area or unit with a certain grade or no grade, it is not a wrong concept.

(3) This article studied 140 mountainous areas, listed the natural geographic information of each mountainous area, and correspondingly formed 140 local mountainous floras, thereby showcasing their differences in natural geography and floristic phyto-geography.

Furthermore, this article successfully classified 140 mountains into five geomorphic types (by searching for detailed geological survey data, and special investigation report on Floras), revealing and analyzing the relationship and possible reasons between these local floras and mountainous lithology. We believe that this is a progress. On this basis, it will be possible to further study endemism, geographical elements of those floras, and diffusion, migration of floristic elements between different geomorphic types and different bedrock, floras.

(4) Of course, in our response to the reviewer's question, we wrote biogeographic zones, which are indeed not strict enough and have a larger scope. We just wanted to explain that each mountain has its unique characteristics. Wu Zhengyi et al. (1996) considered that an area of every relative independent mountainous flora covered should not be less than 100 square kilometers. Due to the large amount of content and limited space, these concepts and data were not included in the main text.

RESPONSE:

Thanks for further explanation. We accept your view on using of biogeography terms. In the revised MS, we removed the “, which is a relatively independent biogeographical unit^{Error! Reference source not found.}” order to create ambiguity.

The sentence is revised as: “*Here, we used the term “flora” to refer to the collection of all angiosperm species growing on a specific mountain or in a well-delimited area*^{Error! Reference source not found.}”

First, mountain is geographical barrier to plants diffusion.

REV: This is not correct (as I had stated in the previous rounds of review already), phrased in this general manner. It depends on the orientation of the mountain. Only if a mountain has W-E orientation, it can act as barrier (e.g. Himalayas, Alps). For N-S-oriented mountains (such as the Andes, or the Hengduan mountains), this is not correct.

RESPONSE:

We believe that it is unnecessary for the reviewers to oppose the above views. The author's viewpoint

Commented [A11]: The authors present some general textbook biogeographic contents here. I assume many readers of *Nature* will be familiar with these.

However, I could not find a clear connection to my previous claim that I would not generally agree to “the flora of a mountain being a relatively independent biogeographical unit”.

My point was to call for caution when using terms such as “independent biogeographical unit”, or “biogeographic zone”.

This needs to be taken care of as it might lead to confusion among readers. As the authors point out themselves in their reply, biogeography has its own terminology and concepts.

is simply that mountains can affect the diffusion of species or serve as a barrier of species diffusion. In fact, mountains are both a barrier for species diffusion and may also play a promoting role in species diffusion. For example, the north-south direction mountains play a promoting role in the north-south migration of species, while serving as a barrier for the east-west migration of species; Due to the existence of Hengduan Mountains, many genera and species in Chinese Mainland have formed China-Himalaya distribution subtypes and China-Japan distribution subtypes, with Hengduan Mountains as the boundary. Similarly, due to the barrier effect of the Nanling Mountain in China, many tropical genera and species cannot exceed the Nanling Mountain, such as *Pandanus*, *Endospermum*, etc.

The reviewer mentioned or emphasized the local migration, endemism, and differentiation of species within a mountainous area, especially in high mountains. Our focus is to emphasize that the migration of species between different mountainous areas will be hindered, especially between Danxia landforms, karst landforms, and granite landforms, where species diffusion is not easy.

RESPONSE:

We have made modifications in the previous MS. Please see Line 347-349.

“For many biogeographers, mountains are regarded as both barriers and bridges of species dispersal^{Error! Reference source not found.}. The role of mountains as corridors has been documented in several mountains has a North-South orientation, such as the Andes^{Error! Reference source not found.} and Hengduan Mountains^{Error! Reference source not found.}”

Second, a mountain usually shows sky island effects and contain their own unique clades or several endemic species.

REV: This is not correct. First, there is no such thing like a “sky island effect” (phrased this way). Second, only a small fraction of the world’s mountains qualify as sky islands – only, if their vegetation is drastically different from the surrounding lowlands. This may only be the case if plants living on the mountains are different from the lowlands and were therefore not likely recruited/evolved from lowlands in which case the nearest relatives would come from other mountains (example: some mountains located in drylands, e.g. Africa).

RESPONSE:

Here is just a brief discussion with the reviewer. The “sky island effect” of mountain flora is a question worth further study and discussing. As our understand, in essence the sky islands mountains is qualified by its flora showed discontinuous distribution. In generally, in mountains vertical zonality of vegetation on the elevation gradient is widespread obserded. We suspect the plants or community occurs on the top of the mountain should be treated as a result of sky islands effects? These community differs from the low land, and could represents the early stage of species assemble of mountain flora.

RESPONSE:

The content of this section is tangential to the concepts presented in our current manuscript; therefore, we have refrained from making corresponding amendments within the manuscript. The following is a concise discussion aimed at providing reviewers with a clearer understanding of our perspectives.

We agree with you, no such a term or concept “sky island effect” has been formally proposed. In our perspective, the "sky island effect" reflects species diffuse form one mountain top to another mountain top is restricted. Although the constraints on diffusion are comparatively less pronounced

Commented [A12]: My point was simply that the statement is not correct, when phrased in this general manner. This is still valid, and thus I suggest, now more concretely, to re-phrase this:

... mountains can act as barriers to plant diffusion ...

Commented [A13]: What I was intending to suggest to the authors by my previous comment was to tune down “a mountain usually shows sky island effects”. First, because “usually” is not correct, as this would imply this statement holds true for the majority of mountains, which is not the case. Secondly, “sky island effects” is unclear to me. What is a sky island effect? Has this even been defined, and by whom?

In short, what I had intended with my previous comment was to re-formulate this.

than that in a typical sky islands mountains. The “sky island effect” of mountain flora is a question worth further study and discussing. This is an issue worthy of further study.

Anyway, we appreciate your comments on precise use of terminology.

There are also some literature to suggest that mountain region could be viewed as a biogeographical unit. For instance, Rahbek et al. (2019, Science 365, 1108-1113) wrote that “Like an island, a mountain region may be viewed as a biogeographical unit in itself, with in situ speciation and extinction playing a key role in building the regional species assemblage”.

REV: Is anyone else except this cited paper claiming this as well? I could imagine that most mountain biogeographers would disagree with this oversimplification.

RESPONSE:

We cannot agree with the reviewer's opinion, as stated in the previous response.

RESPONSE:

We accept your view on using of biogeography terms, and revised this sentence as bellow.

“Here, we used the term “flora” to refer to the collection of all angiosperm species growing on a specific mountain or in a well-delimited area”^{Error! Reference source not found.,Error! Reference source not found.,}

“The mountains of sedimentary bedrock have much stronger environmental filtering effect, as these mountain ecosystems are more sensitive to rainfall. The environmental filtering effect further promoted the clustering of phylogentic structures of mountain floras as predicted by the phylogenetic niche conservatism (PNC)”^{Error! Reference source not found.,Error! Reference source not found.,Error! Reference source not found.,}

REV: It would be interesting to discuss here radiations more generally, e.g., how do the studies of plant radiations included in the global study by Muellner-Riehl et al. 2019 (JBI) compare to these statements/findings? Are the findings of these studies (and other studies published since then, i.e. after 2019) comparable to this? I suggest looking at the mountain systems (and their bedrocks) where these studies were undertaken.

Response: These are really good advice. The potential radiation speciation occurs when plants colonize from one landform to another which maybe an important path of plant diversification. Here, we document several cases (not all) on adaptative radiation that occur on the bedrocks. But the relations between radiations and bedrocks/landform are still poorly known. We suspect that the effect of landform and bedrock, which promote species differentiation, is more or less neglected under the shadow of “climate change”. Nevertheless, we have reorganized this paragraph carefully.

REV: I don't understand the reasoning here, as currently phrased. It is unclear what the authors mean (potential speciation radiation?). There exist different kinds of radiations, not all of them are “adaptive”.

RESPONSE:

Here we would like to shown adaptive evolution (associate with morphological traits and physiological traits) maybe the most important mechanism for plant to radiation in a new environment. Although the mechanisms of radiative evolution are diverse, plants ultimately need to adapt to their local environment in order to survive. Furthermore, evolutionary radiations underpinned by variation in physiological or behavioral traits can more easily be perceived as non-adaptive, compared to those involving more conspicuous morphological traits, causing a bias in our

Commented [A14]: See my answers before. My point was to call for caution when using terms such as “independent biogeographical unit”, or “biogeographic zone”.

The citation from Rahbek et al. “may be viewed as a biogeographical unit in itself” seems like an agreeable cautionary phrasing.

It is different from the authors' statement “independent biogeographical unit”, or “biogeographic zone”.

understanding of the extent and distribution of adaptive radiations in nature. That is reason a large number of traits of convergence (such as Alpine flora, Arid flora, and mangrove plants) and adaptive evolution have been broadly observed (Nevado et al. 2019, doi: 10.1016/j.cub.2019.07.059; Xia et al. 2021, doi: 10.1093/molbev/msab314).

RESPONSE:

The role of plant radiation evolution on floristic assemblage is discussed in the section “Bedrock promotes local speciation resulting in floristic differentiation between landforms” of ms. In our view, species differentiation caused by bedrock differences is an important source of mountain floristic differences.

Please see Line 332-344 “*On the other hand, adaptive evolution, encompassing both morphological and physiological traits, could play a pivotal role in facilitating the diversification of plants in novel environments*”. In plants, radiative evolution often accompanies habitat and landform shifts, as seen in Old World gesneriads and North American deserts rock daisies (Compositae tribe Perityleae). Similar patterns can also be observed in insects. For example, two radiated clades (nodes 14 to 16 and nodes 31 to 33) of *Exocelina* resulted from the transition from uplifted Australian Plate bedrock to ultramafic/ophiolite. Although, many previous studies also document climate change as an important driver of species radiations. The adaptive radiation of plants is more or less associated with bedrock type. For example, the development of a key innovation (lime-secreting hydathodes) may have made *Saxifraga* sect. *Porphyrium* better suited to limestone habitats, and the low specific leaf area (SLA) exhibited by *Erica* is an adaptation to oligotrophic habitats (quartzite/sandstone) in Cape. These obvious landform and bedrock effects could strongly promote both species and floristic differentiation between different regions.

“Dispersal also contributes to the mountain diversity. A well-known example is climatic fluctuations during the Quaternary ice age drove changes in the distribution of species around the globe. The spatial configuration of mountain range affect their functional connectivity, influencing species dispersal. In fact, dispersal more easily occurs at initial stage of landform develop mountain, as the slope of mountain still gentle and geographical barrier still relatively low. Such as the plants dispersal events in Qinghai-Tibet Plateau (QTP, low barrier) is obvious higher than that in Hengduan Mountains (high barrier). Furthermore, the effects of dispersal on species diversity in mountainous areas mainly occurred at lowland, and is limited at highlands where has more local endemic species. This means that dispersal has weak influence on the unique composition of mountain flora of different landforms, especially considering that landform have a significant filtering effect on the species.”

REV: The entire paragraph needs re-writing. Dispersal between mountains is not limited to times of climate change, which somehow is suggested here as currently written. Dispersal and establishment was easier in the Hengduan mountains at times of climate change as they have a N-S orientation, enabling plants to change their latitudinal distribution range more easily.

Reading this, I am not sure the authors have read the papers I had suggested to consult, and which they cited in the paper already before (e.g. Ding et al.).

Commented [A15]: From this response I cannot not see how my previous request (for discussing this in the ms) is now actually tackled in the manuscript.

Response: We accept this advice and wrote this paragraph. We would like to clarify the questions about dispersal events and diffusion barrier. We had carefully readed the paper by Ding et al (2020). It is not difficult to deduce the conclusion “*dispersal events in Qinghai-Tibet Plateau (QTP, low barrier) is obvious higher than that in Hengduan Mountains (high barrier)*” (see Ding et al, 2020, figure 2). On the other hand, we thought dispersal would reduce the β -diversity between different floras. Such as the proportion of endemic species in Hengduan Mountains is the highest is largely affect by its low rate of colonization (<0.05), and high rate of *in situ* speciation and local and recruitment.

REV: This could also/additionally be the effect of higher levels of extinction on the QTP, in contrast to the Hengduan mountains.

RESPONSE: This may be right as the QTP has lower habitat heterogeneity than the Hengduan Mountains. The role of extinction rate QTP and Hengduan Mountains are not mentioned in the paper by Ding et al (2020). A region with low habitat consistency are associated with lower biodiversity and also more species with strong ecological adaptation. Distinguishing the roles of history, speciation and extinction in the mountain flora assamble still represents a major challenge to future research.

RESPONSE:

Sorry for any misunderstanding caused by “of low habitat consistency ...”. This is a language translation error. Our point is that areas characterized by low habitat heterogeneity exhibit correspondingly low levels of biodiversity.

We further clarify the relationship between diffusion and mountain landform development. Please see in line 355-362 : “*The dispersal process is generally less constrained during the initial stages of mountain landform development, which are characterized by gentle slopes and limited geographical barriers*”. This scenario is well demonstrated by the assembly of alpine biotas. Since the Miocene, the colonization rate in gentle elevation gradient Qinghai-Tibet Plateau (QTP) (0.06-0.25) is always larger than that in the QHM (<0.05). The role of local species recruitment is most important during the early stages of mountain floristic assembly, subsequently being supplanted by local adaptations or *in situ* speciation due to the emergence of heterogeneous mountain environments.

In contrast, the low proportion of endemic species in Qinghai-Tibet Plateau has the highest rate of colonization since early Miocene. It is clearly the dispersal is negative correlation to the proportion of endemic species in flora.

REV: I am not pleased with this entire section of text (incl. the one further above), as the main points of criticism still remain to be addressed.

RESPONSE:

We reorganized the section to make our point clear. Our main idea is that differences in landform type have a constraining effect on species dispersal between mountains. For example, species could not spread freely between limestone and non-limestone mountains. Such restrictive effect caused by landform are general associated with the differs of bedrock, soil, water cycle processes.

The revised section is please see line338-359: ***Restricted dispersal between landforms as the result of environment filtering*** “*For many biogeographers, mountains are regarded as both barriers and bridges of species dispersal*”

Commented [A16]: I don't fully understand* this answer (due to use of English language and terminology), but will need to check the manuscript text whether any relevant amendments have been made to the actual text.

*e.g. which region are the authors referring to as being “of low habitat consistency ...”? Roles of – which/what kind of? – history?]

Commented [A17]: They can “spread” (better: disperse), but may not become established if they are not generalist, but specialist species adapted to specific bedrock. Authors need to take care of precise wording (importantly, in the manuscript text itself, for which this is even more important than in the reply text here).

Commented [A18]: See my previous comment. Dispersal and establishment are different matters.

Better/more precise:
Restricted dispersal with establishment ...
Restricted range expansion ...

Although we have made significant changes to this section. We responded to the reviewer's concerns accordingly thereafter.

RESPONSE:

We appreciate for this suggestion, which made our view more clear. We made corresponding changes in the manuscript.

See Line 346 “*Restricted dispersal with establishment between landforms as the result of environment filtering*”.

And line 349-352 “*However; the contribution of dispersal to montane floristic diversity largely depends on the ecological and physiological requirements of the species, as well as their disperse ability. Because, dispersal is only effective affects regional species diversity if it is followed by successful establishment.*”

The re-wrote paragraph please see lines 349-367. “*Dispersal also contributes to montane floristic diversity, although dispersal is only effective if it is followed by successful establishment. Dispersal occurs more freely during the initial stages of landform development in mountains, when slopes are still gentle and geographical barriers are minimal. For example, the plant dispersal events across the Qinghai-Tibet Plateau (low barrier) have been considerably greater than that in Hengduan Mountains (high barrier). The role of local recruitment was most important during the early stages of mountain flora assembly, a role which was subsequently replaced by local adaptations or in situ speciation. Mountains in different landforms will recruit different plant species, because each species is differently adapted to specific types of bedrock. The spatial configuration of mountain ranges alters their functional connectivity, thus influencing the species dispersal process. Furthermore, during the middle stages of landform development, the effects of dispersal on species diversity in mountainous areas are primarily restricted to lowlands, and are limited in highlands where there are more local endemic species. Taken together, this suggests that dispersal only weakly influences the unique composition of mountain flora associated with different landforms, especially considering that landforms exhibit a significant filtering effect. Variation in the species composition of mountain floras between different landforms will increase under the significant, combined landform environment filtering effect, and local endemic speciation. Therefore, we propose that the patchy spatial distribution of different mountain landforms is an important factor in shaping surface biogeographical zoning.*”

REV: Again, this text’s meaning is still largely unclear.

Qinghai-Tibet Plateau (low barrier) have been considerably greater than that in Hengduan Mountains (high barrier) – which kind of barrier do the authors means – and barrier to what?

RESPONSE:

“*For example, the plant dispersal events across the Qinghai-Tibet Plateau (low barrier) have been considerably greater than that in Hengduan Mountains (high barrier)*”.

– which kind of barrier do the authors means – and barrier to what?”.

The “barrier” here means the change of elevation gradient in Qinghai-Tibet Plateau (QTP) is

Commented [A19]: When I pose these questions, what I inherently mean is that those points are unclear to the reader from the text as currently presented in the manuscript. Thus, my recommendation in these cases is that the text in the manuscript itself to be clarified by re-phrasing by the authors.

relatively small than Hengduan Mountains (HDM) (see the fig below, left, cited from Ding et al., 2020). Such low-barrier of QTP is clearly more conducive to species dispersal, which is tested from the assembly of alpine biotas. Since the Miocene, the colonization rate QTP (0.06-0.25) is always large than the QHM (<0.05) (in Ding et al., 2020, Fig.2). In addition, there is no significant increase in species dispersal rate was detected in the Hengduan Mountains (although N-S-oriented) during the Quaternary in Ding's result (see the fig below, right).

[editorial note: figures redacted]

RESPONSE:

Done. Please see in line 355-362 : “ *The dispersal process is generally less constrained during the initial stages of mountain landform development, which are characterized by gentle slopes and limited geographical barriers*^{Error! Reference source not found.}. This scenario is well demonstrated by the assembly of alpine biotas. Since the Miocene, the colonization rate in gentle elevation gradient Qinghai-Tibet Plateau (QTP) (0.06-0.25) is always larger than that in the QHM (<0.05)^{Error! Reference source not found.}. The role of local species recruitment is most important during the early stages of mountain floristic assembly, subsequently being supplanted by local adaptations or in situ speciation due to the emergence of heterogeneous mountain environments^{Error! Reference source not found.} ”

REV: Role of local recruitment: In times of climate change, local recruitment from lowlands may play a potentially important role.

RESPONSE:

“The role of local recruitment was most important during the early stages of mountain flora assembly, a role which was subsequently replaced by local adaptations or in situ speciation^{24,57}.”

We understand the point “local recruitment in climate change” made by the reviewers. The range of species is expect shrinks during colding climate and expands during interglacial periods. But that doesn't contradict our viewpoint. Because no matter how the climate changes, species are always more easily dispersal in low geographical barriers. In the early stages of mountain flora assembly, the topographic fluctuation in the mountain is always relatively small.

RESPONSE:

We appreciate your comment. The sentence “Because no matter how the climate changes, species are always more easily dispersal in low geographical barriers” means the barrier effect of topological on species diffusion does not change in response to climate fluctuations. In other words, irrespective of climatic fluctuations, species exhibit enhanced dispersal capabilities in regions with minimal geographical barriers.

We further revised this section as follows: Line 359-362 “*The role of local species recruitment is most important during the early stages of mountain floristic assembly, subsequently being supplanted by local adaptations or in situ speciation due to the emergence of heterogeneous mountain environments*^{Error! Reference source not found.} ”

REV: “Spatial configuration” – meaning unclear.

RESPONSE:

Commented [A20]: As mentioned by me further above, the meaning needs to get through to the reader, by the way this is explained in the actual text.

I appreciate this is explained by the authors here in their reply, but it also – and importantly – needs to be understood from the actual text as presented in the manuscript.

Commented [A21]: This is not generally correct, it depends on whether in a given region/mountain, the climate change will lead to either an increase or decrease of suitable habitat area. This is dependent on mountain morphology (hypsography class).

Again, when I marked something as unclear or in need to more clarity in presentation, what I meant was that the text in the manuscript should be adjusted to make it easily understandable by readers.

Commented [A22]: This sentence is hard to follow.

“The spatial configuration of mountain ranges alters their functional connectivity, thus influencing the species dispersal process⁸⁷⁻⁸⁸.”

Here, “Spatial configuration” means topographical configuration, which is associated with the available ecological niches of a mountain along elevation gradient.

RESPONSE:

We have reorganized the sentence in order to make it clearer. Please see line 366-371: “The landform restriction effect on species diffusion gradually strengthens when more bedrock is exposed and the connectivity between mountains of different landforms is greatly reduced. Variation in the species composition of mountain floras between different landforms increases under the combined effects of landform, environmental filtering, and local endemic speciation. Therefore, we propose that the patchy spatial distribution of different mountain landforms is an important factor in shaping biogeographical zoning.”

“Geodiversity, the variability of Earth’s surface materials, forms, and physical processes, is an integral part of nature and crucial for sustaining ecosystems and their services. Several studies proposed to use geodiversity to model biodiversity based on the hypotheses that specific geo-sites should support unique biota and that high geodiversity is coupled with high biodiversity. However, the predictive power of geodiversity has not been high. For example, the geodiversity index (GD) was found to be no better than elevational range for predicting mountain species diversity. Another recent study reported that combinations of environmental variables better explained tropical species diversity than GD. In fact, the contribution of geodiversity component (GDC) showed similar patterns to topography, and usually effects at smaller extents and finer grain size. The GD is a poor predictor of mountain diversity in large part because interactions between mountains and biodiversity are complex, and thus geodiversity could not be treated as an index. In contrast, landform type is an objective description of the present state of erosion of a mountain or bedrocks. When we stand in front of a karst mountain, we can see quite clearly that the plants are different from those in a granite mountain. We propose that landform identity rather than GD, is a more suitable indicators of geodiversity for tracking changes in mountain floras along geological time scales.”

REV: I see a lot of problems with this text, for various reasons:

- Basically none of the sentences is clearly and understandably formulated
- scientific terms are not properly used
- the text, in parts, is misleading and, in parts, contains wrong claims

To illustrate this, I have above left some comments – many more would have been possible. I hope the authors will see what I mean when commenting about the necessity to improve this text considerable.

RESPONSE:

In this section, the reviewer #3 raised many questions in the form of annotation. We have carefully considered these review comments. In the new manuscript, we have decided to delete this part. First, delete this section does not affect the conclusion of our paper and makes the manuscript more brief and intelligible. Second, the definition, measurement and practical application of geodiversity

Commented [A23]:

Here again, when I marked something as unclear or in need to more clarity in presentation, what I meant was that the text in the manuscript should be adjusted to make it easily understandable for readers.

“Topographical” bears more information than “spatial”, but I still don’t fully understand the meaning of the sentence if “Topographical” is exchanged against “spatial”.

Just as a note aside: An ecological niche (sensu Hutchinson) is a characteristic of a species, not of the abiotic world. So “available ecological niches of a mountain” is, strictly speaking, not very precise.

Commented [A24]: I have looked at the comments by the authors below and have inserted my answers. However, since the authors say that they deleted the text altogether, this does not seem to have any impact on the ms anyway. Nevertheless, I rate that some of my answers may be useful for the authors to improve their future work and hopefully understand my comments and the reasoning behind a little better. It seems to me that the authors often did not get the main points I was hinting at.

remain confusion. Third, some questions raised by the reviewers here are ambiguous.

Hereafter are the point to point replies to the reviewer's questions raised in this section.

RESPONSE:

We greatly appreciate your continued patience in providing further clarification on previous comments. Furthermore, we kindly request your understanding regarding the removal of this section in our final manuscripts. Please note that we have not responded to your comments on geological diversity point by point. Because we did not strictly follow the concept of geological diversity to discuss its impact on mountain biodiversity. The inclusion of a discussion on geodiversity would be incongruous within the context of the article. Conversely, we firmly believe that the current organizational structure of the article enhances its comprehensibility.

However, we still highly appreciate the previous rounds of discussions we have had with you on this topic, during which we have dedicated significant thought to understanding the impact of geodiversity on mountain biodiversity. Our perspective on geodiversity is outlined as follows.

The introduction of the concept of geological diversity represents a significant advancement in the field of biodiversity development and conservation. Geodiversity forms a basis for biological diversity because organisms depend on the abiotic components of their environment. However, as an emerging key concept in the scientific community, its acceptance compared to biodiversity still needs to improve.

The meaning of geological diversity needs to be further clarified. Geodiversity, maybe mostly accept as the abiotic diversity of the Earth surface and sub-surface (Alahuhta, et al. 2020). It is still debated which factors should be included in geodiversity. The definition of “geodiversity” by Bailey (2017) maybe the suitable. However, in our view Bailey's “geodiversity” left out some important factors that reflect the geographical characteristics of the region, such as elevation. The truth is, elevation plays a crucial role in assessing the diversity of geological features (see Zarnetske et al., 2019, doi: 10.1111/geb.12887; Vernham et al., 2023, doi: 10.1016/j.tree.2023.02.010).

The ambiguous connotation of the concept of geodiversity makes different researchers confused about the which variables should be employed in the empirical study as so far (at least for us). This also significantly diminishes the comparability among the findings of diverse studies.

“Based on our results, we put forward the "geological lithology hypothesis of flora" to explain the assemble and differentiation of mountain floras. In this theoretical framework, floristic assembly in mountains is driven by the lithospheric cycle, which refers to the bedrock-constrained developmental processes of landforms. Specifically, under this hypothesis, the mountain species differentiation closely related to the type of bedrock and degree of erosion, species richness and species composition in mountain flora are interaction result of landform and environment, and phylogenetic niche evolution^{Error! Reference source not found.} can promote the spread of species among different landform floras. Overall, our study provides a novel framework and approach for determining the mechanisms of species diversity within mountains and the distributional patterns of some of the world's richest floras.”

REV: This section needs to be improved and provide a more balanced view on previous work by other authors and suggestions put forward here. As I had already suggested in my review of the first manuscript draft, the manuscript here needs to go into some more depth concerning previous hypotheses put forward, importantly the MGH, and I still don't see this has been done. While only briefly mentioned in the intro, the MGH does not show up here in the discussion. I would like to

see some of what the authors have answered in their rebuttal be actually also included here in the manuscript text.

Response:

This is a great suggestion. We rewrote this section. Please see line 389-413.

“..... different landform floras. This differs from the MGH, which assumes that the montane biodiversity hotspots require three key boundary conditions: 1) the presence of lowland, montane and alpine zones, 2) climatic fluctuations to produce a “species pump” effect, and 3) high-relief terrain with environmental in a given mountain region)^{Error! Reference source not found.}. In contrast, our hypothesis suggests that montane bedrocks and landform processes determine the geographic distribution of plants. The MGH effectively explains the high biodiversity of mountains characterized by large elevational differences (such as the Himalayan and Andes mountains), but such restrictive boundary conditions constrain the applicability of the hypothesis.”

REV: Similar to what I mentioned for the text above does also apply here. The authors need to pay attention to content correctness (e.g., when referring to the MGH, and comparing to their hypothesis), precision in formulation and use of terminology (e.g. concerning PNC) and claims.

In addition, it appears to me that especially in the last third of the text, different topics are being confused/intermixed. The MGH refers to mountains, but the authors talk about the “Namib desert” and “SE Asian karst landforms” for which the MGH would not apply anyway. They also talk about “global diversity patterns” when referring to their hypothesis. The MGH specifically deals with mountains, the hypothesis by the authors is supposed now to refer to global phenomena? If so, do the hypotheses actually deal with different matters? I strongly encourage the authors to carefully revise the text, and stick to what the hypotheses were developed for. Also, point in time geographic distribution of plants, and processes acting in concert, appear to be compared despite being different. As such, the text is not clear for the reader.

RESPONSE:

Thanks for the questions raised by the reviewers. We consider the differences between our new hypothesis and MGH in the spatial scale. In terms of scale differences. The MGH focus the biodiversity of a mountain which usually the presence of lowland, montane and alpine zones. This is a hypothesis of alpha diversity. In contrast our theory is more concerned with the process of mountain flora differentiation, which is a hypothesis of beta diversity. In this study, we investigate the flora assemble differences between mountains with different landform type.

We made further revisions to the manuscript to mad it clear. Please see line 376-381 “*To explain montane species diversity, this hypothesis differs from previous assumptions, such as MGH^{Error! Reference source not found.}, which is more focus on the biodiversity hotspots in high-altitude mountainous areas and the cause of diversity. Our hypothesis focuses more on mountainous areas with different altitudes and geological and lithological types, as well as their floristic compositions and differences, which is important for explaining differences in biodiversity hotspots from both low- to high-altitude area^{Error! Reference source not found.}.”*

RESPONSE:

Thanks for these helpful comments. We agree with you and made further changes to this section.

Please see line 388-396: “*To explain montane species diversity, this hypothesis differs from previous ones, such as MGH^{Error! Reference source not found.}, which focus more on the origination of high levels of biodiversity found in mountain systems. The MGH is invoked to explain the cause of alpha diversity. In contrast, our hypothesis is more concerned with the process*

Commented [A25]: Thanks for this clarification. I suggest this should then be explained as such in the ms.

The text following below would not fully meet this (apart from language issues), and introduces some further error. I include my comments below.

Commented [A26]: Care needs to be taken when using the term “biodiversity hotspots”. The MGH was originally proposed for the Tibet-Himalaya-Hengduan region, for which not all mountain systems qualify as biodiversity hotspots. This should be deleted, and instead could read something like:

.. such as the MGH, which focuses more on the origination of high levels of biodiversity found in mountain systems.

Commented [A27]: Why not formulate this the way it was explained before, in terms of alpha and beta diversity?

In my view, this would make the text’s content more accessible to the readers.

of mountain flora differentiation, which is a hypothesis of beta diversity. Here, we would like to introduce a new concept “landform flora” for mountain biodiversity study, meaning a unique flora formed under the influence of bedrock erosion and mountain landform development processes. Recognizing the differences that exist between different “landform flora” (eg., granitic flora, karst flora, Danxia flora) will benefit future study in prediction of mountain biodiversity, speciation, and also species protection^{Error! Reference source not found.,Error! Reference source not found.,Error! Reference source not found.,}

.....

The comments raised in the manuscript is also revised as below.

Line 44-45 “*In China, ten mountainous hotspot ecoregions were found to contain 92%.....*”

Line 278-280 “*Moreover, environmental filtering effects further contribute to the aggregation of species tolerant of cold and alpine environmental conditions, as anticipated by the phylogenetic niche conservatism (PNC)*^{Error! Reference source not found.,}”

REVIEWERS' COMMENTS

Reviewer #3 (Remarks to the Author):

I appreciate the further improvements of the manuscript effected by the authors, and their replies to my previous comments.

For convenience, I have inserted my comments, suggestions and corrections directly into the rebuttal. The corrections will need to be transferred to the actual manuscript file as well.

[editorial note: mentioned rebuttal is at the end of this page]

Reviewer:

I appreciate the further improvements of the manuscript effected by the authors, and their replies to my previous comments.

For convenience, I have inserted my comments, suggestions and corrections directly into this rebuttal. The corrections will need to be transferred to the actual manuscript file as well.

Formatted: Font: (Default) +Body (Calibri), Font color: Auto

Thanks for taking the time to review our manuscript. The new comments and suggestions from Reviewer #3 are helpful, and we appreciate them a lot. The manuscript has been furtherly revised following your comments marked in yellow (in both response letter and ms). The language in the manuscript and response has also been revised to make our point clearer. We provided point-by-point responses to your comments as following.

Our latest response is in red colour which began with "RESPONSE: ".

Deleted: ¶

Reviewer #3 (Remarks to the Author):

"Although it is well documented that mountains tend to exhibit high biodiversity, whether and how geological processes affect the assemblage of montane floras remains unclear. Here,"

RESPONSE:

Done. This sentence has been revised as "Although it is well documented that mountains tend to exhibit high biodiversity, how geological processes affect the assemblage of montane floras is a matter of ongoing research."

Commented [A1]: Delete "whether and" "remains unclear" -- > "is a matter of ongoing research"

"Furthermore, the evolutionary history of each biological taxa also affects local biodiversity. Reference source not found. Error! Reference source not found. ."

RESPONSE:

Done. This sentence has been revised as "Moreover, the unique evolutionary history of each biological taxon in mountainous regions could exert a profound influence on local biodiversity. Reference source not found. Error! Reference source not found. ."

Commented [A2]: taxon, not taxa

Commented [A3]: This is a rather awkwardly formulated sentence, I suggest to re-formulate this.

"Then, the species richness (SR), phylogenetic diversity (PD=Faith's PD), phylogenetic structure indices (PDI, NRI, NTI)"

RESPONSE:

Done. "Then, species richness (SR), phylogenetic diversity (PD=Faith's PD), phylogenetic structure indices (PDI, NRI, NTI) .."

Commented [A4]: delete "the"

"Thus, species with the deepest phylogenetic divergences occur in karst landforms, while species with the shallowest divergences occur in desert landforms. This is consistent with some fossil evidence. For example, the fossil flora discovered in southwestern China indicate that local karst

vegetation may have existed since the early Oligocene^{Error! Reference source not found.},^{Error! Reference source not found.}.”

Here, we would like to indicated the species in the limestone mountains are generally had earlier diverge age and could survive for a long time. Especially those species can adapted to the arid habitat of the limestone mountain top. We present an example, the Oligocene fossil flora Wenshan basin which located in Yunnan, China. The vegetation (infer from fossil data) and genera composition of this fossil flora in is very similar to the current karst flora in Wenshan, China.

Commented [A5]: Poor language (use of tenses), up to a degree which makes understanding of the meaning difficult for readers.

For the second question, “Discussion” occurs in Result section. The reason we did this is we want to give a brief explanation of the results or compared to previous study. We thought this would help readers understand the meaning of our results more quickly, and also avoid repetition before and after the article. We notes this approach is generally be accept in the articles published in Nature Communications. In the later “Discussion” section (line 286), we further discussed the relationship between landform process, species richness, local speciation, dispersal in mountain flora.

Commented [A6]: Due to poor language really hard to understand.

RESPONSE:

Here, we would like to indicate that the higher phylogenetic diversity index (PDI) of karst flora may be attributed to the long-term ecological stability. To be brief, karst flora exhibits long-term stability. As an example, we present the Oligocene fossil flora from Wenshan basin located in Yunnan, China. The vegetation (inferred from fossil data) and genera composition of this fossil flora closely resemble the current karst flora in Wenshan, China.

The following revision has been made to this part. See the Line 175-180 : “The PDI results serve as an indicator of the level of floristic stability. Here, we would like to indicate that the species inhabiting the arid limestone mountains generally exhibit an earlier divergence age and possess a remarkable capacity for long-term survival. Take for example, an Oligocene fossil flora discovered in Wenshan basin located in Yunnan, China, revealed a fossil assemblage (e.g., *Burretiodendron*, *Ficus microtrivialis*), of which clearly indicated current local karst vegetation, may have existed since the early Oligocene^{Error! Reference source not found.},^{Error! Reference source not found.}.”

Commented [A7]: I suggest to change this to:

generally → in our data

Commented [A8]: minor editorial adjustment from my side

Commented [A9]: Due to use of language, the meaning of this sentence is hard to understand.

Would the following convey the message the authors intend to put forward?

“...), which clearly indicates close taxonomic affinities to the current karst vegetation, the latter which may have existed in its basic composition since the early Oligocene ...”

The second question pertains to the paper writing format. In our manuscript, we have incorporated several concise discussions within the “Results” section. This approach has been adopted to provide a succinct explanation of our findings and compare them with previous studies, aiming to enhance readers' comprehension without unnecessary repetition throughout the article. Such an writing format is accepted by Nature Communications.

Question about previous response.

Second, based on the technical requirements (this is based on the null model assumption) of the software package, flora data is extracted from the mountain species database to reflect the phylogenetic structure of the flora. Thus, a global or regional tree is not needed here. In fact, as we know there isn't currently an approach that is calculates a regional phylogenetic structure based on a global tree.

REV: The meaning of this sentence is not clear to me. Of course, there exist studies that have been using e.g. large angiosperm-wide trees as source for creating much smaller trees with only regional species samples representation. This needs clarification.

RESPONSE:

“Using e.g. large angiosperm-wide trees as source for creating much smaller trees with only regional species samples representation”, which is also the method used in our study.

However, the meaning of “In fact, as we know there isn’t currently an approach that calculates a regional phylogenetic structure based on a global tree.” is we can’t use a large phylogenetic tree (included about 30000 species) to calculate the PDI, NRI and NTI of floristic distribution dataset included less species (such as 10 floras in total included 3000 species). The species in the phylogenetic tree should be consistent to the floristic distribution dataset, otherwise would violate the assumption of the null model.

So that the software warning reason here is not the inconsistencies of taxonomic name. The true reason is species number in the phylogenetic tree not consistent to the floristic distribution dataset (more species included in phylogenetic tree than floristic distribution dataset, as the case we shown above, less species in floristic dataset than in phylogenetic tree).

“ Warning: the input matrix has fewer columns than the number of species in the tree. ”

Please also refer to the previous reply. We hope our response made it clear.

RESPONSE:

Here, the reviewer expressed concerns regarding the phylogenetic tree employed in this research. It seems like the reviewer is not familiar with phylogenetic or community phylogenetic approaches, which has led to some misunderstandings. Further elaboration will be provided below. The key point seems to be “why not use large angiosperm-wide trees” and “why R packages warning”. To illustrate this, we have presented a brief case below, accompanied by detailed instructions.

The phylogenetic structure indices (PDI, NRI and NTI) is calculated by R packages 'PhyloMeasures' and 'picante' based on phylogenetic tree and species distribution data. The consistency between species occurs in the phylogenetic tree and those in the distribution dataset is a fundamental prerequisite for ensuring the operational functionality of R packages 'PhyloMeasures' (otherwise, it would violate the null model).

In the following case, species *Pertya cormbosa* (in red frame,) not occurs in phylogenetic tree2 and distribution data2. When utilizing the combination of phylogenetic tree2 and distribution data1, or phylogenetic tree1 and distribution data2 for calculating phylogenetic structure indices (PDI, NRI, and NTI), the 'PhyloMeasures' package will generate a 'warning'.

The relevant research methods have been extensively accepted, and the details have been elaborated upon in our manuscript (please see manuscript line 448-475). Therefore, we did not make further changes on methods in the new manuscript (to maintain the manuscript's simplicity). If you want to know more details about phylogenies and ecology, several literature is suggested here (Webb et al. 2002, doi: 10.1146/annurev.ecolsys.33.010802.150448; Tsirigiannis, C. & Sandel, B. 2016, doi: 10.1111/ecog.01814; Kembel et al., 2010, doi: 10.1093/bioinformatics/btq166).

We hope our response will answer your concerns.

Commented [A10]: Unfortunately, I don't understand the reply (due to poor quality of English language).

Commented [A11]: I still don't fully understand from this reply why the error message would complain about the missing “species names” if the names would have been resolved prior or during analysis, and – I assume - measures been taken to include missing data? I am missing the context between the previous and the new replies. I still don't fully understand the connections from the reply here.

Commented [A12]: If authors do, at first, not recognise the background to a reviewer's questions, it would be a fallacy to claim that the reviewer has no expertise in this specialist area. It is the authors' task to explain the facts so clearly that they can be understood by the readership.

As there has already been considerable previous communication about the issue of taxon sampling and use of trees with the reviewer(s) (see also my previous comments further below regarding related issues), I will leave it there. The chief editor may have the last word to decide whether the issue is now satisfactorily dealt with in the manuscript, or further justification is needed.

The topic here is related to issues dealt with further below. Since issues have been dealt with and improvements are evident, this may be viewed in favour.

We suspect there are no differences between “species distinctiveness” and species age, because both of them are based on the in-clade in the phylogenetic tree. Besides, species age (or species divergence time) is more well known than “species distinctiveness” in evolutionary biology and biogeography.

REV: What do the authors mean by “in-clade”?

RESPONSE:

Here, “in-clade” just represents the branches of each species contained in the phylogenetic tree.

RESPONSE:

We are here for further clarification. The so called “in-clade” proposed by us is used to explain the term “species distinctiveness” (which raised by review#2). The meaning of “in-clade” is a evolution branch of a phylogenetic tree. And, the branch length is closely associated with species age. The words “in-clade” does not occurs in our MS. We are sorry for any confusion have caused for you.

In our study, we used the mean divergence times (MDT) proposed by Lu et al.(2018) to represent the “flora age”. As such, the MDT reflected the age composition of species within a flora. A flora with larger MDT has more ancient species and hence is expected to have older floristic ages. Overall, we believe that our use of MDT to measure the species age structure of flora is consistent with the existing literature and therefore is appropriate.

REV: Not clear to me. Only because many people have been doing something does not mean it is correct or the most appropriate way to do it.

RESPONSE:

Reviewer # 3 mentioned that “many people are doing the same thing, which does not necessarily mean that this approach is correct”. We cannot fully agree with the comments of Reviewer #3. Many people have used this method to carry out a lot of work, although it cannot be considered right, it cannot be considered wrong, and it cannot be considered that several journals currently

Commented [A13]: I still don't understand from this reply what the authors mean by this.

publish the wrong method? Especially, if there is no better update method available, traditional methods can only be used.

However, frankly, it is difficulty of develop a better or appropriate method to measure the species age structure of flora. Because species age occurs in a flora is still can't be well defined. At present, we can only study the floristic assemble process on a macroscale of statistical significance. So that in our study we used the MDT method proposed by Lu et al. (2018), which is not so bad. Development new methods to understand the species assemble process of flora over time scale is an important topic for future research.

In addition, in this article, we only hope to calculate the aggregation and divergence of plant flora at a macro scale. Therefore, we used the MDT method proposed by Lu et al. (2018). The limitations of this method have been explained and revised as previous reviewer comments (such as MS Line 461-470). Developing new methods to reveal the species composition of plant flora on a temporal scale is an important topic for future research. Of course, we would greatly appreciate it if the reviewer could recommend a better method to us.

RESPONSE:

Thank you for further clarification of your concerns. We acknowledge that it is still impossible to determine the true age of each species in the flora based on current research approach. The limitations of MDT method have been explained and mentioned in previous manuscript (section "Divergence time estimation"), and also Extended Data Fig. 8.

In the new revised manuscript, we further clarify this issue. Please see Line 481-492:

"In this approach, divergence time of species is expected to be overestimated, as the branch of some species in a local phylogeny is usually longer than that in global phylogeny (included all species). For instance, if a lineage became extinct, the divergence time of its existing closest relative species would be dated at the point of their last common ancestor. To assess the robustness and the effect of this sampling bias on the final results of species age structure of a mountain flora, we used four divergence time datasets and found similar MDT patterns between mountains of different landforms (Extended Data Fig. 8). This result is consistent with a study^{Error! Reference source not found.}, which found that "in large-scale biodiversity and phylogenetic analyses, sources of noise in divergence time estimation are to be expected, but they did not affect the reliability of the results". We believe that our dated megaphylogenetic tree was suitable for this study because our aim was to reveal the general patterns of landform influence on the formation of mountain flora rather than focusing on the age of each species. "

"Here, we apply the term "flora" to refer to the sum of all angiosperm families, genera, and species growing on a specific mountain or in a well-delimited area^{Error! Reference source not found.},^{Error! Reference source not found.}, which is a relatively independent biogeographical unit^{Error! Reference source not found.}."

REV: I would argue against the flora of a mountain being a *relatively independent biogeographical unit*, if this is what is meant here (but I may have misunderstood the sentence). First, because related lineages are shared between mountains, and second, because most biogeographers, in terms of terminology, would not consider the flora of a mountain as a "biogeographical unit".

Also in **Pre-response:** "we mentioned is not only a unit, but als" I am not sure how "unit" here would refer to the biogeographic units mentioned in the ms. This may be worth of clarification.

Commented [A14]: Unfortunately, I don't fully understand all parts of this reply (due to poor quality of English language).

However, from what I understand the authors claim here, I want to assure the authors that I was actually never asking to "develop a new method". Rather, I was pointing out already previously the limitations of the approach, that when using a local geographical species pool for divergence time estimations, "true" ages of species cannot possibly be inferred. Why? Because the closest relatives of those species (which may occur elsewhere and are therefore not included in the tree) would be required to be added in such a dating analysis, to infer better estimates of the species' ages. Likewise, extinction will likely be different in different taxa/clades (e.g. due to taxon age, and/or geological/climatic events in different mountain regions), and a tree which is missing closest relatives will then be even more prone to overlooking this effect. This was the reason I was referring to "random" distribution of extinction.

The authors previously replied that they do not intend to present the "true" ages, but just the relative ages. Nevertheless, even then, because of uneven taxonomic sampling, bias will naturally be introduced. This limitation should be mentioned.

The authors claim that other studies have done similar as they do. However, at the very least, the limitations of this approach should be mentioned in the manuscript. The answer by the authors does not invalidate my calls for caution.

Commented [A15]: including

Commented [A16]: I suggest to re-write this last part of the sentence, as it is unclear what "reliability of results" actually means. How would "reliability" be defined?

Response: We appreciate the key question raised by the reviewer. In general, the term “biogeographic unit” is used in the context of biogeographic regionalization.

Here we consider that the flora of mountain and biogeographic zones is equivalent at small scale.

REV: This is unclear. Which biogeographic zones are meant here?

RESPONSE:

(1) Here, the mountains wasn't refer to a single peak, but rather to a mountainous area or a certain nature reserve, including many peaks and adjacent areas. Correspondingly, all the plants growing in this area consist of many different species, genera, and families, whose was called on a mountain flora, it is a relatively independent natural geographical area. Due to differences in longitude, latitude, altitude, and climate factors, biological factor (or floristic compose) among different mountainous regions, these geographical regions are also bound to have differences.

(2) The biogeographical units mentioned in the main text cannot be equated with biogeographical divisions. Biogeographic zoning often divides any geographical areas into several level or grades, such as the global flora, which is often divided into kingdom, region, province, and county. These are different biogeographical units (hierarchical units). The unit itself does not have hierarchical significance, and only after studying all species in the region, namely Flora, its hierarchical level or grade be determined, such as the local flora, the geographical elements of the families, genera, or species (tropical, temperate elements), historical elements (antiquity), originative elements (endemic families, genera, species etc.), ecological elements, migration elements, etc., Only through floristic phyto-geography analysis, a detain units such as a kingdom, region, province, and county be divided.

A certain biogeographical unit, such as a mountain area, may also be divided into different geographical regions, such as the Hengduan Mountains in China and the Himalayan region, which are divided into the East Asian Flora and the China-Himalayan Forest Subregion. In the east, it is divided into the East Asian Flora and the China-Japan Forest Subregion; Correspondingly, the latter can be divided into South China Province, Central China Province, etc. (Wu Zhengyi et al., 1996). A certain mountain area, depending on its nature geography and biogeography property, they may divided into a hierarchical unit (division) or a subunit. Or they could be belonged a province, or a county.

Therefore, when we mention a mountain region, or a mountain flora, similar to a nature geographic or biogeographical area or unit with a certain grade or no grade, it is not a wrong concept.

(3) This article studied 140 mountainous areas, listed the natural geographic information of each mountainous area, and correspondingly formed 140 local mountainous floras, thereby showcasing their differences in natural geography and floristic phyto-geography.

Furthermore, this article successfully classified 140 mountains into five geomorphic types (by searching for detailed geological survey data, and special investigation report on Floras), revealing and analyzing the relationship and possible reasons between these local floras and mountainous lithology. We believe that this is a progress. On this basis, it will be possible to further study endemism, geographical elements of those floras, and diffusion, migration of floristic elements between different geomorphic types and different bedrock, floras.

(4) Of course, in our response to the reviewer's question, we wrote biogeographic zones, which are indeed not strict enough and have a larger scope. We just wanted to explain that each mountain has its unique characteristics. Wu Zhengyi et al. (1996) considered that an area of every relative independent mountainous flora covered should not be less than 100 square kilometers. Due to the large amount of content and limited space, these concepts and data were not included in the main text.

RESPONSE:

Thanks for further explanation. We accept your view on using of biogeography terms. In the revised MS, we removed the “, which is a relatively independent biogeographical unit” order to create ambiguity.

The sentence is revised as: “Here, we used the term “flora” to refer to the collection of all angiosperm species growing on a specific mountain or in a well-delimited area”

First, mountain is geographical barrier to plants diffusion.

REV: This is not correct (as I had stated in the previous rounds of review already), phrased in this general manner. It depends on the orientation of the mountain. Only if a mountain has W-E orientation, it can act as barrier (e.g. Himalayas, Alps). For N-S-oriented mountains (such as the Andes, or the Hengduan mountains), this is not correct.

RESPONSE:

We believe that it is unnecessary for the reviewers to oppose the above views. The author's viewpoint is simply that mountains can affect the diffusion of species or serve as a barrier of species diffusion. In fact, mountains are both a barrier for species diffusion and may also play a promoting role in species diffusion. For example, the north-south direction mountains play a promoting role in the north-south migration of species, while serving as a barrier for the east-west migration of species; Due to the existence of Hengduan Mountains, many genera and species in Chinese Mainland have formed China-Himalaya distribution subtypes and China-Japan distribution subtypes, with Hengduan Mountains as the boundary. Similarly, due to the barrier effect of the Nanling Mountain in China, many tropical genera and species cannot exceed the Nanling Mountain, such as *Pandanus*, *Endospermum*, etc.

The reviewer mentioned or emphasized the local migration, endemism, and differentiation of species within a mountainous area, especially in high mountains. Our focus is to emphasize that the migration of species between different mountainous areas will be hindered, especially between Danxia landforms, karst landforms, and granite landforms, where species diffusion is not easy.

RESPONSE:

We have made modifications in the previous MS. Please see Line 347-349.

“For many biogeographers, mountains are regarded as both barriers and bridges of species dispersal. The role of mountains as corridors has been documented in several mountains has a North-South orientation, such as the Andes and Hengduan Mountains”

Commented [A17]: The authors present some general textbook biogeographic contents here. I assume many readers of *Nature* will be familiar with these.

However, I could not find a clear connection to my previous claim that I would not generally agree to “the flora of a mountain being a relatively independent biogeographical unit”.

My point was to call for caution when using terms such as “independent biogeographical unit”, or “biogeographic zone”.

This needs to be taken care of as it might lead to confusion among readers. As the authors point out themselves in their reply, biogeography has its own terminology and concepts.

Commented [A18]: My point was simply that the statement is not correct, when phrased in this general manner. This is still valid, and thus I suggest, now more concretely, to rephrase this:

... mountains can act as barriers to plant diffusion ...

Commented [A19]: This sentence is grammatically not correct and needs re-wording.

I assume authors mean something like:

The role of mountains as corridors has been documented in those of North-South orientation, such as the Andes⁸⁸ and Hengduan Mountains⁸⁹.”

OR

The role of mountains as corridors has been documented in several with North-South orientation, such as the Andes⁸⁸ and Hengduan Mountains⁸⁹.”

Second, a mountain usually shows sky island effects and contain their own unique clades or several endemic species.

REV: This is not correct. First, there is no such thing like a “sky island effect” (phrased this way). Second, only a small fraction of the world’s mountains qualify as sky islands – only, if their vegetation is drastically different from the surrounding lowlands. This may only be the case if plants living on the mountains are different from the lowlands and were therefore not likely recruited/evolved from lowlands in which case the nearest relatives would come from other mountains (example: some mountains located in drylands, e.g. Africa).

RESPONSE:

Here is just a brief discussion with the reviewer. The “sky island effect” of mountain flora is a question worth further study and discussing. As our understand, in essence the sky islands mountains is qualified by its flora showed discontinuous distribution. In generally, in mountains vertical zonality of vegetation on the elevation gradient is widespread obserded. We suspect the plants or community occurs on the top of the mountain should be treated as a result of sky islands effects? These community differs from the low land, and could represents the early stage of species assemble of mountain flora.

RESPONSE:

The content of this section is tangential to the concepts presented in our current manuscript; therefore, we have refrained from making corresponding amendments within the manuscript. The following is a concise discussion aimed at providing reviewers with a clearer understanding of our perspectives.

We agree with you, no such a term or concept “sky island effect” has been formally proposed. In our perspective, the “sky island effect” reflects species diffuse from one mountain top to another mountain top is restricted. Although the constraints on diffusion are comparatively less pronounced than that in a typical sky islands mountains. The “sky island effect” of mountain flora is a question worth further study and discussing. This is an issue worthy of further study. Anyway, we appreciate your comments on precise use of terminology.

There are also some literature to suggest that mountain region could be viewed as a biogeographical unit. For instance, Rahbek et al. (2019, Science 365, 1108-1113) wrote that “Like an island, a mountain region may be viewed as a biogeographical unit in itself, with in situ speciation and extinction playing a key role in building the regional species assemblage”.

REV: Is anyone else except this cited paper claiming this as well? I could imagine that most mountain biogeographers would disagree with this oversimplification.

RESPONSE:

We cannot agree with the reviewer's opinion, as stated in the previous response.

RESPONSE:

We accept your view on using of biogeography terms, and revised this sentence as bellow.

“Here, we used the term “flora” to refer to the collection of all angiosperm species growing on a specific mountain or in a well-delimited area^{Error! Reference source not found.}^{Error! Reference source not found.}.”

“The mountains of sedimentary bedrock have much stronger environmental filtering effect, as

Commented [A20]: What I was intending to suggest to the authors by my previous comment was to tune down “a mountain usually shows sky island effects”. First, because “usually” is not correct, as this would imply this statement holds true for the majority of mountains, which is not the case. Secondly, “sky island effects” is unclear to me. What is a sky island effect? Has this even been defined, and by whom?

In short, what I had intended with my previous comment was to re-formulate this.

Commented [A21]: from

Commented [A22]: Something is wrong with this sentence – due to language, the meaning is unclear.

Commented [A23]: Same here. Meaning not entirely clear.

Commented [A24]: See my answers before. My point was to call for caution when using terms such as “independent biogeographical unit”, or “biogeographic zone”.

The citation from Rahbek et al. “may be viewed as a biogeographical unit in itself” seems like an agreeable cautionary phrasing.

It is different from the authors’ statement “independent biogeographical unit”, or “biogeographic zone”.

these mountain ecosystems are more sensitive to rainfall. The environmental filtering effect further promoted the clustering of phylogenetic structures of mountain floras as predicted by the phylogenetic niche conservatism (PNC) Error! Reference source not found., Error! Reference source not found., Error! Reference source not found.

REV: It would be interesting to discuss here radiations more generally, e.g., how do the studies of plant radiations included in the global study by Muellner-Riehl et al. 2019 (JBI) compare to these statements/findings? Are the findings of these studies (and other studies published since then, i.e. after 2019) comparable to this? I suggest looking at the mountain systems (and their bedrocks) where these studies were undertaken.

Response: These are really good advice. The potential radiation speciation occurs when plants colonize from one landform to another which maybe an important path of plant diversification. Here, we document several cases (not all) on adaptative radiation that occur on the bedrocks. But the relations between radiations and bedrocks/landform are still poorly known. We suspect that the effect of landform and bedrock, which promote species differentiation, is more or less neglected under the shadow of “climate change”. Nevertheless, we have reorganized this paragraph carefully.

REV: I don’t understand the reasoning here, as currently phrased. It is unclear what the authors mean (potential speciation radiation?). There exist different kinds of radiations, not all of them are “adaptive”.

RESPONSE:

Here we would like to shown adaptive evolution (associate with morphological traits and physiological traits) maybe the most important mechanism for plant to radiation in a new environment. Although the mechanisms of radiative evolution are diverse, plants ultimately need to adapt to their local environment in order to survive. Furthermore, evolutionary radiations underpinned by variation in physiological or behavioral traits can more easily be perceived as non-adaptive, compared to those involving more conspicuous morphological traits, causing a bias in our understanding of the extent and distribution of adaptive radiations in nature. That is reason a large number of traits of convergence (such as Alpine flora, Arid flora, and mangrove plants) and adaptive evolution have been broadly observed (Nevado et al. 2019, doi: 10.1016/j.cub.2019.07.059; Xia et al. 2021, doi: 10.1093/molbev/msab314).

RESPONSE:

The role of plant radiation evolution on floristic assemblage is discussed in the section “Bedrock promotes local speciation resulting in floristic differentiation between landforms” of ms. In our view, species differentiation caused by bedrock differences is an important source of mountain floristic differences.

Please see Line 332-344 “On the other hand, adaptive evolution, encompassing both morphological and physiological traits, could play a pivotal role in facilitating the diversification of plants in novel environments Error! Reference source not found. In plants, radiative evolution often accompanies habitat and landform shifts, as seen in Old World gesneriads Error! Reference source not found. and North American **deserts** rock daisies (Compositae tribe Perityleae) Error! Reference source not found. Similar patterns can also be observed in insects. For example, two radiated clades (nodes 14 to 16 and nodes 31 to 33) of Exocelina **resulted from the transition from uplifted Australian Plate bedrock to ultramafic/ophiolite** Error! Reference source not found. Although many previous studies also document climate change as an important driver of species radiations Error! Reference source not found., Error! Reference source

Commented [A25]: From this response I cannot not see how my previous request (for discussing this in the ms) is now actually tackled in the manuscript.

Commented [A26]: desert

Commented [A27]: phrasing sounds unusual, maybe better use “radiating clades”

Commented [A28]: “clades ... result from” – here again, I suggest more precise re-phrasing, as the meaning is here is not entirely clear. Do authors mean something like:

“the radiation of clades ... has been suggested .. as having been connected – to the ecological transition from ...”?

Commented [A29]: edit

not found.,Error! Reference source not found.,Error! Reference source not found.. The adaptive radiation of plants is more or less associated with bedrock type. For example, the development of a key innovation (lime-secreting hydathodes) may have made Saxifraga sect. Porphyron better suited to limestone habitats,Error! Reference source not found., and the low specific leaf area (SLA) exhibited by Erica is an adaptation to oligotrophic habitats (quartzite/sandstone) in the Cape,Error! Reference source not found.. These obvious landform and bedrock effects could strongly promote both species and floristic differentiation between different regions,Error! Reference source not found.,Error! Reference source not found.,Error! Reference source not found..”

“Dispersal also contributes to the mountain diversity,Error! Reference source not found.,Error! Reference source not found.,Error! Reference source not found.. A well-known example is climatic fluctuations during the Quaternary ice age drove changes in the distribution of species around the globe,Error! Reference source not found.,Error! Reference source not found.. The spatial configuration of mountain range affect their functional connectivity, influencing species dispersal,Error! Reference source not found.,Error! Reference source not found.. In fact, dispersal more easily occurs at initial stage of landform develop mountain, as the slope of mountain still gentle and geographical barrier still relatively low,Error! Reference source not found.. Such as the plants dispersal events in Qinghai-Tibet Plateau (QTP, low barrier) is obvious higher than that in Hengduan Mountains (high barrier),Error! Reference source not found.. Furthermore, the effects of dispersal on species diversity in mountainous areas mainly occurred at lowland, and is limited at highlands where has more local endemic species,Error! Reference source not found.,Error! Reference source not found.. This means that dispersal has weak influence on the unique composition of mountain flora of different landforms, especially considering that landform have a significant filtering effect on the species.”

REV: The entire paragraph needs re-writing. Dispersal between mountains is not limited to times of climate change, which somehow is suggested here as currently written. Dispersal and establishment was easier in the Hengduan mountains at times of climate change as they have a N-S orientation, enabling plants to change their latitudinal distribution range more easily. Reading this, I am not sure the authors have read the papers I had suggested to consult, and which they cited in the paper already before (e.g. Ding et al.).

Response: We accept this advice and wrote this paragraph. We would like to clarify the questions about dispersal events and diffusion barrier. We had carefully readed the paper by Ding et al (2020). It is not difficult to deduce the conclusion “*dispersal events in Qinghai-Tibet Plateau (QTP, low barrier) is obvious higher than that in Hengduan Mountains (high barrier)*” (see Ding et al, 2020, fugure 2). On the other hand, we thought dispersal would reduce the β -diversity between different floras. Such as the proportion of endemic species in Hengduan Mountains is the highest is largely affect by its low rate of colonization (<0.05), and high rate of *in situ* speciation and local and recruitment.

REV: This could also/additionally be the effect of higher levels of extinction on the QTP, in contrast to the Hengduan mountains.

RESPONSE: This may be right as the QTP has lower habitat heterogeneity than the Hengduan Mountains. The role of extinction rate QTP and Hengduan Mountains are not mentioned in the paper by Ding et al (2020). A region with low habitat consistency are associated with lower biodiversity and also more species with strong ecological adaptation. Distinguishing the roles of history, speciation and extinction in the mountain flora assamble still represents a major challenge

Commented [A30]: The first sentence of the two sentences here is not complete. Is it supposed to be connected to the next sentence? This needs re-phrasing.

Commented [A31]: This is a too general statement and needs re-phrasing.

Commented [A32]: Or, “may be”?

Commented [A33]: edit

Commented [A34]: I suggest to tune down this formulation slightly (caution for difference between correlation and causality), i.e., delete “obvious”.

to future research.

RESPONSE:

Sorry for any misunderstanding caused by “of low habitat consistency ...”. This is a language translation error. Our point is that areas characterized by low habitat heterogeneity exhibit correspondingly low levels of biodiversity.

We further clarify the relationship between diffusion and mountain landform development. Please see in line 355-362 : “The dispersal process is generally less constrained during the initial stages of mountain landform development, which are characterized by gentle slopes and limited geographical barriers. This scenario is well demonstrated by the assembly of alpine biotas. Since the Miocene, the colonization rate in gentle elevation gradient Qinghai-Tibet Plateau (QTP) (0.06-0.25) is always larger than that in the QHM (<0.05). The role of local species recruitment is most important during the early stages of mountain floristic assembly, subsequently being supplanted by local adaptations or in situ speciation due to the emergence of heterogeneous mountain environments.”

In contrast, the low proportion of endemic species in Qinghai-Tibet Plateau has the highest rate of colonization since early Miocene. It is clearly the dispersal is negative correlation to the proportion of endemic species in flora.

REV: I am not pleased with this entire section of text (incl. the one further above), as the main points of criticism still remain to be addressed.

RESPONSE:

We reorganized the section to make our point clear. Our main idea is that differences in landform type have a constraining effect on species dispersal between mountains. For example, species could not spread freely between limestone and non-limestone mountains. Such restrictive effect caused by landform are general associated with the differs of bedrock, soil, water cycle processes. The revised section is please see line338-359: **Restricted dispersal between landforms as the result of environment filtering** “For many biogeographers, mountains are regarded as both barriers and bridges of species dispersal.”

Although we have made significant changes to this section. We responded to the reviewer's concerns accordingly thereafter.

RESPONSE:

We appreciate for this suggestion, which made our view more clear. We made corresponding changes in the manuscript.

See Line 346 “Restricted dispersal with establishment between landforms as the result of environment filtering”.

And line 349-352 “However, the contribution of dispersal to montane floristic diversity largely depends on the ecological and physiological requirements of the species, as well as their disperse ability. Because, dispersal is only effective affects regional species diversity if it is followed by successful establishment.”

Commented [A35]: I don't fully understand* this answer (due to use of English language and terminology), but will need to check the manuscript text whether any relevant amendments have been made to the actual text.

[*e.g. which region are the authors referring to as being “of low habitat consistency ...”? Roles of – which/what kind of? – history?]

Commented [A36]: This needs re-phrasing. The QTP had Andean shape at times of its earlier geological evolution, so “gentle elevation gradient” may be viewed as potentially misleading, also given that different parts of the QTP (which first had no plateau shape) had different “shape” at different times in geological history.

Commented [A37]: There is a very recent paper by Carruthers et al. (2024) on this topic, precisely disentangle the contribution of these different processes through time (“quantify the processes that generate alpine plant diversity and their changing dynamics through time”). Carruthers, T., Moerland, M.S., Ebersbach, J. et al. Repeated upslope biome shifts in *Saxifraga* during late-Cenozoic climate cooling. *Nat Commun* 15, 1100 (2024). <https://doi.org/10.1038/s41467-024-45289-w>

Commented [A38]: They can “spread” (better: disperse), but may not become established if they are not generalist, but specialist species adapted to specific bedrock. Authors need to take care of precise wording (importantly, in the manuscript text itself, for which this is even more important than in the reply text here).

Commented [A39]: See my previous comment. Dispersal and establishment are different matters.

Better/more precise:
Restricted dispersal with establishment ...
Restricted range expansion ...

Commented [A40]: dispersal

The re-wrote paragraph please see lines 349-367. “Dispersal also contributes to montane floristic diversity^{Error! Reference source not found.,Error! Reference source not found.,Error! Reference source not found.}, although dispersal is only effective if it is followed by successful establishment^{Error! Reference source not found.}. Dispersal occurs more freely during the initial stages of landform development in mountains, when slopes are still gentle and geographical barriers are minimal^{Error! Reference source not found.}. For example, the plant dispersal events across the Qinghai-Tibet Plateau (low barrier) have been considerably greater than that in Hengduan Mountains (high barrier)^{Error! Reference source not found.}. The role of local recruitment was most important during the early stages of mountain flora assembly, a role which was subsequently replaced by local adaptations or in situ speciation^{Error! Reference source not found.,Error! Reference source not found.}. Mountains in different landforms will recruit different plant species, because each species is differently adapted to specific types of bedrock^{Error! Reference source not found.,Error! Reference source not found.}. The spatial configuration of mountain ranges alters their functional connectivity, thus influencing the species dispersal process^{Error! Reference source not found.,Error! Reference source not found.}. Furthermore, during the middle stages of landform development, the effects of dispersal on species diversity in mountainous areas are primarily restricted to lowlands, and are limited in highlands where there are more local endemic species^{Error! Reference source not found.,Error! Reference source not found.}. Taken together, this suggests that dispersal only weakly influences the unique composition of mountain flora associated with different landforms, especially considering that landforms exhibit a significant filtering effect. Variation in the species composition of mountain floras between different landforms will increase under the significant, combined landform environment filtering effect^{Error! Reference source not found.,Error! Reference source not found.}, and local endemic speciation^{Error! Reference source not found.}. Therefore, we propose that the patchy spatial distribution of different mountain landforms is an important factor in shaping surface biogeographical zoning.”

REV: Again, this text’s meaning is still largely unclear.

Qinghai-Tibet Plateau (low barrier) have been considerably greater than that in Hengduan Mountains (high barrier) – which kind of barrier do the authors means – and barrier to what?

RESPONSE:

“For example, the plant dispersal events across the Qinghai-Tibet Plateau (low barrier) have been considerably greater than that in Hengduan Mountains (high barrier)⁵⁷.” – which kind of barrier do the authors means – and barrier to what?”

The “barrier” here means the change of elevation gradient in Qinghai-Tibet Plateau (QTP) is relatively small than Hengduan Mountains (HDM) (see the fig below, left, cited from Ding et al., 2020). Such low-barrier of QTP is clearly more conducive to species dispersal, which is tested from the assembly of alpine biotas. Since the Miocene, the colonization rate QTP (0.06-0.25) is always large than the QHM (<0.05) (in Ding et al., 2020, Fig.2). In addition, there is no significant increase in species dispersal rate was detected in the Hengduan Mountains (although N-S-oriented) during the Quaternary in Ding’s result (see the fig below, right).

[editorial note: figures redacted]

RESPONSE:

Done. Please see in line 355-362 : “The dispersal process is generally less constrained during the

Commented [A41]: When I pose these questions, what I inherently mean is that those points are unclear to the reader from the text as currently presented in the manuscript. Thus, my recommendation in these cases is that the text in the manuscript itself to be clarified by re-phrasing by the authors.

Commented [A42]: As mentioned by me further above, the meaning needs to get through to the reader, by the way this is explained in the actual text.

I appreciate this is explained by the authors here in their reply, but it also – and importantly – needs to be understood from the actual text as presented in the manuscript.

initial stages of mountain landform development, which are characterized by gentle slopes and limited geographical barriers^{Error! Reference source not found.,Error! Reference source not found.}. This scenario is well demonstrated by the assembly of alpine biotas. Since the Miocene, the colonization rate in gentle elevation gradient Qinghai-Tibet Plateau (QTP) (0.06-0.25) is always larger than that in the QHM (<0.05)^{Error! Reference source not found.}. The role of local species recruitment is most important during the early stages of mountain floristic assembly, subsequently being supplanted by local adaptations or in situ speciation due to the emergence of heterogeneous mountain environments^{Error! Reference source not found.,Error! Reference source not found.}.”

Commented [A43]: See my comments to the same text before.

REV: Role of local recruitment: In times of climate change, local recruitment from lowlands may play a potentially important role.

RESPONSE:

“The role of local recruitment was most important during the early stages of mountain flora assembly, a role which was subsequently replaced by local adaptations or in situ speciation^{24,57.}” We understand the point “local recruitment in climate change” made by the reviewers. The range of species is expect shrinks during colding climate and expands during interglacial periods. But that doesn't contradict our viewpoint. Because no matter how the climate changes, species are always more easily dispersal in low geographical barriers. In the early stages of mountain flora assembly, the topographic fluctuation in the mountain is always relatively small.

RESPONSE:

We appreciate your comment. The sentence “Because no matter how the climate changes, species are always more easily dispersal in low geographical barriers” means the barrier effect of topological on species diffusion does not change in response to climate fluctuations. In other words, irrespective of climatic fluctuations, species exhibit enhanced dispersal capabilities in regions with minimal geographical barriers.

We further revised this section as follows: Line 359-362 “The role of local species recruitment is most important during the early stages of mountain floristic assembly, subsequently being supplanted by local adaptations or in situ speciation due to the emergence of heterogeneous mountain environments^{Error! Reference source not found.,Error! Reference source not found.}.”

Commented [A44]: This is not generally correct, it depends on whether in a given region/mountain, the climate change will lead to either an increase or decrease of suitable habitat area. This is dependent on mountain morphology (hypsography class).

Again, when I marked something as unclear or in need to more clarity in presentation, what I meant was that the text in the manuscript should be adjusted to make it easily understandable by readers.

Commented [A45]: This sentence is hard to follow.

Commented [A46]: See comment before

REV: “Spatial configuration” – meaning unclear.

RESPONSE:

“The spatial configuration of mountain ranges alters their functional connectivity, thus influencing the species dispersal process^{87-88.}”

Here, “Spatial configuration” means topographical configuration, which is associated with the available ecological niches of a mountain along elevation gradient.

RESPONSE:

We have reorganized the sentence in order to make it clearer. Please see line 366-371: “The landform restriction effect on species diffusion gradually strengthens when more bedrock is exposed and the connectivity between mountains of different landforms is greatly reduced^{Error! Reference source not found.,Error! Reference source not found.,Error! Reference source not found.}. Variation in the species composition of mountain floras between different landforms increases under the combined effects

Commented [A47]:

Here again, when I marked something as unclear or in need to more clarity in presentation, what I meant was that the text in the manuscript should be adjusted to make it easily understandable for readers.

“Topographical” bears more information than “spatial”, but I still don't fully understand the meaning of the sentence if “Topographical” is exchanged against “spatial”.

Just as a note aside: An ecological niche (sensu Hutchinson) is a characteristic of a species, not of the abiotic world. So “available ecological niches of a mountain” is, strictly speaking, not very precise.

of landform, environmental filtering^{Error! Reference source not found.}, and local endemic speciation^{Error! Reference source not found.}. Therefore, we propose that the patchy spatial distribution of different mountain landforms is an important factor in shaping biogeographical zoning.”

“Geodiversity, the variability of Earth’s surface materials, forms, and physical processes, is an integral part of nature and crucial for sustaining ecosystems and their services^{Error! Reference source not found.}. Several studies proposed to use geodiversity to model biodiversity based on the hypotheses that specific geo-sites should support unique biota and that high geodiversity is coupled with high biodiversity^{Error! Reference source not found.}. However, the predictive power of geodiversity has not been high. For example, the geodiversity index (GD) was found to be no better than elevational range for predicting mountain species diversity^{Error! Reference source not found.}. Another recent study reported that combinations of environmental variables better explained tropical species diversity than GD^{Error! Reference source not found.}. In fact, the contribution of geodiversity component (GDC) showed similar patterns to topography, and usually effects at smaller extents and finer grain size^{Error! Reference source not found.}. The GD is a poor predictor of mountain diversity in large part because interactions between mountains and biodiversity are complex^{Error! Reference source not found.}, and thus geodiversity could not be treated as an index. In contrast, landform type is an objective description of the present state of erosion of a mountain or bedrocks^{Error! Reference source not found.}. When we stand in front of a karst mountain, we can see quite clearly that the plants are different from those in a granite mountain. We propose that landform identity rather than GD, is a more suitable indicators of geodiversity^{Error! Reference source not found.} for tracking changes in mountain floras along geological time scales.”

REV: I see a lot of problems with this text, for various reasons:

- Basically none of the sentences is clearly and understandably formulated
- scientific terms are not properly used
- the text, in parts, is misleading and, in parts, contains wrong claims

To illustrate this, I have above left some comments – many more would have been possible. I hope the authors will see what I mean when commenting about the necessity to improve this text considerable.

RESPONSE:

In this section, the reviewer #3 raised many questions in the form of annotation. We have carefully considered these review comments. In the new manuscript, we have decided to delete this part. First, delete this section does not affect the conclusion of our paper and makes the manuscript more brief and intelligible. Second, the definition, measurement and practical application of geodiversity remain confusion. Third, some questions raised by the reviewers here are ambiguous. Hereafter are the point to point replies to the reviewer’s questions raised in this section.

RESPONSE:

We greatly appreciate your continued patience in providing further clarification on previous comments. Furthermore, we kindly request your understanding regarding the removal of this section in our final manuscripts. Please note that we have not responded to the your comments on geological diversity point by point. Because we did not strictly follow the concept of geological diversity to discuss its impact on mountain biodiversity. The inclusion of a discussion on

Commented [A48]: I have looked at the comments by the authors below and have inserted my answers. However, since the authors say that they deleted the text altogether, this does not seem to have any impact on the ms anyway. Nevertheless, I rate that some of my answers may be useful for the authors to improve their future work and hopefully understand my comments and the reasoning behind a little better. It seems to me that the authors often did not get the main points I was hinting at.

geodiversity would be incongruous within the context of the article. Conversely, we firmly believe that the current organizational structure of the article enhances its comprehensibility.

However, we still highly appreciate the previous rounds of discussions we have had with you on this topic, during which we have dedicated significant thought to understanding the impact of geodiversity on mountain biodiversity. Our perspective on geodiversity is outlined as follows.

The introduction of the concept of geological diversity represents a significant advancement in the field of biodiversity development and conservation. Geodiversity forms a basis for biological diversity because organisms depend on the abiotic components of their environment. However, as an emerging key concept in the scientific community, its acceptance compared to biodiversity still needs to improve.

The meaning of geological diversity needs to be further clarified. Geodiversity, maybe mostly accepted as the abiotic diversity of the Earth surface and sub-surface (Alahuhta, et al. 2020). It is still debated which factors should be included in geodiversity. The definition of "geodiversity" by Bailey (2017) maybe the suitable. However, in our view Bailey's "geodiversity" left out some important factors that reflect the geographical characteristics of the region, such as elevation. The truth is, elevation plays a crucial role in assessing the diversity of geological features (see Zarnetske et al., 2019, doi: 10.1111/geb.12887; Vernham et al., 2023, doi: 10.1016/j.tree.2023.02.010).

The ambiguous connotation of the concept of geodiversity makes different researchers confused about the which variables should be employed in the empirical study as so far (at least for us). This also significantly diminishes the comparability among the findings of diverse studies.

"Based on our results, we put forward the "geological lithology hypothesis of flora" to explain the assemble and differentiation of mountain floras. In this theoretical framework, floristic assembly in mountains is driven by the lithospheric cycle, which refers to the bedrock-constrained developmental processes of landforms. Specifically, under this hypothesis, the mountain species differentiation closely related to the type of bedrock and degree of erosion, species richness and species composition in mountain flora are interaction result of landform and environment, and phylogenetic niche evolution^{Error! Reference source not found.} can promote the spread of species among different landform floras. Overall, our study provides a novel framework and approach for determining the mechanisms of species diversity within mountains and the distributional patterns of some of the world's richest floras."

REV: This section needs to be improved and provide a more balanced view on previous work by other authors and suggestions put forward here. As I had already suggested in my review of the first manuscript draft, the manuscript here needs to go into some more depth concerning previous hypotheses put forward, importantly the MGH, and I still don't see this has been done. While only briefly mentioned in the intro, the MGH does not show up here in the discussion. I would like to see some of what the authors have answered in their rebuttal be actually also included here in the manuscript text.

Response:

This is a great suggestion. We rewrote this section. Please see line 389-413.

".... different landform floras. This differs from the MGH, which assumes that the montane biodiversity hotspots require three key boundary conditions: 1) the presence of lowland, montane and alpine zones, 2) climatic fluctuations to produce a "species pump" effect, and 3) high-relief

terrain with environmental in a given mountain region)^{Error! Reference source not found.,Error! Reference source not found.}. In contrast, our hypothesis suggests that montane bedrocks and landform processes determine the geographic distribution of plants. The MGH effectively explains the high biodiversity of mountains characterized by large elevational differences (such as the Himalayan and Andes mountains), but such restrictive boundary conditions constrain the applicability of the hypothesis.”

REV: Similar to what I mentioned for the text above does also apply here. The authors need to pay attention to content correctness (e.g., when referring to the MGH, and comparing to their hypothesis), precision in formulation and use of terminology (e.g. concerning PNC) and claims. In addition, it appears to me that especially in the last third of the text, different topics are being confused/intermixed. The MGH refers to mountains, but the authors talk about the “Namib desert” and “SE Asian karst landforms” for which the MGH would not apply anyway. They also talk about “global diversity patterns” when referring to their hypothesis. The MGH specifically deals with mountains, the hypothesis by the authors is supposed now to refer to global phenomena? If so, do the hypotheses actually deal with different matters? I strongly encourage the authors to carefully revise the text, and stick to what the hypotheses were developed for. Also, point in time geographic distribution of plants, and processes acting in concert, appear to be compared despite being different. As such, the text is not clear for the reader.

RESPONSE:

Thanks for the questions raised by the reviewers. We consider the differences between our new hypothesis and MGH in the spatial scale. In terms of scale differences. The MGH focus the biodiversity of a mountain which usually the presence of lowland, montane and alpine zones. This is a hypothesis of alpha diversity. In contrast our theory is more concerned with the process of mountain flora differentiation, which is a hypothesis of beta diversity. In this study, we investigate the flora assemble differences between mountains with different landform type.

We made further revisions to the manuscript to mad it clear. Please see line 376-381 “To explain montane species diversity, this hypothesis differs from previous assumptions, such as MGH^{Error! Reference source not found.,Error! Reference source not found.}, which is more focus on the biodiversity hotspots in high-altitude mountainous areas and the cause of diversity. Our hypothesis focuses more on mountainous areas with different altitudes and geological and lithological types, as well as their floristic compositions and differences, which is important for explaining differences in biodiversity hotspots from both low- to high-altitude area^{Error! Reference source not found.,Error! Reference source not found.}.”

RESPONSE:

Thanks for these helpful comments. We agree with you and made further changes to this section. Please see line 388-396: “To explain montane species diversity, this hypothesis differs from previous ones, such as the MGH^{Error! Reference source not found.,Error! Reference source not found.}, which focus more on the origination of high levels of biodiversity found in mountain systems. The MGH is invoked to explain the cause of alpha diversity. In contrast, our hypothesis is more concerned with the process of mountain flora differentiation, which is a hypothesis of beta diversity. Here, we would like to introduce a new concept “landform flora” for mountain biodiversity study, meaning a unique flora formed under the influence of bedrock erosion and mountain landform development processes. Recognizing the differences that exist between different “landform flora” (e.g., granitic flora, karst flora, Danxia flora) will benefit future study in prediction of mountain biodiversity, speciation, and also species protection^{Error! Reference source not found.,Error! Reference source not found.,Error! Reference source not found.}.”

Commented [A49]: Thanks for this clarification. I suggest this should then be explained as such in the ms.

The text following below would not fully meet this (apart from language issues), and introduces some further error. I include my comments below.

Commented [A50]: Care needs to be taken when using the term “biodiversity hotspots”. The MGH was originally proposed for the Tibet-Himalaya-Hengduan region, for which not all mountain systems qualify as biodiversity hotspots. This should be deleted, and instead could read something like:

.. such as the MGH, which focuses more on the origination of high levels of biodiversity found in mountain systems.

Commented [A51]: Why not formulate this the way it was explained before, in terms of alpha and beta diversity?

In my view, this would make the text’s content more accessible to the readers.

Commented [A52]: the

Commented [A53]: edit

.....
The comments raised in the manuscript is also revised as below.

Line 44-45 "In China, ten mountainous hotspot ecoregions were found to contain 92%....."

Line 278-280 "Moreover, environmental filtering effects further contribute to the aggregation of species tolerant of cold and alpine environmental conditions, as anticipated by phylogenetic niche conservation (PNC)^{Error! Reference source not found.}"

Commented [A54]: edit

Deleted: the